# A Block Coordinate Descent Method for Nonsmooth Composite Optimization under Orthogonality Constraints

## Abstract

Nonsmooth composite optimization with orthogonality constraints is crucial in statistical learning and data science, but it presents challenges due to its nonsmooth objective and computationally expensive, non-convex constraints. In this paper, we propose a new approach called **OBCD**, which leverages Block Coordinate Descent (BCD) to address these challenges. **OBCD** is a feasible method with a small computational footprint. In each iteration, it updates $k$ rows of the solution matrix, where $k \geq 2$, while globally solving a small nonsmooth optimization problem under orthogonality constraints. We prove that the limiting points of **OBCD**, referred to as (global) block-$k$ stationary points, offer stronger optimality than standard critical points. Furthermore, we show that **OBCD** converges to $\epsilon$-block-$k$ stationary points with an ergodic convergence rate of $\mathcal{O}(1/\epsilon)$. Additionally, under the Kurdyka-Lojasiewicz (KL) inequality, we establish the non-ergodic convergence rate of **OBCD**. We also extend **OBCD** with breakpoint searching methods for subproblem solving and greedy strategies for working set selection. Comprehensive experiments demonstrate the superior performance of our approach across various tasks.

## 1 Introduction

We consider the following nonsmooth composite optimization problem under orthogonality constraints ('$\triangleq$' means define):

$$\min_{\mathbf{X} \in \mathbb{R}^{n \times r}} F(\mathbf{X}) \triangleq f(\mathbf{X}) + h(\mathbf{X}), \ s.t. \ \mathbf{X}^{\mathsf{T}}\mathbf{X} = \mathbf{I}_r. \tag{1}$$

Here, $n \geq r$ and $\mathbf{I}_r$ is a $r \times r$ identity matrix. We do not assume convexity of $f(\mathbf{X})$ and $h(\mathbf{X})$. For brevity, the orthogonality constraints $\mathbf{X}^{\mathsf{T}}\mathbf{X} = \mathbf{I}_r$ in Problem (1) is rewritten as $\mathbf{X} \in \mathrm{St}(n, r) \triangleq \{\mathbf{X} \in \mathbb{R}^{n \times r} \mid \mathbf{X}^{\mathsf{T}}\mathbf{X} = \mathbf{I}_r\}$, where $\mathcal{M} \triangleq \mathrm{St}(n, r)$ is the Stiefel manifold in the literature (Edelman et al., 1998; Absil et al., 2008; Wen & Yin, 2013; Hu et al., 2020). We impose the following assumptions on Problem (1) throughout this paper. (Asm-i) For any $\mathbf{X}$ and $\mathbf{X}^+$, where $\mathbf{X}$ and $\mathbf{X}^+$ only differ at most by $k$ rows with $k \geq 2$, we assume $f : \mathbb{R}^{n \times r} \mapsto \mathbb{R}$ is $\mathbf{H}$-smooth with $\mathbf{0} \preceq \mathbf{H} \in \mathbb{R}^{nr \times nr}$ such that:

$$f(\mathbf{X}^+) \leq \mathcal{Q}(\mathbf{X}^+; \mathbf{X}) \triangleq f(\mathbf{X}) + \langle \mathbf{X}^+ - \mathbf{X}, \nabla f(\mathbf{X}) \rangle + \tfrac{1}{2}\|\mathbf{X}^+ - \mathbf{X}\|_{\mathbf{H}}^2, \tag{2}$$

where $\|\mathbf{H}\|_{\mathsf{sp}} \leq L_f$ for some constant $L_f > 0$ and $\|\mathbf{X}\|_{\mathbf{H}}^2 \triangleq \mathrm{vec}(\mathbf{X})^{\mathsf{T}}\mathbf{H}\mathrm{vec}(\mathbf{X})$ [1]. Here, $\|\mathbf{H}\|_{\mathsf{sp}}$ is the spectral norm of $\mathbf{H}$. Notably, when $\mathbf{H} = L_f \cdot \mathbf{I}_{nr}$, this condition simplifies to the standard $L_f$-smoothness (Nesterov, 2003). (Asm-ii) The function $h(\mathbf{X}) : \mathbb{R}^{n \times r} \mapsto \mathbb{R}$ is closed, proper, and lower semicontinuous, and potentially non-smooth. Additionally, it is coordinate-wise separable, such that $h(\mathbf{X}) = \sum_{i,j} h(\mathbf{X}_{ij})$. Typical examples of $h(\mathbf{X})$ include the $\ell_p$ norm function $h(\mathbf{X}) = \|\mathbf{X}\|_p$ with $p \in \{0, 1\}$, and the indicator function for non-negativity constraints $h(\mathbf{X}) = \mathcal{I}_{\geq 0}(\mathbf{X})$. (Asm-iii) The following small-sized subproblem can be solved exactly and efficiently:

$$\min_{\mathbf{V} \in \mathrm{St}(k,k)} \mathcal{P}(\mathbf{V}) \triangleq \tfrac{1}{2}\|\mathbf{V}\|_{\mathbf{Q}}^2 + \langle \mathbf{V}, \mathbf{P} \rangle + h(\mathbf{V}\mathbf{Z}), \tag{3}$$

---

[1] Given any symmetric matrices $\mathbf{C} \in \mathbb{R}^{n \times n}$ and $\mathbf{D} \in \mathbb{R}^{r \times r}$, we let $\mathbf{H} = \mathbf{D} \otimes \mathbf{C}$. The function $f(\mathbf{X}) = \tfrac{1}{2}\mathrm{tr}(\mathbf{X}^{\mathsf{T}}\mathbf{C}\mathbf{X}\mathbf{D}) = \tfrac{1}{2}\|\mathbf{X}\|_{\mathbf{H}}^2$ satisfies (2) with equality, as $f(\mathbf{X}^+) = \mathcal{Q}(\mathbf{X}^+; \mathbf{X})$ holds for all $\mathbf{X}$ and $\mathbf{X}^+$.

for any given $\mathbf{Z} \in \mathbb{R}^{k \times r}$, $\mathbf{P} \in \mathbb{R}^{k \times k}$, and $\mathbf{Q} \in \mathbb{R}^{k^2 \times k^2}$. Here, we employ a notational simplification by defining $h(\mathbf{V}\mathbf{Z}) \triangleq \sum_{i,j} h([\mathbf{V}\mathbf{Z}]_{ij})$, given the coordinate-wise separability of the function $h(\cdot)$.

Problem (1) is an optimization framework that plays a crucial role in a variety of statistical learning and data science models, such as sparse Principal Component Analysis (PCA) (Journée et al., 2010; Shalit & Chechik, 2014), nonnegative PCA (Zass & Shashua, 2006; Qian et al., 2021), deep neural networks (Cogswell et al., 2016; Cho & Lee, 2017; Xie et al., 2017; Bansal et al., 2018; Massart & Abrol, 2022; Huang & Gao, 2023), electronic structure calculation (Zhang et al., 2014; Liu et al., 2014), Fourier transforms approximation (Frerix & Bruna, 2019), phase synchronization (Liu et al., 2017), orthogonal nonnegative matrix factorization (Jiang et al., 2022), $K$-indicators clustering (Jiang et al., 2016), and dictionary learning (Zhai et al., 2020).

## 1.1 RELATED WORK

We now present some related algorithms in the literature.

▶ **Minimizing Smooth Functions under Orthogonality Constraints.** One difficulty in solving Problem (1) arises from the nonconvexity of the orthogonality constraints. Existing methods for handling this issue can be divided into three classes. *(i)* Geodesic-like methods (Abrudan et al., 2008; Edelman et al., 1998; Absil et al., 2008; Jiang & Dai, 2015). Since calculating geodesics involves solving ordinary differential equations, which may cause computational complexity, geodesic-like methods iteratively compute the geodesic logarithm using simple linear algebra calculations. The work of (Wen & Yin, 2013) develops a simple and efficient constraint preserving update scheme and achieves low computation complexity per iteration. They combine the feasible update scheme with the Barzilai-Borwein (BB) nonmonotonic line search for optimization with orthogonality constraints. *(ii)* Projection-like methods (Absil et al., 2008; Golub & Van Loan, 2013). These methods preserve the orthogonality constraints by projection. They decrease the objective value using its current Euclidean gradient direction or Riemannian tangent direction, followed by an orthogonal projection operation. This can be calculated by polar decomposition or approximated by QR factorization. *(iii)* Multiplier correction methods (Gao et al., 2018; 2019; Xiao et al., 2022). Since the Lagrangian multiplier associated with the orthogonality constraint is symmetric and has an explicit closed-form expression at the first-order optimality condition, multiplier correction methods update the multiplier after achieving sufficient reduction in the objective function. This leads to efficient first-order feasible or infeasible approaches.

▶ **Minimizing Nonmooth Functions under Orthogonality Constraints.** Another difficulty of solving Problem (1) comes from the nonsmoothness of the objective function. Existing methods for addressing this problem can be classified into three categories. *(i)* Subgradient methods (Hwang et al., 2015; Li et al., 2021; Cheung et al., 2024). Subgradient methods are analogous to gradient descent methods. Most of the aforementioned geodesic-like and projection-like strategies can be incorporated into the subgradient methods. However, the step size in subgradient methods needs to be diminishing to guarantee convergence. *(ii)* Proximal gradient methods (Chen et al., 2020; Li et al., 2024). They solve a strongly convex minimization problem over the tangent space using a semi-smooth Newton method to find a descent direction. Subsequently, they maintain the orthogonality constraint through a retraction operation. *(iii)* Block Majorization Minimization (BMM) or BCD on Riemannian manifolds (Li et al., 2024; 2023; Breloy et al., 2021; Gutman & Ho-Nguyen, 2023; Cheung et al., 2024). This class of methods iteratively constructs a tangential majorizing surrogate for a block of the objective function, takes an approximate descent step in the resulting direction within the tangent space, and then applies retraction to project back onto the manifold. Notably, their subproblems are often solved approximately, whereas our method can solve them exactly due to the small size of the subproblems. *(iv)* Operator splitting methods (Lai & Osher, 2014; Chen et al., 2016; Zhang et al., 2019). Operator splitting methods introduce linear constraints and decompose the original problem into simpler subproblems, which can be solved separately and exactly. Alternating Direction Methods of Multipliers (ADMM) (He & Yuan, 2012) and Smoothing Penalty Methods (SPM) (Chen, 2012) represent two prominent variants of operator splitting methods.

▶ **Block Coordinate Descent Methods.** (Block) coordinate descent is a classical and powerful algorithm that solves optimization problems by iteratively performing minimization along (block) coordinate directions (Tseng & Yun, 2009; Xu & Yin, 2013). The BCD methods have recently gained attention in solving nonconvex optimization problems, including sparse optimization (Yuan, 2024), $k$-means clustering (Nie et al., 2022), structured nonconvex minimization (Yuan, 2023), recurrent neural network (Massart & Abrol, 2022), and multi-layer convolutional networks (Bibi et al.,

2019; Zeng et al., 2019). BCD methods have also been used in (Shalit & Chechik, 2014; Massart & Abrol, 2022) for solving optimization problems with orthogonal group constraints. However, their column-wise BCD methods are limited only to solve smooth minimization problems with $k = 2$ and $r = n$ (Refer to Section 4.2 in (Shalit & Chechik, 2014)). Our row-wise BCD methods can solve general nonsmooth problems with $k \geq 2$ and $r \leq n$. The work of (Gao et al., 2019) proposes a parallelizable column-wise BCD scheme for solving the subproblems of their proximal linearized augmented Lagrangian algorithm. Impressive parallel scalability in a parallel environment of their algorithm is demonstrated. We stress that our **row-wise** BCD methods differ from the two **column-wise** counterparts.

▶ **Summary.** Existing solutions have one or more of the following limitations: *(i)* They rely on full gradient information, incurring high computational costs per iteration. *(ii)* They cannot handle general nonsmooth composite problems. *(iii)* They lack descent properties, even worse, they are infeasible methods, achieving solution feasibility only at the limit point. *(iv)* They often lack rigorous convergence guarantees. *(v)* They only establish weak optimality at critical points. ★ To our knowledge, this represents the first application of BCD methods to solve nonsmooth composite optimization problems under orthogonality constraints, demonstrating strong optimality and convergence guarantees.

## 1.2 Contributions

This paper makes the following contributions. *(i)* Algorithmically: We propose a Block Coordinate Descent (BCD) algorithm tailored for nonsmooth composite optimization under orthogonality constraints (Section 2). *(ii)* Theoretically: We provide comprehensive optimality and convergence analyses of our methods (Sections 3 and 4). *(iii)* Side Contributions: We introduce breakpoint searching methods for solving subproblems when $k = 2$ (Section 5), and present two working set selection greedy strategies to improve the computational efficiency of our methods (Section D in the Appendix). *(iv)* Empirically: Extensive experiments demonstrate that our methods surpass existing solutions in terms of accuracy and/or efficiency (Section 6).

## 2 The Proposed **OBCD** Algorithm

In this section, we introduce **OBCD**, a Block Coordinate Descent algorithm for solving general nonsmooth composite problems under Orthogonality constraints, as defined in Problem 1.

We start by presenting a new update scheme designed to maintain the orthogonality constraint.

▶ **A New Constraint-Preserving Update Scheme**. For any partition of the index vector $[1, 2, ..., n]$ into $[\text{B}, \text{B}^c]$ with $\text{B} \in \mathbb{N}^k$, $\text{B}^c \in \mathbb{N}^{n-k}$, we define $\text{U}_\text{B} \in \mathbb{R}^{n \times k}$ and $\text{U}_{\text{B}^c} \in \mathbb{R}^{n \times (n-k)}$ as: $(\text{U}_\text{B})_{ji} = \left\{ \begin{array}{ll} 1, & \text{B}_i = j; \\ 0, & \text{else.} \end{array} \right.$ , $(\text{U}_{\text{B}^c})_{ji} = \left\{ \begin{array}{ll} 1, & \text{B}_i^c = j; \\ 0, & \text{else.} \end{array} \right.$ . Therefore, we have the following variable splitting for any $\mathbf{X} \in \mathbb{R}^{n \times r}$: $\mathbf{X} = \mathbf{I}_n \mathbf{X} = (\text{U}_\text{B} \text{U}_\text{B}^\mathsf{T} + \text{U}_{\text{B}^c} \text{U}_{\text{B}^c}^\mathsf{T}) \mathbf{X} = \text{U}_\text{B} \mathbf{X}(\text{B}, :) + \text{U}_{\text{B}^c} \mathbf{X}(\text{B}^c, :)$, where $\mathbf{X}(\text{B}, :) = \text{U}_\text{B}^\mathsf{T} \mathbf{X} \in \mathbb{R}^{k \times r}$ and $\mathbf{X}(\text{B}^c, :) = \text{U}_{\text{B}^c}^\mathsf{T} \mathbf{X} \in \mathbb{R}^{(n-k) \times r}$.

In each iteration $t$, the indices $\{1, 2, ..., n\}$ of the rows of decision variable $\mathbf{X} \in \text{St}(n, r)$ are separated to two sets $\text{B}$ and $\text{B}^c$, where $\text{B}$ is the working set with $|\text{B}| = k$ and $\text{B}^c = \{1, 2, ..., n\} \setminus \text{B}$. To simplify notation, we use $\text{B}$ instead of $\text{B}^t$, as $t$ can be inferred from the context. We only update $k$ rows of the variable $\mathbf{X}$ via $\mathbf{X}^{t+1}(\text{B}, :) \Leftarrow \mathbf{V} \mathbf{X}^t(\text{B}, :)$ for some appropriate matrix $\mathbf{V} \in \mathbb{R}^{k \times k}$. The following equivalent expressions hold:

$$\mathbf{X}^{t+1}(\text{B}, :) = \mathbf{V} \mathbf{X}^t(\text{B}, :) \quad \Leftrightarrow \quad \mathbf{X}^{t+1} = (\text{U}_\text{B} \mathbf{V} \text{U}_\text{B}^\mathsf{T} + \text{U}_{\text{B}^c} \text{U}_{\text{B}^c}^\mathsf{T}) \mathbf{X}^t \tag{4}$$

$$\Leftrightarrow \quad \mathbf{X}^{t+1} = \mathbf{X}^t + \text{U}_\text{B}(\mathbf{V} - \mathbf{I}_k) \text{U}_\text{B}^\mathsf{T} \mathbf{X}^t. \tag{5}$$

We consider the following minimization procedure to iteratively solve Problem (1):

$$\min_{\mathbf{V}} F(\mathcal{X}_\text{B}^t(\mathbf{V})), \ s.t. \ \mathcal{X}_\text{B}^t(\mathbf{V}) \in \text{St}(n, r), \ \text{where } \mathcal{X}_\text{B}^t(\mathbf{V}) \triangleq \mathbf{X}^t + \text{U}_\text{B}(\mathbf{V} - \mathbf{I}_k) \text{U}_\text{B}^\mathsf{T} \mathbf{X}^t. \tag{6}$$

The following lemma shows that the orthogonality constraint for $\mathbf{X}^+ = \mathbf{X} + \text{U}_\text{B}(\mathbf{V} - \mathbf{I}_k) \text{U}_\text{B}^\mathsf{T} \mathbf{X}$ can be preserved by choosing suitable $\mathbf{V}$ and $\mathbf{X}$.

**Lemma 2.1.** *(Proof in Appendix E.1) We let* $\text{B} \in \{\mathcal{B}_i\}_{i=1}^{\text{C}_n^k}$, *where the set* $\{\mathcal{B}_1, \mathcal{B}_2, ..., \mathcal{B}_{\text{C}_n^k}\}$ *denotes all possible combinations of the index vectors choosing $k$ items from $n$ without repetition. We let*

$\mathbf{V} \in \text{St}(k,k)$. *We define* $\mathbf{X}^+ \triangleq \mathcal{X}_{\text{B}}(\mathbf{V}) \triangleq \mathbf{X} + \text{U}_{\text{B}}(\mathbf{V} - \mathbf{I}_k)\text{U}_{\text{B}}^{\mathsf{T}}\mathbf{X}$. *(a) For any* $\mathbf{X} \in \mathbb{R}^{n \times r}$, *we have* $[\mathbf{X}^+]^{\mathsf{T}}\mathbf{X}^+ = \mathbf{X}^{\mathsf{T}}\mathbf{X}$. *(b) If* $\mathbf{X} \in \text{St}(n,r)$, *then* $\mathbf{X}^+ \in \text{St}(n,r)$.

Thanks to Lemma 2.1, we can now explore the following alternative formulation for Problem (6).

$$\bar{\mathbf{V}}^t \in \arg\min_{\mathbf{V}} F(\mathcal{X}_{\text{B}}^t(\mathbf{V})), \ s.t. \ \mathbf{V} \in \text{St}(k,k). \tag{7}$$

Then the solution matrix is updated via: $\mathbf{X}^{t+1} = \mathcal{X}_{\text{B}}^t(\bar{\mathbf{V}}^t)$.

The following lemma offers important properties for the update rule $\mathbf{X}^+ = \mathbf{X} + \text{U}_{\text{B}}(\mathbf{V} - \mathbf{I}_k)\text{U}_{\text{B}}^{\mathsf{T}}\mathbf{X}$.

**Lemma 2.2.** *(Proof in Appendix E.2) We define* $\mathbf{X}^+ = \mathbf{X} + \text{U}_{\text{B}}(\mathbf{V} - \mathbf{I}_k)\text{U}_{\text{B}}^{\mathsf{T}}\mathbf{X}$. *For any* $\mathbf{X} \in \text{St}(n,r)$, $\mathbf{V} \in \text{St}(k,k)$, $\text{B} \in \{\mathcal{B}_i\}_{i=1}^{\text{C}_n^k}$, *and symmetric matrix* $\mathbf{H} \in \mathbb{R}^{nr \times nr}$, *we have:*

*(a)* $\frac{1}{2}\|\mathbf{X}^+ - \mathbf{X}\|_{\mathbf{H}}^2 = \frac{1}{2}\|\mathbf{V} - \mathbf{I}_k\|_{\underline{\mathbf{Q}}}^2$, *where* $\underline{\mathbf{Q}} \triangleq (\mathbf{Z}^{\mathsf{T}} \otimes \text{U}_{\text{B}})^{\mathsf{T}}\mathbf{H}(\mathbf{Z}^{\mathsf{T}} \otimes \text{U}_{\text{B}})$, *and* $\mathbf{Z} \triangleq \text{U}_{\text{B}}^{\mathsf{T}}\mathbf{X} \in \mathbb{R}^{k \times r}$.

*(b)* $\frac{1}{2}\|\mathbf{X}^+ - \mathbf{X}\|_{\mathsf{F}}^2 = \langle \mathbf{I}_k - \mathbf{V}, \text{U}_{\text{B}}^{\mathsf{T}}\mathbf{X}\mathbf{X}^{\mathsf{T}}\text{U}_{\text{B}}\rangle$.

*(c)* $\frac{1}{2}\|\mathbf{X}^+ - \mathbf{X}\|_{\mathsf{F}}^2 \leq \frac{1}{2}\|\mathbf{V} - \mathbf{I}_k\|_{\mathsf{F}}^2 = \langle \mathbf{I}_k, \mathbf{I}_k - \mathbf{V}\rangle$.

▶ **The Main Algorithm**. The proposed algorithm **OBCD** is an iterative procedure that sequentially minimizes the objective function along block coordinate directions within a sub-manifold of $\mathcal{M}$.

Starting with an initial feasible solution, **OBCD** iteratively determines a working set $\text{B}^t$ using specific strategies. It then solves the small-sized subproblem in Problem (7) through successive majorization minimization. This method iteratively constructs a surrogate function that majorizes the objective function, driving it to decrease as expected (Mairal, 2013; Razaviyayn et al., 2013; Sun et al., 2016; Breloy et al., 2021), and it has proven effective for minimizing complex functions.

We now demonstrate how to derive the majorization function for $F(\mathcal{X}_{\text{B}}^t(\mathbf{V}))$ in Problem (7). Initially, for any $\mathbf{X}^t \in \text{St}(n,r)$ and $\mathbf{V} \in \text{St}(k,k)$, we establish following inequalities: $f(\mathcal{X}_{\text{B}}^t(\mathbf{V})) - f(\mathbf{X}^t) \overset{①}{\leq} \langle \mathcal{X}_{\text{B}}^t(\mathbf{V}) - \mathbf{X}^t, \nabla f(\mathbf{X}^t)\rangle + \frac{1}{2}\|\mathcal{X}_{\text{B}}^t(\mathbf{V}) - \mathbf{X}^t\|_{\mathbf{H}}^2 \overset{②}{=} \langle \text{U}_{\text{B}}(\mathbf{V} - \mathbf{I}_k)\text{U}_{\text{B}}^{\mathsf{T}}\mathbf{X}^t, \nabla f(\mathbf{X}^t)\rangle + \frac{1}{2}\|\mathbf{V} - \mathbf{I}_k\|_{\underline{\mathbf{Q}}}^2 \overset{③}{\leq} \langle \mathbf{V} - \mathbf{I}_k, [\nabla f(\mathbf{X}^t)(\mathbf{X}^t)^{\mathsf{T}}]_{\text{BB}}\rangle + \frac{1}{2}\|\mathbf{V} - \mathbf{I}_k\|_{\mathbf{Q}+\alpha\mathbf{I}}^2$, where step ① uses Inequality (2); step ② uses Claim (*a*) of Lemma 2.2; step ③ uses $\alpha > 0$ and $\underline{\mathbf{Q}} \preceq \mathbf{Q}$, which can be ensured by choosing $\mathbf{Q}$ using one of the following methods:

$$\mathbf{Q} = \underline{\mathbf{Q}} \triangleq (\mathbf{Z}^{\mathsf{T}} \otimes \text{U}_{\text{B}})^{\mathsf{T}}\mathbf{H}(\mathbf{Z}^{\mathsf{T}} \otimes \text{U}_{\text{B}}), \ \text{with} \ \mathbf{Z} \triangleq \text{U}_{\text{B}}^{\mathsf{T}}\mathbf{X}^t, \tag{8}$$

$$\mathbf{Q} = \varsigma\mathbf{I}, \ \text{with} \ \|\underline{\mathbf{Q}}\|_{\mathsf{sp}} \leq \varsigma \leq L_f. \tag{9}$$

Then, we construct the function $\mathcal{K}(\mathbf{V}; \mathbf{X}^t, \text{B})$ that majorizes $F(\mathcal{X}_{\text{B}}^t(\mathbf{V})) = f(\mathcal{X}_{\text{B}}^t(\mathbf{V})) + h(\mathcal{X}_{\text{B}}^t(\mathbf{V}))$:

$$F(\mathcal{X}_{\text{B}}^t(\mathbf{V})) \leq f(\mathbf{X}^t) + \langle \mathbf{V} - \mathbf{I}_k, [\nabla f(\mathbf{X}^t)(\mathbf{X}^t)^{\mathsf{T}}]_{\text{BB}}\rangle + \frac{1}{2}\|\mathbf{V} - \mathbf{I}_k\|_{\mathbf{Q}+\alpha\mathbf{I}}^2 + h(\mathbf{V}\text{U}_{\text{B}}^{\mathsf{T}}\mathbf{X}^t)$$

$$\leq \underbrace{\frac{1}{2}\|\mathbf{V} - \mathbf{I}_k\|_{\mathbf{Q}+\alpha\mathbf{I}}^2 + \langle \mathbf{V}, [\nabla f(\mathbf{X}^t)(\mathbf{X}^t)^{\mathsf{T}}]_{\text{BB}}\rangle + h(\mathbf{V}\text{U}_{\text{B}}^{\mathsf{T}}\mathbf{X}^t) + \ddot{c}}_{\mathcal{K}(\mathbf{V}; \mathbf{X}^t, \text{B})}, \tag{10}$$

where $\ddot{c} = f(\mathbf{X}^t) + h(\text{U}_{\text{B}^c}^{\mathsf{T}}\mathbf{X}^t) - \langle \mathbf{I}_k, [\nabla f(\mathbf{X}^t)(\mathbf{X}^t)^{\mathsf{T}}]_{\text{BB}}\rangle$ is a constant. Here, we use the coordinate-wise separable property of $h(\cdot)$ as follows: $h(\mathcal{X}_{\text{B}}^t(\mathbf{V})) = h(\text{U}_{\text{B}^c}\text{U}_{\text{B}^c}^{\mathsf{T}}\mathbf{X}^t + \text{U}_{\text{B}}\mathbf{V}\text{U}_{\text{B}}^{\mathsf{T}}\mathbf{X}^t) = h(\text{U}_{\text{B}^c}^{\mathsf{T}}\mathbf{X}^t) + h(\mathbf{V}\text{U}_{\text{B}}^{\mathsf{T}}\mathbf{X}^t)$. We minimize the upper bound of the right-hand side of Inequality (10), resulting in the minimization problem that $\bar{\mathbf{V}}^t \in \arg\min_{\mathbf{V} \in \text{St}(k,k)} \mathcal{K}(\mathbf{V}; \mathbf{X}^t, \text{B})$, which can be efficiently and exactly solved due to our assumption.

Three strategies to find the working set $\text{B}$ with $|\text{B}| = k$ can be considered. *(i)* Random strategy: $\text{B}$ is randomly selected from $\{\mathcal{B}_1, \mathcal{B}_2, ..., \mathcal{B}_{\text{C}_n^k}\}$ with equal probability $1/\text{C}_n^k$. *(ii)* Cyclic strategy: $\text{B}^t$ takes all possible combinations in cyclic order, such as $\mathcal{B}_1 \to \mathcal{B}_2 \to ... \to \mathcal{B}_{\text{C}_n^k} \to \mathcal{B}_1 \to ....$ *(iii)* Greedy strategy: We propose two novel greedy strategies to find a good working set. Due to space limitation, we have included them in Appendix D.

The proposed **OBCD** algorithm is summarized in Algorithm 1. Importantly, **OBCD** is a partial gradient method with low iterative computational complexity as it only assesses $k$ rows of the Euclidean gradient of $\nabla f(\mathbf{X}^t)$ and the solution $\mathbf{X}^t$ to compute the linear term $\langle [\nabla f(\mathbf{X}^t)(\mathbf{X}^t)^{\mathsf{T}}]_{\text{BB}}, \mathbf{V}\rangle = \langle [\nabla f(\mathbf{X}^t)]_{\text{B},:}^{\mathsf{T}}[\mathbf{X}^t]_{\text{B},:}, \mathbf{V}\rangle$, as shown in Equation (10).

▶ **Solving the General OBCD Subproblems**. The following lemma outlines key properties of the **OBCD** subproblems.

---

**Algorithm 1:**  **OBCD**, The Proposed Block Coordinate Descent Algorithm for Problem (1).

---

**Input:** an initial feasible solution $\mathbf{X}^0$. Set $k \geq 2$, $t = 0$.

**for** $t$ from 0 to $T$ **do**

    (**S1**) Use some strategy to find a working set $\mathtt{B}^t$ for the $t$-it iteration with
      $\mathtt{B}^t \in \{1, 2, ..., n\}^k$. Let $\mathtt{B} = \mathtt{B}^t$ and $\mathtt{B}^c = \{1, 2, ..., n\} \setminus \mathtt{B}$.

    (**S2**) Choose a suitable matrix $\mathbf{Q} \in \mathbb{R}^{k^2 \times k^2}$ using Equation (8) or Equation (9):

    (**S3**) Find a **global** (or **local**) optimal solution $\bar{\mathbf{V}}^t$ for the following problem:

$$\bar{\mathbf{V}}^t \in \arg\min_{\mathbf{V} \in \mathrm{St}(k,k)} \mathcal{K}(\mathbf{V}; \mathbf{X}^t, \mathtt{B})$$

    satisfying $\mathcal{K}(\bar{\mathbf{V}}^t; \mathbf{X}^t, \mathtt{B}) \leq \mathcal{K}(\mathbf{I}_k; \mathbf{X}^t, \mathtt{B})$, where $\mathcal{K}(\cdot; \cdot, \cdot)$ is define in Inequality (10).

    (**S4**) $\mathbf{X}^{t+1}(\mathtt{B}, :) = \bar{\mathbf{V}}^t \mathbf{X}^t(\mathtt{B}, :)$

**end**

---

**Lemma 2.3.** *(Proof in Appendix E.3) We define $\mathbf{P} \triangleq [\nabla f(\mathbf{X}^t)(\mathbf{X}^t)^{\mathsf{T}}]_{\mathtt{BB}} - \mathrm{mat}(\mathbf{Q}\mathrm{vec}(\mathbf{I}_k)) - \alpha\mathbf{I}_k$, and $\mathbf{Z} = \mathbf{U}_\mathtt{B}^{\mathsf{T}} \mathbf{X}^t$. We have: (**a**) The subproblem $\bar{\mathbf{V}}^t \in \arg\min_{\mathbf{V} \in \mathrm{St}(k,k)} \mathcal{K}(\mathbf{V}; \mathbf{X}^t, \mathtt{B})$ in Algorithm 1 is equivalent to Problem (3). (**b**) Assume that Formula (9) is used to choose $\mathbf{Q}$. Problem (3) further reduces to the following problem: $\bar{\mathbf{V}}^t \in \arg\min_{\mathbf{V} \in \mathrm{St}(k,k)} \mathcal{P}(\mathbf{V}) \triangleq \langle \mathbf{V}, \mathbf{P} \rangle + h(\mathbf{VZ})$. In particular, when $h(\mathbf{X}) \triangleq 0$, we obtain: $\bar{\mathbf{V}}^t = -\mathbb{P}_\mathcal{M}(\mathbf{P})$. Here, $\mathbb{P}_\mathcal{M}(\mathbf{P})$ is the nearest orthogonality matrix to $\mathbf{P}$.*

**Remark 2.4.** *(**a**) By Claim (**b**) of Lemma 2.3, when $k > 2$, $h(\mathbf{X}) = 0$, and $\mathbf{Q}$ is chosen to be a diagonal matrix as in Equation (9), the subproblem $\bar{\mathbf{V}}^t \in \arg\min_{\mathbf{V} \in \mathrm{St}(k,k)} \mathcal{K}(\mathbf{V}; \mathbf{X}^t, \mathtt{B})$ in Algorithm 1 can be solved exactly and efficiently due to our assumption, see Remark 2.6. (**b**) For general $k$ and $h(\cdot)$, the subproblem may not be solved globally, but a **local** stationary solution $\bar{\mathbf{V}}^t$ satisfying $(\bar{\mathbf{V}}^t; \mathbf{X}^t, \mathtt{B}) \leq \mathcal{K}(\mathbf{I}_k; \mathbf{X}^t, \mathtt{B})$ can be achieved. Although strong optimality may be compromised, convergence to a critical point (as discussed later) for the final solution $\mathbf{X}^\infty$ remains achievable.*

▶ **Smallest Possible Subproblems When** $k = 2$. We now discuss how to solve the subproblems exactly when $k = 2$ and $h(\cdot) \neq 0$. The following lemma reveals an equivalent expression for any $\mathbf{V} \in \mathrm{St}(2, 2)$.

**Lemma 2.5.** *(Proof in Appendix E.4) Any orthogonal matrix $\mathbf{V} \in \mathrm{St}(2,2)$ can be expressed as $\mathbf{V} = \mathbf{V}_\theta^{\mathrm{rot}}$ or $\mathbf{V} = \mathbf{V}_\theta^{\mathrm{ref}}$ for some $\theta \in \mathbb{R}$, where $\mathbf{V}_\theta^{\mathrm{rot}} \triangleq \left( \begin{smallmatrix} \cos(\theta) & \sin(\theta) \\ -\sin(\theta) & \cos(\theta) \end{smallmatrix} \right)$, $\mathbf{V}_\theta^{\mathrm{ref}} \triangleq \left( \begin{smallmatrix} -\cos(\theta) & \sin(\theta) \\ \sin(\theta) & \cos(\theta) \end{smallmatrix} \right)$. We have $\det(\mathbf{V}_\theta^{\mathrm{rot}}) = 1$ and $\det(\mathbf{V}_\theta^{\mathrm{ref}}) = -1$ for any $\theta$.*

Using Lemma 2.5, we can reformulate Problem (3) as the following one-dimensional problem: $\bar{\theta} \in \arg\min_\theta \mathcal{P}(\mathbf{V})$, $s.t. \mathbf{V} \in \{\mathbf{V}_\theta^{\mathrm{rot}}, \mathbf{V}_\theta^{\mathrm{ref}}\}$. The optimal solution $\bar{\theta}$ can be identified even if $h(\cdot) \neq 0$ using a novel breakpoint searching method, which is discussed later in Section 5.

**Remark 2.6.** *(**i**) $\mathbf{V}_\theta^{\mathrm{rot}}$ and $\mathbf{V}_\theta^{\mathrm{ref}}$ are called Givens rotation matrix and Jacobi reflection matrix respectively in the literature (Sun & Bischof, 1995). Previous research only considered $\{\mathbf{V}_\theta^{\mathrm{rot}}\}$ for solving symmetric linear eigenvalue problems (Golub & Van Loan, 2013) and sparse PCA problems (Shalit & Chechik, 2014), while we use $\{\mathbf{V}_\theta^{\mathrm{ref}}, \mathbf{V}_\theta^{\mathrm{rot}}\}$ for solving Problem (1). (**ii**) We show the necessity of using $\{\mathbf{V}_\theta^{\mathrm{ref}}, \mathbf{V}_\theta^{\mathrm{rot}}\}$ in the following two examples of $2 \times 2$ optimization problems with orthogonality constraints: $\min_{\mathbf{V} \in \mathrm{St}(2,2)} F(\mathbf{V}) \triangleq \|\mathbf{V} - \mathbf{A}\|_\mathsf{F}^2$, and $\min_{\mathbf{V} \in \mathrm{St}(2,2)} F(\mathbf{V}) \triangleq \|\mathbf{V} - \mathbf{B}\|_\mathsf{F}^2 + 5\|\mathbf{V}\|_1$, where $\mathbf{A} = \left( \begin{smallmatrix} 1 & 0 \\ -1 & -1 \end{smallmatrix} \right)$ and $\mathbf{B} = \left( \begin{smallmatrix} 1 & 0 \\ 1 & 2 \end{smallmatrix} \right)$. The use of the reflection matrix $\mathbf{V}_\theta^{\mathrm{ref}}$ is essential in these examples because it results in lower objective values. See Section C.1 in the Appendix for more details.*

## 3  OPTIMALITY ANALYSIS

This section provides some optimality analysis for the proposed algorithm.

▶ **Basis Representation of Orthogonal Matrices**. The following theorem is used to characterize any orthogonal matrix $\mathbf{D} \in \mathrm{St}(n, n)$ and $\mathbf{X} \in \mathrm{St}(n, r)$.

**Theorem 3.1.** *(Proof in Appendix F.1, Basis Representation of Orthogonal Matrices) Assume $k = 2$. For all $i \in [\mathrm{C}_n^k]$, we define $\mathcal{W}_i \triangleq \mathbf{I}_n + \mathbf{U}_{\mathcal{B}_i}(\mathcal{V}_i - \mathbf{I}_k)\mathbf{U}_{\mathcal{B}_i}^{\mathsf{T}} = \mathbf{U}_{\mathcal{B}_i}\mathcal{V}_i\mathbf{U}_{\mathcal{B}_i}^{\mathsf{T}} + \mathbf{U}_{\mathcal{B}_i^c}\mathbf{U}_{\mathcal{B}_i^c}^{\mathsf{T}}$, where*

$\mathcal{V}_i \in \mathrm{St}(2,2)$. We have: **(a)** Any matrix $\mathbf{D} \in \mathrm{St}(n,n)$ can be expressed as $\mathbf{D} = \mathcal{W}_{\mathrm{C}_n^k} ... \mathcal{W}_2 \mathcal{W}_1$ using suitable $\mathcal{W}_i$ (which depends on $\mathcal{V}_i$). Furthermore, if $\forall i$, $\mathcal{V}_i = \mathbf{I}_2$, then $\mathbf{D} = \mathbf{I}_n^n$. **(b)** Any matrix $\mathbf{X} \in \mathrm{St}(n,r)$ can be expressed as $\mathbf{X} = \mathcal{W}_{\mathrm{C}_n^k} ... \mathcal{W}_2 \mathcal{W}_1 \mathbf{X}^0$ using suitable $\mathcal{W}_i$ and any fixed constant matrix $\mathbf{X}^0 \in \mathrm{St}(n,r)$.

**Remark 3.2.** *(i) We use both Givens rotation and Jacobi reflection matrices to compute $\mathbf{D} \in \mathrm{St}(n,n)$. This is necessary since a reflection matrix cannot be represented through a sequence of rotations. (ii) The result in Claim (**b**) of Theorem 3.1 indicates that the proposed update scheme $\mathbf{X}^+ \Leftarrow \mathbf{X} + \mathrm{U}_{\mathrm{B}}(\mathbf{V} - \mathbf{I}_k)\mathrm{U}_{\mathrm{B}}^{\mathsf{T}}\mathbf{X}$ as shown in Formula (5) can reach any orthogonal matrix $\mathbf{X} \in \mathrm{St}(n,r)$ for any starting solution $\mathbf{X}^0 \in \mathrm{St}(n,r)$.*

▶ **First-Order Optimality Conditions for Problem (1)**. We provide the first-order optimality condition of Problem (1) (Wen & Yin, 2013; Chen et al., 2020). We use $\partial F(\mathbf{X})$ to denote the limiting subdifferential of $F(\mathbf{X})$ (Mordukhovich, 2006; Rockafellar & Wets., 2009), which is always non-empty since $F(\mathbf{X})$ is closed, proper, and lower semicontinuous. Given $f(\mathbf{X})$ is differentiable, we have $\partial F(\mathbf{X}) = \partial(f + h)(\mathbf{X}) = \nabla f(\mathbf{X}) + \partial h(\mathbf{X})$. We extend the definition of *limiting subdifferential* to introduce $\partial_{\mathcal{M}} F(\mathbf{X})$ as the *Riemannian limiting subdifferential* of $F(\mathbf{X})$ at $\mathbf{X}$, defined as $\partial_{\mathcal{M}} F(\mathbf{X}) \triangleq \partial F(\mathbf{X}) \ominus (\mathbf{X}[\partial F(\mathbf{X})]^{\mathsf{T}}\mathbf{X})$, where $\ominus$ is the element-wise subtraction between sets.

Introducing a Lagrangian multiplier matrix $\mathbf{\Lambda} \in \mathbb{R}^{r \times r}$ for the orthogonality constraint, we define the following Lagrangian function of Problem (1): $\mathcal{L}(\mathbf{X}, \mathbf{\Lambda}) = F(\mathbf{X}) + \frac{1}{2}\langle \mathbf{I}_r - \mathbf{X}^{\mathsf{T}}\mathbf{X}, \mathbf{\Lambda} \rangle$. Notable, the matrix $\mathbf{\Lambda}$ is symmetric, as $\mathbf{X}^{\mathsf{T}}\mathbf{X}$ is symmetric. We state the following definition of first-order optimality condition.

**Definition 3.3.** *Critical Point (Wen & Yin, 2013; Chen et al., 2020). A solution $\check{\mathbf{X}} \in \mathrm{St}(n,r)$ is a critical point of Problem (1) if: $\mathbf{0} \in \partial_{\mathcal{M}} F(\check{\mathbf{X}}) \triangleq \partial F(\check{\mathbf{X}}) \ominus (\check{\mathbf{X}}[\partial F(\check{\mathbf{X}})]^{\mathsf{T}}\check{\mathbf{X}})$, where $(\partial F(\check{\mathbf{X}}) \ominus \check{\mathbf{X}}[\partial F(\check{\mathbf{X}})]^{\mathsf{T}}\check{\mathbf{X}}) \triangleq \{\mathbf{G} - \check{\mathbf{X}}\mathbf{G}^{\mathsf{T}}\check{\mathbf{X}} \,|\, \mathbf{G} \in \partial F(\check{\mathbf{X}})\}$. Furthermore, $\mathbf{\Lambda} \in [\partial F(\check{\mathbf{X}})]^{\mathsf{T}}\check{\mathbf{X}}$.*

**Remark 3.4.** *The critical point condition in Lemma 3.3 can be equivalently expressed as (Absil et al., 2008; Jiang & Dai, 2015; Liu et al., 2016): $\mathbf{0} \in \mathbb{P}_{\mathrm{T}_{\check{\mathbf{X}}}\mathcal{M}}(\partial F(\check{\mathbf{X}}))$. Here, $\mathrm{T}_{\mathbf{X}}\mathcal{M}$ is the tangent space to $\mathcal{M}$ at $\mathbf{X} \in \mathcal{M}$ with $\mathrm{T}_{\mathbf{X}}\mathcal{M} = \{\mathbf{Y} \in \mathbb{R}^{n \times r} \,|\, \mathbf{X}^{\mathsf{T}}\mathbf{Y} + \mathbf{Y}^{\mathsf{T}}\mathbf{X} = \mathbf{0}\}$.*

▶ **Optimality Conditions for the Subproblems**. The Euclidean subdifferential of $\mathcal{K}(\mathbf{V}; \mathbf{X}^t, \mathrm{B}^t)$ *w.r.t.* $\mathbf{V}$ can be computed as follows: $\ddot{\mathbf{G}}(\mathbf{V}) \triangleq \ddot{\mathbf{\Delta}} + \mathrm{U}_{\mathrm{B}}^{\mathsf{T}}[\nabla f(\mathbf{X}^t) + \partial h(\mathbf{X}^{t+1})](\mathbf{X}^t)^{\mathsf{T}}\mathrm{U}_{\mathrm{B}}$, where $\ddot{\mathbf{\Delta}} = \mathrm{mat}((\mathbf{Q} + \alpha\mathbf{I}_k)\mathrm{vec}(\mathbf{V} - \mathbf{I}_k))$, and $\mathbf{X}^{t+1} = \mathbf{X}^t + \mathrm{U}_{\mathrm{B}}(\mathbf{V} - \mathbf{I}_k)\mathrm{U}_{\mathrm{B}}^{\mathsf{T}}\mathbf{X}^t$. Using Lemma 3.3, we set the Riemannian subdifferential of $\mathcal{K}(\mathbf{V}; \mathbf{X}^t, \mathrm{B}^t)$ *w.r.t.* $\mathbf{V}$ to zero and obtain the following first-order optimality condition for $\bar{\mathbf{V}}^t$: $\mathbf{0} \in \partial_{\mathcal{M}}\mathcal{K}(\bar{\mathbf{V}}^t; \mathbf{X}^t, \mathrm{B}^t) \triangleq \ddot{\mathbf{G}}(\bar{\mathbf{V}}^t) \ominus \bar{\mathbf{V}}^t\ddot{\mathbf{G}}(\bar{\mathbf{V}}^t)^{\mathsf{T}}\bar{\mathbf{V}}^t$.

▶ **Optimality Conditions and Their Hierarchy**. We introduce the following new optimality condition of block-$k$ stationary points.

**Definition 3.5.** *(Global) Block-$k$ Stationary Point, abbreviated as $\mathrm{BS}_k$-point. Let $\alpha > 0$ and $k \geq 2$. A solution $\dddot{\mathbf{X}} \in \mathrm{St}(n,r)$ is called a block-$k$ stationary point if: $\forall \mathrm{B} \in \{\mathcal{B}_i\}_{i=1}^{\mathrm{C}_n^k}$, $\mathbf{I}_k \in \arg\min_{\mathbf{V} \in \mathrm{St}(k,k)} \mathcal{K}(\mathbf{V}; \dddot{\mathbf{X}}, \mathrm{B})$, where $\mathcal{K}(\cdot; \cdot, \cdot)$ is defined in Equation (10).*

**Remarks**. $\mathrm{BS}_k$-point states that if we *globally* minimize the majorization function $\mathcal{K}(\mathbf{V}; \dddot{\mathbf{X}}, \mathrm{B})$, there is no possibility of improving the objective function value for $\mathcal{K}(\mathbf{V}; \dddot{\mathbf{X}}, \mathrm{B})$ across all $\mathrm{B} \in \{\mathcal{B}_i\}_{i=1}^{\mathrm{C}_n^k}$.

The following theorem establishes the relation between $\mathrm{BS}_k$-points, standard critical points, and global optimal points.

**Theorem 3.6.** *(Proof in Appendix F.2) We establish the following relationships:*

*(a)* $\{\text{critical points } \check{\mathbf{X}}\} \supseteq \{\mathrm{BS}_2\text{-points } \dddot{\mathbf{X}}\}$.

*(b)* $\{\mathrm{BS}_2\text{-points } \dddot{\mathbf{X}}\} \supseteq \{\text{global optimal points } \bar{\mathbf{X}}\}$.

*(c)* $\{\mathrm{BS_k}\text{-points } \dddot{\mathbf{X}}\} \supseteq \{\mathrm{BS_{k+1}}\text{-points } \dddot{\mathbf{X}}\}$, *where $k \in \{2, 3, \ldots, n-1\}$*.

*(d) The reverse of the above three inclusions may not always hold true.*

**Remark 3.7.** *The optimality of $\mathrm{BS}_2$-points is stronger than that of standard critical points (Wen & Yin, 2013; Chen et al., 2020; Absil et al., 2008).*

## 4 CONVERGENCE ANALYSIS

This section presents the ergodic and non-ergodic (or last-iterate) convergence rates of the proposed **OBCD** algorithm.

We denote any point of the limit point set of **OBCD** (which is not necessarily a singleton) as $\ddot{\mathbf{X}}$. For the case where a random strategy is used to find the working set, **OBCD** generates a random output $(\bar{\mathbf{V}}^t, \mathbf{X}^{t+1})$ with $t = 0, 1, \ldots, \infty$ which depends on the observed realization of the random variable: $\xi^t \triangleq (\mathrm{B}^1, \mathrm{B}^2, \mathrm{B}^3, \ldots, \mathrm{B}^t)$.

### 4.1 ERGODIC CONVERGENCE RATE

Initially, we introduce the notation of $\epsilon$-$\mathrm{BS}_k$-*point* as follows.

**Definition 4.1.** ($\epsilon$-$\mathrm{BS}_k$-*point*) Given any constant $\epsilon > 0$, a point $\ddot{\mathbf{X}}$ is called an $\epsilon$-$\mathrm{BS}_k$-*point* if: $\frac{1}{\mathrm{C}_n^k} \sum_{i=1}^{\mathrm{C}_n^k} \mathrm{dist}(\mathbf{I}_k, \arg\min_{\mathbf{V}} \mathcal{K}(\mathbf{V}; \mathbf{X}, \mathcal{B}_i))^2 \leq \epsilon$, where $\mathcal{K}(\cdot; \cdot, \cdot)$ is defined in Equation (10).

Using the optimality measure from Definition 4.1, we establish the ergodic convergence rates of **OBCD**.

**Theorem 4.2.** *(Proof in Appendix G.1) We define* $\tilde{c} \triangleq \frac{2}{\alpha} \cdot (F(\mathbf{X}^0) - F(\ddot{\mathbf{X}}))$. *We have:*

*(a)* *The following sufficient decrease condition holds for all* $t \geq 0$:
$$\tfrac{\alpha}{2}\|\mathbf{X}^{t+1} - \mathbf{X}^t\|_{\mathsf{F}}^2 \leq \tfrac{\alpha}{2}\|\bar{\mathbf{V}}^t - \mathbf{I}_k\|_{\mathsf{F}}^2 \leq F(\mathbf{X}^t) - F(\mathbf{X}^{t+1}).$$

*(b)* *If the* $\mathrm{B}^t$ *is selected from* $\{\mathcal{B}_i\}_{i=1}^{\mathrm{C}_n^k}$ *randomly and uniformly,* **OBCD** *finds an* $\epsilon$-$\mathrm{BS}_k$-*point of Problem (1) in at most* $T$ *iterations in the sense of expectation, where* $T \geq \lceil \frac{\tilde{c}}{\epsilon} \rceil$.

*(c)* *If the* $\mathrm{B}^t$ *is selected from* $\{\mathcal{B}_i\}_{i=1}^{\mathrm{C}_n^k}$ *cyclically,* **OBCD** *finds an* $\epsilon$-$\mathrm{BS}_k$-*point of Problem (1) in at most* $T$ *iterations deterministically, where* $T \geq \lceil \frac{\tilde{c}}{\epsilon} + \mathrm{C}_n^k \rceil$.

**Remark 4.3.** *Theorem 4.2 shows that* **OBCD** *converges to* $\epsilon$-*block-k stationary points with an ergodic convergence rate of* $\mathcal{O}(1/\epsilon)$, *which is typical for general nonconvex optimization.*

Apart from Definition 4.1, another common optimality measure relies on the Riemannian subgradient. To this end, we present the following lemma. For simplicity, we assume that a random strategy is employed to determine the working set in the remainder of this paper.

**Lemma 4.4.** *(Proof in Appendix G.2,* **Riemannian Subgradient Lower Bound for the Iterates Gap**) *Assume* $\|\nabla f(\mathbf{X})\|_{\mathsf{sp}} \leq l_f, \|\partial h(\mathbf{X})\|_{\mathsf{sp}} \leq l_h$ *for all* $\mathbf{X} \in \mathrm{St}(n, r)$ *with* $l_f, l_h > 0$. *The Riemannian subdifferential of* $\mathcal{K}(\mathbf{V}; \mathbf{X}^t, \mathrm{B}^t)$ *at the point* $\mathbf{V} = \mathbf{I}_k$ *can be computed as:* $\partial_{\mathcal{M}} \mathcal{K}(\mathbf{I}_k; \mathbf{X}^t, \mathrm{B}^t) = \mathrm{U}_{\mathrm{B}^t}^{\mathsf{T}}(\mathbb{D} \ominus \mathbb{D}^{\mathsf{T}})\mathrm{U}_{\mathrm{B}^t}$, *where* $\mathbb{D} = [\nabla f(\mathbf{X}^t) + \partial h(\mathbf{X}^t)][\mathbf{X}^t]^{\mathsf{T}}$. *(a) It holds that:* $\mathbb{E}_{\xi^{t+1}}[\mathrm{dist}(\mathbf{0}, \partial_{\mathcal{M}} \mathcal{K}(\mathbf{I}_k; \mathbf{X}^{t+1}, \mathrm{B}^{t+1}))] \leq \phi \cdot \mathbb{E}_{\xi^t}[\|\bar{\mathbf{V}}^t - \mathbf{I}_k\|_{\mathsf{F}}]$, *where* $\phi \triangleq 4(l_f + l_h + L_f) + 2\alpha$. *(b)* $\mathbb{E}_{\xi^t}[\mathrm{dist}(\mathbf{0}, \partial_{\mathcal{M}} F(\mathbf{X}^t))] \leq \gamma \cdot \mathbb{E}_{\xi^t}[\mathrm{dist}(\mathbf{0}, \partial_{\mathcal{M}} \mathcal{K}(\mathbf{I}_k; \mathbf{X}^t, \mathrm{B}^t))]$, *where* $\gamma \triangleq (\mathrm{C}_n^k/\mathrm{C}_{n-2}^{k-2})^{1/2}$.

**Remark 4.5.** *The important class of nonsmooth* $\ell_1$ *norm function* $h(\mathbf{X}) = \|\mathbf{X}\|_1$ *(Chen et al., 2020; 2024) satisfies the assumption made in Lemma 4.4.*

We establish the ergodic convergence rates of **OBCD** using the optimality measure of Riemannian subgradient (Chen et al., 2020; Cheung et al., 2024; Li et al., 2024).

**Theorem 4.6.** *(Proof in Appendix G.3) We define* $\tilde{c} \triangleq \frac{2}{\alpha} \cdot (F(\mathbf{X}^0) - F(\ddot{\mathbf{X}}))$, *and* $\{\phi, \gamma\}$ *as in Lemma 4.4.* **OBCD** *finds an* $\epsilon$-*critical point of Problem (1) satisfying* $\mathbb{E}_{\xi^{\bar{t}}}[\mathrm{dist}^2(\mathbf{0}, \partial_{\mathcal{M}} F(\mathbf{X}^{\bar{t}+1}))] \leq \epsilon$ *in at most* $T + 1$ *iterations in the sense of expectation, where* $T \geq \lceil \frac{\gamma^2 \phi^2 \tilde{c}}{\epsilon} \rceil$.

### 4.2 NON-ERGODIC CONVERGENCE RATE UNDER KL ASSUMPTION

We establish the non-ergodic convergence rate of **OBCD** using the Kurdyka-Łojasiewicz inequality, a key tool in non-convex analysis (Attouch et al., 2010; Bolte et al., 2014; Liu et al., 2016).

Initially, we make the following additional assumption.

**Assumption 4.7.** *The function* $F^{\circ}(\mathbf{X}) = F(\mathbf{X}) + \mathcal{I}_{\mathcal{M}}(\mathbf{X})$ *is a KL function.*

**Remark 4.8.** *Semi-algebraic functions are a class of functions that satisfy the KL property. These functions are widely used in applications, and they include real polynomial functions, finite sums and products of semi-algebraic functions, and indicator functions of semi-algebraic sets (Attouch et al., 2010; Xu & Yin, 2013).*

We present the following useful proposition, due to (Attouch et al., 2010; Bolte et al., 2014).

**Proposition 4.9.** *(**Kurdyka-Łojasiewicz Property**). For a KL function $F^\circ(\mathbf{X})$ with $\mathbf{X} \in \text{dom } F^\circ$, there exists $\sigma \in [0,1)$, $\eta \in (0, +\infty]$, a neighborhood $\Upsilon$ of $\ddot{\mathbf{X}}$, and a concave continuous function $\varphi(t) = ct^{1-\sigma}$, $c > 0$, $t \in [0, \eta)$ such that for all $\mathbf{X}' \in \Upsilon$ and satisfies $F^\circ(\mathbf{X}') \in (F^\circ(\ddot{\mathbf{X}}), F^\circ(\ddot{\mathbf{X}}) + \eta)$, the following inequality holds: $\text{dist}(\mathbf{0}, \partial F^\circ(\mathbf{X}'))\varphi'(F^\circ(\mathbf{X}') - F^\circ(\ddot{\mathbf{X}})) \geq 1$.*

Utilizing the Kurdyka-Łojasiewicz property, one can establish a finite-length property of **OBCD**, a result considerably stronger than that of Theorem 4.2.

**Theorem 4.10.** *(Proof in Appendix G.4, **A Finite Length Property**). We define $e^{t+1} \triangleq \mathbb{E}_{\xi^t}[\|\bar{\mathbf{V}}^t - \mathbf{I}_k\|_\mathsf{F}]$, and $d^i = \sum_{j=i}^\infty e^{j+1}$. Based on the continuity assumption made in Lemma 4.4, We have:*

**(a)** It holds that $(e^{t+1})^2 \leq \kappa e^t(\varphi^t - \varphi^{t+1})$, where $\varphi^t \triangleq \varphi(F(\mathbf{X}^t) - F(\ddot{\mathbf{X}}))$, $\kappa \triangleq \frac{2\gamma\phi}{\alpha}$ is a positive constant, $\gamma \triangleq (\mathrm{C}_n^k/\mathrm{C}_{n-2}^{k-2})^{1/2}$, $\phi$ is defined in Lemma 4.4, and $\varphi(\cdot)$ is the desingularization function defined in Proposition 4.9.

**(b)** It holds that $\forall t \geq 1$, $d^t \leq e^t + 2\kappa\varphi^t$. The sequence $\{e^t\}_{t=1}^\infty$ has the finite length property that $d^t \triangleq \sum_{j=t}^\infty e^{j+1}$ is always upper-bounded by a certain constant.

Finally, we establish the last-iterate convergence rate for **OBCD**.

**Theorem 4.11.** *(Proof in Appendix G.5). Based on the continuity assumption made in Lemma 4.4, there exists $t'$ such that for all $t \geq t'$, we have:*

*(a) If $\sigma = 0$, then the sequence $\mathbf{X}^t$ converges in a finite number of steps in expectation.*

*(b) If $\sigma \in (0, \frac{1}{2}]$, then there exist $\dot{c} > 0$ and $\dot{\tau} \in [0,1)$ such that $\mathbb{E}_{\xi^{t-1}}[\|\mathbf{X}^t - \mathbf{X}^\infty\|_\mathsf{F}] \leq \dot{c}\dot{\tau}^t$.*

*(c) If $\sigma \in (\frac{1}{2}, 1)$, then there exist $\dot{c} > 0$ such that $\mathbb{E}_{\xi^{t-1}}[\|\mathbf{X}^t - \mathbf{X}^\infty\|_\mathsf{F}] \leq \mathcal{O}(t^{-(1-\sigma)/(2\sigma-1)})$.*

**Remark 4.12.** *When $F(\mathbf{X})$ is a semi-algebraic function and the desingularising function is $\varphi(t) = ct^{1-\sigma}$ for some $c > 0$ and $\sigma \in [0,1)$, Theorem 4.11 shows that **OBCD** converges in finite iterations when $\sigma = 0$, with linear convergence when $\sigma \in (0, \frac{1}{2}]$, and sublinear convergence when $\sigma \in (\frac{1}{2}, 1)$ for the gap $\|\mathbf{X}^t - \mathbf{X}^\infty\|_\mathsf{F}$ in expectation. These results are consistent with those in (Attouch et al., 2010).*

## 5 SOLVING THE SUBPROBLEM WHEN $k = 2$

This section presents a novel Breakpoint Searching Method (**BSM**) to find the *global optimal solution* of Problem (3) when $k = 2$.

Initially, Problem (3) boils down to the following one-dimensional subproblem: $\min_\theta \frac{1}{2}\|\mathbf{V}\|_\mathbf{Q}^2 + \langle \mathbf{V}, \mathbf{P} \rangle + h(\mathbf{VZ})$, $s.t.\, \mathbf{V} \in \{\mathbf{V}_\theta^{\text{rot}}, \mathbf{V}_\theta^{\text{ref}}\}$, which can be further rewritten as: $\bar{\theta} \in \arg\min_\theta \frac{1}{2}\text{vec}(\mathbf{V})^\mathsf{T}\mathbf{Q}\text{vec}(\mathbf{V}) + \langle \mathbf{V}, \mathbf{P} \rangle + h(\mathbf{VZ})$, $s.t.\, \mathbf{V} \triangleq \left(\begin{smallmatrix} \pm\cos(\theta) & \sin(\theta) \\ \mp\sin(\theta) & \cos(\theta) \end{smallmatrix}\right)$, where $\mathbf{Q} \in \mathbb{R}^{4\times4}$, $\mathbf{P} \in \mathbb{R}^{2\times2}$, and $\mathbf{Z} \in \mathbb{R}^{2\times r}$. Given $h(\cdot)$ is coordinate-wise separable, we have the following equivalent optimization problem:

$$\min_\theta h\left(\cos(\theta)\mathbf{x} + \sin(\theta)\mathbf{y}\right) + a\cos(\theta) + b\sin(\theta) + c\cos^2(\theta) + d\cos(\theta)\sin(\theta) + e\sin^2(\theta), \quad (11)$$

where $a = \mathbf{P}_{22} \pm \mathbf{P}_{11}$, $b = \mathbf{P}_{12} \mp \mathbf{P}_{21}$, $c = 0.5(\mathbf{Q}_{11} + \mathbf{Q}_{44}) \pm \mathbf{Q}_{14}$, $d = -\mathbf{Q}_{12} \pm \mathbf{Q}_{13} \mp \mathbf{Q}_{24} + \mathbf{Q}_{34}$, $e = 0.5(\mathbf{Q}_{22} + \mathbf{Q}_{33}) \mp \mathbf{Q}_{23}$, $\mathbf{r} = \pm\mathbf{Z}(1,:)$, $\mathbf{s} = \mathbf{Z}(2,:)$, $\mathbf{p} = \mathbf{Z}(2,:)$, $\mathbf{u} = \mp\mathbf{Z}(1,:)$, $\mathbf{x} \triangleq [\mathbf{r}; \mathbf{p}] \in \mathbb{R}^{2r\times1}$, and $\mathbf{y} \triangleq [\mathbf{s}; \mathbf{u}] \in \mathbb{R}^{2r\times1}$.

Our key strategy is to perform a variable substitution to convert Problem (11) into an equivalent problem that depends on the variable $\tan(\theta) \triangleq t$. The substitution is based on the trigonometric identities that $\cos(\theta) = \pm1/\sqrt{1 + \tan^2(\theta)}$ and $\sin(\theta) = \pm\tan(\theta)/\sqrt{1 + \tan^2(\theta)}$.

The following lemma provides a characterization of the global optimal solution for Problem (11).

**Lemma 5.1.** *(Proof in Appendix H.1) We define $\breve{F}(\tilde{c}, \tilde{s}) \triangleq a\tilde{c} + b\tilde{s} + c\tilde{c}^2 + d\tilde{c}\tilde{s} + e\tilde{s}^2 + h(\tilde{c}\mathbf{x} + \tilde{s}\mathbf{y})$, and $w \triangleq c - e$. The optimal solution $\bar{\theta}$ to (11) can be computed as: $[\cos(\bar{\theta}), \sin(\bar{\theta})] \in \arg\min_{[c,s]} \breve{F}(c,s), \ s.t. [c,s] \in \{[c_1, s_1], [c_2, s_2], [0, 1], [0, -1]\}$, where $c_1 \triangleq \frac{1}{\sqrt{1+(\bar{t}_+)^2}}$, $s_1 = \frac{\bar{t}_+}{\sqrt{1+(\bar{t}_+)^2}}$, $c_2 \triangleq \frac{-1}{\sqrt{1+(\bar{t}_-)^2}}$, and $s_2 \triangleq \frac{-\bar{t}_-}{\sqrt{1+(\bar{t}_-)^2}}$. Furthermore, $\bar{t}_+$ and $\bar{t}_-$ are respectively defined as:*

$$\bar{t}_+ \in \arg\min_t \ p(t) \triangleq \frac{a+bt}{\sqrt{1+t^2}} + \frac{w+dt}{1+t^2} + h(\frac{\mathbf{x}+t\mathbf{y}}{\sqrt{1+t^2}}), \tag{12}$$

$$\bar{t}_- \in \arg\min_t \ \tilde{p}(t) \triangleq \frac{-a-bt}{\sqrt{1+t^2}} + \frac{w+dt}{1+t^2} + h(\frac{-\mathbf{x}-t\mathbf{y}}{\sqrt{1+t^2}}). \tag{13}$$

We describe our **BSM** to solve Problem (12); our approach can be naturally extended to tackle Problem (13). **BSM** first identifies all the possible breakpoints / critical points $\Theta$, and then picks the solution that leads to the lowest value as the optimal solution $\bar{t}$, i.e., $\bar{t} \in \arg\min_t \ p(t), \ s.t. \ t \in \Theta$.

We assume $\mathbf{y}_i \neq 0$. If this is not true and there exists $\mathbf{y}_i = 0$ for some $i$, then $\{\mathbf{x}_i, \mathbf{y}_i\}$ can be removed since it does not affect the minimizer of the problem.

We now show that how to find the breakpoint set $\Theta$ for $h(\mathbf{x}) = \lambda\|\mathbf{x}\|_0$, where $\lambda \geq 0$. We also provide additional examples of **BSM** for other different $h(\mathbf{x})$. Due to space limitation, we have included them in Appendix B.

▶ **Finding the Breakpoint Set for** $h(\mathbf{x}) \triangleq \lambda\|\mathbf{x}\|_0$

Since the function $h(\mathbf{x}) \triangleq \lambda\|\mathbf{x}\|_0$ is scale-invariant and symmetric with $\| \pm t\mathbf{x}\|_0 = \|\mathbf{x}\|_0$ for all $t > 0$, Problem (12) reduces to the following problem:

$$\min_t p(t) \triangleq \frac{a+bt}{\sqrt{1+t^2}} + \frac{w+dt}{1+t^2} + \lambda\|\mathbf{x} + t\mathbf{y}\|_0. \tag{14}$$

Given the limiting subdifferential of the $\ell_0$ norm function can be computed as $\partial\|t\|_0 \in \{ \begin{smallmatrix} \mathbb{R}, & t = 0; \\ \{0\}, & \text{else.} \end{smallmatrix} \}$ (see Appendix C.5), we consider the following two cases. *(i)* We assume $(\mathbf{x} + t\mathbf{y})_i = 0$ for some $i$. Then the solution $\bar{t}$ can be determined using $\bar{t} = \frac{\mathbf{x}_i}{\mathbf{y}_i}$. There are $2r$ breakpoints $\{\frac{\mathbf{x}_1}{\mathbf{y}_1}, \frac{\mathbf{x}_2}{\mathbf{y}_2}, ..., \frac{\mathbf{x}_{2r}}{\mathbf{y}_{2r}}\}$ for this case. *(ii)* We now assume $(\mathbf{x} + t\mathbf{y})_i \neq 0$ for all $i$. Then $\lambda\|\mathbf{x} + t\mathbf{y}\|_0 = 2r\lambda$ becomes a constant. Setting the subgradient of $p(t)$ to zero yields: $0 = \nabla p(t) = [b(1+t^2) - (a+bt)t] \cdot \sqrt{1+t^2} \cdot t^\circ + [d(1+t^2) - (w+dt)(2t)] \cdot t^\circ$, where $t^\circ = (1+t^2)^{-2}$. Since $t^\circ > 0$, we obtain: $d(1+t^2) - (w+dt)2t = -(b-at) \cdot \sqrt{1+t^2}$. Squaring both sides, we obtain the following quartic equation: $c_4t^4 + c_3t^3 + c_2t^2 + c_1t + c_0 = 0$ for some suitable $c_4, c_3, c_2, c_1$ and $c_0$. Solving this equation analytically using Lodovico Ferrari's method (WikiContributors), we obtain all its real roots $\{\bar{t}_1, \bar{t}_2, ..., \bar{t}_j\}$ with $1 \leq j \leq 4$. There are at most 4 breakpoints for this case. Therefore, Problem (14) contains at most $2r + 4$ breakpoints $\Theta = \{\frac{\mathbf{x}_1}{\mathbf{y}_1}, \frac{\mathbf{x}_2}{\mathbf{y}_2}, ..., \frac{\mathbf{x}_{2r}}{\mathbf{y}_{2r}}, \bar{t}_1, \bar{t}_2, ..., \bar{t}_j\}$.

## 6 EXPERIMENTS

This section provides numerical comparisons of **OBCD** against state-of-the-art methods on both real-world and synthetic data. We describe the application of $L_0$ norm-based Sparse PCA (SPCA) in the sequel, while additional applications for nonnegative PCA and $\ell_1$ norm-based SPCA can be found in Appendix J.

▶ **Application to $L_0$ Norm-based SPCA**. $L_0$ norm-based Sparse PCA (SPCA) is a method that uses $\ell_0$ norm to produce modified principal components with sparse loadings, which helps reduce model complexity and increase model interpretability (d'Aspremont et al., 2008; Chen et al., 2016). It can be formulated as: $\min_{\mathbf{X} \in \text{St}(n,r)} -\frac{1}{2}\langle\mathbf{X}, \mathbf{C}\mathbf{X}\rangle + \lambda\|\mathbf{X}\|_0$, where $\mathbf{C} = \mathbf{A}^\mathsf{T}\mathbf{A} \in \mathbb{R}^{n \times n}$ is the covariance of the data matrix $\mathbf{A} \in \mathbb{R}^{m \times n}$ and $\lambda > 0$.

▶ **Data Sets**. To generate the data matrix $\mathbf{A}$, we consider 10 publicly available real-world or random data sets: 'w1a', 'TDT2', '20News', 'sector', 'E2006', 'MNIST', 'Gisette', 'Caltech', 'Cifar', 'randn'. We randomly select a subset of examples from the original data set. The size of $\mathbf{A} \in \mathbb{R}^{m \times n}$ are are chosen from the following set $(m, n) \in \{(2477, 300), (500, 1000), (8000, 1000), (6412, 1000), (2000, 1000), (60000, 784), (3000, 1000), (1000, 1000), (500, 1000)\}$.

▶ **Compared Methods**. We compare with two existing operator splitting methods: Linearized ADMM (LADMM) (Lai & Osher, 2014; He & Yuan, 2012) and Smoothing Penalty Method (SPM)

| data-m-n | $F_{\min}$ | LADMM (id) | SPM (id) | LADMM (rnd) | SPM (rnd) | OBCD-R (id) |
|---|---|---|---|---|---|---|
| $r = 20, \lambda = 1000$, time limit=30 | | | | | | |
| w1a-2477-300 | 1.5e+04 | 2.60e+03 | 3.90e+03 | 1.48e+03 | 8.02e+03 | 0.00e+00 |
| TDT2-500-1000 | 2.0e+04 | 4.00e+03 | 6.71e-01 | 2.00e+03 | 7.00e+03 | 0.00e+00 |
| 20News-8000-1000 | 2.0e+04 | 3.00e+03 | 3.00e+03 | 5.00e+03 | 6.00e+03 | 0.00e+00 |
| sector-6412-1000 | 2.0e+04 | 1.01e+03 | 3.00e+03 | 1.02e+03 | 1.30e+04 | 0.00e+00 |
| E2006-2000-1000 | 2.0e+04 | 2.00e+03 | 1.16e-01 | 4.00e+03 | 1.20e+04 | 0.00e+00 |
| MNIST-60000-784 | -6.7e+04 | 6.38e+04 | 8.68e+04 | 2.28e+03 | 4.30e+04 | 0.00e+00 |
| Gisette-3000-1000 | -2.1e+05 | 4.11e+05 | 2.02e+05 | 1.19e+05 | 8.65e+04 | 0.00e+00 |
| CnnCaltech-3000-1000 | 1.9e+04 | 9.09e+03 | 3.09e+04 | 2.40e+04 | 3.09e+04 | 0.00e+00 |
| Cifar-1000-1000 | 1.6e+04 | 1.80e+04 | 9.99e+02 | 2.40e+04 | 1.10e+05 | 0.00e+00 |
| randn-500-1000 | 1.4e+04 | 2.53e+04 | 5.81e+04 | 2.22e+04 | 4.92e+04 | 0.00e+00 |

Table 1: Comparisons of relative objective values $(F(\mathbf{X}) - F_{\min})$ for $L_0$ norm-based SPCA across all the compared methods. The $1^{st}$, $2^{nd}$, and $3^{rd}$ best results are colored with red, green and blue, respectively.

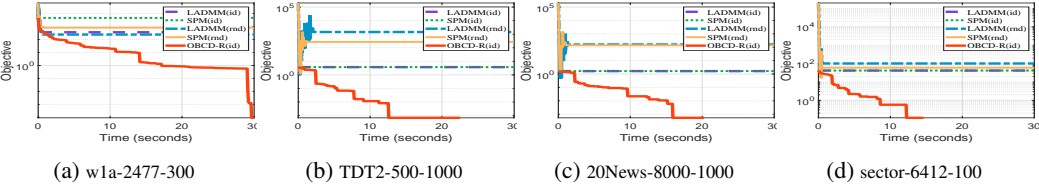

(a) w1a-2477-300    (b) TDT2-500-1000    (c) 20News-8000-1000    (d) sector-6412-100

Figure 1: The convergence curve of the compared methods for solving $L_0$ norm-based SPCA with $\lambda = 100$. No matter how long the algorithms run, the other methods remain trapped in poor local minima.

(Lai & Osher, 2014; Chen, 2012), initialized differently with random and identity matrices, resulting in four variants: LADMM(id), SPM(id), LADMM(rnd), and SPM(rnd). We use a random strategy to find the working set for **OBCD**, initializing it with the identity matrix, resulting in **OBCD-R**(id).

▶ **Implementations**. All methods are implemented in MATLAB on an Intel 2.6 GHz CPU with 32 GB RAM. However, our breakpoint searching procedure is developed in C++ and integrated into the MATLAB environment [2], as it requires inefficient element-wise loops in native MATLAB. Code to reproduce the experiments can be found in the **supplemental material**.

▶ **Experiment Settings**. We compare objective values $(F(\mathbf{X}) - F_{\min})$ for different methods after running for 30 seconds, where $F_{\min}$ represents the smallest objective among all methods. For numerical stability in reporting the objectives, we use the count of elements with absolute values greater than a threshold of $10^{-6}$ instead of the original $\ell_0$ norm function $\|\mathbf{X}\|_0$. We set $\alpha = 10^{-5}$ for **OBCD**. Full-gradient methods have higher per-iteration complexity but require fewer iterations, while **OBCD**, as a partial-gradient method, has lower per-iteration costs but needs more iterations. Thus, we compare based on CPU time rather than iteration count.

▶ **Experiment Results**. Table 1 and Figure 1 display accuracy and computational efficiency results for $L_0$ norm-based SPCA, yielding the following observations: *(i)* **OBCD-R** delivers the best performance. *(ii)* Unlike other methods where objectives fluctuate during iterations, **OBCD-R** monotonically decreases the objective function while maintaining the orthogonality constraint. This is because **OBCD** is a greedy descent method for this problem class. *(iii)* While other methods often get stuck in poor local minima, **OBCD-R** escapes from such minima and generally finds lower objectives, aligning with our theory that our methods locate *stronger stationary points*.

# 7 CONCLUSIONS

In this paper, we introduced **OBCD**, a new block coordinate descent method for nonsmooth composite optimization under orthogonality constraints. **OBCD** operates on $k$ rows of the solution matrix, offering lower computational complexity per iteration for $k \geq 2$. We also provide a novel optimality analysis, showing how **OBCD** exploits problem structure to escape bad local minima and find better stationary points than methods focused on critical points. Under the Kurdyka-Lojasiewicz (KL) inequality, we establish strong limit-point convergence. Additionally, we present two extensions: efficient subproblem solvers for $k = 2$ and new greedy strategies for working set selection. Extensive experiments demonstrate that **OBCD** outperforms existing methods.

---

[2]Though we prioritize accuracy over speed, the comparisons remain fair despite using different programming languages. The other methods, based on matrix multiplication and SVD, utilize highly optimized BLAS and LAPACK libraries for the computational platform and compilation architecture.

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

# Appendix

The appendix section is organized as follows.

Section A covers notations, technical preliminaries, and relevant lemmas.

Section B presents additional examples of breakpoint searching methods.

Section C offers further discussions on the proposed algorithm.

Section D introduces greedy strategies for working set selection.

Section E contains proofs from Section 2.

Section F contains proofs from Section 3.

Section G contains proofs from Section 4.

Section H contains proofs from Section 5.

Section I contains proofs from Section D.

Section J showcases additional experiments.

## A    NOTATIONS, TECHNICAL PRELIMINARIES, AND RELEVANT LEMMAS

### A.1    NOTATIONS

Throughout this paper, $\mathcal{M} \triangleq \mathrm{St}(n, r)$ denotes the Stiefel manifold, which is an embedded submanifold of the Euclidean space $\mathbb{R}^{n \times r}$. Boldfaced lowercase letters denote vectors and uppercase letters denote real-valued matrices. We adopt the Matlab colon notation to denote indices that describe submatrices. For given natual numbers $n$ and $k$, we use $\{\mathcal{B}_1, \mathcal{B}_2, ..., \mathcal{B}_{\mathrm{C}_n^k}\}$ to denote all the possible combinations of the index vectors choosing $k$ items from $n$ without repetition, where $\mathrm{C}_n^k$ is the total number of such combinations and $\mathcal{B}_i \in \mathbb{N}^k$, $\forall i \in [\mathrm{C}_n^k]$. For any one-dimensional function $p(t) : \mathbb{R} \mapsto \mathbb{R}$, we define: $p(\pm x \mp y) \triangleq \min\{p(x - y), p(-x + y)\}$. We use the following notations in this paper.

- $[n]$: $\{1, 2, ..., n\}$
- $\|\mathbf{x}\|$: Euclidean norm: $\|\mathbf{x}\| = \|\mathbf{x}\|_2 = \sqrt{\langle \mathbf{x}, \mathbf{x} \rangle}$
- $\mathbf{x}_i$: the $i$-th element of vector $\mathbf{x}$
- $\mathbf{X}_{i,j}$ or $\mathbf{X}_{ij}$ : the ($i^{\mathrm{th}}$, $j^{\mathrm{th}}$) element of matrix $\mathbf{X}$
- $\mathrm{vec}(\mathbf{X})$ : $\mathrm{vec}(\mathbf{X}) \in \mathbb{R}^{nr \times 1}$, the vector formed by stacking the column vectors of $\mathbf{X}$
- $\mathrm{mat}(\mathbf{x})$ : $\mathrm{mat}(\mathbf{x}) \in \mathbb{R}^{n \times r}$, Convert $\mathbf{x} \in \mathbb{R}^{nr \times 1}$ into a matrix with $\mathrm{mat}(\mathrm{vec}(\mathbf{X})) = \mathbf{X}$
- $\mathbf{X}^\mathsf{T}$ : the transpose of the matrix $\mathbf{X}$
- $\mathrm{sign}(t)$ : the signum function, $\mathrm{sign}(t) = 1$ if $t \geq 0$ and $\mathrm{sign}(t) = -1$ otherwise
- $\det(\mathbf{D})$ : Determinant of a square matrix $\mathbf{D} \in \mathbb{R}^{n \times n}$
- $\mathrm{C}_n^k$ : the number of possible combinations choosing $k$ items from $n$ without repetition
- $\mathbf{0}_{n,r}$ : A zero matrix of size $n \times r$; the subscript is omitted sometimes
- $\mathbf{I}_r$ : $\mathbf{I}_r \in \mathbb{R}^{r \times r}$, Identity matrix
- $\mathbf{X} \succeq \mathbf{0}$(or $\succ \mathbf{0}$) : the Matrix $\mathbf{X}$ is symmetric positive semidefinite (or definite)
- $\mathrm{tr}(\mathbf{A})$ : Sum of the elements on the main diagonal $\mathbf{X}$: $\mathrm{tr}(\mathbf{A}) = \sum_i \mathbf{A}_{i,i}$
- $\langle \mathbf{X}, \mathbf{Y} \rangle$ : Euclidean inner product, i.e., $\langle \mathbf{X}, \mathbf{Y} \rangle = \sum_{ij} \mathbf{X}_{ij} \mathbf{Y}_{ij}$
- $\mathbf{X} \otimes \mathbf{Y}$ : Kronecker product of $\mathbf{X}$ and $\mathbf{Y}$
- $\|\mathbf{X}\|_{\mathsf{sp}}$ : Operator/Spectral norm: the largest singular value of $\mathbf{X}$
- $\|\mathbf{X}\|_{\mathsf{F}}$ : Frobenius norm: $(\sum_{ij} \mathbf{X}_{ij}^2)^{1/2}$
- $\nabla f(\mathbf{X})$ : Euclidean gradient of $f(\mathbf{X})$ at $\mathbf{X}$
- $\nabla_{\mathcal{M}} f(\mathbf{X})$ : Riemannian gradient of $f(\mathbf{X})$ at $\mathbf{X}$

- $\partial F(\mathbf{X})$ : limiting Euclidean subdifferential of $F(\mathbf{X})$ at $\mathbf{X}$

- $\partial_{\mathcal{M}} F(\mathbf{X})$ : limiting Riemannian subdifferential of $F(\mathbf{X})$ at $\mathbf{X}$

- $\mathcal{I}_{\Xi}(\mathbf{X})$ : the indicator function of a set $\Xi$ with $\mathcal{I}_{\Xi}(\mathbf{X}) = 0$ if $\mathbf{X} \in \Xi$ and otherwise $+\infty$

- $\mathbb{P}_{\Xi}(\mathbf{Z})$ : Orthogonal projection of $\mathbf{Z}$ with $\mathbb{P}_{\Xi}(\mathbf{Z}) = \arg\min_{\mathbf{X} \in \Xi} \|\mathbf{Z} - \mathbf{X}\|_F^2$

- $\mathbb{P}_{\mathcal{M}}(\mathbf{Y})$ : Nearest orthogonal matrix of $\mathbf{Y}$ with $\mathbb{P}_{\mathcal{M}}(\mathbf{Y}) = \arg\min_{\mathbf{X}^\top \mathbf{X} = \mathbf{I}_r} \|\mathbf{X} - \mathbf{Y}\|_F^2$

- $\mathrm{dist}(\Xi, \Xi')$ : the distance between two sets with $\mathrm{dist}(\Xi, \Xi') \triangleq \inf_{\mathbf{X} \in \Xi, \mathbf{X}' \in \Xi'} \|\mathbf{X} - \mathbf{X}'\|_F$

- $\mathcal{I}_{\geq 0}(\mathbf{X})$: indicator function of non-negativity constraint with $\mathcal{I}_{\geq 0}(\mathbf{X}) = \{ \begin{smallmatrix} 0, & \mathbf{X} \geq \mathbf{0}; \\ \infty, & \text{else.} \end{smallmatrix} \}$

- $\|\mathbf{X}\|_0$: the number of non-zero elements in the matrix $\mathbf{X}$

- $\|\mathbf{X}\|_1$: the absolute sum of the elements in the matrix $\mathbf{X}$ with $\|\mathbf{X}\|_1 = \sum_{i,j} |\mathbf{X}_{i,j}|$

- $\mathbb{A} + \mathbb{B}$, $\mathbb{A} - \mathbb{B}$: standard Minkowski addition and subtraction between sets $\mathbb{A}$ and $\mathbb{B}$

- $\mathbb{A} \oplus \mathbb{B}$, $\mathbb{A} \ominus \mathbb{B}$: element-wise addition and subtraction between sets $\mathbb{A}$ and $\mathbb{B}$

- $\|\partial F(\mathbf{X})\|_F$: the distance from the origin $\mathbf{0}$ to the boundary of the set $\partial F(\mathbf{X})$ with $\|\partial F(\mathbf{X})\|_F = \inf_{\mathbf{Y} \in \partial F(\mathbf{X})} \|\mathbf{Y}\|_F = \mathrm{dist}(\mathbf{0}, \partial F(\mathbf{X}))$

## A.2 TECHNICAL PRELIMINARIES

As the function $F(\cdot)$ can be non-convex and non-smooth, we introduce some tools in non-smooth analysis (Mordukhovich, 2006; Rockafellar & Wets., 2009). The domain of any extended real-valued function $F : \mathbb{R}^{n \times r} \to (-\infty, +\infty]$ is defined as $\mathrm{dom}(F) \triangleq \{\mathbf{X} \in \mathbb{R}^{n \times r} : |F(\mathbf{X})| < +\infty\}$. The Fréchet subdifferential of $F$ at $\mathbf{X} \in \mathrm{dom}(F)$ is defined as $\hat{\partial} F(\mathbf{X}) \triangleq \{\boldsymbol{\xi} \in \mathbb{R}^{n \times r} : \lim_{\mathbf{Z} \to \mathbf{X}} \inf_{\mathbf{Z} \neq \mathbf{X}} \frac{F(\mathbf{Z}) - F(\mathbf{X}) - \langle \boldsymbol{\xi}, \mathbf{Z} - \mathbf{X} \rangle}{\|\mathbf{Z} - \mathbf{X}\|_F} \geq 0\}$, while the limiting subdifferential of $F(\mathbf{X})$ at $\mathbf{X} \in \mathrm{dom}(F)$ is denoted as $\partial F(\mathbf{X}) \triangleq \{\boldsymbol{\xi} \in \mathbb{R}^n : \exists \mathbf{X}^t \to \mathbf{X}, F(\mathbf{X}^t) \to F(\mathbf{X}), \boldsymbol{\xi}^t \in \hat{\partial} F(\mathbf{X}^t) \to \boldsymbol{\xi}, \forall t\}$. We denote $\nabla F(\mathbf{X})$ as the gradient of $F(\cdot)$ at $\mathbf{X}$ in the Euclidean space. We have the following relation between $\hat{\partial} F(\mathbf{X})$, $\partial F(\mathbf{X})$, and $\nabla F(\mathbf{X})$. (i) It holds that $\hat{\partial} F(\mathbf{X}) \subseteq \partial F(\mathbf{X})$. (ii) If the function $F(\cdot)$ is convex, $\partial F(\mathbf{X})$ and $\hat{\partial} F(\mathbf{X})$ essentially the classical subdifferential for convex functions, i.e., $\partial F(\mathbf{X}) = \hat{\partial} F(\mathbf{X}) = \{\boldsymbol{\xi} \in \mathbb{R}^{n \times r} : F(\mathbf{Z}) \geq F(\mathbf{X}) + \langle \boldsymbol{\xi}, \mathbf{Z} - \mathbf{X} \rangle, \forall \mathbf{Z} \in \mathbb{R}^{n \times r}\}$. (iii) If the function $F(\cdot)$ is differentiable, then $\hat{\partial} F(\mathbf{X}) = \partial F(\mathbf{X}) = \{\nabla F(\mathbf{X})\}$.

We need some prerequisite knowledge in optimization with orthogonality constraints (Absil et al., 2008). The nearest orthogonality matrix to an arbitrary matrix $\mathbf{Y} \in \mathbb{R}^{n \times r}$ is given by $\mathbb{P}_{\mathcal{M}}(\mathbf{Y}) = \hat{\mathbf{U}} \hat{\mathbf{V}}^\top$, where $\mathbf{Y} = \hat{\mathbf{U}} \mathrm{Diag}(\mathbf{s}) \hat{\mathbf{V}}^\top$ is the singular value decomposition of $\mathbf{Y}$. We use $\mathcal{N}_{\mathcal{M}}(\mathbf{X})$ to denote the limiting normal cone to $\mathcal{M}$ at $\mathbf{X}$, leading to $\mathcal{N}_{\mathcal{M}}(\mathbf{X}) = \partial \mathcal{I}_{\mathcal{M}}(\mathbf{X}) = \{\mathbf{Z} \in \mathbb{R}^{n \times r} : \langle \mathbf{Z}, \mathbf{X} \rangle \geq \langle \mathbf{Z}, \mathbf{Y} \rangle, \forall \mathbf{Y} \in \mathcal{M}\}$. The tangent and norm space to $\mathcal{M}$ at $\mathbf{X} \in \mathcal{M}$ are denoted as $\mathrm{T}_{\mathbf{X}} \mathcal{M}$ and $\mathrm{N}_{\mathbf{X}} \mathcal{M}$, respectively. For a given $\mathbf{X} \in \mathcal{M}$, we let $\mathcal{A}_{\mathbf{X}}(\mathbf{Y}) \triangleq \mathbf{X}^\top \mathbf{Y} + \mathbf{Y}^\top \mathbf{X}$ for $\mathbf{Y} \in \mathbb{R}^{n \times r}$, and we have $\mathrm{T}_{\mathbf{X}} \mathcal{M} = \{\mathbf{Y} \in \mathbb{R}^{n \times r} | \mathcal{A}_X(\mathbf{Y}) = \mathbf{0}\}$ and $\mathrm{N}_{\mathbf{X}} \mathcal{M} = 2\mathbf{X}\boldsymbol{\Lambda} \,|\, \boldsymbol{\Lambda} = \boldsymbol{\Lambda}^\top, \boldsymbol{\Lambda} \in \mathbb{R}^{r \times r}\}$. For any non-convex and non-smooth function $F(\mathbf{X})$, we use $\partial_{\mathcal{M}} F(\mathbf{X})$ to denote the limiting Riemannian gradient of $F(\mathbf{X})$ at $\mathbf{X}$, and obtain $\partial_{\mathcal{M}} F(\mathbf{X}) = \mathbb{P}_{\mathrm{T}_{\mathbf{X}} \mathcal{M}}(\partial F(\mathbf{X}))$. We denote $\partial F(\mathbf{X}) \ominus \mathbf{X}[\partial F(\mathbf{X})]^\top \mathbf{X} \triangleq \{\mathbb{E} \,|\, \mathbb{E} = \mathbf{G} - \mathbf{X}\mathbf{G}^\top \mathbf{X}, \mathbf{G} \in \partial F(\mathbf{X})\}$.

## A.3 RELEVANT LEMMAS

We offer a set of useful lemmas, each of which stands independently of context and specific methodology.

**Lemma A.1.** *Let $k \geq 2$ and $\mathbf{W} \in \mathbb{R}^{n \times n}$. If $\mathbf{0}_{k,k} = \mathrm{U}_{\mathbb{B}}^\top \mathbf{W} \mathrm{U}_{\mathbb{B}}$ for all $\mathbb{B} \in \{\mathcal{B}_i\}_{i=1}^{\mathrm{C}_n^k}$, then $\mathbf{W} = \mathbf{0}$. Here, the set $\{\mathcal{B}_1, \mathcal{B}_2, ..., \mathcal{B}_{\mathrm{C}_n^k}\}$ represents all possible combinations of the index vectors choosing $k$ items from $n$ without repetition.*

*Proof.* This result is based on elementary deductions. Notably, the conclusion of this lemma does not necessarily hold if $|\mathbb{B}| = k = 1$. This is because any matrix $\mathbf{W} \in \mathbb{R}^{n \times n}$ with $\mathbf{W}_{ii} = 0$ for all $i \in [n]$ satisfies the condition of this lemma but is not necessary a zero matrix. $\square$

**Lemma A.2.** *For any matrices $\mathbf{A} \in \mathbb{R}^{k \times k}$ and $\mathbf{C} \in \mathbb{R}^{k \times k}$, we have: $\|\mathbf{A} - \mathbf{A}^\top\|_F \leq 2\|\mathbf{A} - \mathbf{C}\|_F + \|\mathbf{C} - \mathbf{C}^\top\|_F$.*

*Proof.* We derive: $\|\mathbf{A} - \mathbf{A}^\mathsf{T}\|_\mathsf{F} = \|(\mathbf{A} - \mathbf{C}) + (\mathbf{C} - \mathbf{C}^\mathsf{T}) + (\mathbf{C}^\mathsf{T} - \mathbf{A}^\mathsf{T})\|_\mathsf{F} \overset{①}{\leq} \|\mathbf{A} - \mathbf{C}\|_\mathsf{F} + \|\mathbf{C} - \mathbf{C}^\mathsf{T}\|_\mathsf{F} + \|\mathbf{C}^\mathsf{T} - \mathbf{A}^\mathsf{T}\|_\mathsf{F} = 2\|\mathbf{A} - \mathbf{C}\|_\mathsf{F} + \|\mathbf{C} - \mathbf{C}^\mathsf{T}\|_\mathsf{F}$, where step ① uses the triangle inequality.

$\square$

**Lemma A.3.** *Let $\tau \in \mathbb{R}$, and $\mathbf{A} \in \mathbb{R}^{2 \times 2}$ be any skey-symmetric matrix with $\mathbf{A}^\mathsf{T} = -\mathbf{A}$. The matrix $\mathbf{Q} = [(\mathbf{I}_2 + \frac{\tau}{2}\mathbf{A})^{-1}(\mathbf{I}_k - \frac{\tau}{2}\mathbf{A})]$ is always a rotation matrix with $\det(\mathbf{Q}) = 1$.*

*Proof.* Since $\mathbf{A}$ is a two-dimensional matrix, it can be expressed in the form: $\mathbf{A} = \left( \begin{smallmatrix} 0 & a \\ -a & 0 \end{smallmatrix} \right)$ for some $a \in \mathbb{R}$. Letting $b = \frac{\tau}{2}a$, we derive:

$$\mathbf{Q} = (\mathbf{I}_2 + \tfrac{\tau}{2}\mathbf{A})^{-1}(\mathbf{I}_k - \tfrac{\tau}{2}\mathbf{A}) \overset{①}{=} \left( \begin{smallmatrix} 1 & b \\ -b & 1 \end{smallmatrix} \right)^{-1} \left( \begin{smallmatrix} 1 & -b \\ b & 1 \end{smallmatrix} \right) \overset{②}{=} \tfrac{1}{1+b^2} \left( \begin{smallmatrix} 1 & -b \\ b & 1 \end{smallmatrix} \right) \left( \begin{smallmatrix} 1 & -b \\ b & 1 \end{smallmatrix} \right) = \tfrac{1}{1+b^2} \left( \begin{smallmatrix} 1-b^2 & -2b \\ 2b & 1-b^2 \end{smallmatrix} \right),$$

where step ① uses $\frac{\tau}{2}\mathbf{A} = \left( \begin{smallmatrix} 0 & b \\ -b & 0 \end{smallmatrix} \right)$; step ② uses the fact that $\left( \begin{smallmatrix} a & b \\ c & d \end{smallmatrix} \right)^{-1} = \frac{1}{ad-bc} \left( \begin{smallmatrix} d & -b \\ -c & a \end{smallmatrix} \right)^{-1}$ for all $a, b, c, d \in \mathbb{R}$. We further obtain: $\det(\mathbf{Q}) \overset{①}{=} \frac{1-b^2}{1+b^2} \cdot \frac{1-b^2}{1+b^2} - \frac{2b}{1+b^2} \cdot \frac{-2b}{1+b^2} = \frac{(1-b^2)^2 + 4b^2}{(1+b^2)^2} = \frac{(1+b^2)^2}{(1+b^2)^2} = 1$, where step ① uses the fact that $\det\left( \begin{smallmatrix} a & b \\ c & d \end{smallmatrix} \right) = ad - bc$ for all $a, b, c, d \in \mathbb{R}$.

$\square$

**Lemma A.4.** *For any $\mathbf{W} \in \mathbb{R}^{n \times n}$, we have $\sum_{i=1}^{\mathrm{C}_n^k} \|\mathbf{W}(\mathcal{B}_i, \mathcal{B}_i)\|_\mathsf{F}^2 = \mathrm{C}_{n-2}^{k-2} \sum_i \sum_{j,j \neq i} \mathbf{W}_{ij}^2 + \frac{k}{n} \mathrm{C}_n^k \sum_i \mathbf{W}_{ii}^2$. Here, the set $\{\mathcal{B}_1, \mathcal{B}_2, ..., \mathcal{B}_{\mathrm{C}_n^k}\}$ represents all possible combinations of the index vectors choosing $k$ items from $n$ without repetition.*

*Proof.* For any matrix $\mathbf{W} \in \mathbb{R}^{n \times n}$, we define: $\mathbf{w} \triangleq \mathrm{diag}(\mathbf{W}) \in \mathbb{R}^n$, and $\mathbf{W}' \triangleq \mathbf{W} - \mathrm{Diag}(\mathbf{w})$.

We have: $\mathbf{W} = \mathrm{Diag}(\mathbf{w}) + \mathbf{W}'$, this leads to the following decomposition:

$$
\begin{aligned}
\sum_{i=1}^{\mathrm{C}_n^k} \|\mathbf{U}_{\mathcal{B}_i}^\mathsf{T} \mathbf{W} \mathbf{U}_{\mathcal{B}_i}\|_\mathsf{F}^2 &= \sum_{i=1}^{\mathrm{C}_n^k} \|\mathbf{U}_{\mathcal{B}_i}^\mathsf{T} (\mathrm{Diag}(\mathbf{w}) + \mathbf{W}') \mathbf{U}_{\mathcal{B}_i}\|_\mathsf{F}^2 \\
&= \underbrace{\sum_{i=1}^{\mathrm{C}_n^k} \|\mathbf{U}_{\mathcal{B}_i}^\mathsf{T} \mathrm{Diag}(\mathbf{w}) \mathbf{U}_{\mathcal{B}_i}\|_\mathsf{F}^2}_{\Gamma_1} + \underbrace{\sum_{i=1}^{\mathrm{C}_n^k} \|\mathbf{U}_{\mathcal{B}_i}^\mathsf{T} \mathbf{W}' \mathbf{U}_{\mathcal{B}_i}\|_\mathsf{F}^2}_{\Gamma_2}. \quad (15)
\end{aligned}
$$

We first focus on the term $\Gamma_1$. We have:

$$\Gamma_1 = \sum_{i=1}^{\mathrm{C}_n^k} \|\mathbf{U}_{\mathcal{B}_i}^\mathsf{T} \mathrm{Diag}(\mathbf{w}) \mathbf{U}_{\mathcal{B}_i}\|_\mathsf{F}^2 \overset{①}{=} \sum_{i=1}^{\mathrm{C}_n^k} \|\mathbf{w}_{\mathcal{B}_i}\|_2^2 \overset{②}{=} \mathrm{C}_n^k \cdot \tfrac{k}{n} \cdot \|\mathbf{w}\|_2^2 = \mathrm{C}_n^k \cdot \tfrac{k}{n} \cdot \sum_i \mathbf{W}_{ii}^2, \quad (16)$$

where step ① uses the fact that $\|\mathtt{B}^\mathsf{T} \mathrm{Diag}(\mathbf{w}) \mathtt{B}\|_\mathsf{F}^2 = \|[\mathrm{Diag}(\mathbf{w})]_{\mathtt{B}\mathtt{B}}\|_\mathsf{F}^2 = \|\mathbf{w}_\mathtt{B}\|_2^2$ for any $\mathtt{B} \in \{\mathcal{B}_i\}_{i=1}^{\mathrm{C}_n^k}$; step ② uses the observation that $\mathbf{w}_i$ appears in the term $\sum_{i=1}^{\mathrm{C}_n^k} \|\mathbf{w}_{\mathcal{B}_i}\|_2^2$ a total of $(\mathrm{C}_n^k \cdot \frac{k}{n})$ times, which can be deduced using basic induction.

We now focus on the term $\Gamma_2$. Noticing that $\mathbf{W}'_{ii} = 0$ for all $i \in [n]$, we have:

$$\Gamma_2 = \sum_{i=1}^{\mathrm{C}_n^k} \|\mathbf{U}_{\mathcal{B}_i}^\mathsf{T} \mathbf{W}' \mathbf{U}_{\mathcal{B}_i}\|_\mathsf{F}^2 \overset{①}{=} \sum_i \sum_{j,j \neq i} [\mathrm{C}_{n-2}^{k-2} (\mathbf{W}'_{ij})^2] \overset{②}{=} \mathrm{C}_{n-2}^{k-2} \sum_i \sum_{j,j \neq i} (\mathbf{W}_{ij})^2, \quad (17)$$

where step ① uses the fact that the term $\sum_{i=1}^{\mathrm{C}_n^k} \|\mathbf{U}_{\mathcal{B}_i}^\mathsf{T} \mathbf{W}' \mathbf{U}_{\mathcal{B}_i}\|_\mathsf{F}^2$ comprises $\mathrm{C}_{n-2}^{k-2}$ distinct patterns, each including $\{i, j\}$ with $i \neq j$; step ② uses $\sum_{i,j \neq i} (\mathbf{W}_{ij})^2 = \sum_{i,j \neq i} (\mathbf{W}'_{ij})^2$.

In view of Equalities (15), (16), and (17), we complete the proof of this lemma. $\square$

**Lemma A.5.** *Assume $\mathbf{Q}\mathbf{R} = \mathbf{X} \in \mathbb{R}^{n \times n}$, where $\mathbf{Q} \in \mathrm{St}(n, n)$ and $\mathbf{R}$ is a lower triangular matrix with $\mathbf{R}_{i,j} = 0$ for all $i < j$. If $\mathbf{X} \in \mathrm{St}(n, n)$, then $\mathbf{R}$ is a diagonal matrix with $\mathbf{R}_{i,i} \in \{-1, +1\}$ for all $i \in [n]$.*

*Proof.* We derive: $\mathbf{R}\mathbf{R}^\mathsf{T} \overset{①}{=} (\mathbf{Q}\mathbf{X})(\mathbf{Q}\mathbf{X})^\mathsf{T} = \mathbf{Q}\mathbf{X}\mathbf{X}^\mathsf{T}\mathbf{Q}^\mathsf{T} \overset{②}{=} \mathbf{I}$, where step ① uses $\mathbf{R} = \mathbf{Q}^\mathsf{T}\mathbf{X}$; step ② uses $\mathbf{X} \in \mathrm{St}(n, n)$ and $\mathbf{Q} \in \mathrm{St}(n, n)$. First, given $\|\mathbf{R}(1, :)\| = 1$ and $\mathbf{R}(1, 2 : n) = 0$, we have $\mathbf{R}_{1,1} \in \{-1, +1\}$. Second, we have $\|\mathbf{R}(2, :)\| = 1$ and $\mathbf{R}(1, :)^\mathsf{T}\mathbf{R}(:, 2) = 0$, leading to $\mathbf{R}_{1,2} = 0$ and $\mathbf{R}_{2,2} \in \{-1, +1\}$. Finally, using similar recursive strategy, we conclude that $\mathbf{R}$ is a diagonal matrix with $\mathbf{R}_{i,i} \in \{-1, +1\}$ for all $i \in [n]$. $\square$

**Lemma A.6.** *We define* $T_{\mathbf{X}}\mathcal{M} \triangleq \{\mathbf{Y} \in \mathbb{R}^{n \times r} \mid \mathcal{A}_X(\mathbf{Y}) = \mathbf{0}\}$ *and* $\mathcal{A}_{\mathbf{X}}(\mathbf{Y}) \triangleq \mathbf{X}^{\mathsf{T}}\mathbf{Y} + \mathbf{Y}^{\mathsf{T}}\mathbf{X}$. *For any* $\mathbf{G} \in \mathbb{R}^{n \times r}$ *and* $\mathbf{X} \in \mathrm{St}(n, k)$, *we have:* $(\mathbf{G} - \frac{1}{2}\mathbf{X}\mathcal{A}_{\mathbf{X}}(\mathbf{G})) = \arg\min_{\mathbf{Y} \in T_{\mathbf{X}}\mathcal{M}} \|\mathbf{Y} - \mathbf{G}\|_{\mathsf{F}}^2$.

*Proof.* The conclusion of this lemma can be found in (Absil et al., 2008). For completeness, we present a short proof.

Consider the convex problem: $\bar{\mathbf{Y}} = \arg\min_{\mathbf{Y}} \|\mathbf{Y} - \mathbf{G}\|_{\mathsf{F}}^2$, $s.t.\, \mathbf{X}^{\mathsf{T}}\mathbf{Y} + \mathbf{Y}^{\mathsf{T}}\mathbf{X} = \mathbf{0}$. Introducing a multiplier $\mathbf{\Lambda} \in \mathbb{R}^{r \times r}$ for the linear constraints leads to the following Lagrangian function: $\tilde{\mathcal{L}}(\mathbf{Y}; \mathbf{\Lambda}) = \|\mathbf{Y} - \mathbf{G}\|_{\mathsf{F}}^2 + \langle \mathbf{X}^{\mathsf{T}}\mathbf{Y} + \mathbf{Y}^{\mathsf{T}}\mathbf{X}, \mathbf{\Lambda} \rangle$. We derive the subsequent first-order optimality condition: $2(\mathbf{Y} - \mathbf{G}) + \mathbf{X}(\mathbf{\Lambda} + \mathbf{\Lambda}^{\mathsf{T}}) = \mathbf{0}$, and $\mathbf{X}^{\mathsf{T}}\mathbf{Y} + \mathbf{Y}^{\mathsf{T}}\mathbf{X} = \mathbf{0}$. Given $\mathbf{\Lambda}$ is symmetric, we have $\mathbf{Y} = \mathbf{G} - \mathbf{X}\mathbf{\Lambda}$. Incorporating this result into $\mathbf{X}^{\mathsf{T}}\mathbf{Y} + \mathbf{Y}^{\mathsf{T}}\mathbf{X} = \mathbf{0}$, we obtain: $\mathbf{X}^{\mathsf{T}}(\mathbf{G} - \mathbf{X}\mathbf{\Lambda}) + (\mathbf{G} - \mathbf{X}\mathbf{\Lambda})^{\mathsf{T}}\mathbf{X} = \mathbf{0}$. Given $\mathbf{X} \in \mathrm{St}(n, r)$, we have $\mathbf{X}^{\mathsf{T}}\mathbf{G} - \mathbf{\Lambda} + \mathbf{G}^{\mathsf{T}}\mathbf{X} - \mathbf{\Lambda}^{\mathsf{T}} = \mathbf{0}$, leading to: $\mathbf{\Lambda} = \frac{1}{2}(\mathbf{X}^{\mathsf{T}}\mathbf{G} + \mathbf{G}^{\mathsf{T}}\mathbf{X})$. Therefore, the optimal solution $\bar{\mathbf{Y}}$ can be computed as $\bar{\mathbf{Y}} = \mathbf{G} - \mathbf{X}\mathbf{\Lambda} = \mathbf{G} - \frac{1}{2}\mathbf{X}(\mathbf{X}^{\mathsf{T}}\mathbf{G} + \mathbf{G}^{\mathsf{T}}\mathbf{X})$.

$\square$

**Lemma A.7.** *Consider the following problem:* $\min_{\mathbf{X}} F^{\circ}(\mathbf{X}) \triangleq F(\mathbf{X}) + \mathcal{I}_{\mathcal{M}}(\mathbf{X})$, *where* $F(\mathbf{X})$ *is defined in Equation (1). For any* $\mathbf{X} \in \mathrm{St}(n, r)$, *it holds that* $\mathrm{dist}(\mathbf{0}, \partial F^{\circ}(\mathbf{X})) \leq \mathrm{dist}(\mathbf{0}, \partial_{\mathcal{M}} F(\mathbf{X}))$.

*Proof.* We let $\mathbf{G} \in \partial F(\mathbf{X})$ and define $\mathcal{A}_{\mathbf{X}}(\mathbf{G}) \triangleq \mathbf{X}^{\mathsf{T}}\mathbf{G} + \mathbf{G}^{\mathsf{T}}\mathbf{X}$.

Recall that the following first-order optimality conditions are equivalent for all $\mathbf{X} \in \mathrm{St}(n, r)$: $(\mathbf{0} \in \partial F^{\circ}(\mathbf{X})) \Leftrightarrow (\mathbf{0} \in \mathbb{P}_{T_{\mathbf{X}}\mathcal{M}}(\partial F(\mathbf{X})))$. Therefore, we derive the following results:

$$
\begin{aligned}
\mathrm{dist}(\mathbf{0}, \partial F^{\circ}(\mathbf{X})) &= \inf_{\mathbf{Y} \in \partial F^{\circ}(\mathbf{X})} \|\mathbf{Y}\|_{\mathsf{F}} = \inf_{\mathbf{Y} \in \mathbb{P}_{(T_{\mathbf{X}}\mathcal{M})}(\partial F(\mathbf{X}))} \|\mathbf{Y}\|_{\mathsf{F}} \\
&\overset{①}{=} \|\mathbb{P}_{(T_{\mathbf{X}}\mathcal{M})}(\mathbf{G})\|_{\mathsf{F}} \\
&\overset{②}{=} \|\mathbf{G} - \tfrac{1}{2}\mathbf{X}\mathcal{A}_{\mathbf{X}}(\mathbf{G})\|_{\mathsf{F}} \\
&\overset{③}{=} \|\mathbf{G} - \tfrac{1}{2}\mathbf{X}(\mathbf{X}^{\mathsf{T}}\mathbf{G} + \mathbf{G}^{\mathsf{T}}\mathbf{X})\|_{\mathsf{F}} \\
&\overset{④}{=} \|(\mathbf{I} - \tfrac{1}{2}\mathbf{X}\mathbf{X}^{\mathsf{T}})(\mathbf{G} - \mathbf{X}\mathbf{G}^{\mathsf{T}}\mathbf{X})\|_{\mathsf{F}} \\
&\overset{⑤}{\leq} \|\mathbf{G} - \mathbf{X}\mathbf{G}^{\mathsf{T}}\mathbf{X}\|_{\mathsf{F}},
\end{aligned}
$$

where step ① uses $\mathbf{G} \in \partial F(\mathbf{X})$; step ② uses Lemma A.6; step ③ uses the definition of $\mathcal{A}_{\mathbf{X}}(\mathbf{G})$; step ④ uses the identity that $\mathbf{G} - \frac{1}{2}\mathbf{X}(\mathbf{X}^{\mathsf{T}}\mathbf{G} + \mathbf{G}^{\mathsf{T}}\mathbf{X}) = (\mathbf{I} - \frac{1}{2}\mathbf{X}\mathbf{X}^{\mathsf{T}})(\mathbf{G} - \mathbf{X}\mathbf{G}^{\mathsf{T}}\mathbf{X})$; step ⑤ uses the norm inequality and fact that the matrix $\mathbf{I} - \frac{1}{2}\mathbf{X}\mathbf{X}^{\mathsf{T}}$ only contains eigenvalues that are $\frac{1}{2}$ or 1.

$\square$

**Lemma A.8.** *Assume* $\cos(\theta) \neq 0$. *Any pair of trigonometric functions* $(\cos(\theta), \sin(\theta))$ *can be represented as follows:*

*a)* $\cos(\theta) = \frac{1}{\sqrt{1 + \tan^2(\theta)}}$, *and* $\sin(\theta) = \frac{\tan(\theta)}{\sqrt{1 + \tan^2(\theta)}}$.

*b)* $\cos(\theta) = \frac{-1}{\sqrt{1 + \tan^2(\theta)}}$, *and* $\sin(\theta) = \frac{-\tan(\theta)}{\sqrt{1 + \tan^2(\theta)}}$.

*Proof.* For all values of $\theta$ where $\cos(\theta) \neq 0$, the trigonometric functions $\{\sin(\theta), \cos(\theta), \tan(\theta)\}$ are well-defined. Utilizing the identity $\sin^2(\theta) + \cos^2(\theta) = 1$ and $\tan(\theta)\cos(\theta) = \sin(\theta)$, we derive: $(\tan(\theta) \cdot \cos(\theta))^2 + \cos^2(\theta) = 1$. Consequently, we find: $\cos(\theta) = \frac{\pm 1}{\sqrt{\tan^2(\theta) + 1}}$. Finally, we can express $\sin(\theta)$ as $\sin(\theta) = \tan(\theta) \cdot \cos(\theta) = \frac{\tan(\theta)}{\sqrt{\tan^2(\theta) + 1}}$.

$\square$

**Lemma A.9.** *Let* $A \in \mathbb{R}$ *and* $B \in \mathbb{R}$. *The minimizer of the following one-dimensional problem:*

$$
\bar{\theta} \in \arg\min_{\theta} h(\theta) = A\cos(\theta) + B\sin(\theta) \tag{18}
$$

*will be achieved at* $\bar{\theta}$, *where* $\cos(\bar{\theta}) = -\frac{A}{\sqrt{A^2 + B^2}}$, $\sin(\bar{\theta}) = -\frac{B}{\sqrt{A^2 + B^2}}$, *and* $h(\bar{\theta}) = -\sqrt{A^2 + B^2}$.

*Proof.* Initially, we consider the special case when $\cos(\theta) = 0$ or $A = 0$. Problem (18) reduces to:

$$\bar{\theta} \in \arg\min_\theta h(\theta) = B\sin(\theta).$$

Clearly, we have: $\sin(\bar{\theta}) = -\frac{B}{|B|}$, $\cos(\bar{\theta}) = 0$, and $h(\bar{\theta}) = -|B|$. The conclusion of this lemma holds.

We now assume that $A \neq 0$ and $\cos(\theta) \neq 0$ for all $\theta$. Using the fact that $\tan(\theta) = \frac{\sin(\theta)}{\cos(\theta)}$ and $\cos(\theta)^2 + \sin(\theta)^2 = 1$, we have the following two cases for $\cos(\theta)$ and $\sin(\theta)$:

**a)** $\cos(\theta) = \frac{1}{\sqrt{1+\tan^2(\theta)}}$, and $\sin(\theta) = \frac{\tan(\theta)}{\sqrt{1+\tan^2(\theta)}}$

**b)** $\cos(\theta) = \frac{-1}{\sqrt{1+\tan^2(\theta)}}$, and $\sin(\theta) = \frac{-\tan(\theta)}{\sqrt{1+\tan^2(\theta)}}$.

Therefore, Problem (18) reduces to the following equivalent minimization problem:

$$\bar{\theta} \in \arg\min_\theta \frac{\pm A \pm \tan(\theta)B}{\sqrt{1+\tan^2(\theta)}}.$$

Using the variable substitution that $\tan(\theta) = t$, we have the following equivalent problem:

$$\bar{t} \in \arg\min_t h(t) \triangleq \frac{\pm(A+Bt)}{\sqrt{1+t^2}}.$$

For any optimal solution $\bar{t}$, we have the following necessary first-order optimality condition:

$$0 \in \partial h(\bar{t}) = \frac{\pm B\sqrt{1+\bar{t}^2} \mp (A+B\bar{t})\cdot(1+\bar{t}^2)^{-1/2}\bar{t}}{1+\bar{t}^2}$$
$$\Rightarrow \quad 0 \in \pm B\sqrt{1+\bar{t}^2} \mp \frac{(A+B\bar{t})\bar{t}}{\sqrt{1+\bar{t}^2}} \Rightarrow B\sqrt{1+\bar{t}^2} = \frac{(A+B\bar{t})\bar{t}}{\sqrt{1+\bar{t}^2}} \Rightarrow \bar{t} = \frac{B}{A}$$

Therefore, we have: $\bar{t} = \frac{B}{A} = \tan(\bar{\theta})$. The optimal solution pair $[\cos(\bar{\theta}), \sin(\bar{\theta})]$ for Problem (18) can be computed as one of the following two cases:

**a)** $\cos(\bar{\theta}) = \frac{A}{\sqrt{A^2+B^2}}$, and $\sin(\bar{\theta}) = \frac{B}{\sqrt{A^2+B^2}}$.

**b)** $\cos(\bar{\theta}) = \frac{-A}{\sqrt{A^2+B^2}}$, and $\sin(\bar{\theta}) = \frac{-B}{\sqrt{A^2+B^2}}$.

In view of the original problem $\bar{\theta} = \arg\min_\theta h(\theta) = A\cos(\theta) + B\sin(\theta)$, we conclude that $\cos(\bar{\theta}) = \frac{-A}{\sqrt{A^2+B^2}}$, and $\sin(\bar{\theta}) = \frac{-B}{\sqrt{A^2+B^2}}$.

$\square$

**Lemma A.10.** *Assume $(e^{t+1})^2 \leq e^t(p^t - p^{t+1})$ and $p^t \geq p^{t+1}$, where $\{e^t, p^t\}_{t=0}^\infty$ are two nonnegative sequences. For all $i \geq 1$, we have: $\sum_{t=i}^\infty e^{t+1} \leq e^i + 2p^i$.*

*Proof.* We define $w_t \triangleq p^t - p^{t+1}$. We let $1 \leq i < T$.

First, for any $i \geq 1$, we have:

$$\sum_{t=i}^T w_t = \sum_{t=i}^T (p^t - p^{t+1}) = p^i - p^{T+1} \overset{①}{\leq} p^i, \tag{19}$$

where step ① uses $p^i \geq 0$ for all $i$.

Second, we obtain:

$$e^{t+1} \overset{①}{\leq} \sqrt{e^t w_t}$$
$$\overset{②}{\leq} \sqrt{\tfrac{\alpha}{2}(e^t)^2 + (w_t)^2/(2\alpha)}, \forall\alpha > 0$$
$$\overset{③}{\leq} \sqrt{\tfrac{\alpha}{2}} \cdot e^t + w_t\sqrt{1/(2\alpha)}, \forall\alpha > 0. \tag{20}$$

Here, step ① uses $(e^{t+1})^2 \leq e^t(p^t - p^{t+1})$ and $w_t \triangleq p^t - p^{t+1}$; step ② uses the fact that $ab \leq \frac{\alpha}{2}a^2 + \frac{1}{2\alpha}b^2$ for all $\alpha > 0$; step ③ uses the fact that $\sqrt{a+b} \leq \sqrt{a} + \sqrt{b}$ for all $a, b \geq 0$.

Assume $1 - \sqrt{\frac{\alpha}{2}} > 0$. Telescoping Inequality (20) over $t$ from $i$ to $T$, we have:

$$\sum_{t=i}^{T} w_t \sqrt{1/(2\alpha)}$$

$$\geq \{\sum_{t=i}^{T} e^{t+1}\} - \sqrt{\tfrac{\alpha}{2}}\{\sum_{t=i}^{T} e^t\}$$

$$= \{e^{T+1} + \sum_{t=i}^{T-1} e^{t+1}\} - \sqrt{\tfrac{\alpha}{2}}\{e^i + \sum_{t=i}^{T-1} e^{t+1}\}$$

$$= e^{T+1} - \sqrt{\tfrac{\alpha}{2}}e^i + (1 - \sqrt{\tfrac{\alpha}{2}})\sum_{t=i}^{T-1} e^{t+1}$$

$$\overset{①}{\geq} -\sqrt{\tfrac{\alpha}{2}}e^i + (1 - \sqrt{\tfrac{\alpha}{2}})\sum_{t=i}^{T-1} e^{t+1},$$

where step ① uses $e^{T+1} \geq 0$ and $1 - \sqrt{\frac{\alpha}{2}} > 0$. This leads to:

$$\sum_{t=i}^{T-1} e^{t+1} \quad \leq \quad (1 - \sqrt{\tfrac{\alpha}{2}})^{-1} \cdot \{\sqrt{\tfrac{\alpha}{2}}e^i + \sqrt{\tfrac{1}{2\alpha}}\sum_{t=i}^{T} w_t\}$$

$$\overset{①}{=} \quad e^i + 2\sum_{t=i}^{T} w_t$$

$$\overset{②}{\leq} \quad e^i + 2p^i,$$

step ① uses the fact that $(1 - \sqrt{\frac{\alpha}{2}})^{-1} \cdot \sqrt{\frac{\alpha}{2}} = 1$ and $(1 - \sqrt{\frac{\alpha}{2}})^{-1} \cdot \sqrt{\frac{1}{2\alpha}} = 2$ when $\alpha = \frac{1}{2}$; step ② uses Inequalities (19). Letting $T \to \infty$, we conclude this lemma.

$\square$

**Lemma A.11.** *Let $\{d^t\}_{t=0}^{\infty}$ be any nonnegative sequence. Assume that $[d^t]^{\tau+1} \leq a(d^{t-1} - d^t)$, where $\tau, a > 0$. We have: $d^T \leq \mathcal{O}(T^{-1/\tau})$.*

*Proof.* We let $\kappa > 1$ be any constant. We define $h(s) = s^{-\tau-1}$, where $\tau > 0$.

We consider two cases for $r^t \triangleq h(d^t)/h(d^{t-1})$.

**Case (1)**. $r^t \leq \kappa$. We define $\breve{h}(s) \triangleq -\frac{1}{\tau} \cdot s^{-\tau}$. We derive:

$$1 \quad \overset{①}{\leq} \quad a(d^{t-1} - d^t) \cdot h(d^t)$$

$$\overset{②}{\leq} \quad a(d^{t-1} - d^t) \cdot \kappa h(d^{t-1})$$

$$\overset{③}{\leq} \quad a\kappa \int_{d^t}^{d^{t-1}} h(s)ds$$

$$\overset{④}{=} \quad a\kappa \cdot (\breve{h}(d^{t-1}) - \breve{h}(d^t))$$

$$\overset{⑤}{=} \quad a\kappa \cdot \tfrac{1}{\tau} \cdot ([d^t]^{-\tau} - [d^{t-1}]^{-\tau}),$$

where step ① uses $[d^t]^{\tau+1} \leq a(d^{t-1} - d^t)$; step ② uses $h(d^t) \leq \kappa h(d^{t-1})$; step ③ uses the fact that $h(s)$ is a nonnegative and increasing function that $(a - b)h(a) \leq \int_b^a h(s)ds$ for all $a, b \in [0, \infty)$; step ④ uses the fact that $\nabla \breve{h}(s) = h(s)$; step ⑤ uses the definition of $\breve{h}(\cdot)$. This leads to:

$$[d^t]^{-\tau} - [d^{t-1}]^{-\tau} \geq \tfrac{\tau}{\kappa\alpha}. \tag{21}$$

**Case (2)**. $r^t > \kappa$. We have:

$$h(d^t) > \kappa h(d^{t-1}) \quad \overset{①}{\Rightarrow} \quad [d^t]^{-(\tau+1)} > \kappa \cdot [d^{t-1}]^{-(\tau+1)}$$

$$\overset{②}{\Rightarrow} \quad ([d^t]^{-(\tau+1)})^{\frac{\tau}{\tau+1}} > \kappa^{\frac{\tau}{\tau+1}} \cdot ([d^{t-1}]^{-(\tau+1)})^{\frac{\tau}{\tau+1}}$$

$$\Rightarrow \quad [d^t]^{-\tau} > \kappa^{\frac{\tau}{\tau+1}} \cdot [d^{t-1}]^{-\tau}, \tag{22}$$

where step ① uses the definition of $h(\cdot)$; step ② uses the fact that if $a > b > 0$, then $a^{\dot{\tau}} > b^{\dot{\tau}}$ for any exponent $\dot{\tau} \triangleq \frac{\tau}{\tau+1} \in (0, 1)$. For any $t \geq 1$, we derive:

$$[d^t]^{-\tau} - [d^{t-1}]^{-\tau} \quad \overset{①}{\geq} \quad (\kappa^{\frac{\tau}{\tau+1}} - 1) \cdot [d^{t-1}]^{-\tau}$$

$$\overset{②}{\geq} \quad (\kappa^{\frac{\tau}{\tau+1}} - 1) \cdot [d^0]^{-\tau}, \tag{23}$$

where step ① uses Inequality (22); step ② uses $\tau > 0$ and $d^{t-1} \leq d^0$ for all $t \geq 1$.

In view of Inequalities (21) and (23), we have:

$$[d^t]^{-\tau} - [d^{t-1}]^{-\tau} \geq \underbrace{\min(\tfrac{\tau}{\kappa\alpha}, (\kappa^{\frac{\tau}{\tau+1}} - 1) \cdot [d^0]^{-\tau})}_{\triangleq \ddot{c}}. \tag{24}$$

Telescoping Inequality (24) over $t$ from 1 to $T$, we have:

$$[d^T]^{-\tau} - [d^0]^{-\tau} \geq T\ddot{c}.$$

This leads to:

$$d^T = ([d^T]^{-\tau})^{-1/\tau} \leq \mathcal{O}(T^{-1/\tau}).$$

$\square$

# B  ADDITIONAL EXAMPLES OF THE BREAKPOINT SEARCHING METHOD

In this section, we provide additional examples of **BSM** for other different $h(\mathbf{x})$.

▶ **Finding the Breakpoint Set for** $h(\mathbf{x}) \triangleq \lambda\|\mathbf{x}\|_1$

Since the function $h(\mathbf{x}) \triangleq \lambda\|\mathbf{x}\|_1$ is symmetric, Problem (12) reduces to the following problem:

$$\bar{t} \in \arg\min_t p(t) \triangleq \tfrac{a+bt}{\sqrt{1+t^2}} + \tfrac{w+dt}{1+t^2} + \tfrac{\lambda\|\mathbf{x}+t\mathbf{y}\|_1}{\sqrt{1+t^2}}. \tag{25}$$

Setting the subgradient of $p(\cdot)$ to zero yields: $0 \in \partial p(t) = t^\circ[d(1+t^2) - (w+dt)2t + (b-at) \cdot \sqrt{1+t^2}] + t^\circ\lambda \cdot \sqrt{1+t^2} \cdot [\langle\text{sign}(\mathbf{x}+t\mathbf{y}), \mathbf{y}\rangle(1+t^2) - \|\mathbf{x}+t\mathbf{y}\|_1 t]$, where $t^\circ = (1+t^2)^{-2}$. We consider the following two cases. *(i)* We assume $(\mathbf{x}+t\mathbf{y})_i = 0$ for some $i$. Then the solution $\bar{t}$ can be determined using $\bar{t} = \tfrac{\mathbf{x}_i}{\mathbf{y}_i}$. There are $2r$ breakpoints $\{\tfrac{\mathbf{x}_1}{\mathbf{y}_1}, \tfrac{\mathbf{x}_2}{\mathbf{y}_2}, ..., \tfrac{\mathbf{x}_{2r}}{\mathbf{y}_{2r}}\}$ for this case. *(ii)* We now assume $(\mathbf{x}+t\mathbf{y})_i \neq 0$ for all $i$. We define $\mathbf{z} \triangleq \{+\tfrac{\mathbf{x}_1}{\mathbf{y}_1}, -\tfrac{\mathbf{x}_1}{\mathbf{y}_1}, +\tfrac{\mathbf{x}_2}{\mathbf{y}_2}, -\tfrac{\mathbf{x}_2}{\mathbf{y}_2}, ..., +\tfrac{\mathbf{x}_{2r}}{\mathbf{y}_{2r}}, -\tfrac{\mathbf{x}_{2r}}{\mathbf{y}_{2r}}\} \in \mathbb{R}^{4r\times 1}$, and sort $\mathbf{z}$ in non-descending order. Given $\bar{t} \neq \mathbf{z}_i$ for all $i$ in this case, the domain $p(t)$ can be divided into $(4r+1)$ non-overlapping intervals: $(-\infty, \mathbf{z}_1), (\mathbf{z}_1, \mathbf{z}_2), ..., (\mathbf{z}_{4r}, +\infty)$. In each interval, $\text{sign}(\mathbf{x}+t\mathbf{y}) \triangleq \mathbf{o}$ can be determined. Combining with the fact that $t^\circ > 0$ and $\|\mathbf{x}+t\mathbf{y}\|_1 = \langle\mathbf{o}, \mathbf{x}+t\mathbf{y}\rangle$, the first-order optimality condition reduces to: $0 = [d(1+t^2) - (w+dt)2t + (b-at) \cdot \sqrt{1+t^2}] + \lambda \cdot \sqrt{1+t^2} \cdot [\langle\mathbf{o}, \mathbf{y}\rangle(1+t^2) - \langle\mathbf{o}, \mathbf{x}+t\mathbf{y}\rangle t]$, which can be simplified as: $(at-b) \cdot \sqrt{1+t^2} - \lambda \cdot \sqrt{1+t^2} \cdot [\langle\mathbf{o}, \mathbf{y}-t\mathbf{x}\rangle] = [d(1+t^2) - (w+dt)2t]$. We square both sides and then solve the quartic equation. We obtain obtain all its real roots $\{\bar{t}_1, \bar{t}_2, ..., \bar{t}_j\}$ with $1 \leq j \leq 4$. Therefore, Problem (25) contains at most $2r + (4r+1) \times 4$ breakpoints.

▶ **Finding the Breakpoint Set for** $h(\mathbf{x}) \triangleq I_{\geq 0}(\mathbf{x})$

Since the function $h(\mathbf{x}) \triangleq \mathcal{I}_{\geq 0}(\mathbf{x})$ is scale-invariant with $h(t\mathbf{x}) = h(\mathbf{x})$ forall $t \geq 0$, Problem (12) reduces to the following problem:

$$\bar{t} \in \arg\min_t p(t) \triangleq \tfrac{a+bt}{\sqrt{1+t^2}} + \tfrac{w+dt}{1+t^2}, \, s.t. \, \mathbf{x}+t\mathbf{y} \geq \mathbf{0}. \tag{26}$$

We define $I \triangleq \{i|\mathbf{y}_i > 0\}$ and $J \triangleq \{i|\mathbf{y}_i < 0\}$. It is not difficult to verity that $\{x+t\mathbf{y} \geq 0\} \Leftrightarrow \{-\tfrac{\mathbf{x}_I}{\mathbf{y}_I} \leq t, t \leq -\tfrac{\mathbf{x}_J}{\mathbf{y}_J}\} \Leftrightarrow \{lb \triangleq \max(-\tfrac{\mathbf{x}_I}{\mathbf{y}_I}) \leq t \leq \min(-\tfrac{\mathbf{x}_J}{\mathbf{y}_J}) \triangleq ub\}$. When $lb > ub$, we can directly conclude that the problem has no solution for this case. Now we assume $ub \geq lb$ and define $P(t) \triangleq \min(ub, \max(t, lb))$. We omit the bound constraint and set the gradient of $p(t)$ to zero, which yields: $0 = \nabla p(t) = [b(1+t^2) - (a+bt)t] \cdot \sqrt{1+t^2} \cdot t^\circ + [d(1+t^2) - (w+dt)(2t)] \cdot t^\circ$, where $t^\circ = (1+t^2)^{-2}$. We obtain all its real roots $\{\bar{t}_1, \bar{t}_2, ..., \bar{t}_j\}$ with $1 \leq j \leq 4$ after squaring both sides and solving the quartic equation. Combining with the bound constraints, we conclude that Problem (26) contains at most $(4+2)$ breakpoints $\{P(\bar{t}_1), P(\bar{t}_2), ..., P(\bar{t}_j), lb, ub\}$ with $1 \leq j \leq 4$.

## C ADDITIONAL DISCUSSIONS

This section encompasses various discussions, covering topics such as: (*i*) simple examples for the optimality hierarchy, (*ii*) the computation of the matrix $\mathbf{Q}$, (*iii*) a complexity comparison with full gradient methods, (*iv*) generalization to multiple row updates, and (*v*) the subdifferential of the cardinality function.

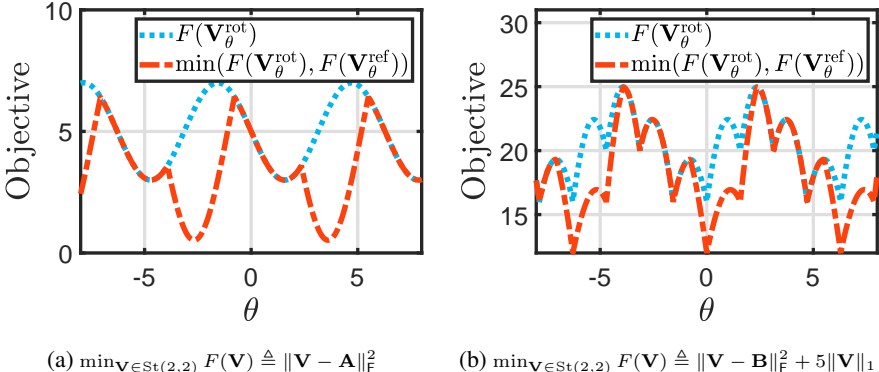

(a) $\min_{\mathbf{V}\in\mathrm{St}(2,2)} F(\mathbf{V}) \triangleq \|\mathbf{V} - \mathbf{A}\|_{\mathsf{F}}^2$     (b) $\min_{\mathbf{V}\in\mathrm{St}(2,2)} F(\mathbf{V}) \triangleq \|\mathbf{V} - \mathbf{B}\|_{\mathsf{F}}^2 + 5\|\mathbf{V}\|_1$

Figure 2: Geometric Visualizations of Two Examples of $2 \times 2$ Optimization Problems with Orthogonality Constraints with $\mathbf{A} = \left(\begin{smallmatrix} 1 & 0 \\ -1 & -1 \end{smallmatrix}\right)$ and $\mathbf{B} = \left(\begin{smallmatrix} 1 & 0 \\ 1 & 2 \end{smallmatrix}\right)$.

### C.1 SIMPLE EXAMPLES FOR THE OPTIMALITY HIERARCHY

To demonstrate the strong optimality of $\mathrm{BS}_2$-points and the advantages of the proposed method, we examine the following simple examples of $2 \times 2$ optimization problems mentioned in the paper:

$$\min_{\mathbf{V}\in\mathrm{St}(2,2)} F(\mathbf{V}) \triangleq \|\mathbf{V} - \mathbf{A}\|_{\mathsf{F}}^2, \text{ with } \mathbf{A} = \left(\begin{smallmatrix} 1 & 0 \\ -1 & -1 \end{smallmatrix}\right). \tag{27}$$

$$\min_{\mathbf{V}\in\mathrm{St}(2,2)} F(\mathbf{V}) \triangleq \|\mathbf{V} - \mathbf{B}\|_{\mathsf{F}}^2 + 5\|\mathbf{V}\|_1, \text{ with } \mathbf{B} = \left(\begin{smallmatrix} 1 & 0 \\ 1 & 2 \end{smallmatrix}\right). \tag{28}$$

Figure 2 shows the geometric visualizations of Problems (27) and (28) using the relation $\min_\theta \min(F(\mathbf{V}_\theta^{\mathrm{rot}}), F(\mathbf{V}_\theta^{\mathrm{ref}})) = \min_{\mathbf{V}\in\mathrm{St}(2,2)} F(\mathbf{V})$. The two objective functions exhibit periodicity with a period of $2\pi$. Within the interval $[0, 2\pi)$, each of them contains one unique $\mathrm{BS}_2$-*point*, while the two respective examples contain 4 and 8 critical points. Therefore, the optimality condition of $\mathrm{BS}_2$-points might be much stronger than that of critical points.

**$\mathrm{BS}_2$-points vs. Critical Point: Algorithm Instance Study**. We briefly analyze methods that find critical points of Problem (27), and demonstrate how they may lead to suboptimal results for Problem (27). We illustrate this with the notable feasible method based on the Cayley transformation (Wen & Yin, 2013). According to Equation (7) from (Wen & Yin, 2013), the update rule is defined as: $\mathbf{X}^{t+1} \Leftarrow \mathbf{Q}\mathbf{X}^t$, where $\mathbf{Q} \triangleq [(\mathbf{I}_2 + \frac{\tau}{2}\mathbf{A})^{-1}(\mathbf{I}_2 - \frac{\tau}{2}\mathbf{A})]$. Here, $\tau \in \mathbb{R}$, and $\mathbf{Q} \in \mathbb{R}^{2\times 2}$ is a suitable skew-symmetric matrix. Lemma A.3 shows that the matrix $\mathbf{Q}$ consistently functions as a rotation matrix. Consequently, if $\mathbf{X}^0$ is initialized as a rotation matrix, the resulting solution $\mathbf{X}^{t+1}$ will remain confined to this rotation matrix for all $t$.

### C.2 COMPUTING THE MATRIX $\mathbf{Q}$

Computing the matrix $\mathbf{Q} \in \mathbb{R}^{k^2 \times k^2}$ as in (8) can be a challenging task because it involves the matrix $\mathbf{H} \in \mathbb{R}^{nr \times nr}$. However, in practice, $\mathbf{H}$ often has some special structure that enables fast matrix computation. For example, $\mathbf{H}$ might take a diagonal matrix that is equal to $L\mathbf{I}_{nr}$ for some $L \geq 0$ or has a Kronecker structure where $\mathbf{H} = \mathbf{H}_1 \otimes \mathbf{H}_2$ for some $\mathbf{H}_1 \in \mathbb{R}^{r \times r}$ and $\mathbf{H}_2 \in \mathbb{R}^{n \times n}$. The lemmas provided below demonstrate how to compute the matrix $\mathbf{Q}$.

**Lemma C.1.** *Assume (8) is used to find* $\mathbf{Q}$. *(a) If* $\mathbf{H} = \mathbf{H}_1 \otimes \mathbf{H}_2$, *we have:* $\mathbf{Q} = \mathbf{Q}_1 \otimes \mathbf{Q}_2$, *where* $\mathbf{Q}_1 = \mathbf{Z}\mathbf{H}_1\mathbf{Z}^\mathsf{T} \in \mathbb{R}^{k \times k}$ *and* $\mathbf{Q}_2 = \mathrm{U}_{\mathrm{B}}^\mathsf{T}\mathbf{H}_2\mathrm{U}_{\mathrm{B}} \in \mathbb{R}^{k \times k}$. *(b) If* $\mathbf{H} = L\mathbf{I}_{nr}$, *we have* $\mathbf{Q} = (L\mathbf{Z}\mathbf{Z}^\mathsf{T}) \otimes \mathbf{I}_k$.

*Proof.* Recall that for any matrices $\bar{\mathbf{A}}, \bar{\mathbf{B}}, \bar{\mathbf{C}}, \bar{\mathbf{D}}$ of suitable dimensions, we have the following equality: $(\bar{\mathbf{A}} \otimes \bar{\mathbf{B}})(\bar{\mathbf{C}} \otimes \bar{\mathbf{D}}) = (\bar{\mathbf{A}}\bar{\mathbf{C}}) \otimes (\bar{\mathbf{B}}\bar{\mathbf{D}})$.

**(a)** If $\mathbf{H} = \mathbf{H}_1 \otimes \mathbf{H}_2$, we derive: $\mathbf{Q} \triangleq (\mathbf{Z}^\mathsf{T} \otimes \mathrm{U}_\mathtt{B})^\mathsf{T} \mathbf{H} (\mathbf{Z}^\mathsf{T} \otimes \mathrm{U}_\mathtt{B}) = (\mathbf{Z}^\mathsf{T} \otimes \mathrm{U}_\mathtt{B})^\mathsf{T} (\mathbf{H}_1 \otimes \mathbf{H}_2)(\mathbf{Z}^\mathsf{T} \otimes \mathrm{U}_\mathtt{B}) = (\mathbf{Z}^\mathsf{T} \otimes \mathrm{U}_\mathtt{B})^\mathsf{T} [(\mathbf{H}_1 \mathbf{Z}^\mathsf{T}) \otimes (\mathbf{H}_2 \mathrm{U}_\mathtt{B})] = (\mathbf{Z}\mathbf{H}_1\mathbf{Z}^\mathsf{T}) \otimes (\mathrm{U}_\mathtt{B}^\mathsf{T} \mathbf{H}_2 \mathrm{U}_\mathtt{B}) = \mathbf{Q}_1 \otimes \mathbf{Q}_2$.

**(b)** If $\mathbf{H} = L\mathbf{I}_{nr}$, we have: $\mathbf{Q} \triangleq (\mathbf{Z}^\mathsf{T} \otimes \mathrm{U}_\mathtt{B})^\mathsf{T} \mathbf{H} (\mathbf{Z}^\mathsf{T} \otimes \mathrm{U}_\mathtt{B}) = L(\mathbf{Z}^\mathsf{T} \otimes \mathrm{U}_\mathtt{B})^\mathsf{T} (\mathbf{Z}^\mathsf{T} \otimes \mathrm{U}_\mathtt{B}) = L(\mathbf{Z}\mathbf{Z}^\mathsf{T}) \otimes \mathbf{I}_k$.

$\square$

**Lemma C.2.** *Assume (9) is used to find* $\mathbf{Q}$. *(a) If* $\mathbf{H} = \mathbf{H}_1 \otimes \mathbf{H}_2$, *we have* $\mathbf{Q} = \|\mathbf{Q}_1\|_{\mathsf{sp}} \cdot \|\mathbf{Q}_2\|_{\mathsf{sp}} \cdot \mathbf{I}$, *where* $\mathbf{Q}_1$ *and* $\mathbf{Q}_2$ *are defined in Lemma C.1. (b) If* $\mathbf{H} = L\mathbf{I}_{nr}$, *we have* $\mathbf{Q} = L\|\mathbf{Z}\|_{\mathsf{sp}}^2 \cdot \mathbf{I}$.

*Proof.* **(a)** Using the results in Claim **(a)** of Lemma C.1, we have: $(\mathbf{Z}^\mathsf{T} \otimes \mathrm{U}_\mathtt{B})^\mathsf{T} \mathbf{H} (\mathbf{Z}^\mathsf{T} \otimes \mathrm{U}_\mathtt{B}) = \mathbf{Q}_1 \otimes \mathbf{Q}_2 \preceq \|\mathbf{Q}_1\|_{\mathsf{sp}} \cdot \|\mathbf{Q}_2\|_{\mathsf{sp}} \cdot \mathbf{I}$.

**(b)** Using the results in Claim **(b)** of Lemma C.1, we have: $(\mathbf{Z}^\mathsf{T} \otimes \mathrm{U}_\mathtt{B})^\mathsf{T} \mathbf{H} (\mathbf{Z}^\mathsf{T} \otimes \mathrm{U}_\mathtt{B}) = L\mathbf{Z}\mathbf{Z}^\mathsf{T} \otimes \mathbf{I}_k \preceq L\|\mathbf{Z}\|_{\mathsf{sp}}^2 \cdot \mathbf{I}$.

$\square$

## C.3 A Computational Complexity Comparison with Full Gradient Methods

We present a computational complexity comparison with full gradient methods using the linear eigenvalue problem: $\min_{\mathbf{X}} F(\mathbf{X}) \triangleq \frac{1}{2}\langle \mathbf{X}, \mathbf{C}\mathbf{X} \rangle$, *s.t.* $\mathbf{X}^\mathsf{T}\mathbf{X} = \mathbf{I}_r$, where $\mathbf{C} \in \mathbb{R}^{n \times n}$ is given.

We first examine full gradient methods such as the Riemannian gradient method (Jiang & Dai, 2015; Liu et al., 2016). The computation of the Riemannian gradient $\nabla_\mathcal{M} F(\mathbf{X}) = \mathbf{C}\mathbf{X} - \mathbf{X}[\mathbf{C}\mathbf{X}]^\mathsf{T}\mathbf{X}$ requires $\mathcal{O}(n^2 r)$ time, while the retraction step using SVD, QR, or polar decomposition demands $\mathcal{O}(nr^2)$. Consequently, the overall complexity for Riemannian gradient method is $N_1 \times \mathcal{O}(n^2 r)$, where $N_1$ is the number of iterations required for convergence.

We now consider the proposed **OBCD** method where the matrix $\mathbf{Q}$ is chosen to be a diagonal matrix as in Equality (9). (*i*) We adopt an incremental update strategy for computing the Euclidean gradient $\nabla F(\mathbf{X}) = \mathbf{C}\mathbf{X}$, maintaining the relationship $\mathbf{Y}^t = \mathbf{C}\mathbf{X}^t$ for all $t$. The initialization $\mathbf{Y}^0 = \mathbf{C}\mathbf{X}^0$ occurs only once. When $\mathbf{X}^t$ is updated via a $k$-row change, resulting in $\mathbf{X}^{t+1} = \mathbf{X}^t + \mathrm{U}_\mathtt{B}(\mathbf{V} - \mathbf{I})\mathrm{U}_\mathtt{B}^\mathsf{T}\mathbf{X}^t$, we efficiently reconstruct $\mathbf{C}\mathbf{X}^{t+1}$ by updating $\mathbf{Y}^{t+1} = \mathbf{Y}^t + \mathbf{C}\mathrm{U}_\mathtt{B}(\mathbf{V} - \mathbf{I})\mathrm{U}_\mathtt{B}^\mathsf{T}\mathbf{X}^t$ in $\mathcal{O}(nr)$ time. (*ii*) Computing the matrix $\mathbf{P}$ as shown in (3) involves matrix multiplication between matrices $[\nabla f(\mathbf{X}^t)]_{\mathtt{B}:} \in \mathbb{R}^{k \times r}$ and $[[\mathbf{X}^t]_{\mathtt{B}:}]^\mathsf{T} \in \mathbb{R}^{r \times k}$, which can be done in $\mathcal{O}(rk^2)$. (*iii*) Solving the subproblem using small-size SVD takes $\mathcal{O}(k^3)$ time. Thus, the total complexity for **OBCD** is $N_2 \times \mathcal{O}(nr + rk^2 + k^3)$, with $N_2$ denoting the number of **OBCD** iterations.

## C.4 Generalization to Multiple Row Updates

The proposed **OBCD** algorithm can be generalized to multiple row updates scheme.

Assume that $n$ is an even number, and $k = 2$. As mentioned in Lemma 2.3, when (9) is used to find $\mathbf{Q}$, the subproblem $\bar{\mathbf{V}}^t \in \arg\min_{\mathbf{V} \in \mathrm{St}(k,k)} \mathcal{K}(\mathbf{V}; \mathbf{X}^t, \mathtt{B})$ in Algorithm 1 reduces to:

$$\min_{\mathbf{V} \in \mathrm{St}(2,2)} \langle \mathbf{V}, (\nabla f(\mathbf{X}^t)[\mathbf{X}^t]^\mathsf{T})_{\mathtt{BB}} \rangle + h(\mathbf{V}\mathrm{U}_\mathtt{B}\mathbf{X}^t). \tag{29}$$

One can independently solve $(n/2)$ subproblems, each formulated as follows:

$\min_{\mathbf{V} \in \mathrm{St}(2,2)} \langle \mathbf{V}, (\nabla f(\mathbf{X}^t)[\mathbf{X}^t]^\mathsf{T})_{\mathtt{BB}} \rangle + h(\mathbf{V}\mathrm{U}_\mathtt{B}\mathbf{X}^t)$ with $\mathtt{B} = [1, 2]$.

$\min_{\mathbf{V} \in \mathrm{St}(2,2)} \langle \mathbf{V}, (\nabla f(\mathbf{X}^t)[\mathbf{X}^t]^\mathsf{T})_{\mathtt{BB}} \rangle + h(\mathbf{V}\mathrm{U}_\mathtt{B}\mathbf{X}^t)$ with $\mathtt{B} = [3, 4]$.

$\cdots$

$\min_{\mathbf{V} \in \mathrm{St}(2,2)} \langle \mathbf{V}, (\nabla f(\mathbf{X}^t)[\mathbf{X}^t]^\mathsf{T})_{\mathtt{BB}} \rangle + h(\mathbf{V}\mathrm{U}_\mathtt{B}\mathbf{X}^t)$ with $\mathtt{B} = [n - 1, n]$.

This approach, known as the Jacobi update in the literature, allows for the parallel update of $n$ rows of the matrix $\mathbf{X}$.

Notably, one can consider $k \triangleq |\mathtt{B}| > 2$ when $h(\cdot) = 0$, and the associated subproblems can be solved using SVD.

## C.5   Limiting Subdifferential of the Cardinality Function

We demonstrate how to calculate the limiting subdifferential of the cardinality function $h(\mathbf{X}) = \|\mathbf{X}\|_0$. Given that $h(\mathbf{X}) = \|\mathbf{X}\|_0$ is coordinate-wise separable, we focus only on the scalar function $h(x) = |x|_0$, where $|x|_0 = \left\{ \begin{smallmatrix} 0; & x = 0; \\ 1, & \text{else.} \end{smallmatrix} \right\}$.

The Fréchet subdifferential of the function $h(x) = |x|_0$ at $x \in \text{dom}(h)$ is defined as $\hat{\partial} h(x) \triangleq \{\xi \in \mathbb{R} : \lim_{z \to x} \inf_{z \neq x} \frac{h(z) - h(x) - \langle \xi, z - x \rangle}{|z - x|} \geq 0\}$, while the limiting subdifferential of $h(x)$ at $x \in \text{dom}(h)$ is denoted as $\partial h(x) \triangleq \{\xi \in \mathbb{R} : \exists x^t \to x, h(x^t) \to h(x), \xi^t \in \hat{\partial} h(x^t) \to \xi, \forall t\}$. We consider the following two cases. (***i***) $x \neq 0$. We have: $\hat{\partial} h(x) = \{\xi \in \mathbb{R} : \lim_{z \to x} \inf_{z \neq x} \frac{-\langle \xi, z - x \rangle}{|z - x|} \geq 0\} = \{0\}$. (***ii***) $x = 0$. We have: $\hat{\partial} h(x) = \{\xi \in \mathbb{R} : \lim_{z \to x} \inf_{z \neq x} \frac{|z|_0 - \langle \xi, z - x \rangle}{|z - x|} \geq 0\} = \{\xi \in \mathbb{R} : \lim_{z \to x} \inf_{z \neq x} \frac{1 - \langle \xi, z \rangle}{|z|} \geq 0\} = \mathbb{R}$.

We therefore conclude that $[\partial \|\mathbf{X}\|_0]_{i,j} \in \left\{ \begin{smallmatrix} \mathbb{R}, \\ \{0\}, \end{smallmatrix} \begin{smallmatrix} \mathbf{X}_{i,j} = 0; \\ \text{else.} \end{smallmatrix} \right\}$ for all $i \in [n]$ and $j \in [r]$.

## D   Greedy Strategies for Working Set Selection

In this section, we introduce two novel greedy strategies designed to identify an effective working set to enhance the practical computational efficiency of **ODBC** for $k = 2$, as shown in Algorithm 2. These methods exclusively utilize the current solution $\mathbf{X}^t$ and its associated subgradient $\mathbf{G}^t \in \partial F(\mathbf{X}^t)$. Notably, our subsequent discussion relies on an additional variable matrix denoted as the scoring matrix $\mathbf{S}$.

Our first Working Set Selection (**WSS**) strategy is based on the maximum Stationarity Violation pair, denoted as **WWS-SV**. It selects the index $\mathtt{B} = [\bar{i}, \bar{j}]$ that most violates the first-order optimality condition.

Our second working set selection strategy is rooted in the maximum Objective Reduction pair, denoted as **WWS-OR**. It chooses the index $\mathtt{B} = [\bar{i}, \bar{j}]$ that leads to the maximum objective reduction under certain criteria.

We have the following results for the theoretical properties of **WWS-SV** and **WWS-OR**.

**Lemma D.1.** *(Proof in Appendix I.1,* **Properties of WSS-SV**). Assume that the scoring matrix $\mathbf{S}$ is computed using (30), we have: (***a***) $\mathbf{X}^t \in \text{St}(n, r)$ is a critical point $\Leftrightarrow \mathbf{S} = \mathbf{0}$. (***b***) $\mathbf{S} = \mathbf{0}$ $\Leftrightarrow \mathbf{S}(\bar{i}, \bar{j}) = 0$.

---

**Algorithm 2:  WSS: Working Set Selection via Greedy Strategies.**

**Input:** $\mathbf{X}^t$ and $\mathbf{G}^t \in \partial F(\mathbf{X}^t)$.
(**S1**) Compute the scoring matrix $\mathbf{S} \in \mathbb{R}^{n \times n}$ using one of the following two strategies:
● Option **WSS-SV** (using Maximum Stationarity Violation Pair):

$$\mathbf{S} = \mathbf{X}^t [\mathbf{G}^t]^\mathsf{T} - \mathbf{G}^t [\mathbf{X}^t]^\mathsf{T}. \tag{30}$$

● Option **WSS-OR** (using Maximum Objective Reduction Pair):

$$\mathbf{S}_{ij} = \min_{\mathbf{V}^\mathsf{T} \mathbf{V} = \mathbf{I}_2} \langle \mathbf{V} - \mathbf{I}_2, \mathbf{T}_{\mathtt{BB}} \rangle, \mathtt{B} = [i, j], \tag{31}$$

where $\mathbf{T} = (\mathbf{G}^t - L_f \mathbf{X}^t)(\mathbf{X}^t)^\mathsf{T} - \alpha \mathbf{I}_n \in \mathbb{R}^{n \times n}$.
(**S2**) Output: $\mathtt{B} = [\bar{i}, \bar{j}] = \arg \max_{i \in [n], j \in [n], i \neq j} |\mathbf{S}_{ij}|$

---

**Theorem D.2.** *(Proof in Appendix I.2,* **Properties of WSS-OR**). Assume that the scoring matrix $\mathbf{S}$ is computed using (31). Assume $h(\mathbf{X}) = 0$ and Equation (8) is used to choose the matrix $\mathbf{Q}$. We have:

(**a**) The value of $\mathbf{S}_{ij}$ for any given $[i,j]$ can be computed as $\mathbf{S}_{ij} = \min(w_1, w_2)$, where $w_1 \triangleq -c_1 - \sqrt{c_1^2 + c_2^2}$, $w_2 \triangleq -c_1 - \sqrt{c_3^2 + c_4^2}$, $c_1 \triangleq \mathbf{T}_{ii} + \mathbf{T}_{jj}$, $c_2 \triangleq \mathbf{T}_{ij} - \mathbf{T}_{ji}$, $c_3 \triangleq \mathbf{T}_{jj} - \mathbf{T}_{ii}$ and $c_4 \triangleq \mathbf{T}_{ij} + \mathbf{T}_{ji}$.

(**b**) If $\mathbf{X}^t$ is not a critical point, it holds that: $\mathbf{S}(\bar{i}, \bar{j}) < 0$ and $F(\mathbf{X}^{t+1}) < F(\mathbf{X}^t)$.

**Remarks**. (**i**) The computational complexity of both **WSS-MV** and **WSS-OR** for a given pair $[i,j]$ is $\mathcal{O}(r)$. Therefore, the overall computational complexity for all $\mathrm{C}_n^2$ pairs is $\mathcal{O}(n^2 r)$. Such computational complexity could be high when $n$ is large. We consider the following more practical approach for $k = 2$ in our experiments. We randomly and uniformly sample $p \triangleq \min(n, 200)$ elements from the set $\{\mathcal{B}_i\}_{i=1}^{\mathrm{C}_n^2}$ as $\{\bar{\mathcal{B}}_i\}_{i=1}^{p}$, and then we pick the working set using $\mathbb{B} = [\bar{i}, \bar{j}] = \arg\max_{i,j,i \neq j} |\mathbf{S}_{ij}|$, $s.t.\ [i,j] \in \{\bar{\mathcal{B}}_i\}_{i=1}^{p}$. This strategy leads to a significant reduction in computational complexity to $\mathcal{O}(pr)$ when $p \ll \mathrm{C}_n^2$. (**ii**) When choosing $k$ coordinates with $k > 2$, one can simply pick the top-$k$ nonoverlapping coordinates according $|\mathbf{S}|$ as the working set.

# E   PROOF FOR SECTION 2

## E.1   PROOF FOR LEMMA 2.1

*Proof.* **Part (a)**. For any $\mathbf{V} \in \mathbb{R}^{k \times k}$ and $\mathbb{B} \in \{\mathcal{B}_i\}_{i=1}^{\mathrm{C}_n^k}$, we have:

$$
\begin{aligned}
& [\mathbf{X}^+]^\mathsf{T} \mathbf{X}^+ - \mathbf{X}^\mathsf{T} \mathbf{X} \\
\overset{\text{①}}{=}\ & [\mathbf{X} + \mathrm{U}_\mathbb{B}(\mathbf{V} - \mathbf{I}_k)\mathrm{U}_\mathbb{B}^\mathsf{T}\mathbf{X}]^\mathsf{T}[\mathbf{X} + \mathrm{U}_\mathbb{B}(\mathbf{V} - \mathbf{I}_k)\mathrm{U}_\mathbb{B}^\mathsf{T}\mathbf{X}] - \mathbf{X}^\mathsf{T}\mathbf{X} \\
=\ & \mathbf{X}^\mathsf{T}\mathrm{U}_\mathbb{B}(\mathbf{V} - \mathbf{I}_k)\mathrm{U}_\mathbb{B}^\mathsf{T}\mathbf{X} + [\mathrm{U}_\mathbb{B}(\mathbf{V} - \mathbf{I}_k)\mathrm{U}_\mathbb{B}^\mathsf{T}\mathbf{X}]^\mathsf{T}\mathbf{X} + [\mathrm{U}_\mathbb{B}(\mathbf{V} - \mathbf{I}_k)\mathrm{U}_\mathbb{B}^\mathsf{T}\mathbf{X}]^\mathsf{T}[\mathrm{U}_\mathbb{B}(\mathbf{V} - \mathbf{I}_k)\mathrm{U}_\mathbb{B}^\mathsf{T}\mathbf{X}] \\
=\ & \mathbf{X}^\mathsf{T}\mathrm{U}_\mathbb{B}\left[(\mathbf{V} - \mathbf{I}_k + \mathbf{V}^\mathsf{T} - \mathbf{I}_k) + (\mathbf{V} - \mathbf{I}_k)^\mathsf{T}\mathrm{U}_\mathbb{B}^\mathsf{T}\mathrm{U}_\mathbb{B}(\mathbf{V} - \mathbf{I}_k)\right]\mathrm{U}_\mathbb{B}^\mathsf{T}\mathbf{X} \\
\overset{\text{②}}{=}\ & \mathbf{X}^\mathsf{T}\mathrm{U}_\mathbb{B}\left[(\mathbf{V} - \mathbf{I}_k + \mathbf{V}^\mathsf{T} - \mathbf{I}_k) + (\mathbf{V} - \mathbf{I}_k)^\mathsf{T}(\mathbf{V} - \mathbf{I}_k)\right]\mathrm{U}_\mathbb{B}^\mathsf{T}\mathbf{X} \\
=\ & \mathbf{X}^\mathsf{T}\mathrm{U}_\mathbb{B}(\mathbf{V} - \mathbf{I}_k + \mathbf{V}^\mathsf{T} - \mathbf{I}_k + \mathbf{V}^\mathsf{T}\mathbf{V} - \mathbf{V}^\mathsf{T} - \mathbf{V} + \mathbf{I}_k)\mathrm{U}_\mathbb{B}^\mathsf{T}\mathbf{X} \\
=\ & \mathbf{X}^\mathsf{T}\mathrm{U}_\mathbb{B}(-\mathbf{I}_k + \mathbf{V}^\mathsf{T}\mathbf{V})\mathrm{U}_\mathbb{B}^\mathsf{T}\mathbf{X} \\
\overset{\text{③}}{=}\ & \mathbf{X}^\mathsf{T}\mathrm{U}_\mathbb{B} \cdot \mathbf{0} \cdot \mathrm{U}_\mathbb{B}^\mathsf{T}\mathbf{X} \\
=\ & \mathbf{0},
\end{aligned}
$$

where step ① uses $\mathbf{X}^+ = \mathbf{X} + \mathrm{U}_\mathbb{B}(\mathbf{V} - \mathbf{I}_k)\mathrm{U}_\mathbb{B}^\mathsf{T}\mathbf{X}$; step ② uses $\mathrm{U}_\mathbb{B}^\mathsf{T}\mathrm{U}_\mathbb{B} = \mathbf{I}_k$; step ③ uses $\mathbf{V}^\mathsf{T}\mathbf{V} = \mathbf{I}_k$.

**Part (b)**. Obvious.

$\square$

## E.2   PROOF OF LEMMA 2.2

*Proof.* We define $\mathbf{X}^+ \triangleq \mathbf{X} + \mathrm{U}_\mathbb{B}(\mathbf{V} - \mathbf{I}_k)\mathrm{U}_\mathbb{B}^\mathsf{T}\mathbf{X}$, $\underline{\mathbf{Q}} \triangleq (\mathbf{Z}^\mathsf{T} \otimes \mathrm{U}_\mathbb{B})^\mathsf{T}\mathbf{H}(\mathbf{Z}^\mathsf{T} \otimes \mathrm{U}_\mathbb{B})$, and $\mathbf{Z} \triangleq \mathrm{U}_\mathbb{B}^\mathsf{T}\mathbf{X}$.

**Part (a)**. We derive the following results:

$$
\begin{aligned}
\|\mathbf{X}^+ - \mathbf{X}\|_\mathbf{H}^2 \ \overset{\text{①}}{=}\ & \|\mathrm{U}_\mathbb{B}(\mathbf{V} - \mathbf{I}_k)\mathbf{Z}\|_\mathbf{H}^2 \\
\overset{\text{②}}{=}\ & \mathrm{vec}(\mathrm{U}_\mathbb{B}(\mathbf{V} - \mathbf{I}_k)\mathbf{Z})^\mathsf{T}\mathbf{H}\mathrm{vec}(\mathrm{U}_\mathbb{B}(\mathbf{V} - \mathbf{I}_k)\mathbf{Z}) \\
\overset{\text{③}}{=}\ & \mathrm{vec}(\mathbf{V} - \mathbf{I}_k)^\mathsf{T}(\mathbf{Z}^\mathsf{T} \otimes \mathrm{U}_\mathbb{B})^\mathsf{T}\mathbf{H}(\mathbf{Z}^\mathsf{T} \otimes \mathrm{U}_\mathbb{B})\mathrm{vec}(\mathbf{V} - \mathbf{I}_k) \\
\overset{\text{④}}{=}\ & \|\mathbf{V} - \mathbf{I}_k\|_{(\mathbf{Z}^\mathsf{T} \otimes \mathrm{U}_\mathbb{B})^\mathsf{T}\mathbf{H}(\mathbf{Z}^\mathsf{T} \otimes \mathrm{U}_\mathbb{B})}^2 \\
\overset{\text{⑤}}{=}\ & \|\mathbf{V} - \mathbf{I}_k\|_{\underline{\mathbf{Q}}}^2,
\end{aligned}
$$

where step ① uses $\mathbf{X}^+ \triangleq \mathbf{X} + \mathrm{U}_\mathbb{B}(\mathbf{V} - \mathbf{I}_k)\mathbf{Z}$; step ② uses $\|\mathbf{X}\|_\mathbf{H}^2 = \mathrm{vec}(\mathbf{X})^\mathsf{T}\mathbf{H}\mathrm{vec}(\mathbf{X})$; step ③ uses $(\mathbf{Z}^\mathsf{T} \otimes \mathbf{R})\mathrm{vec}(\mathbf{U}) = \mathrm{vec}(\mathbf{R}\mathbf{U}\mathbf{Z})$ for all $\mathbf{R}$, $\mathbf{Z}$, and $\mathbf{U}$ of suitable dimensions; step ④ uses $\|\mathbf{X}\|_\mathbf{H}^2 = \mathrm{vec}(\mathbf{X})^\mathsf{T}\mathbf{H}\mathrm{vec}(\mathbf{X})$ again; step ⑤ uses the definition of $\underline{\mathbf{Q}}$.

**Part (b)**. We derive the following equalities:

$$
\begin{aligned}
\|\mathbf{X}^+ - \mathbf{X}\|_{\mathsf{F}}^2 \;\overset{①}{=}\;& \|\mathrm{U}_{\mathrm{B}}(\mathbf{V} - \mathbf{I}_k)\mathbf{Z}\|_{\mathsf{F}}^2 \\
\overset{②}{=}\;& \|(\mathbf{V} - \mathbf{I}_k)\mathbf{Z}\|_{\mathsf{F}}^2 \\
=\;& \langle (\mathbf{V} - \mathbf{I}_k)^{\mathsf{T}}(\mathbf{V} - \mathbf{I}_k), \mathbf{Z}\mathbf{Z}^{\mathsf{T}} \rangle \\
\overset{③}{=}\;& 2\langle \mathbf{I}_k - \mathbf{V}, \mathbf{Z}\mathbf{Z}^{\mathsf{T}} \rangle + \langle \mathbf{V} - \mathbf{V}^{\mathsf{T}}, \mathbf{Z}\mathbf{Z}^{\mathsf{T}} \rangle. \\
\overset{④}{=}\;& 2\langle \mathbf{I}_k - \mathbf{V}, \mathbf{Z}\mathbf{Z}^{\mathsf{T}} \rangle + 0.
\end{aligned}
$$

where step ① uses $\mathbf{X}^+ \triangleq \mathbf{X} + \mathrm{U}_{\mathrm{B}}(\mathbf{V} - \mathbf{I}_k)\mathbf{Z}$; step ② uses the fact that $\|\mathrm{U}_{\mathrm{B}}\mathbf{V}\|_{\mathsf{F}}^2 = \|\mathbf{V}\|_{\mathsf{F}}^2$ for any $\mathbf{V} \in \mathbb{R}^{k \times k}$; step ③ uses

$$
(\mathbf{V} - \mathbf{I}_k)^{\mathsf{T}}(\mathbf{V} - \mathbf{I}_k) = \mathbf{I}_k - \mathbf{V}^{\mathsf{T}} - \mathbf{V} + \mathbf{I}_k = 2(\mathbf{I}_k - \mathbf{V}) + (\mathbf{V} - \mathbf{V}^{\mathsf{T}});
$$

step ④ uses the fact that $\langle \mathbf{V}, \mathbf{Z}\mathbf{Z}^{\mathsf{T}} \rangle = \langle \mathbf{V}^{\mathsf{T}}, (\mathbf{Z}\mathbf{Z}^{\mathsf{T}})^{\mathsf{T}} \rangle = \langle \mathbf{V}^{\mathsf{T}}, \mathbf{Z}\mathbf{Z}^{\mathsf{T}} \rangle$ which holds true as the matrix $\mathbf{Z}\mathbf{Z}^{\mathsf{T}}$ is symmetric.

**Part (c)**. We have:

$$
\begin{aligned}
\|\mathbf{X}^+ - \mathbf{X}\|_{\mathsf{F}}^2 \;=\;& \|\mathrm{U}_{\mathrm{B}}(\mathbf{V} - \mathbf{I}_k)\mathrm{U}_{\mathrm{B}}^{\mathsf{T}}\mathbf{X}\|_{\mathsf{F}}^2 \\
\overset{①}{\leq}\;& \|\mathrm{U}_{\mathrm{B}}\|_{\mathsf{sp}}^2 \cdot \|(\mathbf{V} - \mathbf{I}_k)\mathrm{U}_{\mathrm{B}}^{\mathsf{T}}\mathbf{X}\|_{\mathsf{F}}^2 \\
\overset{②}{\leq}\;& \|\mathrm{U}_{\mathrm{B}}\|_{\mathsf{sp}}^2 \cdot \|\mathbf{V} - \mathbf{I}_k\|_{\mathsf{F}}^2 \cdot \|\mathrm{U}_{\mathrm{B}}^{\mathsf{T}}\|_{\mathsf{sp}}^2 \cdot \|\mathbf{X}\|_{\mathsf{sp}}^2 \\
\overset{③}{=}\;& \|\mathbf{V} - \mathbf{I}_k\|_{\mathsf{F}}^2 \\
\overset{④}{=}\;& 2\langle \mathbf{I}_k - \mathbf{V}, \mathbf{I}_k \rangle,
\end{aligned}
$$

where step ① and step ② uses the norm inequality that $\|\mathbf{A}\mathbf{X}\|_{\mathsf{F}} \leq \|\mathbf{A}\|_{\mathsf{F}} \cdot \|\mathbf{X}\|_{\mathsf{sp}}$ for any $\mathbf{A}$ and $\mathbf{X}$; step ③ uses $\|\mathrm{U}_{\mathrm{B}}\|_{\mathsf{sp}} = \|\mathrm{U}_{\mathrm{B}}^{\mathsf{T}}\|_{\mathsf{sp}} = \|\mathbf{X}\|_{\mathsf{sp}} = 1$ for any $\mathbf{X} \in \mathrm{St}(n, r)$; step ④ uses the following equalities for any $\mathbf{V} \in \mathrm{St}(k, k)$:

$$
\|\mathbf{V} - \mathbf{I}_k\|_{\mathsf{F}}^2 = \|\mathbf{V}\|_{\mathsf{F}}^2 + \|\mathbf{I}_k\|_{\mathsf{F}}^2 - 2\langle \mathbf{I}_k, \mathbf{V} \rangle = \|\mathbf{I}_k\|_{\mathsf{F}}^2 + \|\mathbf{I}_k\|_{\mathsf{F}}^2 - 2\langle \mathbf{I}_k, \mathbf{V} \rangle = 2\langle \mathbf{I}_k, \mathbf{I}_k - \mathbf{V} \rangle.
$$

$\square$

### E.3 Proof of Lemma 2.3

*Proof.* We define $\mathcal{K}(\mathbf{V}; \mathbf{X}^t, \mathrm{B}) \triangleq \frac{1}{2}\|\mathbf{V} - \mathbf{I}_k\|_{\mathbf{Q}+\alpha\mathbf{I}}^2 + h(\mathbf{V}\mathbf{Z}) + \langle \mathbf{V}, [\nabla f(\mathbf{X}^t)(\mathbf{X}^t)^{\mathsf{T}}]_{\mathrm{BB}} \rangle + \ddot{c}$, where $\mathbf{Z} \triangleq \mathrm{U}_{\mathrm{B}}^{\mathsf{T}}\mathbf{X}^t$ and $\ddot{c} = h(\mathrm{U}_{\mathrm{B}^c}^{\mathsf{T}}\mathbf{X}^t) + f(\mathbf{X}^t) - \langle \mathbf{I}_k, [\nabla f(\mathbf{X}^t)(\mathbf{X}^t)^{\mathsf{T}}]_{\mathrm{BB}} \rangle$ is a constant.

**Part (a)**. Using the definition of $\mathcal{K}(\mathbf{V}; \mathbf{X}^t, \mathrm{B})$, we have the following equalities for all $\mathbf{V} \in \mathrm{St}(k, k)$:

$$
\begin{aligned}
& \mathcal{K}(\mathbf{V}; \mathbf{X}^t, \mathrm{B}) - \ddot{c} \\
\triangleq\;& \tfrac{1}{2}\|\mathbf{V} - \mathbf{I}\|_{\mathbf{Q}+\alpha\mathbf{I}}^2 + \langle \mathbf{V}, [\nabla f(\mathbf{X}^t)(\mathbf{X}^t)^{\mathsf{T}}]_{\mathrm{BB}} \rangle + h(\mathbf{V}\mathbf{Z}) \\
=\;& \tfrac{1}{2}\|\mathbf{V} - \mathbf{I}\|_{\mathbf{Q}}^2 + \tfrac{\alpha}{2}\|\mathbf{V} - \mathbf{I}\|_{\mathsf{F}}^2 + \langle \mathbf{V}, [\nabla f(\mathbf{X}^t)(\mathbf{X}^t)^{\mathsf{T}}]_{\mathrm{BB}} \rangle + h(\mathbf{V}\mathbf{Z}) \\
\overset{①}{=}\;& \tfrac{1}{2}\|\mathbf{V}\|_{\mathbf{Q}}^2 - \langle \mathbf{V}, \mathrm{mat}(\mathbf{Q}\mathrm{vec}(\mathbf{I}_k)) \rangle + \tfrac{1}{2}\|\mathbf{I}_k\|_{\mathbf{Q}}^2 + \alpha\langle \mathbf{I}, \mathbf{I} - \mathbf{V} \rangle + \langle \mathbf{V}, [\nabla f(\mathbf{X}^t)(\mathbf{X}^t)^{\mathsf{T}}]_{\mathrm{BB}} \rangle + h(\mathbf{V}\mathbf{Z}) \\
\overset{②}{=}\;& \tfrac{1}{2}\|\mathbf{V}\|_{\mathbf{Q}}^2 + \langle \mathbf{V}, \underbrace{[\nabla f(\mathbf{X}^t)(\mathbf{X}^t)^{\mathsf{T}}]_{\mathrm{BB}} - \mathrm{mat}(\mathbf{Q}\mathrm{vec}(\mathbf{I}_k)) - \alpha\mathbf{I}_k}_{\triangleq \mathbf{P}} \rangle + h(\mathbf{V}\mathbf{Z}) + \tfrac{1}{2}\|\mathbf{I}_k\|_{\mathbf{Q}}^2,
\end{aligned}
$$

where step ① uses Claim (*c*) of Lemma 2.2 that: $\frac{\alpha}{2}\|\mathbf{V} - \mathbf{I}_k\|_{\mathsf{F}}^2 = \alpha\langle \mathbf{I}, \mathbf{I} - \mathbf{V} \rangle$; step ② uses the definition of $\mathbf{P}$.

**Part (b)**. We consider the case that $\mathbf{Q}$ is chosen to be a diagonal matrix that $\mathbf{Q} = \varsigma\mathbf{I}$, where $\varsigma$ is defined in Equation (9). Using $\mathbf{V} \in \mathrm{St}(k, k)$, the term $\frac{1}{2}\|\mathbf{V}\|_{\mathbf{Q}}^2$ simplifies to a constant with $\frac{1}{2}\|\mathbf{V}\|_{\mathbf{Q}}^2 = \frac{\varsigma}{2}k$. We can deduce from (3):

$$
\bar{\mathbf{V}}^t \in \arg\min_{\mathbf{V} \in \mathrm{St}(k,k)} \mathcal{P}(\mathbf{V}) \triangleq \langle \mathbf{V}, \mathbf{P} \rangle + h(\mathbf{X}). \tag{32}
$$

In particular, when $h(\mathbf{X}) = 0$, Problem (32) becomes the nearest orthogonality matrix problem and can be solved analytically, yielding a closed-form solution that:

$$\bar{\mathbf{V}}^t \in \arg\min_{\mathbf{V}\in\mathrm{St}(k,k)} \tfrac{1}{2}\|\mathbf{V} - (-\mathbf{P})\|_\mathsf{F}^2 = \mathbb{P}_\mathcal{M}(-\mathbf{P}) = -\mathbb{P}_\mathcal{M}(\mathbf{P}) = -\tilde{\mathbf{U}}\tilde{\mathbf{V}}^\mathsf{T}.$$

Here, $\mathbf{P} = \tilde{\mathbf{U}}\mathrm{Diag}(\mathbf{s})\tilde{\mathbf{V}}^\mathsf{T}$ is the singular value decomposition of $\mathbf{P}$ with $\tilde{\mathbf{U}}, \tilde{\mathbf{V}} \in \mathrm{St}(k,k)$, $\mathbf{s} \in \mathbb{R}^k$, and $\mathbf{s} \geq \mathbf{0}$.

Notably, the multiplier for the orthogonality constraint $\mathbf{V}^\mathsf{T}\mathbf{V} = \mathbf{I}_k$ can be computed as: $\mathbf{\Lambda} = -\mathbf{P}^\mathsf{T}\bar{\mathbf{V}}^t \overset{①}{=} -[\tilde{\mathbf{U}}\mathrm{Diag}(\mathbf{s})\tilde{\mathbf{V}}^\mathsf{T}]^\mathsf{T} \cdot [-\tilde{\mathbf{U}}\tilde{\mathbf{V}}^\mathsf{T}] = \tilde{\mathbf{V}}\mathrm{Diag}(\mathbf{s})\tilde{\mathbf{U}}^\mathsf{T}\tilde{\mathbf{U}}\tilde{\mathbf{V}}^\mathsf{T} \overset{②}{=} \tilde{\mathbf{V}}\mathrm{Diag}(\mathbf{s})\tilde{\mathbf{V}}^\mathsf{T} \overset{③}{\succeq} \mathbf{0}$, where step ① uses $\mathbf{P} = \tilde{\mathbf{U}}\mathrm{Diag}(\mathbf{s})\tilde{\mathbf{V}}^\mathsf{T}$ and $\bar{\mathbf{V}}^t = -\tilde{\mathbf{U}}\tilde{\mathbf{V}}^\mathsf{T}$; step ② uses $\tilde{\mathbf{U}}^\mathsf{T}\tilde{\mathbf{U}} = \mathbf{I}_k$; step ③ uses $\mathbf{s} \geq 0$.

$\square$

### E.4 PROOF OF LEMMA 2.5

*Proof.* Any $2 \times 2$ matrix takes the form $\mathbf{V} = \left(\begin{smallmatrix} a & b \\ c & d \end{smallmatrix}\right)$. The orthogonality constraint implies that $\mathbf{V} \in \mathrm{St}(2,2)$ meets the following three equations: $1 = a^2 + b^2$, $1 = c^2 + d^2$, $0 = ac + bd$. Without loss of generality, we let $c = \sin(\theta)$ and $d = \cos(\theta)$ with $\theta \in \mathbb{R}$. Then we obtain either *(i)* $a = \cos(\theta), b = -\sin(\theta)$ or *(ii)* $a = -\cos(\theta), b = \sin(\theta)$. Therefore, we have the following Givens rotation matrix $\mathbf{V}_\theta^{\mathrm{rot}}$ and Jacobi reflection matrix $\mathbf{V}_\theta^{\mathrm{ref}}$:

$$\mathbf{V}_\theta^{\mathrm{rot}} \triangleq \begin{bmatrix} \cos(\theta) & -\sin(\theta) \\ \sin(\theta) & \cos(\theta) \end{bmatrix}, \ \mathbf{V}_\theta^{\mathrm{ref}} \triangleq \begin{bmatrix} -\cos(\theta) & \sin(\theta) \\ \sin(\theta) & \cos(\theta) \end{bmatrix}.$$

Note that for any $a, b, c, d \in \mathbb{R}$, we have: $\det\left(\begin{smallmatrix} a & b \\ c & d \end{smallmatrix}\right) = ad - bc$. Therefore, we obtain: $\det(\mathbf{V}_\theta^{\mathrm{rot}}) = \cos^2(\theta) + \sin^2(\theta) = 1$ and $\det(\mathbf{V}_\theta^{\mathrm{rot}}) = -\cos^2(\theta) - \sin^2(\theta) = -1$ for any $\theta \in \mathbb{R}$.

$\square$

## F PROOF FOR SECTION 3

### F.1 PROOF OF THEOREM 3.1

*Proof.* **Part (a)**. First, recall the classical **Givens-QR** algorithm, which is detailed in Section 5.2.5 of (Golub & Van Loan, 2013)). This algorithm can decompose any matrix $\mathbf{X} \in \mathbb{R}^{n \times n}$ (not necessarily orthogonal) into the form $\mathbf{X} = \mathbf{Q}\mathbf{R}$, where $\mathbf{Q}$ is an orthogonal matrix ($\mathbf{Q} \in \mathrm{St}(n,n)$) and $\mathbf{R}$ is a lower triangular matrix with $\mathbf{R}_{ij} = 0$ for all $i < j$, achieved through $\mathrm{C}_n^2 = \frac{n(n-1)}{2}$ Givens rotation steps.

Combining the result from Lemma A.5, we can conclude that classical **Givens-QR** algorithm can decompose any orthogonal matrix into the form $\mathbf{X} = \mathbf{Q}\mathbf{R}$, where $\mathbf{Q} \in \mathrm{St}(n,n)$ and $\mathbf{R}$ is diagonal matrix with $\mathbf{R}_{i,i} \in \{-1, +1\}$ for all $i \in [n]$.

We introduce a modification to the **Givens-QR** algorithm, resulting in our **Jacobi-Givens-QR** algorithm as presented in Listing 1. This algorithm can decompose any matrix $\mathbf{X} \in \mathrm{St}(n,n)$ into the form $\mathbf{X} = \mathbf{Q}\mathbf{R}$, where $\mathbf{Q} = \mathbf{X}$ and $\mathbf{R} = \mathbf{I}_n$, using a sequence of $\mathrm{C}_n^k$ Givens rotation or Jacobi reflection steps.

```
function [Q,R] = JacobiGivensQR(X)                                      1
n = size(X,1); Q = eye(n); R = X;                                       2
for j=1:n                                                               3
    for i=n:-1:(j+1)                                                    4
        B = [i-1;i]; V = Givens(R(i-1,j),R(i,j));                       5
        R(B,:) = V'*R(B,:); Q(:,B) = Q(:,B)*V;                          6
        if (i==j+1 && R(j,j)<0)                                         7
            V = [-1 0; 0 -1]; % or V = [-1 0; 0 1];                     8
            R(B,:) = V'*R(B,:); Q(:,B) = Q(:,B)*V;                      9
        end                                                            10
    end                                                                11
end                                                                    12
if(R(n,n)<0)                                                           13
    V = [1 0;0 -1]; R(B,:) = V'*R(B,:); Q(:,B) = Q(:,B)*V;             14
end                                                                    15
                                                                       16
function V = Givens(a,b)                                               17
% Find a Givens rotation that V'*[a;b] = [r;0]                         18
if (b==0)                                                              19
    c = 1; s = 0;                                                     20
else                                                                   21
    if (abs(b) > abs(a))                                              22
        tau = -a/b; s = 1/sqrt(1+tau^2); c = s*tau;                   23
    else                                                               24
        tau = -b/a; c = 1/sqrt(1+tau^2); s = c*tau;                   25
    end                                                                26
end                                                                    27
V = [c s;-s c];                                                       28
```

Listing 1: Matlab implementation for our **Jacobi-Givens-QR** algorithm.

Please take note of the following four important points in Listing 1.

**a)** When we remove Lines 7-10 and Lines 13-15 from Listing 1, it essentially reverts to the classical **Givens-QR** algorithm. **Givens-QR** operates by selecting an appropriate Givens rotation matrix $\mathbf{V} = \begin{bmatrix} \cos(\theta) & \sin(\theta) \\ -\sin(\theta) & \cos(\theta) \end{bmatrix}$ with a suitable rotation angle $\theta$ to zero-out the matrix element $\mathbf{R}_{ij}$ systematically from left to right ($j = 1 \rightarrow n$) and bottom to top ($i = n \rightarrow (j+1)$) within every pair of neighboring columns.

**b)** Lines 7-10 and Lines 13-15 can be viewed as correction steps to ensure that the entries $\mathbf{R}_{j,j} = 1$ for all $j = n$.

**c)** Line 7-10 is executed for $(n-2)$ times. In Line 7-10, when **Jacobi-Givens-QR** detects a negative entry $\mathbf{R}_{i-1,i-1}$ with $i = j + 1$, it applies a rotation matrix $\mathbf{V} \triangleq \begin{pmatrix} -1 & 0 \\ 0 & -1 \end{pmatrix}$ to the two rows B $= [i-1, i]$ to ensure that[3] $\mathbf{R}_{i-1,i-1} = 1$.

**d)** Line 13-15 is executed only once when $\det(\mathbf{X}) = -1$. In such cases, we have $\mathbf{R}_{\text{BB}} = \begin{pmatrix} 1 & 0 \\ 0 & -1 \end{pmatrix}$ and $\det(\mathbf{R}_{\text{BB}}) = -1$, where B $= [n-1, n]$ is the two indices for the final rotation or reflection step. To ensure that the resulting $\mathbf{R}_{\text{BB}}$ is an identify matrix, **Jacobi-Givens-QR** employs a reflection matrix $\mathbf{V} = \begin{pmatrix} 1 & 0 \\ 0 & -1 \end{pmatrix}$, leading to $\mathbf{V}^{\mathsf{T}}\mathbf{R}_{\text{BB}} = \mathbf{I}_2$.

Therefore, we establish the conclusion that any orthogonal matrix $\mathbf{X} \in \text{St}(n,n)$ can be expressed as $\mathbf{D} = \mathcal{W}_{C_n^k}...\mathcal{W}_2\mathcal{W}_1$, where $\mathcal{W}_i = \mathbf{U}_{\mathcal{B}_i}\mathcal{V}_i\mathbf{U}_{\mathcal{B}_i}^{\mathsf{T}} + \mathbf{U}_{\mathcal{B}_i^c}\mathbf{U}_{\mathcal{B}_i^c}^{\mathsf{T}}$, and $\mathcal{V}_i \in \text{St}(2,2)$ is a suitable matrix associated with $\mathcal{B}_i$. Furthermore, if $\forall i, \mathcal{V}_i = \mathbf{I}_2$, we have $\forall i, \mathcal{W}_i = \mathbf{I}_n$, leading to $\mathbf{D} = \mathbf{I}_n$. This concludes the proof of the first part of this theorem.

**Part (b)**. For any given $\mathbf{X} \in \text{St}(n,r)$ and $\mathbf{X}^0 \in \text{St}(n,r)$, we let:

$$\bar{\mathbf{D}} = \mathbb{P}_{\text{St}(n,n)}(\mathbf{X}[\mathbf{X}^0]^{\mathsf{T}}), \tag{33}$$

where $\mathbb{P}_{\text{St}(n,n)}(\mathbf{Y})$ denotes the nearest orthogonality matrix to the given matrix $\mathbf{Y}$.

---

[3]Alternatively, one can use the reflection matrix $\mathbf{V} \triangleq \begin{pmatrix} -1 & 0 \\ 0 & 1 \end{pmatrix}$ instead of the rotation matrix $\mathbf{V} \triangleq \begin{pmatrix} -1 & 0 \\ 0 & -1 \end{pmatrix}$ to ensure that $\mathbf{R}_{i-1,i-1} = 1$.

Assume that the matrix $\mathbf{X}[\mathbf{X}^0]^\mathsf{T}$ has the following singular value decomposition:

$$\mathbf{X}(\mathbf{X}^0)^\mathsf{T} = \mathbf{U}\mathrm{Diag}(\mathbf{z})\mathbf{V}^\mathsf{T}, \ \mathbf{z} \in \{0,1\}^n, \ \mathbf{U} \in \mathrm{St}(n,n), \ \mathbf{V} \in \mathrm{St}(n,n).$$

Therefore, we have the following equalities:

$$\mathrm{Diag}(\mathbf{z}) = \mathbf{U}^\mathsf{T}\mathbf{X}[\mathbf{X}^0]^\mathsf{T}\mathbf{V}. \qquad (34)$$

$$\bar{\mathbf{D}} = \mathbf{U}\mathbf{V}^\mathsf{T} \in \mathrm{St}(n,n). \qquad (35)$$

Furthermore, we derive the following results:

$$\mathbf{z} \in \{0,1\}^n$$
$$\Rightarrow \ \mathrm{Diag}(\mathbf{z})^\mathsf{T} = \mathrm{Diag}(\mathbf{z})\mathrm{Diag}(\mathbf{z})^\mathsf{T}$$
$$\Rightarrow \ \mathbf{U}[\mathrm{Diag}(\mathbf{z})^\mathsf{T} - \mathrm{Diag}(\mathbf{z})\mathrm{Diag}(\mathbf{z})^\mathsf{T}]\mathbf{U}^\mathsf{T}\mathbf{X} = \mathbf{0}$$
$$\overset{\text{\textcircled{1}}}{\Rightarrow} \ \mathbf{U}[\mathbf{V}^\mathsf{T}\mathbf{X}^0\mathbf{X}^\mathsf{T}\mathbf{U} - \mathbf{U}^\mathsf{T}\mathbf{X}(\mathbf{X}^0)^\mathsf{T}\mathbf{V}\mathbf{V}^\mathsf{T}\mathbf{X}^0\mathbf{X}^\mathsf{T}\mathbf{U}]\mathbf{U}^\mathsf{T}\mathbf{X} = \mathbf{0}$$
$$\Rightarrow \ \mathbf{U}\mathbf{V}^\mathsf{T}\mathbf{X}^0\mathbf{X}^\mathsf{T}\mathbf{U}\mathbf{U}^\mathsf{T}\mathbf{X} - \mathbf{U}\mathbf{U}^\mathsf{T}\mathbf{X}(\mathbf{X}^0)^\mathsf{T}\mathbf{V}\mathbf{V}^\mathsf{T}\mathbf{X}^0\mathbf{X}^\mathsf{T}\mathbf{U}\mathbf{U}^\mathsf{T}\mathbf{X} = \mathbf{0}$$
$$\overset{\text{\textcircled{2}}}{\Rightarrow} \ \mathbf{U}\mathbf{V}^\mathsf{T}\mathbf{X}^0 - \mathbf{X} = \mathbf{0}$$
$$\overset{\text{\textcircled{3}}}{\Rightarrow} \ \bar{\mathbf{D}} \cdot \mathbf{X}^0 - \mathbf{X} = \mathbf{0},$$

where step \textcircled{1} uses (34); step \textcircled{2} uses $\mathbf{U}\mathbf{U}^\mathsf{T} = \mathbf{I}_n$, $\mathbf{V}\mathbf{V}^\mathsf{T} = \mathbf{I}_n$, $\mathbf{X}^\mathsf{T}\mathbf{X} = \mathbf{I}_r$, and $[\mathbf{X}^0]^\mathsf{T}\mathbf{X}^0 = \mathbf{I}_r$; step \textcircled{3} uses (35). We conclude that, for any given $\mathbf{X} \in \mathrm{St}(n,r)$ and $\mathbf{X}^0 \in \mathrm{St}(n,r)$, we can always find a matrix $\bar{\mathbf{D}} \in \mathrm{St}(n,n)$ such that $\bar{\mathbf{D}}\mathbf{X}^0 = \mathbf{X}$.

Since the matrix $\bar{\mathbf{D}} \in \mathrm{St}(n,n)$ can be represented as $\bar{\mathbf{D}} = \mathcal{W}_{\mathrm{C}_n^k}...\mathcal{W}_2\mathcal{W}_1$, where $\mathcal{W}_i = \mathbf{U}_{\mathcal{B}_i}\mathcal{V}_i\mathbf{U}_{\mathcal{B}_i}^\mathsf{T} + \mathbf{U}_{\mathcal{B}_i^c}\mathbf{U}_{\mathcal{B}_i^c}^\mathsf{T}$ for some suitable $\mathcal{V}_i \in \mathrm{St}(2,2)$ (as established in the first part of this theorem), we can conclude that any matrix $\mathbf{X} \in \mathrm{St}(n,r)$ can be expressed as $\mathbf{X} = \bar{\mathbf{D}}\mathbf{X}^0 = \mathcal{W}_{\mathrm{C}_n^k}...\mathcal{W}_2\mathcal{W}_1\mathbf{X}^0$.

$\square$

## F.2 Proof for Theorem 3.6

*Proof.* We use $\bar{\mathbf{X}}$, $\ddot{\mathbf{X}}$, and $\check{\mathbf{X}}$ to denote the *global optimal point*, $\mathrm{BS}_k$*-point*, and *critical point* of Problem (1), respectively.

Setting the Riemannian subgradient of $\mathcal{K}(\mathbf{V}; \ddot{\mathbf{X}}, \mathrm{B})$ *w.r.t.* $\mathbf{V}$ to zero, we have $\mathbf{0} \in \partial_\mathcal{M}\mathcal{K}(\mathbf{V}; \ddot{\mathbf{X}}, \mathrm{B}) = \ddot{\mathbf{G}}(\mathbf{V}) \ominus \mathbf{V}[\ddot{\mathbf{G}}(\mathbf{V})]^\mathsf{T}\mathbf{V}$, where $\ddot{\mathbf{G}}(\mathbf{V}) = \alpha(\mathbf{V} - \mathbf{I}_k) + \mathbf{U}_\mathrm{B}^\mathsf{T}[\mathrm{mat}(\mathbf{H}\mathrm{vec}(\mathbf{X}^+ - \ddot{\mathbf{X}})) + \nabla f(\ddot{\mathbf{X}}) + \partial h(\mathbf{X}^+)]\ddot{\mathbf{X}}^\mathsf{T}\mathbf{U}_\mathrm{B}$ and $\mathbf{X}^+ = \ddot{\mathbf{X}} + \mathbf{U}_\mathrm{B}(\mathbf{V} - \mathbf{I}_k)\mathbf{U}_\mathrm{B}^\mathsf{T}\ddot{\mathbf{X}}$. Letting $\mathbf{V} = \mathbf{I}_k$, we have the following **necessary but not sufficient** condition for any $\mathrm{BS}_k$*-point*:

$$\forall \mathrm{B} \in \{\mathcal{B}_i\}_{i=1}^{\mathrm{C}_n^k}, \ \mathbf{0} = \mathbf{U}_\mathrm{B}^\mathsf{T}(\mathbf{G}\ddot{\mathbf{X}}^\mathsf{T} - \ddot{\mathbf{X}}\mathbf{G}^\mathsf{T})\mathbf{U}_\mathrm{B}, \ \text{with} \ \mathbf{G} \in \nabla f(\ddot{\mathbf{X}}) + \partial h(\ddot{\mathbf{X}}). \qquad (36)$$

**Part (a)**. We now show that $\{\text{critical points } \check{\mathbf{X}}\} \supseteq \{\mathrm{BS_k}\text{-points } \ddot{\mathbf{X}}\}$ for all $k \geq 2$. We let $\mathbf{G} \in \nabla f(\ddot{\mathbf{X}}) + \partial h(\ddot{\mathbf{X}})$. Using Lemma A.1, we have:

$$\mathbf{0}_{n,n} = \mathbf{G}\ddot{\mathbf{X}}^\mathsf{T} - \ddot{\mathbf{X}}\mathbf{G}^\mathsf{T} \ \Rightarrow \ (\mathbf{0}_{n,n} \cdot \ddot{\mathbf{X}}) = (\mathbf{G}\ddot{\mathbf{X}}^\mathsf{T} - \ddot{\mathbf{X}}\mathbf{G}^\mathsf{T})\ddot{\mathbf{X}}$$
$$\overset{\text{\textcircled{1}}}{\Rightarrow} \ \mathbf{0}_{n,r} = \mathbf{G} - \ddot{\mathbf{X}}\mathbf{G}^\mathsf{T}\ddot{\mathbf{X}}, \qquad (37)$$
$$\Rightarrow \ \ddot{\mathbf{X}}^\mathsf{T} \cdot \mathbf{0}_{n,r} = \ddot{\mathbf{X}}^\mathsf{T}(\mathbf{G} - \ddot{\mathbf{X}}\mathbf{G}^\mathsf{T}\ddot{\mathbf{X}})$$
$$\overset{\text{\textcircled{2}}}{\Rightarrow} \ \mathbf{0}_{r,r} = \ddot{\mathbf{X}}^\mathsf{T}\mathbf{G} - \mathbf{G}^\mathsf{T}\ddot{\mathbf{X}}$$
$$\Rightarrow \ \mathbf{0}_{n,n} = \ddot{\mathbf{X}}(\ddot{\mathbf{X}}^\mathsf{T}\mathbf{G} - \mathbf{G}^\mathsf{T}\ddot{\mathbf{X}})\ddot{\mathbf{X}}^\mathsf{T}$$
$$\overset{\text{\textcircled{3}}}{\Rightarrow} \ \mathbf{0}_{n,n} = \ddot{\mathbf{X}}\underbrace{\ddot{\mathbf{X}}^\mathsf{T}\mathbf{G}\ddot{\mathbf{X}}^\mathsf{T}}_{\triangleq\mathbf{G}^\mathsf{T}} - \underbrace{\ddot{\mathbf{X}}\mathbf{G}^\mathsf{T}\ddot{\mathbf{X}}}_{\triangleq\mathbf{G}}\ddot{\mathbf{X}}^\mathsf{T},$$

where steps \textcircled{1} and \textcircled{2} use $\ddot{\mathbf{X}}^\mathsf{T}\ddot{\mathbf{X}} = \mathbf{I}_r$; step \textcircled{3} uses Equality (37) that $\mathbf{G} = \ddot{\mathbf{X}}\mathbf{G}^\mathsf{T}\ddot{\mathbf{X}}$. We conclude that the necessary condition in Equation (36) is equivalent to the optimality condition of critical points.

**Part (b)**. We now show that $\{\text{BS}_2\text{-points }\ddot{\mathbf{X}}\} \supseteq \{\text{global optimal points }\bar{\mathbf{X}}\}$. We define $\mathcal{X}_{\text{B}}^{\star}(\mathbf{V}) \triangleq \bar{\mathbf{X}} + \mathbf{U}_{\text{B}}(\mathbf{V} - \mathbf{I})\mathbf{U}_{\text{B}}^{\mathsf{T}}\bar{\mathbf{X}}$, and $\mathcal{K}(\mathbf{V}; \mathbf{X}, \text{B}) \triangleq f(\mathbf{X}) + \langle \mathbf{V} - \mathbf{I}_k, [\nabla f(\mathbf{X})(\mathbf{X})^{\mathsf{T}}]_{\text{BB}} \rangle + \frac{1}{2}\|\mathbf{V} - \mathbf{I}_k\|_{\mathbf{Q}+\alpha\mathbf{I}}^2 + h(\mathbf{U}_{\text{B}^c}^{\mathsf{T}}\mathbf{X}) + h(\mathbf{V}\mathbf{U}_{\text{B}}^{\mathsf{T}}\mathbf{X})$. We let $\mathbf{V}_{(i)} \in \text{St}(2, 2)$ and $\mathcal{B}_i \in \{\mathcal{B}_i\}_{i=1}^{C_n^k}$. We derive:

$$\mathcal{K}(\mathbf{I}_2; \bar{\mathbf{X}}, \mathcal{B}_i), \ \forall \mathcal{B}_i$$

$$\overset{①}{=} F(\bar{\mathbf{X}}) = h(\bar{\mathbf{X}}) + f(\bar{\mathbf{X}})$$

$$\overset{②}{=} h(\mathbf{X}) + f(\mathbf{X}), \forall \mathbf{X} \in \text{St}(n, r)$$

$$\overset{③}{\leq} h(\bar{\mathbf{X}} + \mathbf{U}_{\mathcal{B}_i}(\mathbf{V}_{(i)} - \mathbf{I})\mathbf{U}_{\mathcal{B}_i}^{\mathsf{T}}\bar{\mathbf{X}}) + f(\bar{\mathbf{X}} + \mathbf{U}_{\mathcal{B}_i}(\mathbf{V}_{(i)} - \mathbf{I})\mathbf{U}_{\mathcal{B}_i}^{\mathsf{T}}\bar{\mathbf{X}}), \ \forall \mathbf{V}_{(i)}, \ \forall \mathcal{B}_i$$

$$\overset{④}{=} h(\mathcal{X}_{\mathcal{B}_i}^{\star}(\mathbf{V}_{(i)})) + f(\mathcal{X}_{\mathcal{B}_i}^{\star}(\mathbf{V}_{(i)})), \ \forall \mathbf{V}_{(i)}, \ \forall \mathcal{B}_i$$

$$\overset{⑤}{=} \mathcal{K}(\mathbf{V}_{(i)}; \bar{\mathbf{X}}, \mathcal{B}_i), \ \forall \mathbf{V}_{(i)}, \ \forall \mathcal{B}_i$$

$$= \min_{\mathbf{V} \in \text{St}(2,2)} \mathcal{K}(\mathbf{V}; \bar{\mathbf{X}}, \mathcal{B}_i), \ \forall \mathcal{B}_i, \tag{38}$$

where step ① uses the definition of $\mathcal{K}(\mathbf{V}; \mathbf{X}, \text{B}) \triangleq f(\mathbf{X}) + \langle \mathbf{V} - \mathbf{I}_k, [\nabla f(\mathbf{X})(\mathbf{X})^{\mathsf{T}}]_{\text{BB}} \rangle + \frac{1}{2}\|\mathbf{V} - \mathbf{I}_k\|_{\mathbf{Q}+\alpha\mathbf{I}}^2 + h(\mathbf{U}_{\text{B}^c}^{\mathsf{T}}\mathbf{X}) + h(\mathbf{V}\mathbf{U}_{\text{B}}^{\mathsf{T}}\mathbf{X})$; step ② uses the definition of $\bar{\mathbf{X}}$; step ③ uses the basis representation of orthogonal matrices when $k = 2$, as shown in Theorem 3.1; step ④ uses the definition of $\mathcal{X}_{\text{B}}^{\star}(\mathbf{V})$; step ⑤ uses the same strategy as in deriving Inequality (2). This leads to:

$$\mathbf{I}_2 \in \arg\min_{\mathbf{V} \in \text{St}(2,2)} \mathcal{K}(\mathbf{V}; \bar{\mathbf{X}}, \mathcal{B}_i), \ \forall \mathcal{B}_i.$$

The inclusion above implies that $\{\text{BS}_2\text{-points }\ddot{\mathbf{X}}\} \supseteq \{\text{global optimal points }\bar{\mathbf{X}}\}$.

**Part (c)**. We now show that $\{\text{BS}_{\text{k}}\text{-}points\ \ddot{\mathbf{X}}\} \supseteq \{\text{BS}_{\text{k+1}}\text{-}points\ \ddot{\mathbf{X}}\}$. It is evident that the subproblem of finding $\text{BS}_{\text{k}}\text{-}points$ is encompassed within that of finding $\text{BS}_{\text{k+1}}\text{-}points$ stationary point. Thus, we conclude that the optimality of the latter is stronger.

**Part (d)**. The inclusion $\{\text{critical points }\check{\mathbf{X}}\} \subseteq \{\text{BS}_{\text{k}}\text{-points }\ddot{\mathbf{X}}\}$ may not always hold true. This can be illustrated through simple examples of $2 \times 2$ optimization problems under orthogonality constraints (see Appendix Section C.1 for more details). Lastly, it is also evident that the inclusions $\{\text{BS}_2\text{-points }\ddot{\mathbf{X}}\} \subseteq \{\text{global optimal points }\bar{\mathbf{X}}\}$ and $\{\text{BS}_{\text{k}}\text{-}points\ \ddot{\mathbf{X}}\} \subseteq \{\text{BS}_{\text{k+1}}\text{-}points\ \ddot{\mathbf{X}}\}$ may not always hold true.

$$\square$$

# G PROOF FOR SECTION 4

## G.1 PROOF FOR THEOREM 4.2

*Proof.* We define $\mathcal{K}(\mathbf{V}; \mathbf{X}^t, \text{B}) \triangleq \frac{1}{2}\|\mathbf{V} - \mathbf{I}_k\|_{\mathbf{Q}+\alpha\mathbf{I}}^2 + h(\mathbf{V}\mathbf{Z}) + \langle \mathbf{V}, [\nabla f(\mathbf{X}^t)(\mathbf{X}^t)^{\mathsf{T}}]_{\text{BB}} \rangle + \ddot{c}$, where $\mathbf{Z} \triangleq \mathbf{U}_{\text{B}}^{\mathsf{T}}\mathbf{X}^t$ and $\ddot{c} = h(\mathbf{U}_{\text{B}^c}^{\mathsf{T}}\mathbf{X}^t) + f(\mathbf{X}^t) - \langle \mathbf{I}_k, [\nabla f(\mathbf{X}^t)(\mathbf{X}^t)^{\mathsf{T}}]_{\text{BB}} \rangle$ is a constant.

We define $\tilde{c} \triangleq \frac{2}{\alpha} \cdot (F(\mathbf{X}^0) - F(\ddot{\mathbf{X}}))$.

**Part (a)**. First, we have the following equalities:

$$h(\mathbf{X}^{t+1}) - h(\mathbf{X}^t) \overset{①}{=} h(\mathbf{U}_{\text{B}}\bar{\mathbf{V}}^t\mathbf{U}_{\text{B}}^{\mathsf{T}}\mathbf{X}^t + \mathbf{U}_{\text{B}^c}\mathbf{U}_{\text{B}^c}^{\mathsf{T}}\mathbf{X}^t) - h(\mathbf{U}_{\text{B}}\mathbf{U}_{\text{B}}^{\mathsf{T}}\mathbf{X}^t + \mathbf{U}_{\text{B}^c}\mathbf{U}_{\text{B}^c}^{\mathsf{T}}\mathbf{X}^t)$$

$$\overset{②}{=} h(\mathbf{U}_{\text{B}}\bar{\mathbf{V}}^t\mathbf{U}_{\text{B}}^{\mathsf{T}}\mathbf{X}^t) + h(\mathbf{U}_{\text{B}^c}\mathbf{U}_{\text{B}^c}^{\mathsf{T}}\mathbf{X}^t) - h(\mathbf{U}_{\text{B}}\mathbf{U}_{\text{B}}^{\mathsf{T}}\mathbf{X}^t) - h(\mathbf{U}_{\text{B}^c}\mathbf{U}_{\text{B}^c}^{\mathsf{T}}\mathbf{X}^t)$$

$$\overset{③}{=} h(\bar{\mathbf{V}}^t\mathbf{U}_{\text{B}}^{\mathsf{T}}\mathbf{X}^t) - h(\mathbf{U}_{\text{B}}^{\mathsf{T}}\mathbf{X}^t), \tag{39}$$

where step ① uses $\mathbf{X}^{t+1} = \mathbf{U}_{\text{B}}\mathbf{V}\mathbf{U}_{\text{B}}^{\mathsf{T}}\mathbf{X}^t + \mathbf{U}_{\text{B}^c}\mathbf{U}_{\text{B}^c}^{\mathsf{T}}\mathbf{X}^t$ as in (4) and $\mathbf{I} = \mathbf{U}_{\text{B}}\mathbf{U}_{\text{B}}^{\mathsf{T}} + \mathbf{U}_{\text{B}^c}\mathbf{U}_{\text{B}^c}^{\mathsf{T}}$; step ② and step ③ use the coordinate-wise separable structure of $h(\cdot)$.

Second, given $\bar{\mathbf{V}}^t$ is a global or local optimal solution of the following problem: $\bar{\mathbf{V}}^t \in \arg\min_{\mathbf{V} \in \text{St}(k,k)} \mathcal{K}(\mathbf{V}; \mathbf{X}^t, \text{B})$ satisfying $\mathcal{K}(\bar{\mathbf{V}}^t; \mathbf{X}^t, \text{B}) \leq \mathcal{K}(\mathbf{I}_k; \mathbf{X}^t, \text{B})$, we have:

$$h(\bar{\mathbf{V}}^t\mathbf{U}_{\text{B}}^{\mathsf{T}}\mathbf{X}^t) + \frac{1}{2}\|\bar{\mathbf{V}}^t - \mathbf{I}_k\|_{\mathbf{Q}+\alpha\mathbf{I}}^2 + \langle \bar{\mathbf{V}}^t - \mathbf{I}, [\nabla f(\mathbf{X}^t)(\mathbf{X}^t)^{\mathsf{T}}]_{\text{BB}} \rangle \leq h(\mathbf{U}_{\text{B}}^{\mathsf{T}}\mathbf{X}^t). \tag{40}$$

Third, we denote $\mathbf{X}^{t+1} = \mathcal{X}_{\mathtt{B}}^t(\bar{\mathbf{V}}^t)$ and derive:

$$
\begin{aligned}
f(\mathbf{X}^{t+1}) - f(\mathbf{X}^t) \quad &\overset{①}{\leq} \quad \langle \mathcal{X}_{\mathtt{B}}^t(\bar{\mathbf{V}}^t) - \mathbf{X}^t, \nabla f(\mathbf{X}^t) \rangle + \tfrac{1}{2}\|\mathcal{X}_{\mathtt{B}}^t(\bar{\mathbf{V}}^t) - \mathbf{X}^t\|_{\mathbf{H}}^2 \\
&\overset{②}{=} \quad \langle \mathrm{U}_{\mathtt{B}}(\bar{\mathbf{V}}^t - \mathbf{I}_k)\mathrm{U}_{\mathtt{B}}^{\mathsf{T}}\mathbf{X}^t, \nabla f(\mathbf{X}^t) \rangle + \tfrac{1}{2}\|\bar{\mathbf{V}}^t - \mathbf{I}_k\|_{\underline{\mathbf{Q}}}^2 \\
&\overset{③}{\leq} \quad \langle \bar{\mathbf{V}}^t - \mathbf{I}_k, [\nabla f(\mathbf{X}^t)(\mathbf{X}^t)^{\mathsf{T}}]_{\mathtt{BB}} \rangle + \tfrac{1}{2}\|\bar{\mathbf{V}}^t - \mathbf{I}_k\|_{\mathbf{Q}}^2,
\end{aligned}
\tag{41}
$$

where step ① uses Inequality (2); step ② uses Claim (*a*) of Lemma 2.2; step ③ uses $\mathbf{Q} \succeq \underline{\mathbf{Q}}$.

Adding (39), (40), and (41) together, we obtain the following sufficient decrease condition:

$$
F(\mathbf{X}^{t+1}) - F(\mathbf{X}^t) \leq -\tfrac{\alpha}{2}\|\bar{\mathbf{V}}^t - \mathbf{I}_k\|_{\mathsf{F}}^2 \overset{①}{\leq} -\tfrac{\alpha}{2}\|\mathbf{X}^{t+1} - \mathbf{X}^t\|_{\mathsf{F}}^2,
\tag{42}
$$

where step ① uses Claim (*c*) of Lemma 2.2.

**Part (b)**. We assume that $\mathtt{B}^t$ is selected from $\{\mathcal{B}_i\}_{i=1}^{\mathrm{C}_n^k}$ randomly and uniformly.

Taking the expectation for Inequality (42), we obtain a lower bound on the expected progress made by each iteration:

$$
\mathbb{E}_{\xi^t}[F(\mathbf{X}^{t+1})] - F(\mathbf{X}^t) \leq -\mathbb{E}_{\xi^t}[\tfrac{\alpha}{2}\|\bar{\mathbf{V}}^t - \mathbf{I}_k\|_{\mathsf{F}}^2].
$$

Telescoping the inequality above over $t = 0, 1, ..., T$, we have:

$$
\mathbb{E}_{\xi^T}[\tfrac{\alpha}{2}\textstyle\sum_{t=0}^T \|\bar{\mathbf{V}}^t - \mathbf{I}_k\|_{\mathsf{F}}^2] \leq \mathbb{E}_{\xi^T}[F(\mathbf{X}^0) - F(\mathbf{X}^{T+1})] \leq \mathbb{E}_{\xi^T}[F(\mathbf{X}^0) - F(\ddot{\mathbf{X}})],
$$

where $\ddot{\mathbf{X}}$ denotes the limit point of Algorithm 1. As a result, there exists an index $\bar{t}$ with $0 \leq \bar{t} \leq T$ such that

$$
\mathbb{E}_{\xi^T}[\|\bar{\mathbf{V}}^{\bar{t}} - \mathbf{I}_k\|_{\mathsf{F}}^2] \leq \tfrac{2}{\alpha(T+1)}[F(\mathbf{X}^0) - F(\ddot{\mathbf{X}})] = \tfrac{\tilde{c}}{T+1}.
\tag{43}
$$

Furthermore, for any $t$, $\bar{\mathbf{V}}^t$ is the optimal solution of the following minimization problem at $\mathbf{X}^t$: $\bar{\mathbf{V}}^t \in \arg\min_{\mathbf{V}} \min_{\mathbf{V}} \mathcal{K}(\mathbf{V}; \mathbf{X}^t, \mathtt{B}^t)$. Given $\bar{\mathbf{V}}^t$ is a random output matrix depends on the observed realization of the random variable $\mathtt{B}^t$, we directly obtain the following equality:

$$
\tfrac{1}{\mathrm{C}_n^k}\textstyle\sum_{i=1}^{\mathrm{C}_n^k} \mathrm{dist}(\mathbf{I}_k, \arg\min_{\mathbf{V}} \mathcal{K}(\mathbf{V}; \mathbf{X}^t, \mathcal{B}_i))^2 = \mathbb{E}_{\xi^t}[\|\bar{\mathbf{V}}^t - \mathbf{I}_k\|_{\mathsf{F}}^2].
\tag{44}
$$

Combining (43) and (44), we conclude that there exists an index $\bar{t}$ with $\bar{t} \in [0, T]$ such that the associated solution $\mathbf{X}^{\bar{t}}$ qualifies as an $\epsilon$-BS$_k$-*point* of Problem (1), provided that $T$ is sufficiently large such that $\frac{\tilde{c}}{T+1} \leq \epsilon$. We conclude that **OBCD** finds an $\epsilon$-BS$_k$-*point* of Problem (1) in at most $T$ iterations deterministically, where $T \geq \lceil \frac{\tilde{c}}{\epsilon} - 1 \rceil$.

**Part (c)**. We assume that $\mathtt{B}^t$ is selected from $\{\mathcal{B}_i\}_{i=1}^{\mathrm{C}_n^k}$ cyclically, i.e., $\mathcal{B}_1 \to \mathcal{B}_2 \to \mathcal{B}_3 \to \ldots \to \mathcal{B}_{\mathrm{C}_n^k-1} \to \mathcal{B}_{\mathrm{C}_n^k} \to \mathcal{B}_1 \to \mathcal{B}_2 \to \mathcal{B}_3 \to \ldots$.

Telescoping Inequality (42) over $t$ from 0 to $T$ yields:

$$
\tfrac{\alpha}{2}\textstyle\sum_{t=0}^T \|\bar{\mathbf{V}}^t - \mathbf{I}_k\|_F^2 \leq F(\mathbf{X}^0) - F(\mathbf{X}^{T+1}) \leq F(\mathbf{X}^0) - F(\ddot{\mathbf{X}}),
\tag{45}
$$

For notation simplicity, we define $z \triangleq \mathrm{C}_n^k$ and $e^t \triangleq \|\bar{\mathbf{V}}^t - \mathbf{I}_k\|_{\mathsf{F}}^2$. We have from Inequality (45):

$$
\begin{aligned}
\tilde{c} \quad &\triangleq \quad (F(\mathbf{X}^0) - F(\ddot{\mathbf{X}})) \cdot \tfrac{2}{\alpha} \\
&\geq \quad \textstyle\sum_{t=0}^T e^t \\
&= \quad e^0 + \textstyle\sum_{i=1}^z e^i + \textstyle\sum_{i=z+1}^{2z} e^i + \textstyle\sum_{i=2z+1}^{3z} e^i + \ldots \\
&\quad + \textstyle\sum_{i=[\lfloor T/z \rfloor - 1]z+1}^{\lfloor T/z \rfloor z} e^i + \textstyle\sum_{i=\lfloor T/z \rfloor z+1}^T e^i \\
&\overset{①}{\geq} \quad \textstyle\sum_{i=1}^z e^i + \textstyle\sum_{i=z+1}^{2z} e^i + \textstyle\sum_{i=2z+1}^{3z} e^i + \ldots + \textstyle\sum_{i=[\lfloor T/z \rfloor - 1]z+1}^{\lfloor T/z \rfloor z} e^i \\
&\geq \quad (\min_{k=1}^{\lfloor T/z \rfloor}[\textstyle\sum_{i=(k-1)z+1}^{kz} e^i]) \times \lfloor T/z \rfloor, \\
&\overset{②}{\geq} \quad (\min_{k=1}^{\lfloor T/z \rfloor}[\textstyle\sum_{i=(k-1)z+1}^{kz} e^i]) \times (\tfrac{T-z}{z}),
\end{aligned}
\tag{46}
$$

where step ① uses $e^i \geq 0$ for all $i$, and $T \geq \lfloor T/z \rfloor z$ for all $T \geq 0$; step ② uses $\lfloor T/z \rfloor \geq \frac{T}{z} - 1$ for all $T > z > 0$. Inequality (46) implies that there exists an index $\bar{k}$ with $\bar{k} \in [1, \lfloor T/z \rfloor]$ satisfying

$$\frac{1}{z} \sum_{i=1+(\bar{k}-1)z}^{\bar{k}z} e^i \leq \frac{\tilde{c}}{T-z}. \tag{47}$$

Such inequality further implies the associated solution $\mathbf{X}^{\bar{k}z}$ qualifies as an $\epsilon$-BS$_k$-*point* of Problem (1), provided that $T$ is sufficiently large such that $\frac{\tilde{c}}{T-z} \leq \epsilon$ and $T > z$. We conclude that **OBCD** finds an $\epsilon$-BS$_k$-*point* of Problem (1) in at most $T$ iterations deterministically, where $T \geq \lceil \frac{\tilde{c}}{\epsilon} + z \rceil$.

$\square$

### G.2   PROOF OF LEMMA 4.4

*Proof.* For notation simplicity, we define: $\|\partial F(\mathbf{X})\|_\mathsf{F} = \inf_{\mathbf{Y} \in \partial F(\mathbf{X})} \|\mathbf{Y}\|_\mathsf{F} = \text{dist}(\mathbf{0}, \partial F(\mathbf{X}))$.

We define $\mathbb{A} \ominus \mathbb{B}$ as the element-wise subtraction between sets $\mathbb{A}$ and $\mathbb{B}$.

We let $\mathbb{H}^{t+1} \in \partial h(\mathbf{X}^{t+1})$, and define:

$$\Omega_0 \triangleq \mathrm{U}_{\mathrm{B}^t}^\mathsf{T} [\nabla f(\mathbf{X}^{t+1}) + \mathbb{H}^{t+1}][\mathbf{X}^{t+1}]^\mathsf{T} \mathrm{U}_{\mathrm{B}^t} \in \mathbb{R}^{k \times k}, \tag{48}$$

$$\Omega_1 \triangleq \mathrm{U}_{\mathrm{B}^t}^\mathsf{T} [\nabla f(\mathbf{X}^{t+1}) + \mathbb{H}^{t+1}][\mathbf{X}^t]^\mathsf{T} \mathrm{U}_{\mathrm{B}^t} \in \mathbb{R}^{k \times k}, \tag{49}$$

$$\Omega_2 \triangleq \mathrm{U}_{\mathrm{B}^t}^\mathsf{T} [\nabla f(\mathbf{X}^t) - \nabla f(\mathbf{X}^{t+1})][\mathbf{X}^t]^\mathsf{T} \mathrm{U}_{\mathrm{B}^t} \in \mathbb{R}^{k \times k}. \tag{50}$$

**Part (a)**. First, using the optimality of $\bar{\mathbf{V}}^t$ for the subproblem, we have:

$$\mathbf{0}_{k,k} = \tilde{\mathbf{G}} - \bar{\mathbf{V}}^t \tilde{\mathbf{G}}^\mathsf{T} \bar{\mathbf{V}}^t$$

$$\text{where } \tilde{\mathbf{G}} = \underbrace{\text{mat}((\mathbf{Q} + \alpha \mathbf{I}_k)\text{vec}(\bar{\mathbf{V}}^t - \mathbf{I}_k))}_{\triangleq \Upsilon_1} + \underbrace{\mathrm{U}_{\mathrm{B}^t}^\mathsf{T}[\nabla f(\mathbf{X}^t) + \mathbb{H}^{t+1}](\mathbf{X}^t)^\mathsf{T} \mathrm{U}_{\mathrm{B}^t}}_{\triangleq \Upsilon_2}.$$

Using the relation that $\tilde{\mathbf{G}} = \Upsilon_1 + \Upsilon_2$, we obtain the following results from the above equality:

$$\mathbf{0}_{k,k} = (\Upsilon_1 + \Upsilon_2) - \bar{\mathbf{V}}^t (\Upsilon_1 + \Upsilon_2)^\mathsf{T} \bar{\mathbf{V}}^t$$

$$\overset{①}{\Rightarrow} \mathbf{0}_{k,k} = \Upsilon_1 + \Omega_1 + \Omega_2 - \bar{\mathbf{V}}^t (\Upsilon_1 + \Omega_1 + \Omega_2)^\mathsf{T} \bar{\mathbf{V}}^t$$

$$\Rightarrow \Omega_1 = \bar{\mathbf{V}}^t (\Upsilon_1 + \Omega_1 + \Omega_2)^\mathsf{T} \bar{\mathbf{V}}^t - \Upsilon_1 - \Omega_2, \tag{51}$$

where step ① uses $\Upsilon_2 = \Omega_1 + \Omega_2$.

Second, since both $\mathrm{B}^t$ and $\mathrm{B}^{t+1}$ are randomly and dependently selected from $\{\mathcal{B}_i\}_{i=1}^{\mathrm{C}_n^k}$ with replacement, each with an equal probability of $\frac{1}{\mathrm{C}_n^k}$, for any $\tilde{\mathbf{A}} \in \mathbb{R}^{n \times n}$, we have:

$$\mathbb{E}_{\mathrm{B}^{t+1}}[\|\mathrm{U}_{\mathrm{B}^{t+1}}^\mathsf{T} \tilde{\mathbf{A}} \mathrm{U}_{\mathrm{B}^{t+1}}\|_\mathsf{F}^2] = \frac{1}{\mathrm{C}_n^k} \sum_{i=1}^{\mathrm{C}_n^k} \|\mathrm{U}_{\mathcal{B}_i}^\mathsf{T} \tilde{\mathbf{A}} \mathrm{U}_{\mathcal{B}_i}\|_\mathsf{F}^2 = \mathbb{E}_{\mathrm{B}^t} \|\mathrm{U}_{\mathrm{B}^t}^\mathsf{T} \tilde{\mathbf{A}} \mathrm{U}_{\mathrm{B}^t}\|_\mathsf{F}^2. \tag{52}$$

Third, we derive the following results:

$$\mathbb{E}_{\xi^{t+1}}[\text{dist}(\mathbf{0}, \partial_\mathcal{M} \mathcal{K}(\mathbf{I}_k; \mathbf{X}^{t+1}, \mathrm{B}^{t+1}))] = \mathbb{E}_{\xi^{t+1}}[\|\partial_\mathcal{M} \mathcal{K}(\mathbf{I}_k; \mathbf{X}^{t+1}, \mathrm{B}^{t+1})\|_\mathsf{F}]$$

$$\overset{①}{=} \mathbb{E}_{\xi^{t+1}}[\|\mathrm{U}_{\mathrm{B}^{t+1}}^\mathsf{T} \{\partial F(\mathbf{X}^{t+1})[\mathbf{X}^{t+1}]^\mathsf{T} \ominus \mathbf{X}^{t+1}[\partial F(\mathbf{X}^{t+1})]^\mathsf{T}\} \mathrm{U}_{\mathrm{B}^{t+1}}\|_\mathsf{F}]$$

$$\overset{②}{=} \mathbb{E}_{\xi^t}[\|\mathrm{U}_{\mathrm{B}^t}^\mathsf{T} \{\partial F(\mathbf{X}^{t+1})[\mathbf{X}^{t+1}]^\mathsf{T} \ominus \mathbf{X}^{t+1}[\partial F(\mathbf{X}^{t+1})]^\mathsf{T}\} \mathrm{U}_{\mathrm{B}^t}\|_\mathsf{F}]$$

$$\overset{③}{\leq} \mathbb{E}_{\xi^t}[\|\Omega_0 - \Omega_0^\mathsf{T}\|_\mathsf{F}]$$

$$\overset{④}{\leq} 2\mathbb{E}_{\xi^t}[\|\Omega_0 - \Omega_1\|_\mathsf{F}] + \mathbb{E}_{\xi^t}[\|\Omega_1 - \Omega_1^\mathsf{T}\|_\mathsf{F}]$$

$$\overset{⑤}{=} 2\mathbb{E}_{\xi^t}[\|\Omega_0 - \Omega_1\|_\mathsf{F}] + \mathbb{E}_{\xi^t}[\|\bar{\mathbf{V}}^t (\Upsilon_1 + \Omega_1 + \Omega_2)^\mathsf{T} \bar{\mathbf{V}}^t - \Upsilon_1 - \Omega_2 - \Omega_1^\mathsf{T}\|_\mathsf{F}]$$

$$\overset{⑥}{=} 2\mathbb{E}_{\xi^t}[\|\Omega_0 - \Omega_1\|_\mathsf{F}] + \mathbb{E}_{\xi^t}[\|\bar{\mathbf{V}}^t \Upsilon_1^\mathsf{T} \bar{\mathbf{V}}^t - \Upsilon_1\|_\mathsf{F}] + \mathbb{E}_{\xi^t}[\|\bar{\mathbf{V}}^t \Omega_1^\mathsf{T} \bar{\mathbf{V}}^t - \Omega_1^\mathsf{T}\|_\mathsf{F}]$$

$$+ \mathbb{E}_{\xi^t}[\|\bar{\mathbf{V}}^t \Omega_2^\mathsf{T} \bar{\mathbf{V}}^t - \Omega_2\|_\mathsf{F}] \tag{53}$$

where step ① uses the definition of $\partial_{\mathcal{M}}\mathcal{K}(\mathbf{V}; \mathbf{X}^{t+1}, \mathrm{B}^{t+1})$ at the point $\mathbf{V} = \mathbf{I}_k$; step ② uses Equality (52) with $\tilde{\mathbf{A}} = \partial F(\mathbf{X}^{t+1})(\mathbf{X}^{t+1})^\mathsf{T} \ominus \mathbf{X}^{t+1}(\partial F(\mathbf{X}^{t+1}))^\mathsf{T}$; step ③ uses the definition of $\Omega_0$ in Equation (48); step ④ uses Lemma A.2; step ⑤ uses Equality (51); step ⑥ uses the triangle inequality.

We now establish individual bounds for each term in Inequality (53). For the first term $2\mathbb{E}_{\xi^t}[\|\Omega_0 - \Omega_1\|_\mathsf{F}]$ in (53), we have:

$$2\mathbb{E}_{\xi^t}[\|\Omega_0 - \Omega_1\|_\mathsf{F}]$$
$$\leq \quad 2\mathbb{E}_{\xi^t}[\|\mathrm{U}_{\mathrm{B}^t}^\mathsf{T}[\nabla f(\mathbf{X}^{t+1}) + \mathbb{H}^{t+1}][\mathbf{X}^{t+1} - \mathbf{X}^t]^\mathsf{T}\mathrm{U}_{\mathrm{B}^t}\|_\mathsf{F}]$$
$$\overset{①}{=} \quad 2\mathbb{E}_{\xi^t}[\|\mathrm{U}_{\mathrm{B}^t}^\mathsf{T}[\nabla f(\mathbf{X}^{t+1}) + \mathbb{H}^{t+1}][\mathrm{U}_\mathrm{B}(\bar{\mathbf{V}}^t - \mathbf{I}_k)\mathrm{U}_{\mathrm{B}^t}\mathbf{X}^t]^\mathsf{T}\mathrm{U}_{\mathrm{B}^t}\|_\mathsf{F}]$$
$$\overset{②}{\leq} \quad 2(l_f + l_h)\mathbb{E}_{\xi^t}[\|\bar{\mathbf{V}}^t - \mathbf{I}_k\|_\mathsf{F}], \tag{54}$$

where step ① uses $\mathbf{X}^{t+1} = \mathbf{X}^t + \mathrm{U}_\mathrm{B}(\bar{\mathbf{V}}^t - \mathbf{I}_k)\mathrm{U}_\mathrm{B}^\mathsf{T}\mathbf{X}^t$; step ② uses the inequality $\|\mathbf{X}\mathbf{Y}\|_\mathsf{F} \leq \|\mathbf{X}\|_\mathsf{F}\|\mathbf{Y}\|_\mathsf{sp}$ for all $\mathbf{X}$ and $\mathbf{Y}$ repeatedly, and the fact that $\forall \mathbf{X}, \|\nabla f(\mathbf{X})\|_\mathsf{sp} \leq l_f, \|\partial h(\mathbf{X})\|_\mathsf{sp} \leq l_h$.

For the second term $\mathbb{E}_{\xi^t}[\|\bar{\mathbf{V}}^t\Upsilon_1^\mathsf{T}\bar{\mathbf{V}}^t - \Upsilon_1\|_\mathsf{F}]$ in (53), we have::

$$\mathbb{E}_{\xi^t}[\|\bar{\mathbf{V}}^t\Upsilon_1^\mathsf{T}\bar{\mathbf{V}}^t - \Upsilon_1\|_\mathsf{F}]$$
$$\overset{①}{\leq} \quad \mathbb{E}_{\xi^t}[\|\bar{\mathbf{V}}^t\Upsilon_1^\mathsf{T}\bar{\mathbf{V}}^t\|_\mathsf{F}] + \mathbb{E}_{\xi^t}[\|\Upsilon_1\|_\mathsf{F}]$$
$$\overset{②}{\leq} \quad 2\mathbb{E}_{\xi^t}[\|\Upsilon_1\|_\mathsf{F}]$$
$$\overset{③}{=} \quad 2\mathbb{E}_{\xi^t}[\|\mathrm{mat}((\mathbf{Q} + \alpha\mathbf{I}_k)\mathrm{vec}(\bar{\mathbf{V}}^t - \mathbf{I}_k))\|_\mathsf{F}]$$
$$\leq \quad 2\|\mathbf{Q} + \alpha\mathbf{I}_k\|_\mathsf{sp} \cdot \mathbb{E}_{\xi^t}[\|\bar{\mathbf{V}}^t - \mathbf{I}_k)\|_\mathsf{F}]$$
$$\overset{④}{\leq} \quad 2(L_f + \alpha) \cdot \mathbb{E}_{\xi^t}[\|\bar{\mathbf{V}}^t - \mathbf{I}_k\|_\mathsf{F}] \tag{55}$$

where step ① uses the triangle inequality; step ② uses the inequality $\|\mathbf{X}\mathbf{Y}\|_\mathsf{F} \leq \|\mathbf{X}\|_\mathsf{F}\|\mathbf{Y}\|_\mathsf{sp}$ for all $\mathbf{X}$ and $\mathbf{Y}$; step ③ uses the definition of $\Omega_1$ in (49); step ④ uses the fact that $\|\mathbf{Q}\|_\mathsf{sp} \leq L_f$.

For the third term $\mathbb{E}_{\xi^t}[\|\bar{\mathbf{V}}^t\Omega_1^\mathsf{T}\bar{\mathbf{V}}^t - \Omega_1^\mathsf{T}\|_\mathsf{F}]$ in (53), we have:

$$\mathbb{E}_{\xi^t}[\|\bar{\mathbf{V}}^t\Omega_1^\mathsf{T}\bar{\mathbf{V}}^t - \Omega_1^\mathsf{T}\|_\mathsf{F}]$$
$$\overset{①}{=} \quad \mathbb{E}_{\xi^t}[\|\bar{\mathbf{V}}^t\Omega_1^\mathsf{T}(\bar{\mathbf{V}}^t - \mathbf{I}_k) + (\bar{\mathbf{V}}^t - \mathbf{I})\Omega_1^\mathsf{T}\|_\mathsf{F}]$$
$$\overset{②}{\leq} \quad 2\mathbb{E}_{\xi^t}[\|\Omega_1\|_\mathsf{sp} \cdot \|(\bar{\mathbf{V}}^t - \mathbf{I}_k)\|_\mathsf{F}]$$
$$\overset{③}{\leq} \quad 2\mathbb{E}_{\xi^t}[\|\nabla f(\mathbf{X}^{t+1}) + \mathbb{H}^{t+1}\|_\mathsf{sp} \cdot \|(\bar{\mathbf{V}}^t - \mathbf{I}_k)\|_\mathsf{F}]$$
$$\overset{④}{\leq} \quad 2(l_f + l_h)\mathbb{E}_{\xi^t}[\|(\bar{\mathbf{V}}^t - \mathbf{I}_k)\|_\mathsf{F}] \tag{56}$$

where step ① uses the fact that $-\bar{\mathbf{V}}^t\Omega_1^\mathsf{T}\mathbf{I}_k + \bar{\mathbf{V}}^t\Omega_1^\mathsf{T} = \mathbf{0}$; step ② uses the norm inequality; step ③ uses the fact that $\|\Omega_1\|_\mathsf{sp} = \|\mathrm{U}_{\mathrm{B}^t}^\mathsf{T}[\nabla f(\mathbf{X}^{t+1}) + \mathbb{H}^{t+1}][\mathbf{X}^t]^\mathsf{T}\mathrm{U}_{\mathrm{B}^t}\|_\mathsf{sp} \leq \|\nabla f(\mathbf{X}^{t+1}) + \mathbb{H}^{t+1}\|_\mathsf{sp}$ which can be derived using the norm inequality; step ④ uses the fact that $\forall \mathbf{X}, \|\nabla f(\mathbf{X})\|_\mathsf{sp} \leq l_f, \|\partial h(\mathbf{X})\|_\mathsf{sp} \leq l_h$.

For the fourth term $\mathbb{E}_{\xi^t}[\|\bar{\mathbf{V}}^t\Omega_2^\mathsf{T}\bar{\mathbf{V}}^t - \Omega_2\|_\mathsf{F}]$ in (53), we have:

$$\mathbb{E}_{\xi^t}[\|\bar{\mathbf{V}}^t\Omega_2^\mathsf{T}\bar{\mathbf{V}}^t - \Omega_2\|_\mathsf{F}]$$
$$\overset{①}{\leq} \quad \mathbb{E}_{\xi^t}[\|\bar{\mathbf{V}}^t\Omega_2^\mathsf{T}\bar{\mathbf{V}}^t\|_\mathsf{F}] + \mathbb{E}[\|\Omega_2\|_\mathsf{F}]$$
$$\overset{②}{\leq} \quad 2\mathbb{E}_{\xi^t}[\|\Omega_2\|_\mathsf{F}]$$
$$\overset{③}{=} \quad 2\mathbb{E}_{\xi^t}[\|\mathrm{U}_{\mathrm{B}^t}^\mathsf{T}[\nabla f(\mathbf{X}^t) - \nabla f(\mathbf{X}^{t+1})][\mathbf{X}^t]^\mathsf{T}\mathrm{U}_{\mathrm{B}^t}\|_\mathsf{F}]$$
$$\overset{④}{=} \quad 2\mathbb{E}_{\xi^t}[\|\nabla f(\mathbf{X}^t) - \nabla f(\mathbf{X}^{t+1})\|_\mathsf{F}]$$
$$\overset{⑤}{=} \quad 2L_f\mathbb{E}_{\xi^t}[\|\mathbf{X}^t - \mathbf{X}^{t+1}\|_\mathsf{F}]$$
$$\overset{⑥}{=} \quad 2L_f\mathbb{E}_{\xi^t}[\|\bar{\mathbf{V}}^t - \mathbf{I}_k\|_\mathsf{F}], \tag{57}$$

where step ① uses the triangle inequality; step ② uses the norm inequality; step ③ uses the definition of $\Omega_2 = U_{B^t}^\mathsf{T}[\nabla f(\mathbf{X}^t) - \nabla f(\mathbf{X}^{t+1})][\mathbf{X}^t]^\mathsf{T}U_{B^t}$ in (50); step ④ uses the norm inequality; step ⑤ uses the fact that $\nabla f(\mathbf{X})$ is $L_f$-Lipschitz continuous; step ⑥ uses Claim (*c*) of Lemma 2.2.

In view of (54), (55), (56), (57), and (53), we have:

$$\mathbb{E}_{\xi^{t+1}}[\|\partial_{\mathcal{M}}\mathcal{K}(\mathbf{I}_k; \mathbf{X}^{t+1}, B^{t+1})\|_\mathsf{F}] \leq \underbrace{(c_1 + c_2 + c_3 + c_4)}_{\triangleq \phi} \cdot \mathbb{E}_{\xi^t}[\|\bar{\mathbf{V}}^t - \mathbf{I}_k\|_\mathsf{F}],$$

where $c_1 = 2(l_f + l_h)$, $c_2 = 2(L_f + \alpha)$, $c_3 = 2(l_f + l_h)$, and $c_4 = 2L_f$.

**Part (b)**. we show that $\mathbb{E}_{\xi^t}[\text{dist}(\mathbf{0}, \partial_{\mathcal{M}}F(\mathbf{X}^t))] \leq \gamma \cdot \mathbb{E}_{\xi^t}[\text{dist}(\mathbf{0}, \partial_{\mathcal{M}}\mathcal{K}(\mathbf{I}_k; \mathbf{X}^t, B^t))]$, where $\gamma \triangleq (\mathrm{C}_n^k/\mathrm{C}_{n-2}^{k-2})^{1/2}$. For all $\mathbf{D}^t \triangleq \partial F(\mathbf{X}^t)[\mathbf{X}^t]^\mathsf{T} \ominus \mathbf{X}^t[\partial F(\mathbf{X}^t)]^\mathsf{T}$, we obtain:

$$
\begin{aligned}
\|\mathbf{D}^t\|_\mathsf{F}^2 &= \textstyle\sum_i \sum_{j\neq i}(\mathbf{D}_{ij}^t)^2 + \sum_i \sum_{j=i}(\mathbf{D}_{ij}^t)^2 \\
&\overset{①}{=} \textstyle\sum_i \sum_{j\neq i}(\mathbf{D}_{ij}^t)^2 \\
&\overset{②}{=} \tfrac{1}{\mathrm{C}_{n-2}^{k-2}}\textstyle\sum_{i=1}^{\mathrm{C}_n^k}\|U_{\mathcal{B}_i}^\mathsf{T}\mathbf{D}^tU_{\mathcal{B}_i}\|_\mathsf{F}^2 \\
&\overset{③}{=} \tfrac{1}{\mathrm{C}_{n-2}^{k-2}} \cdot \mathrm{C}_n^k \mathbb{E}_{B^t}[\|U_{B^t}^\mathsf{T}\mathbf{D}^tU_{B^t}\|_\mathsf{F}^2] \\
&\overset{④}{=} \gamma^2 \mathbb{E}_{B^t}[\|U_{B^t}^\mathsf{T}\mathbf{D}^tU_{B^t}\|_\mathsf{F}^2],
\end{aligned}
\tag{58}
$$

where step ① uses the fact that $\mathbf{D}_{ii}^t = 0$ for all $i \in [n]$; step ② uses Claim (*a*) of this lemma with $\mathbf{D}_{ii}^t = 0$ for all $i \in [n]$; step ③ uses $\mathbb{E}_{B^t}[\|U_{B^t}^\mathsf{T}\mathbf{W}U_{B^t}\|_\mathsf{F}^2] = \tfrac{1}{\mathrm{C}_n^k}\sum_{i=1}^{\mathrm{C}_n^k}\|U_{\mathcal{B}_i}^\mathsf{T}\mathbf{W}U_{\mathcal{B}_i}\|_\mathsf{F}^2$ as $B^t$ are chosen from $\{\mathcal{B}_i\}_{i=1}^{\mathrm{C}_n^k}$ randomly and uniformly; ④ uses the definition of $\gamma$. We further derive:

$$
\begin{aligned}
\mathbb{E}_{\xi^t}\|\partial_{\mathcal{M}}F(\mathbf{X}^t)\|_\mathsf{F} &\overset{①}{=} \|\partial F(\mathbf{X}^t) \ominus \mathbf{X}^t[\partial F(\mathbf{X}^t)]^\mathsf{T}\mathbf{X}^t\|_\mathsf{F} \\
&\overset{②}{=} \|\partial F(\mathbf{X}^t)[\mathbf{X}^t]^\mathsf{T}\mathbf{X}^t \ominus \mathbf{X}^t[\partial F(\mathbf{X}^t)]^\mathsf{T}\mathbf{X}^t\|_\mathsf{F} \\
&\overset{③}{\leq} \|\partial F(\mathbf{X}^t)[\mathbf{X}^t]^\mathsf{T} \ominus \mathbf{X}^t[\partial F(\mathbf{X}^t)]^\mathsf{T}\|_\mathsf{F} \\
&\overset{④}{=} \gamma \mathbb{E}_{B^t}[\|U_{B^t}^\mathsf{T}\{\partial F(\mathbf{X}^t)[\mathbf{X}^t]^\mathsf{T} \ominus \mathbf{X}^t[\partial F(\mathbf{X}^t)]^\mathsf{T}\}U_{B^t}\|_\mathsf{F}] \\
&\overset{⑤}{=} \gamma\|\partial_{\mathcal{M}}\mathcal{K}(\mathbf{I}_k; \mathbf{X}^t, B^t)\|_\mathsf{F}
\end{aligned}
\tag{59}
$$

where step ① uses the definition of $\partial_{\mathcal{M}}F(\mathbf{X}^t)$); step ② uses $[\mathbf{X}^t]^\mathsf{T}\mathbf{X}^t = \mathbf{I}_r$; step ③ uses the inequality that $\|\mathbf{A}\mathbf{X}\|_\mathsf{F}^2 \leq \|\mathbf{A}\|_\mathsf{F}^2$ for all $\mathbf{X} \in \text{St}(n, r)$; step ④ uses Equality (58); step ⑤ uses the definition of $\partial_{\mathcal{M}}\mathcal{K}(\mathbf{I}_k; \mathbf{X}^t, B^t)$.

$\square$

## G.3 PROOF OF THEOREM 4.6

*Proof.* We derive the following results:

$$
\begin{aligned}
\mathbb{E}_{\xi^T}[\text{dist}^2(\mathbf{0}, \partial_{\mathcal{M}}F(\mathbf{X}^{T+1}))] &\overset{①}{=} \gamma^2 \cdot \mathbb{E}_{\xi^{T+1}}[\text{dist}^2(\mathbf{0}, \partial_{\mathcal{M}}\mathcal{K}(\mathbf{I}_k; \mathbf{X}^{T+1}, B^{T+1}))] \\
&\overset{②}{\leq} \gamma^2 \cdot \phi^2 \cdot \mathbb{E}_{\xi^T}[\|\bar{\mathbf{V}}^T - \mathbf{I}_k\|_\mathsf{F}^2] \\
&\overset{③}{\leq} \gamma^2 \cdot \phi^2 \cdot \tfrac{\tilde{c}}{T+1},
\end{aligned}
$$

where step ① uses Lemma 4.4(*b*); step ② uses Lemma 4.4(*a*); step ③ uses Inequality (43).

Therefore, we conclude that there exists an index $\bar{t}$ with $\bar{t} \in [0, T]$ such that the associated solution $\mathbf{X}^{\bar{t}}$ qualifies as an $\epsilon$-*critical point* of Problem (1) satisfying $\mathbb{E}_{\xi^{\bar{t}}}[\text{dist}^2(\mathbf{0}, \partial_{\mathcal{M}}F(\mathbf{X}^{\bar{t}+1}))] \leq \epsilon$, provided that $T$ is sufficiently large to ensure $\gamma^2 \cdot \phi^2 \cdot \tfrac{\tilde{c}}{T+1} \leq \epsilon$.

$\square$

### G.4 PROOF OF THEOREM 4.10

*Proof.* Initially, given $F^\circ(\mathbf{X}) \triangleq F(\mathbf{X}) + \mathcal{I}_{\mathcal{M}}(\mathbf{X})$ is a KL function by our assumption, we can conclude, from Proposition 4.9, that:

$$\frac{1}{\varphi'(F^\circ(\mathbf{X}^t) - F^\circ(\ddot{\mathbf{X}}))} \leq \text{dist}(0, \partial F^\circ(\mathbf{X}^t)). \tag{60}$$

Since $\varphi(\cdot)$ is a concave desingularization function, we have: $\varphi(b) + (a-b)\varphi'(a) \leq \varphi(a)$. Applying the inequality above with $a = F(\mathbf{X}^t) - F(\ddot{\mathbf{X}})$ and $b = F(\mathbf{X}^{t+1}) - F(\ddot{\mathbf{X}})$, we have:

$$
\begin{aligned}
&(F(\mathbf{X}^t) - F(\mathbf{X}^{t+1}))\varphi'(F(\mathbf{X}^t) - F(\ddot{\mathbf{X}})) \\
\leq \quad &\mathcal{E}^t \triangleq \varphi(F(\mathbf{X}^t) - F(\ddot{\mathbf{X}})) - \varphi(F(\mathbf{X}^{t+1}) - F(\ddot{\mathbf{X}})).
\end{aligned} \tag{61}
$$

**Part (a)**. We define $\varphi^t \triangleq \varphi(F(\mathbf{X}^t) - F(\ddot{\mathbf{X}}))$. We derive the following inequalities:

$$
\begin{aligned}
(e^{t+1})^2 \triangleq \mathbb{E}_{\xi^t}[\|\bar{\mathbf{V}}^t - \mathbf{I}_k\|_{\mathsf{F}}^2] \quad &\overset{①}{\leq} \quad \frac{2}{\alpha} \cdot \mathbb{E}_{\xi^t}[F(\mathbf{X}^t) - F(\mathbf{X}^{t+1})] \\
&\overset{②}{\leq} \quad \frac{2}{\alpha} \cdot \mathbb{E}_{\xi^t}\left[\frac{\mathcal{E}^t}{\varphi'(F(\mathbf{X}^t) - F(\ddot{\mathbf{X}}))}\right] \\
&\overset{③}{\leq} \quad \frac{2}{\alpha} \cdot \mathbb{E}_{\xi^t}[\mathcal{E}^t \cdot \text{dist}(0, \partial F^\circ(\mathbf{X}^t))] \\
&\overset{④}{\leq} \quad \frac{2}{\alpha} \cdot \mathbb{E}_{\xi^t}[\mathcal{E}^t \cdot \|\partial_{\mathcal{M}} F(\mathbf{X}^t)\|_{\mathsf{F}}] \\
&\overset{⑤}{\leq} \quad \frac{2}{\alpha} \cdot \mathbb{E}_{\xi^t}[\mathcal{E}^t \gamma \|\partial_{\mathcal{M}} \mathcal{K}(\mathbf{I}_k; \mathbf{X}^t, \mathsf{B}^t)\|_{\mathsf{F}}] \\
&\overset{⑥}{\leq} \quad \frac{2}{\alpha} \cdot \mathbb{E}_{\xi^{t-1}}[\mathcal{E}^t \gamma \phi \|\bar{\mathbf{V}}^{t-1} - \mathbf{I}_k\|_{\mathsf{F}}] \\
&\overset{⑦}{=} \quad \underbrace{\frac{2}{\alpha} \cdot \gamma \phi}_{\triangleq \kappa} \cdot (\varphi^t - \varphi^{t+1}) \cdot e^t,
\end{aligned}
$$

where step ① uses the sufficient decrease condition as shown in Theorem 4.2; step ② uses Inequality (61); step ③ uses Inequality (60); step ④ uses Lemma A.7; step ⑤ uses Inequality (59); step ⑥ uses Lemma 4.4; step ⑦ uses the definitions of $\{\kappa, \varphi^t, e^t, \mathcal{E}^t\}$.

**Part (b)**. Applying Lemma A.10 with $p^t = \kappa \varphi^t$ with $p^t \geq p^{t+1}$, for all $i \geq 1$, we have:

$$\sum_{j=i}^{\infty} e^{j+1} \leq e^i + 2p^i.$$

Using the definition of $d^t \triangleq \sum_{j=t}^{\infty} e^{j+1}$ and letting $i = t$, we obtain:

$$d^t \leq e^t + 2p^t \overset{①}{=} e^t + 2\kappa\varphi^t \overset{②}{\leq} e^t + 2\kappa\varphi^1 \overset{③}{\leq} 2\sqrt{k} + 2\kappa\varphi^1,$$

where step ① uses $p^t = \kappa\varphi^t$; step ② uses $\varphi^t \leq \varphi^1$; step ③ uses $e^t \triangleq \mathbb{E}_{\xi^{t-1}}[\|\bar{\mathbf{V}}^{t-1} - \mathbf{I}_k\|_{\mathsf{F}}] \leq \mathbb{E}_{\xi^{t-1}}[\|\bar{\mathbf{V}}^{t-1}\|_{\mathsf{F}}] + \|\mathbf{I}_k\|_{\mathsf{F}} \leq \sqrt{k} + \sqrt{k}$. We conclude that $d^t \triangleq \sum_{j=t}^{\infty} e^{j+1}$ is always upper-bounded.

Using the fact that $\|\mathbf{X}^{t+1} - \mathbf{X}^t\|_{\mathsf{F}}^2 \leq \|\bar{\mathbf{V}}^t - \mathbf{I}_k\|_{\mathsf{F}}^2$ as shown in Lemma 2.2(*c*), we conclude that $\sum_{i=1}^{\infty} \mathbb{E}_{\xi^i}[\|\mathbf{X}^{i+1} - \mathbf{X}^i\|_{\mathsf{F}}]$ is also always upper-bounded.

$\square$

### G.5 PROOF OF THEOREM 4.11

*Proof.* We define $\varphi^t \triangleq \varphi(s^t)$, where $s^t \triangleq F(\mathbf{X}^t) - F(\ddot{\mathbf{X}})$.

We define $e^{t+1} \triangleq \mathbb{E}_{\xi^t}[\|\bar{\mathbf{V}}^t - \mathbf{I}_k\|_{\mathsf{F}}]$, and $d^i = \sum_{j=i}^{\infty} e^{j+1}$.

First, we have:

$$
\begin{aligned}
\|\mathbf{X}^T - \mathbf{X}^\infty\|_{\mathsf{F}} &\stackrel{①}{\leq} \sum_{j=T}^{\infty} \|\mathbf{X}^j - \mathbf{X}^{j+1}\|_{\mathsf{F}} \\
&\stackrel{②}{\leq} \sum_{j=T}^{\infty} \|\bar{\mathbf{V}}^j - \mathbf{I}_k\|_{\mathsf{F}} \\
&\stackrel{③}{=} \sum_{j=T}^{\infty} e^{j+1} \\
&\stackrel{④}{=} d^T,
\end{aligned}
$$

where step ① uses the triangle inequality; step ② uses $\|\mathbf{X}^{t+1} - \mathbf{X}^t\|_{\mathsf{F}}^2 \leq \|\bar{\mathbf{V}}^t - \mathbf{I}_k\|_{\mathsf{F}}^2$, as shown in Lemma 2.2(*c*); step ③ uses the definition of $e^{t+1}$; step ④ uses the definition of $d^T$. Therefore, it suffices to establish the convergence rate of $d^T$.

Second, we obtain the following results:

$$
\begin{aligned}
\frac{1}{\varphi'(s^t)} &\stackrel{①}{\leq} \|\mathrm{dist}(\mathbf{0}, \partial F^\circ(\mathbf{X}^t))\|_{\mathsf{F}} \\
&\stackrel{②}{\leq} \|\partial_{\mathcal{M}} F(\mathbf{X}^t)\|_{\mathsf{F}} \\
&\stackrel{③}{\leq} \mathbb{E}_{\xi^t}[\gamma\|\partial_{\mathcal{M}}\mathcal{K}(\mathbf{I}_k; \mathbf{X}^t, \mathrm{B}^t)\|_{\mathsf{F}} \\
&\stackrel{④}{\leq} \mathbb{E}_{\xi^t}[\gamma\phi\|\bar{\mathbf{V}}^{t-1} - \mathbf{I}_k\|_{\mathsf{F}} \\
&\stackrel{⑤}{\leq} \gamma\phi e^t,
\end{aligned}
$$

where step ① uses uses Proposition 4.9 that $\mathrm{dist}(\mathbf{0}, \partial F^\circ(\mathbf{X}'))\varphi'(F^\circ(\mathbf{X}') - F^\circ(\ddot{\mathbf{X}})) \geq 1$; step ② uses Lemma A.7; step ③ uses Inequality (59); step ④ uses the Riemannian subgradient lower bound for the iterates gap in Lemma 4.4; step ⑤ uses the definition of $e^t \triangleq \mathbb{E}_{\xi^{t-1}}[\|\bar{\mathbf{V}}^{t-1} - \mathbf{I}_k\|_{\mathsf{F}}^2]$.

Third, using the definition of $d^t$, we derive:

$$
\begin{aligned}
d^t &\triangleq \sum_{i=t}^{\infty} e^{i+1} \\
&\stackrel{①}{\leq} e^t + 2\kappa\varphi^t \\
&\stackrel{②}{=} e^t + 2\kappa c \cdot \{[s^t]^\sigma\}^{\frac{1-\sigma}{\sigma}} \\
&\stackrel{③}{=} e^t + 2\kappa c \cdot \{c(1-\sigma) \cdot \frac{1}{\varphi'(s^t)}\}^{\frac{1-\sigma}{\sigma}} \\
&\stackrel{④}{=} e^t + 2\kappa c \cdot \{c(1-\sigma) \cdot \gamma\phi e^t\}^{\frac{1-\sigma}{\sigma}} \\
&\stackrel{⑤}{=} d^{t-1} - d^t + 2\kappa c \cdot \{c(1-\sigma) \cdot \gamma\phi(d^{t-1} - d^t)\}^{\frac{1-\sigma}{\sigma}} \\
&= d^{t-1} - d^t + \underbrace{2\kappa c \cdot [c(1-\sigma)\gamma\phi]^{\frac{1-\sigma}{\sigma}}}_{\triangleq \ddot{\kappa}} \cdot \{d^{t-1} - d^t\}^{\frac{1-\sigma}{\sigma}},
\end{aligned}
\tag{62}
$$

where step ① uses $\sum_{i=t}^{\infty} e^{i+1} \leq e^t + 2\kappa\varphi^t$, as shown in Theorem 4.10(*b*); step ② uses the definitions that $\varphi^t \triangleq \varphi(s^t)$, and $\varphi(s) = cs^{1-\sigma}$; step ③ uses $\varphi'(s) = c(1-\sigma) \cdot [s]^{-\sigma}$, leading to $[s^t]^\sigma = c(1-\sigma) \cdot \frac{1}{\varphi'(s^t)}$; step ④ uses Inequality (62); step ⑤ uses the fact that $e^t = d^{t-1} - d^t$.

We consider three cases for $\sigma \in [0, 1)$.

**Part (a)**. We consider $\sigma = 0$. We have from Inequality (62):

$$
\begin{aligned}
0 &\leq -\frac{1}{\varphi'(s^t)} + \gamma\phi e^t \\
&\stackrel{①}{=} -\frac{1}{c(1-\sigma) \cdot [s^t]^{-\sigma}} + \gamma\phi e^t \\
&\stackrel{②}{=} -\frac{1}{c} + \gamma\phi(d^{t-1} - d^t) \\
&\stackrel{③}{\leq} -\frac{1}{c} + \gamma\phi d^{t-1},
\end{aligned}
$$

where step ① uses $\varphi'(s) = c(1 - \sigma) \cdot [s]^{-\sigma}$; step ② uses $\sigma = 0$ and $e^t = d^{t-1} - d^t$; step ③ uses $-d^t \leq 0$.

Since $d^t \to 0$, and $\gamma, \phi, c > 0$, this results in a contradiction. Therefore, there exists $t'$ such that $d^t = 0$ for all $t > t'$, ensuring that the algorithm terminates in a finite number of steps.

**Part (b)**. We consider $\sigma \in (0, \frac{1}{2}]$. We let $t' \triangleq \{i \,|\, d^{i-1} - d^i \leq 1\}$. For all $t \geq t'$, we have from Inequality (62):

$$
\begin{aligned}
d^t &\leq & d^{t-1} - d^t + (d^{t-1} - d^t)^{\frac{1-\sigma}{\sigma}} \cdot \ddot{\kappa} \\
&\overset{①}{\leq} & d^{t-1} - d^t + (d^{t-1} - d^t) \cdot \ddot{\kappa} \\
&\leq & d^{t-1} \cdot \tfrac{\ddot{\kappa}+1}{\ddot{\kappa}+2},
\end{aligned}
\tag{63}
$$

where step ① uses the fact that $[\Delta^{(1-\sigma)/\sigma}]/\Delta = \Delta^{(1-2\sigma)/\sigma} = \Delta^{(1/\sigma-2)} \leq \Delta^0 = 1$ for all $\Delta = d^{t-1} - d^t \in [0, 1]$ and $\sigma \in (0, \frac{1}{2}]$. Therefore, we have:

$$
d^T \leq d^1 \cdot \left(\tfrac{\ddot{\kappa}+1}{\ddot{\kappa}+2}\right)^{T-1}.
$$

**Part (c)**. We consider $\sigma \in (\frac{1}{2}, 1)$. We define $w \triangleq \frac{1-\sigma}{\sigma} \in (0, 1)$, and $\tau \triangleq 1/w - 1 \in (0, \infty)$.

We let $R$ be any positive constant such that $e^t \leq R$ for all $t \geq 1$.

For all $t \geq 2$, we have from Inequality (62):

$$
\begin{aligned}
d^t &\leq & d^{t-1} - d^t + \ddot{\kappa} \cdot (d^{t-1} - d^t)^{\frac{1-\sigma}{\sigma}} \\
&\overset{①}{=} & \ddot{\kappa}(d^{t-1} - d^t)^w + (d^{t-1} - d^t)^w \cdot (e^t)^{1-w} \\
&\overset{②}{\leq} & \ddot{\kappa}(d^{t-1} - d^t)^w + (d^{t-1} - d^t)^w \cdot R^{1-w} \\
&= & (d^{t-1} - d^t)^w \cdot \underbrace{(\ddot{\kappa} + R^{1-w})}_{\triangleq \dot{\kappa}},
\end{aligned}
$$

where step ① uses the definition of $w$ and the fact that $d^{t-1} - d^t = e^t$; step ② uses the fact that $\max_{x \in (0,R]} x^{1-w} \leq R^{1-w}$ if $w \in (0, 1)$ and $R > 0$. We further obtain:

$$
\underbrace{[d^t]^{1/w}}_{=[d^t]^{\tau+1}} \leq (d^{t-1} - d^t) \cdot \dot{\kappa}^{1/w}.
$$

Applying Lemma A.11 with $a = \dot{\kappa}^{1/w}$, we have:

$$
d^T \leq \mathcal{O}(T^{-1/\tau}) \overset{①}{=} \mathcal{O}(T^{-\frac{1}{1/w-1}}) \overset{②}{=} \mathcal{O}(T^{-\frac{1}{\frac{\sigma}{1-\sigma}-1}}) = \mathcal{O}(T^{-\frac{1-\sigma}{2\sigma-1}}),
$$

where step ① uses $\tau \triangleq 1/w - 1$; step ② uses $w \triangleq \frac{1-\sigma}{\sigma}$.

$\square$

# H PROOF FOR SECTION 5

## H.1 PROOF OF LEMMA 5.1

*Proof.* We define $w \triangleq c - e$. We define $\breve{F}(\tilde{c}, \tilde{s}) \triangleq a\tilde{c} + b\tilde{s} + c\tilde{c}^2 + d\tilde{c}\tilde{s} + e\tilde{s}^2 + h(\tilde{c}\mathbf{x} + \tilde{s}\mathbf{y})$.

Initially, using $\sin^2(\theta) = 1 - \cos^2(\theta)$, we obtain the following problem, which is equivalent to Problem (11):

$$
\bar{\theta} \in \arg\min_{\theta} a\cos(\theta) + b\sin(\theta) + w\cos^2(\theta) + d\cos(\theta)\sin(\theta) + e + h(\cos(\theta)\mathbf{x} + \sin(\theta)\mathbf{y}). \tag{64}
$$

We assume $\cos(\theta) \neq 0$. Using Lemma A.8, we consider the two cases for $(\cos(\theta), \sin(\theta))$ in Problem (64).

**Case a).** $\cos(\theta) = \frac{1}{\sqrt{1+\tan^2(\theta)}}$, and $\sin(\theta) = \frac{\tan(\theta)}{\sqrt{1+\tan^2(\theta)}}$. Problem (11) reduces to:

$$\bar{\theta}_+ \in \arg\min_\theta \frac{a+\tan(\theta)b}{\sqrt{1+\tan^2(\theta)}} + \frac{w+\tan(\theta)d}{1+\tan^2(\theta)} + h(\frac{\mathbf{x}+\tan(\theta)\mathbf{y}}{\sqrt{1+\tan^2(\theta)}}).$$

Defining $t = \tan(\theta)$, we have the following equivalent problem:

$$\bar{t}_+ \in \arg\min_t \frac{a+bt}{\sqrt{1+t^2}} + \frac{w+dt}{1+t^2} + h(\frac{\mathbf{x}+\mathbf{y}t}{\sqrt{1+t^2}}).$$

Therefore, the optimal solution $\bar{\theta}_+$ can be computed as:

$$\cos(\bar{\theta}_+) = \frac{1}{\sqrt{1+(\bar{t}_+)^2}}, \ \sin(\bar{\theta}_+) = \frac{\bar{t}_+}{\sqrt{1+(\bar{t}_+)^2}}. \tag{65}$$

**Case b).** $\cos(\theta) = \frac{-1}{\sqrt{1+\tan(\theta)^2}}$, and $\sin(\theta) = \frac{-\tan(\theta)}{\sqrt{1+\tan(\theta)^2}}$. Problem (11) boils down to:

$$\bar{\theta}_- \in \arg\min_\theta \frac{-a-\tan(\theta)b}{\sqrt{1+\tan(\theta)^2}} + \frac{w+\tan(\theta)d}{1+\tan(\theta)^2} + h(\frac{-\mathbf{x}-\tan(\theta)\mathbf{y}}{\sqrt{1+\tan(\theta)^2}}).$$

Defining $t = \tan(\theta)$, we have the following equivalent problem:

$$\bar{t}_- \in \arg\min_t \frac{-a-bt}{\sqrt{1+t^2}} + \frac{w+dt}{1+t^2} + h(\frac{-\mathbf{x}-\mathbf{y}t}{\sqrt{1+t^2}}).$$

Therefore, the optimal solution $\bar{\theta}_-$ can be computed as:

$$\cos(\bar{\theta}_-) = \frac{-1}{\sqrt{1+(\bar{t}_-)^2}}, \ \sin(\bar{\theta}_-) = \frac{-\bar{t}_-}{\sqrt{1+(\bar{t}_-)^2}} \tag{66}$$

In view of (65) and (66), when $\cos(\theta) \neq 0$, the optimal solution $\bar{\theta}$ for Problem (64) is computed as: $[\cos(\bar{\theta}), \sin(\bar{\theta})] \in \arg\min_{c,s} \breve{F}(c,s), \ s.t. \ [c,s] \in \{[\cos(\bar{\theta}_+), \sin(\bar{\theta}_+)], [\cos(\bar{\theta}_-), \sin(\bar{\theta}_-)]\}$. Taking into account the case when $\cos(\theta) = 0$, the optimal solution $\bar{\theta}$ for Problem (64) is computed as:

$$[\cos(\bar{\theta}), \sin(\bar{\theta})] \in \arg\min_{c,s} \breve{F}(c,s),$$

$$s.t. \ [c,s] \in \{[\cos(\bar{\theta}_+), \sin(\bar{\theta}_+)], [\cos(\bar{\theta}_-), \sin(\bar{\theta}_-)], [0,1], [0,-1]\}.$$

Notably, $\{\cos(\bar{\theta}), \sin(\bar{\theta})\}$ uniquely determines $\bar{\theta}$. Moreover, since the objective function in Problem (11) solely depends on $\{\cos(\theta), \sin(\theta)\}$, computing the exact values of $\bar{\theta}_+$ for (65) and $\bar{\theta}_-$ for (66) is unnecessary.

$\square$

# I PROOF FOR APPENDIX SECTION D

## I.1 PROOF OF LEMMA D.1

*Proof.* **(a)** The proof is similar to that of Theorem 3.6. We omit the proof for brevity.

**(b)** Note that the matrix $\mathbf{S}$ is an anti-symmetric matrix with $\mathbf{S} = -\mathbf{S}^\mathsf{T}$ and $\mathrm{diag}(\mathbf{S}) = \mathbf{0}$. By observing $[\bar{i}, \bar{j}] = \arg\max_{i\in[n], j\in[n], i\neq j} |\mathbf{S}_{ij}|$, we can conclude that:

$$\mathbf{S}(\bar{i}, \bar{j}) = 0 \ \Leftrightarrow \ \mathbf{S} = \mathbf{0}.$$

$\square$

## I.2 PROOF OF THEOREM D.2

*Proof.* We define $\mathtt{B} = [i,j]$.

We define $c_1 \triangleq \mathbf{T}_{ii} + \mathbf{T}_{jj}$, $c_2 \triangleq \mathbf{T}_{ij} - \mathbf{T}_{ji}$, $c_3 \triangleq \mathbf{T}_{jj} - \mathbf{T}_{ii}$ and $c_4 \triangleq \mathbf{T}_{ij} + \mathbf{T}_{ji}$.

**(a)** We now focus on the following optimization problem

$$\mathbf{S}_{ij} = \min_{\mathbf{V} \in \mathrm{St}(2,2)} \langle \mathbf{V} - \mathbf{I}_2, \mathbf{T}_{\mathrm{BB}} \rangle. \tag{67}$$

We consider two cases for $\mathbf{V} \in \mathrm{St}(2,2)$.

**Case a).** When $\mathbf{V}$ is a rotation matrix with $\mathbf{V} = \mathbf{V}_\theta^{\mathrm{rot}} = \begin{bmatrix} \cos(\theta) & \sin(\theta) \\ -\sin(\theta) & \cos(\theta) \end{bmatrix}$ for some suitable $\theta$, we have:

$$
\begin{aligned}
&\min_{\mathbf{V} \in \mathrm{St}(k,k)} \langle \mathbf{V} - \mathbf{I}_2, \mathbf{T}_{\mathrm{BB}} \rangle \\
&= \min_\theta \langle \begin{bmatrix} \cos(\theta) - 1 & \sin(\theta) \\ -\sin(\theta) & \cos(\theta) - 1 \end{bmatrix}, \begin{bmatrix} \mathbf{T}_{ii} & \mathbf{T}_{ij} \\ \mathbf{T}_{ji} & \mathbf{T}_{jj} \end{bmatrix} \rangle. \\
&= \min_\theta \cos(\theta)(\mathbf{T}_{ii} + \mathbf{T}_{jj}) + \sin(\theta)(\mathbf{T}_{ij} - \mathbf{T}_{ji}) - (\mathbf{T}_{ii} + \mathbf{T}_{jj}) \\
&\overset{\text{①}}{=} -\sqrt{(\mathbf{T}_{ii} + \mathbf{T}_{jj})^2 + (\mathbf{T}_{ij} - \mathbf{T}_{ji})^2} - (\mathbf{T}_{ii} + \mathbf{T}_{jj}) \\
&\overset{\text{②}}{=} -\sqrt{c_1^2 + c_2^2} - c_1. 
\end{aligned} \tag{68}
$$

where step ① uses Lemma A.9 with $A = \mathbf{T}_{ii} + \mathbf{T}_{jj}$ and $B = \mathbf{T}_{ij} - \mathbf{T}_{ji}$; step ② uses the definition of $c_1 \triangleq \mathbf{T}_{ii} + \mathbf{T}_{jj}$ and $c_2 \triangleq \mathbf{T}_{ij} - \mathbf{T}_{ji}$.

**Case b).** When $\mathbf{V}$ is a reflection matrix with $\mathbf{V} = \mathbf{V}_\theta^{\mathrm{ref}} = \begin{bmatrix} -\cos(\theta) & \sin(\theta) \\ \sin(\theta) & \cos(\theta) \end{bmatrix}$ for some suitable $\theta$, we have:

$$
\begin{aligned}
&\min_{\mathbf{V} \in \mathrm{St}(k,k)} \langle \mathbf{V} - \mathbf{I}_k, \mathbf{T}_{\mathrm{BB}} \rangle \\
&= \min_\theta \langle \begin{bmatrix} -\cos(\theta) - 1 & \sin(\theta) \\ \sin(\theta) & \cos(\theta) - 1 \end{bmatrix}, \begin{bmatrix} \mathbf{T}_{ii} & \mathbf{T}_{ij} \\ \mathbf{T}_{ji} & \mathbf{T}_{jj} \end{bmatrix} \rangle. \\
&= \min_\theta \cos(\theta)(\mathbf{T}_{jj} - \mathbf{T}_{ii}) + \sin(\theta)(\mathbf{T}_{ij} + \mathbf{T}_{ji}) - (\mathbf{T}_{ii} + \mathbf{T}_{jj}) \\
&\overset{\text{①}}{=} -\sqrt{(\mathbf{T}_{jj} - \mathbf{T}_{ii})^2 + (\mathbf{T}_{ij} + \mathbf{T}_{ji})^2} - (\mathbf{T}_{ii} + \mathbf{T}_{jj}) \\
&\overset{\text{②}}{=} -\sqrt{c_3^2 + c_4^2} - c_1. 
\end{aligned} \tag{69}
$$

where step ① uses Lemma A.9 with $A = \mathbf{T}_{jj} - \mathbf{T}_{ii}$ and $B = \mathbf{T}_{ij} + \mathbf{T}_{ji}$; step ② uses the definition of $c_3 \triangleq \mathbf{T}_{jj} - \mathbf{T}_{ii}$, $c_4 \triangleq \mathbf{T}_{ij} + \mathbf{T}_{ji}$, and $c_1 \triangleq \mathbf{T}_{ii} + \mathbf{T}_{jj}$.

In view of Equations (67), (68), and (69), we have:

$$\mathbf{S}_{ij} = \min(-\sqrt{c_1^2 + c_2^2} - c_1, -\sqrt{c_3^2 + c_4^2} - c_1).$$

**(b)** We note that $h(\mathbf{X}) = 0$ for all $\mathbf{X}$ based on our assumption. If $\mathbf{X}^t$ is not a critical point, then the matrix $\mathbf{G}^t[\mathbf{X}^t]^{\mathsf{T}} \in \mathbb{R}^{n \times n}$ is not symmetric, and the matrix $\mathbf{T} \triangleq \mathbf{G}^t[\mathbf{X}^t]^{\mathsf{T}} - L\mathbf{X}^t[\mathbf{X}^t]^{\mathsf{T}} - \alpha\mathbf{I}_n$ is also not symmetric. There exists $\mathrm{B} = [i, j]$ with $i \neq j$ such that $\mathbf{T}_{ij} \neq \mathbf{T}_{ji}$, and $c_2 \triangleq \mathbf{T}_{ij} - \mathbf{T}_{ji} \neq 0$. Consequently, $\mathbf{S}_{ij} = \min(w_1, w_2)$ becomes *strictly negative*, as $w_1 = -c_1 - \sqrt{c_1^2 + c_2^2} < 0$. Since the pair $[\bar{i}, \bar{j}] \in \arg\min_{i,j} \mathbf{S}(i, j)$ is chosen, we conclude that $\mathbf{S}(\bar{i}, \bar{j}) < 0$.

We now prove that a strict decrease is guaranteed with $F(\mathbf{X}^{t+1}) < F(\mathbf{X}^t)$ for **OBCD** if $\mathbf{X}^t$ is not a critical point. We define $\mathcal{X}_{\mathrm{B}}^t(\mathbf{V}) \triangleq \mathbf{X}^t + \mathrm{U}_{\mathrm{B}}(\mathbf{V} - \mathbf{I}_k)\mathrm{U}_{\mathrm{B}}^{\mathsf{T}}\mathbf{X}^t$. Since $\bar{\mathbf{V}}^t$ is the global optimal solution of the problem $\bar{\mathbf{V}}^t \in \arg\min_{\mathbf{V} \in \mathrm{St}(k,k)} \mathcal{K}(\mathbf{V}; \mathbf{X}^t, \mathrm{B})$, we have:

$$
\begin{aligned}
&\tfrac{1}{2}\|\bar{\mathbf{V}}^t - \mathbf{I}_k\|_{\mathbf{Q}+\alpha\mathbf{I}}^2 + \langle \bar{\mathbf{V}}^t - \mathbf{I}_k, [\nabla f(\mathbf{X}^t)(\mathbf{X}^t)^{\mathsf{T}}]_{\mathrm{BB}} \rangle \\
&\leq \tfrac{1}{2}\|\mathbf{V} - \mathbf{I}_k\|_{\mathbf{Q}+\alpha\mathbf{I}}^2 + \langle \mathbf{V} - \mathbf{I}_k, [\nabla f(\mathbf{X}^t)(\mathbf{X}^t)^{\mathsf{T}}]_{\mathrm{BB}} \rangle, \ \forall \mathbf{V} \in \mathrm{St}(k,k). 
\end{aligned} \tag{70}
$$

We derive the following inequalities:

$$F(\mathbf{X}^{t+1}) - F(\mathbf{X}^t) = f(\mathcal{X}_{\mathrm{B}}^t(\mathbf{V})) - f(\mathbf{X}^t)$$

$$\overset{①}{\le} \langle \bar{\mathbf{V}}^t - \mathbf{I}_k, [\nabla f(\mathbf{X}^t)(\mathbf{X}^t)^\mathsf{T}]_{\mathrm{BB}} + \tfrac{1}{2}\|\bar{\mathbf{V}}^t - \mathbf{I}_k\|_{\mathbf{Q}+\alpha\mathbf{I}}^2$$

$$\overset{②}{\le} \tfrac{1}{2}\|\mathbf{V} - \mathbf{I}_k\|_{\mathbf{Q}+\alpha\mathbf{I}}^2 + \langle \mathbf{V} - \mathbf{I}_k, [\nabla f(\mathbf{X}^t)(\mathbf{X}^t)^\mathsf{T}]_{\mathrm{BB}}\rangle, \ \forall \mathbf{V} \in \mathrm{St}(k,k).$$

$$\overset{③}{\le} \min_{\mathbf{V} \in \mathrm{St}(k,k)} \underbrace{\left( \tfrac{1}{2}\|\mathcal{X}_{\mathrm{B}}^t(\mathbf{V}) - \mathbf{X}^t\|_{\mathbf{H}}^2 + \tfrac{\alpha}{2}\|\mathbf{V} - \mathbf{I}_k\|_{\mathsf{F}}^2 + \langle \mathbf{V} - \mathbf{I}_k, [\nabla f(\mathbf{X}^t)(\mathbf{X}^t)^\mathsf{T}]_{\mathrm{BB}}\rangle \right)}_{\triangleq \Xi(\mathbf{V})}, \quad (71)$$

where step ① uses Inequality (10); step ② uses Inequality (70); step ③ uses the fact that $\tfrac{1}{2}\|\mathbf{V} - \mathbf{I}_k\|_{\mathbf{Q}}^2 = \tfrac{1}{2}\|\mathcal{X}_{\mathrm{B}}^t(\mathbf{V}) - \mathbf{X}^t\|_{\mathbf{H}}^2$.

We now prove that the right-hand side of (71) is consistently negative with the following inequalities:

$$\min_{\mathbf{V} \in \mathrm{St}(k,k)} \Xi(\mathbf{V})$$

$$\overset{①}{\le} \min_{\mathbf{V} \in \mathrm{St}(k,k)} \tfrac{L_f}{2}\|\mathcal{X}_{\mathrm{B}}^t(\mathbf{V}) - \mathbf{X}^t\|_{\mathsf{F}}^2 + \langle \mathbf{V} - \mathbf{I}_k, [\nabla f(\mathbf{X}^t)(\mathbf{X}^t)^\mathsf{T}]_{\mathrm{BB}} - \alpha\mathbf{I}_k\rangle$$

$$\overset{②}{=} \min_{\mathbf{V} \in \mathrm{St}(k,k)} L_f\langle \mathbf{I}_k - \mathbf{V}, \mathrm{U}_{\mathrm{B}}^\mathsf{T}\mathbf{X}^t[\mathbf{X}^t]^\mathsf{T}\mathrm{U}_{\mathrm{B}}\rangle + \langle \mathbf{V} - \mathbf{I}_k, [\nabla f(\mathbf{X}^t)(\mathbf{X}^t)^\mathsf{T}]_{\mathrm{BB}} - \alpha\mathbf{I}_k\rangle$$

$$\overset{③}{=} \min_{\mathbf{V} \in \mathrm{St}(k,k)} \langle \mathbf{V} - \mathbf{I}_k, \mathbf{T}_{\mathrm{BB}}\rangle$$

$$\overset{④}{<} 0,$$

where step ① uses $\tfrac{1}{2}\|\mathcal{X}_{\mathrm{B}}^t(\mathbf{V}) - \mathbf{X}^t\|_{\mathbf{H}}^2 \le \tfrac{L_f}{2}\|\mathcal{X}_{\mathrm{B}}^t(\mathbf{V}) - \mathbf{X}^t\|_{\mathsf{F}}^2$ as $\|\mathbf{H}\|_{\mathsf{sp}} \le L_f$, and the identity $\tfrac{\alpha}{2}\|\mathbf{V} - \mathbf{I}_k\|_{\mathsf{F}}^2 = -\langle \mathbf{V} - \mathbf{I}_k, \alpha\mathbf{I}_k\rangle$, which is due to Claim (**c**) of Lemma 2.2; step ② uses Claim (**b**) of Lemma 2.2; step ③ uses the definition of $\mathbf{T} \triangleq (\mathbf{G}^t - L_f\mathbf{X}^t)(\mathbf{X}^t)^\mathsf{T} - \alpha\mathbf{I}_n$ in Algorithm 2; step ④ uses Claim (**a**) of this theorem.

□

## J   ADDITIONAL EXPERIMENTS

In this section, we present the experimental results of the proposed **OBCD** algorithm on the three tasks, namely $\ell_0$ norm-based SPCA, $\ell_1$ norm-based SPCA, and Nonnegative PCA using different working set selection strategies.

### J.1   APPLICATIONS TO $\ell_0$ NORM-BASED SPCA, NONNEGATIVE PCA, AND $\ell_1$ NORM-BASED SPCA

Since we have introduced $\ell_0$ norm-based SPCA in Section 6, we now present nonnegative PCA and $\ell_1$ norm-based SPCA.

▶ **Nonnegative PCA** Nonnegative PCA is an extension of PCA that imposes nonnegativity constraints on the principal vector (Zass & Shashua, 2006; Qian et al., 2021). This constraint leads to a nonnegative representation of loading vectors and it helps to capture data locality in feature selection. Nonnegative PCA can formulated as:

$$\min_{\mathbf{X} \in \mathrm{St}(n,r)} -\tfrac{1}{2}\langle \mathbf{X}, \mathbf{C}\mathbf{X}\rangle, \ s.t. \ \mathbf{X} \ge \mathbf{0},$$

where $\mathbf{C} \in \mathbb{R}^{n \times n}$ is the covariance matrix of the data.

▶ $L_1$ **Norm-based SPCA.** As the $L_1$ norm provides the tightest convex relaxation for the $L_0$-norm over the unit ball in the sense of $L_\infty$-norm, some researchers replace the non-convex and discontinuous $L_0$ norm function with a convex but non-smooth function (Chen et al., 2016; Vu et al., 2013; Lu & Zhang, 2012). This leads to the following optimization problem of $L_1$ norm-based SPCA:

$$\min_{\mathbf{X} \in \mathrm{St}(n,r)} -\tfrac{1}{2}\langle \mathbf{X}, \mathbf{C}\mathbf{X}\rangle + \lambda\|\mathbf{X}\|_1,$$

where $\mathbf{C} \in \mathbb{R}^{n \times n}$ is the covariance matrix of the data, and $\lambda > 0$.

## J.2 EXPERIMENT SETTING

We compare the objective values $(F(\mathbf{X}) - F_{\min})$ for different methods after running $t$ seconds with $t$ varying from 20 to 60, where the constant $F_{\min}$ denotes the smallest objective of all the methods.

**Initializations**. We use the same initializations for all methods. *(i)* For the $\ell_0$ and $\ell_1$ norm-based SPCA tasks, since the optimal solutions are expected to be sparse, we simply set $\mathbf{X}^0 \in \mathrm{St}(n, r)$ to an identity matrix with $\mathbf{X}^0_{ij} = 1$ if $i = j$ and otherwise 0. *(ii)* For the nonnegative PCA task, we use a random nonnegative orthogonal matrix as $\mathbf{X}^0$, which can be generated using the following strategy. We first randomly and uniformly partition the index vector $[1, 2, ..., n]$ into $r$ nonempty groups $\{\mathcal{G}_i\}_{i=1}^r$ with $\mathcal{G}_i$ being the index vector for the $i$-th group, then we set $\mathbf{X}^0(\mathcal{G}_i, i) = \frac{1}{|\mathcal{G}_i|}$ for all $i \in [r]$, where $|\mathcal{G}_i|$ is the number of elements for the $i$-th group.

**Variants of OBCD**. We consider three variants of **OBCD** using different working set selection strategies: *(i)* **OBCD-R** that uses a simple random strategy; *(ii)* **OBCD-CV** that uses a greedy strategy based on maximum stationarity violation pair, and *(iii)* **OBCD-OR** that uses a greedy strategy based on objective reduction violation pair. We only consider $|\mathbb{B}| = k = 2$. In order to solve the subproblem for the $\ell_0$ norm-based SPCA, $\ell_1$ norm-based SPCA, and nonnegative PCA tasks, we use a breakpoint searching method as presented in Section 5 and Section B.

## J.3 EXPERIMENT RESULTS ON THREE TASKS

▶ **$\ell_0$ Norm-based Sparse PCA**

We compare **OBCD** against two state-of-the-art methods: *(i)* Linearized Alternating Direction Method of Multiplier (LADMM) (Lai & Osher, 2014) and *(ii)* Smoothing Penalty Method (SPM) (Lai & Osher, 2014; Chen, 2012). We also initialize **OBCD-R** with the result of LADMM (or SPM) and ran it for 10 seconds to evaluate its effectiveness in improving the solution, leading to **LADMM+OBCD-R** (or **SPM+OBCD-R**). To compute the subgradient $\mathbf{G}^t \in \partial F(\mathbf{X}^t)$ at $\mathbf{X}^t$ for Algorithm 2, we choose $\mathbf{G}^t = -\mathbf{C}\mathbf{X}^t + \mathbf{0}$ as $\mathbf{0}$ is the subgradient of the function $\lambda\|\mathbf{X}\|_0$.

Figure 3 shows the convergence curve of the compared methods with $\lambda = 100$. Table 2, 3, and 4 show the objective values $(F(\mathbf{X}) - F_{\min})$ for different methods with varying $\lambda \in \{1, 300, 1000\}$. Several conclusions can be drawn. *(i)* Due to the use of greedy strategy, **OBCD-CV** and **OBCD-OR** often lead to faster convergence than **OBCD-R** for this task. *(ii)* **OBCD-R** often greatly improves upon LADMM and SPM; this is because our methods find stronger stationary points than LADMM and SPM. *(iii)* The proposed methods generally deliver the best performance.

▶ **Nonnegative PCA**

We compare **OBCD** against two state-of-the-art methods: *(i)* Linearized Alternating Direction Method of Multiplier (LADMM) (He & Yuan, 2012; Lai & Osher, 2014) and *(ii)* Smoothing Penalty Method (SPM). We also initialize **OBCD-R** with the result of LADMM (or SPM) and ran it for 10 seconds to evaluate its effectiveness in improving the solution, leading to **LADMM+OBCD-R** (or **SPM+OBCD-R**). To compute the subgradient $\mathbf{G}^t \in \partial F(\mathbf{X}^t)$ at $\mathbf{X}^t$ for Algorithm 2, we choose $\mathbf{G}^t = -\mathbf{C}\mathbf{X}^t + \mathbf{0}$ as $\mathbf{0}$ is the subgradient of $\mathcal{I}_{\geq 0}(\mathbf{X}^t)$.

Table 5 shows the comparisons of objective values and the violation of the constraints $(F(\mathbf{X}) - F_{\min}, \| \min(\mathbf{0}, \mathbf{X})\|_\mathsf{F} + \|\mathbf{X}^\mathsf{T}\mathbf{X} - \mathbf{I}_r\|_\mathsf{F})$ for different methods. Two conclusions can be drawn. *(i)* **OBCD-CV** and **OBCD-OR** are not as effective as **OBCD-R** in this task. This is because the matrix $\mathbf{0}$ may not be a suitable choice for the subgradient for the nonsmooth function $\mathcal{I}_{\geq 0}(\mathbf{X})$. *(ii)* Feasibility of our methods is achieved with $\| \min(\mathbf{0}, \mathbf{X})\|_\mathsf{F} + \|\mathbf{X}^\mathsf{T}\mathbf{X} - \mathbf{I}_r\|_\mathsf{F} \leq 10^{-12}$. This is because **OBCD** is a feasible method. *(iii)* The proposed methods generally give the best performance. *(iv)* **OBCD-R** often greatly improve upon LADMM and SPM, as our methods find stronger stationary points than LADMM and SPM.

▶ **$\ell_1$ Norm-based Sparse PCA**

We compare **OBCD** against the following state-of-the-art methods: (i) Linearized Alternating Direction Method of Multiplier (LADMM) (He & Yuan, 2012); (ii) ADMM (Lai & Osher, 2014);

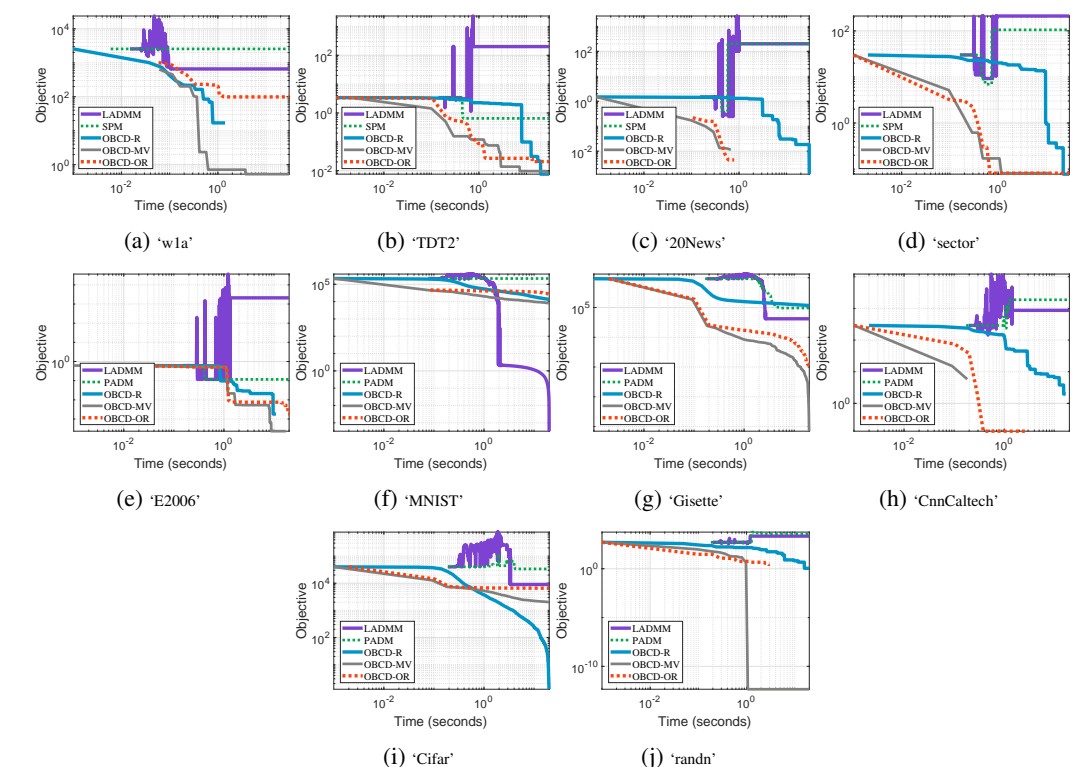

Figure 3: The convergence curve of the compared methods for solving $L_0$ norm-based SPCA with $\lambda = 100$.

(iii) Riemannian Subgradient Method (SubGrad) (Li et al., 2021); (iv) Manifold Proximal Gradient Method (ManPG) (Chen et al., 2020). We also initialize **OBCD-R** with the result of LADMM (or ManPG) and ran it for 10 seconds to evaluate its effectiveness in improving the solution, leading to **LADMM+OBCD-R** (or **ManPG+OBCD-R**). To compute the subgradient $\mathbf{G}^t \in \partial F(\mathbf{X}^t)$ at $\mathbf{X}^t$ for Algorithm 2, we choose $\mathbf{G}^t = -\mathbf{C}\mathbf{X}^t + \lambda \text{sign}(\mathbf{X}^t)$ as $\text{sign}(\mathbf{X})$ is the subgradient of $\|\mathbf{X}\|_1$.

Table 6, 7, and 8 show the comparisons of objective values $(F(\mathbf{X}) - F_{\min})$ for different methods with varying $\lambda \in \{1, 100, 1000\}$. Several conclusions can be drawn. *(i)* ManPG is generally faster than LADMM, ADMM and SubGrad. This is consistent with the reported results in (Chen et al., 2020). *(ii)* **OBCD-OR** outperforms the other methods {LADMM, ADMM, SubGrad, ManPG} by achieving lower objective values. *(iii)* **OBCD-OR** often greatly improve upon the LADMM and SPM.

| data-m-n | $F_{\min}$ | r | LADMM | SPM | OBCD-R | OBCD-SV | OBCD-OR | LADMM + OBCD-R | SPM +OBCD-R |
|---|---|---|---|---|---|---|---|---|---|
| $\lambda = 1.00$, time limit=10 | | | | | | | | | |
| w1a-2477-300 | -6.0e+03 | 20 | 8.70e+02 | 4.98e+03 | 3.67e+01 | 3.39e+02 | 0.00e+00 | 3.45e+02(+) | 7.40e+01(+) |
| TDT2-500-1000 | 1.6e+01 | 20 | 3.75e+00 | 3.75e+00 | 1.85e-02 | 2.06e-03 | 0.00e+00 | 1.85e-02(+) | 1.85e-02(+) |
| 20News-8000-1000 | 1.8e+01 | 20 | 1.66e+00 | 1.66e+00 | 2.19e-02 | 0.00e+00 | 0.00e+00 | 2.19e-02(+) | 2.19e-02(+) |
| sector-6412-1000 | -2.6e+01 | 20 | 4.27e+01 | 4.27e+01 | 1.65e+00 | 0.00e+00 | 2.60e-01 | 1.65e+00(+) | 1.65e+00(+) |
| E2006-2000-1000 | 1.9e+01 | 20 | 6.36e-01 | 6.36e-01 | 5.00e-04 | 0.00e+00 | 3.40e-03 | 5.00e-04(+) | 5.00e-04(+) |
| MNIST-60000-784 | -3.4e+05 | 20 | 2.40e+04 | 3.41e+05 | 4.09e+04 | 0.00e+00 | 5.63e+03 | 1.46e+04(+) | 4.09e+04(+) |
| Gisette-3000-1000 | -1.1e+06 | 20 | 2.39e+04 | 2.28e+04 | 4.49e+04 | 0.00e+00 | 3.18e+04 | 1.47e+04(+) | 1.38e+04(+) |
| CnnCaltech-3000-1000 | -2.4e+03 | 20 | 8.26e+01 | 1.16e+03 | 7.44e+01 | 1.35e-01 | 5.04e+02 | 0.00e+00(+) | 1.07e+01(+) |
| Cifar-1000-1000 | -1.4e+05 | 20 | 6.99e+02 | 6.56e+01 | 1.04e+04 | 2.23e+03 | 5.39e+03 | 6.56e+02 | 0.00e+00(+) |
| randn-500-1000 | -1.7e+04 | 20 | 1.90e-03 | 1.18e+03 | 7.60e+03 | 7.01e+03 | 6.49e+03 | 0.00e+00(+) | 1.17e+03 |
| w1a-2477-300 | -8.6e+03 | 50 | 2.92e+03 | 2.46e+03 | 0.00e+00 | 7.11e+02 | 2.60e+02 | 1.99e+02(+) | 2.73e+02(+) |
| TDT2-500-1000 | 4.6e+01 | 50 | 4.28e+00 | 4.28e+00 | 2.06e-02 | 3.43e-03 | 0.00e+00 | 9.60e-03(+) | 2.06e-02(+) |
| 20News-8000-1000 | 4.8e+01 | 50 | 8.56e-01 | 8.56e-01 | 5.04e-03 | 8.04e-04 | 0.00e+00 | 5.04e-03(+) | 5.04e-03(+) |
| sector-6412-1000 | -1.5e+01 | 50 | 5.99e+01 | 5.99e+01 | 1.18e-02 | 0.00e+00 | 4.60e-02 | 1.18e-02(+) | 1.18e-02(+) |
| E2006-2000-1000 | 4.9e+01 | 50 | 6.64e-01 | 6.64e-01 | 2.13e-04 | 0.00e+00 | 2.51e-04 | 2.13e-04(+) | 2.13e-04(+) |
| MNIST-60000-784 | -3.5e+05 | 50 | 3.09e+04 | 3.28e+04 | 5.88e+04 | 0.00e+00 | 8.81e+02 | 2.21e+04(+) | 2.32e+04(+) |
| Gisette-3000-1000 | -1.1e+06 | 50 | 2.85e+04 | 2.47e+04 | 7.78e+04 | 0.00e+00 | 2.56e+04 | 8.31e+03(+) | 3.41e+03(+) |
| CnnCaltech-3000-1000 | -2.9e+03 | 50 | 4.22e+02 | 1.57e+03 | 1.35e+02 | 0.00e+00 | 5.89e+02 | 1.49e+02(+) | 1.26e+02(+) |
| Cifar-1000-1000 | -1.4e+05 | 50 | 3.34e+04 | 4.67e+01 | 9.69e+03 | 2.88e+03 | 5.36e+03 | 3.34e+03 | 0.00e+00(+) |
| randn-500-1000 | -3.8e+04 | 50 | 2.26e-03 | 3.33e+03 | 1.99e+04 | 1.74e+04 | 1.75e+04 | 0.00e+00(+) | 3.33e+03 |
| w1a-2477-300 | -1.2e+04 | 100 | 3.52e+03 | 3.26e+03 | 0.00e+00 | 4.81e+02 | 4.81e+02 | 7.65e+02(+) | 3.06e+02(+) |
| TDT2-500-1000 | 9.5e+01 | 100 | 4.45e+00 | 4.45e+00 | 0.00e+00 | 1.10e-02 | 2.06e-02 | 0.00e+00(+) | 0.00e+00(+) |
| 20News-8000-1000 | 9.8e+01 | 100 | 9.25e-01 | 9.25e-01 | 0.00e+00 | 7.54e-04 | 8.98e-04 | 0.00e+00(+) | 0.00e+00(+) |
| sector-6412-1000 | 1.6e+01 | 100 | 7.42e+01 | 7.42e+01 | 0.00e+00 | 8.65e-01 | 8.88e-02 | 0.00e+00(+) | 0.00e+00(+) |
| E2006-2000-1000 | 9.9e+01 | 100 | 6.71e-01 | 6.71e-01 | 1.05e-05 | 0.00e+00 | 2.33e-04 | 1.05e-05(+) | 1.05e-05(+) |
| MNIST-60000-784 | -3.5e+05 | 100 | 1.93e+04 | 2.08e+04 | 4.40e+04 | 0.00e+00 | 5.29e+03 | 7.33e+03(+) | 8.46e+03(+) |
| Gisette-3000-1000 | -1.1e+06 | 100 | 4.86e+04 | 7.20e+04 | 8.01e+04 | 0.00e+00 | 1.29e+04 | 1.28e+04(+) | 3.15e+04(+) |
| CnnCaltech-3000-1000 | -3.5e+03 | 100 | 9.37e+02 | 1.98e+03 | 1.29e+01 | 1.57e+01 | 5.08e+02 | 5.26e+02(+) | 0.00e+00(+) |
| Cifar-1000-1000 | -1.4e+05 | 100 | 2.12e+04 | 1.96e+04 | 7.03e+03 | 0.00e+00 | 3.38e+03 | 2.12e+04 | 1.95e+04 |
| randn-500-1000 | -6.7e+04 | 100 | 2.03e-03 | 3.53e+04 | 3.42e+04 | 3.03e+04 | 2.98e+04 | 0.00e+00(+) | 3.53e+04 |
| $\lambda = 1.00$, time limit=20 | | | | | | | | | |
| w1a-2477-300 | -6.0e+03 | 20 | 8.75e+02 | 4.99e+03 | 0.00e+00 | 4.22e+02 | 3.74e+01 | 3.26e+02(+) | 6.99e+01(+) |
| TDT2-500-1000 | 1.6e+01 | 20 | 3.75e+00 | 3.75e+00 | 1.03e-02 | 0.00e+00 | 3.43e-03 | 1.10e-02(+) | 1.10e-02(+) |
| 20News-8000-1000 | 1.8e+01 | 20 | 1.66e+00 | 1.66e+00 | 7.62e-05 | 0.00e+00 | 0.00e+00 | 1.08e-02(+) | 1.08e-02(+) |
| sector-6412-1000 | -2.5e+01 | 20 | 4.25e+01 | 4.25e+01 | 8.84e-01 | 0.00e+00 | 6.83e-02 | 9.95e-01(+) | 9.95e-01(+) |
| E2006-2000-1000 | 1.9e+01 | 20 | 6.38e-01 | 6.38e-01 | 0.00e+00 | 1.20e-03 | 4.92e-04 | 1.73e-03(+) | 1.73e-03(+) |
| MNIST-60000-784 | -3.4e+05 | 20 | 2.63e+04 | 3.43e+05 | 2.03e+04 | 0.00e+00 | 5.73e+03 | 1.47e+04(+) | 3.43e+04(+) |
| Gisette-3000-1000 | -1.1e+06 | 20 | 2.82e+04 | 2.71e+04 | 3.02e+04 | 0.00e+00 | 2.27e+04 | 1.82e+04(+) | 1.71e+04(+) |
| CnnCaltech-3000-1000 | -2.5e+03 | 20 | 1.69e+02 | 1.25e+03 | 5.65e+01 | 0.00e+00 | 4.84e+02 | 7.67e+01(+) | 7.64e+01(+) |
| Cifar-1000-1000 | -1.4e+05 | 20 | 7.09e+02 | 7.31e+01 | 9.22e+03 | 1.34e+03 | 5.01e+03 | 6.61e+02 | 0.00e+00(+) |
| randn-500-1000 | -1.7e+04 | 20 | 2.02e-03 | 1.18e+03 | 7.29e+03 | 6.95e+03 | 6.42e+03 | 0.00e+00(+) | 1.17e+03 |
| w1a-2477-300 | -8.6e+03 | 50 | 2.98e+03 | 2.52e+03 | 0.00e+00 | 5.80e+02 | 7.02e+01 | 2.93e+02(+) | 3.06e+02(+) |
| TDT2-500-1000 | 4.6e+01 | 50 | 4.29e+00 | 4.29e+00 | 0.00e+00 | 2.74e-03 | 4.80e-03 | 1.17e-02(+) | 1.17e-02(+) |
| 20News-8000-1000 | 4.8e+01 | 50 | 8.55e-01 | 8.55e-01 | 2.79e-04 | 0.00e+00 | 0.00e+00 | 1.76e-03(+) | 1.76e-03(+) |
| sector-6412-1000 | -1.5e+01 | 50 | 6.01e+01 | 6.01e+01 | 5.54e-02 | 0.00e+00 | 3.70e-02 | 6.44e-02(+) | 6.44e-02(+) |
| E2006-2000-1000 | 4.9e+01 | 50 | 6.64e-01 | 6.64e-01 | 5.06e-05 | 0.00e+00 | 0.00e+00 | 2.55e-04(+) | 2.55e-04(+) |
| MNIST-60000-784 | -3.6e+05 | 50 | 3.95e+04 | 4.14e+04 | 4.59e+04 | 0.00e+00 | 6.36e+03 | 3.06e+04(+) | 3.16e+04(+) |
| Gisette-3000-1000 | -1.1e+06 | 50 | 3.24e+04 | 2.86e+04 | 7.02e+04 | 0.00e+00 | 2.50e+04 | 1.23e+04(+) | 7.33e+03(+) |
| CnnCaltech-3000-1000 | -2.9e+03 | 50 | 3.67e+02 | 1.51e+03 | 0.00e+00 | 2.46e+01 | 4.73e+02 | 9.37e+01(+) | 7.52e+01(+) |
| Cifar-1000-1000 | -1.4e+05 | 50 | 3.35e+03 | 4.67e+01 | 8.98e+03 | 1.69e+03 | 5.10e+03 | 3.34e+03 | 0.00e+00(+) |
| randn-500-1000 | -3.8e+04 | 50 | 2.25e-03 | 3.33e+03 | 1.98e+04 | 1.76e+04 | 1.74e+04 | 0.00e+00(+) | 3.33e+03 |
| w1a-2477-300 | -1.2e+04 | 100 | 3.54e+03 | 3.28e+03 | 0.00e+00 | 5.14e+02 | 2.95e+02 | 6.96e+02(+) | 3.23e+02(+) |
| TDT2-500-1000 | 9.5e+01 | 100 | 4.45e+00 | 4.45e+00 | 0.00e+00 | 0.00e+00 | 2.06e-02 | 3.43e-03(+) | 3.43e-03(+) |
| 20News-8000-1000 | 9.8e+01 | 100 | 9.26e-01 | 9.26e-01 | 5.67e-04 | 0.00e+00 | 1.10e-02 | 1.40e-03(+) | 1.40e-03(+) |
| sector-6412-1000 | 1.6e+01 | 100 | 7.45e+01 | 7.45e+01 | 0.00e+00 | 1.28e-01 | 2.76e-01 | 2.50e-01(+) | 2.50e-01(+) |
| E2006-2000-1000 | 9.9e+01 | 100 | 6.72e-01 | 6.72e-01 | 3.58e-05 | 0.00e+00 | 1.00e-04 | 3.60e-04(+) | 3.60e-04(+) |
| MNIST-60000-784 | -3.6e+05 | 100 | 2.70e+04 | 2.85e+04 | 4.69e+04 | 0.00e+00 | 6.40e+03 | 1.51e+04(+) | 1.61e+04(+) |
| Gisette-3000-1000 | -1.1e+06 | 100 | 4.51e+04 | 4.27e+04 | 7.71e+04 | 0.00e+00 | 1.38e+04 | 8.70e+03(+) | 3.50e+03(+) |
| CnnCaltech-3000-1000 | -3.6e+03 | 100 | 6.82e+02 | 2.01e+03 | 0.00e+00 | 1.14e+01 | 4.44e+02 | 2.79e+02(+) | 2.52e+01(+) |
| Cifar-1000-1000 | -1.4e+05 | 100 | 8.07e+03 | 2.79e+03 | 7.05e+03 | 0.00e+00 | 3.69e+03 | 8.06e+03 | 2.73e+03 |
| randn-500-1000 | -6.7e+04 | 100 | 2.04e-03 | 7.83e+03 | 3.41e+04 | 3.01e+04 | 3.02e+04 | 0.00e+00(+) | 7.83e+03 |
| $\lambda = 1.00$, time limit=30 | | | | | | | | | |
| w1a-2477-300 | -6.1e+03 | 20 | 8.78e+02 | 4.99e+03 | 0.00e+00 | 4.78e+02 | 1.39e+02 | 3.18e+02(+) | 5.09e+01(+) |
| TDT2-500-1000 | 1.6e+01 | 20 | 3.75e+00 | 3.75e+00 | 6.86e-03 | 0.00e+00 | 0.00e+00 | 1.10e-02(+) | 1.10e-02(+) |
| 20News-8000-1000 | 1.8e+01 | 20 | 1.66e+00 | 1.66e+00 | 0.00e+00 | 0.00e+00 | 0.00e+00 | 1.08e-02(+) | 1.08e-02(+) |
| sector-6412-1000 | -2.6e+01 | 20 | 4.27e+01 | 4.27e+01 | 0.00e+00 | 0.00e+00 | 1.42e-14 | 1.19e+00(+) | 1.19e+00(+) |
| E2006-2000-1000 | 1.9e+01 | 20 | 6.38e-01 | 6.38e-01 | 0.00e+00 | 3.36e-04 | 3.36e-04 | 1.73e-03(+) | 1.73e-03(+) |
| MNIST-60000-784 | -3.4e+05 | 20 | 2.63e+04 | 3.43e+05 | 1.29e+04 | 0.00e+00 | 2.74e+03 | 1.49e+04(+) | 3.61e+04(+) |
| Gisette-3000-1000 | -1.1e+06 | 20 | 2.35e+04 | 2.24e+04 | 2.06e+04 | 0.00e+00 | 1.41e+04 | 1.34e+04(+) | 1.25e+04(+) |
| CnnCaltech-3000-1000 | -2.5e+03 | 20 | 1.43e+02 | 1.22e+03 | 0.00e+00 | 1.53e+01 | 3.15e+02 | 5.35e+01(+) | 4.84e+01(+) |
| Cifar-1000-1000 | -1.4e+05 | 20 | 7.10e+02 | 7.32e+01 | 8.90e+03 | 1.71e+03 | 4.79e+03 | 6.61e+02 | 0.00e+00(+) |
| randn-500-1000 | -1.7e+04 | 20 | 1.99e-03 | 1.18e+03 | 7.15e+03 | 6.91e+03 | 6.73e+03 | 0.00e+00(+) | 1.17e+03 |
| w1a-2477-300 | -8.6e+03 | 50 | 2.99e+03 | 2.53e+03 | 0.00e+00 | 6.30e+02 | 3.24e+01 | 2.98e+02(+) | 3.56e+02(+) |
| TDT2-500-1000 | 4.6e+01 | 50 | 4.29e+00 | 4.29e+00 | 0.00e+00 | 3.43e-03 | 3.43e-03 | 1.23e-02(+) | 1.44e-02(+) |
| 20News-8000-1000 | 4.8e+01 | 50 | 8.55e-01 | 8.55e-01 | 6.77e-05 | 0.00e+00 | 0.00e+00 | 1.76e-03(+) | 1.76e-03(+) |
| sector-6412-1000 | -1.5e+01 | 50 | 6.00e+01 | 6.00e+01 | 3.79e-03 | 0.00e+00 | 1.25e-01 | 4.80e-02(+) | 4.80e-02(+) |
| E2006-2000-1000 | 4.9e+01 | 50 | 6.64e-01 | 6.64e-01 | 0.00e+00 | 1.53e-05 | 0.00e+00 | 8.88e-05(+) | 8.88e-05(+) |
| MNIST-60000-784 | -3.6e+05 | 50 | 4.09e+04 | 4.28e+04 | 3.76e+04 | 0.00e+00 | 5.22e+03 | 3.19e+04(+) | 3.31e+04(+) |
| Gisette-3000-1000 | -1.1e+06 | 50 | 3.45e+04 | 3.07e+04 | 6.56e+04 | 0.00e+00 | 2.24e+04 | 1.43e+04(+) | 9.52e+03(+) |
| CnnCaltech-3000-1000 | -3.0e+03 | 50 | 4.87e+02 | 1.63e+03 | 6.71e+01 | 0.00e+00 | 5.21e+02 | 2.17e+02(+) | 1.93e+02(+) |
| Cifar-1000-1000 | -1.4e+05 | 50 | 3.35e+03 | 4.65e+01 | 8.48e+03 | 1.56e+03 | 4.86e+03 | 3.35e+03 | 0.00e+00(+) |
| randn-500-1000 | -3.8e+04 | 50 | 2.26e-03 | 3.33e+03 | 1.97e+04 | 1.77e+04 | 1.73e+04 | 0.00e+00(+) | 3.33e+03 |
| w1a-2477-300 | -1.2e+04 | 100 | 3.55e+03 | 3.29e+03 | 0.00e+00 | 5.28e+02 | 4.09e+02 | 8.37e+02(+) | 3.17e+02(+) |
| TDT2-500-1000 | 9.5e+01 | 100 | 4.45e+00 | 4.45e+00 | 0.00e+00 | 0.00e+00 | 0.00e+00 | 4.80e-03(+) | 4.80e-03(+) |
| 20News-8000-1000 | 9.8e+01 | 100 | 9.26e-01 | 9.26e-01 | 9.31e-05 | 0.00e+00 | 1.69e-05 | 1.25e-03(+) | 1.25e-03(+) |
| sector-6412-1000 | 1.6e+01 | 100 | 7.45e+01 | 7.45e+01 | 9.80e-03 | 1.50e-03 | 0.00e+00 | 2.72e-01(+) | 2.72e-01(+) |
| E2006-2000-1000 | 9.9e+01 | 100 | 6.72e-01 | 6.72e-01 | 3.81e-05 | 0.00e+00 | 3.56e-04 | 4.16e-04(+) | 4.16e-04(+) |
| MNIST-60000-784 | -3.6e+05 | 100 | 2.64e+04 | 2.80e+04 | 4.42e+04 | 0.00e+00 | 2.36e+03 | 1.46e+04(+) | 1.56e+04(+) |
| Gisette-3000-1000 | -1.1e+06 | 100 | 4.89e+04 | 4.64e+04 | 7.75e+04 | 0.00e+00 | 1.10e+04 | 1.26e+04(+) | 7.26e+03(+) |
| CnnCaltech-3000-1000 | -3.6e+03 | 100 | 7.40e+02 | 2.07e+03 | 2.64e+01 | 0.00e+00 | 5.13e+02 | 3.36e+02(+) | 8.59e+01(+) |
| Cifar-1000-1000 | -1.4e+05 | 100 | 8.91e+03 | 3.56e+03 | 7.52e+03 | 0.00e+00 | 4.47e+03 | 8.90e+03 | 3.49e+03 |
| randn-500-1000 | -6.7e+04 | 100 | 2.03e-03 | 7.83e+03 | 3.41e+04 | 2.97e+04 | 3.02e+04 | 0.00e+00(+) | 7.83e+03 |

Table 2: Comparisons of objective values $(F(\mathbf{X}) - F_{\min})$ of $L_0$ norm-based SPCA for all the compared methods with $\lambda = 1$. The $1^{st}$, $2^{nd}$, and $3^{rd}$ best results are colored with red, green and blue, respectively. If the objective values of 'LADMM+OBCD-R' (or 'SPM+OBCD-R') are smaller than those of 'LADMM' (or 'SPM') by a margin of $0.1 \times a$, where $a$ represents the objective values of 'LADMM' (or 'ManPG'), they will be marked with $(+)$.

| data-m-n | $F_{\min}$ | r | LADMM | SPM | OBCD-R | OBCD-SV | OBCD-OR | LADMM + OBCD-R | SPM +OBCD-R |
|---|---|---|---|---|---|---|---|---|---|
| $\lambda = 300.00$, time limit=10 | | | | | | | | | |
| w1a-2477-300 | 1.3e+03 | 20 | 2.35e+03 | 3.90e+03 | 9.09e-13 | 9.09e-13 | 9.09e-13 | 9.09e-13(+) | 0.00e+00(+) |
| TDT2-500-1000 | 6.0e+03 | 20 | 1.00e+00 | 6.82e-01 | 1.10e-02 | 0.00e+00 | 0.00e+00 | 2.19e-02(+) | 2.19e-02(+) |
| 20News-8000-1000 | 6.0e+03 | 20 | 3.71e-01 | 3.41e-01 | 1.08e-02 | 0.00e+00 | 0.00e+00 | 4.76e-03(+) | 4.76e-03(+) |
| sector-6412-1000 | 6.0e+03 | 20 | 6.19e+02 | 3.05e+02 | 9.27e-01 | 3.06e-01 | 0.00e+00 | 3.01e+02(+) | 3.01e+02 |
| E2006-2000-1000 | 6.0e+03 | 20 | 1.20e+03 | 1.44e-01 | 1.03e-03 | 0.00e+00 | 4.95e-04 | 6.00e+02(+) | 1.03e-03(+) |
| MNIST-60000-784 | -1.8e+05 | 20 | 4.01e+04 | 1.82e+05 | 0.00e+00 | 1.73e+04 | 3.70e+04 | 1.46e+04(+) | 0.00e+00(+) |
| Gisette-3000-1000 | -7.7e+05 | 20 | 2.29e+04 | 4.28e+05 | 1.03e+05 | 0.00e+00 | 1.30e+04 | 1.08e+04(+) | 3.13e+05(+) |
| CnnCaltech-3000-1000 | 5.4e+03 | 20 | 2.59e+03 | 1.42e+04 | 4.63e+00 | 2.62e-01 | 0.00e+00 | 1.15e+03(+) | 9.73e+03(+) |
| Cifar-1000-1000 | 2.0e+03 | 20 | 4.15e+04 | 1.10e+03 | 1.81e+00 | 0.00e+00 | 0.00e+00 | 3.91e+04 | 1.81e+00(+) |
| randn-500-1000 | 1.4e+02 | 20 | 1.62e+04 | 1.51e+04 | 1.69e+01 | 0.00e+00 | 0.00e+00 | 5.09e+03(+) | 3.93e+03(+) |
| w1a-2477-300 | 6.8e+03 | 50 | 5.26e+03 | 4.14e+03 | 3.64e-12 | 3.64e-12 | 3.64e-12 | 0.00e+00(+) | 3.64e-12(+) |
| TDT2-500-1000 | 1.5e+04 | 50 | 9.01e+02 | 1.17e+00 | 7.54e-03 | 0.00e+00 | 6.86e-04 | 3.00e+02(+) | 1.23e-02(+) |
| 20News-8000-1000 | 1.5e+04 | 50 | 2.76e+04 | 3.85e-01 | 1.79e-03 | 0.00e+00 | 5.50e-04 | 1.11e+04(+) | 4.40e-04(+) |
| sector-6412-1000 | 1.5e+04 | 50 | 2.12e+03 | 1.51e+03 | 7.02e-02 | 0.00e+00 | 1.33e-01 | 2.10e+03 | 1.20e+03(+) |
| E2006-2000-1000 | 1.5e+04 | 50 | 2.40e+03 | 1.42e-01 | 0.00e+00 | 1.02e-04 | 2.43e-04 | 2.10e+03(+) | 1.11e-05(+) |
| MNIST-60000-784 | -2.1e+05 | 50 | 6.06e+04 | 1.07e+05 | 0.00e+00 | 1.19e+04 | 3.98e+04 | 1.18e+04(+) | 1.58e+04(+) |
| Gisette-3000-1000 | -7.9e+05 | 50 | 2.18e+05 | 7.30e+05 | 9.65e+04 | 0.00e+00 | 8.62e+03 | 1.89e+05(+) | 1.03e+05(+) |
| CnnCaltech-3000-1000 | 1.4e+04 | 50 | 1.14e+03 | 2.47e+03 | 4.09e-01 | 0.00e+00 | 6.09e-01 | 6.01e+02(+) | 2.09e+03(+) |
| Cifar-1000-1000 | 5.1e+03 | 50 | 3.63e+04 | 2.38e+03 | 9.66e+00 | 0.00e+00 | 3.64e-12 | 3.34e+04 | 9.66e+00(+) |
| randn-500-1000 | 7.6e+02 | 50 | 2.93e+04 | 2.02e+03 | 4.90e+00 | 0.00e+00 | 1.73e+00 | 1.31e+04(+) | 5.58e+02(+) |
| w1a-2477-300 | 1.8e+04 | 100 | 5.87e+03 | 6.01e+03 | 1.17e-05 | 3.00e+00 | 1.17e-05 | 1.17e-05(+) | 0.00e+00(+) |
| TDT2-500-1000 | 3.0e+04 | 100 | 9.90e+03 | 1.20e+03 | 0.00e+00 | 6.86e-04 | 4.12e-03 | 8.70e+03(+) | 6.00e+02(+) |
| 20News-8000-1000 | 3.0e+04 | 100 | 1.20e+04 | 9.01e+02 | 0.00e+00 | 1.02e-03 | 1.80e-03 | 7.20e+03(+) | 9.00e+02 |
| sector-6412-1000 | 3.0e+04 | 100 | 1.23e+04 | 6.02e+03 | 0.00e+00 | 3.82e-01 | 3.60e-01 | 1.05e+04(+) | 3.90e+03(+) |
| E2006-2000-1000 | 3.0e+04 | 100 | 1.17e+04 | 1.31e-01 | 1.51e-04 | 2.44e-04 | 2.79e-04 | 9.90e+03(+) | 0.00e+00(+) |
| MNIST-60000-784 | -2.6e+05 | 100 | 1.01e+05 | 2.25e+05 | 0.00e+00 | 1.79e+04 | 4.50e+04 | 1.92e+04(+) | 3.04e+04(+) |
| Gisette-3000-1000 | -8.3e+05 | 100 | 6.40e+04 | 7.13e+05 | 9.31e+04 | 0.00e+00 | 1.43e+04 | 1.58e+03(+) | 9.92e+04(+) |
| CnnCaltech-3000-1000 | 2.8e+04 | 100 | 1.54e+03 | 4.88e+03 | 0.00e+00 | 9.84e-01 | 5.11e-02 | 6.00e+02(+) | 3.51e+03(+) |
| Cifar-1000-1000 | 1.1e+04 | 100 | 3.43e+04 | 3.82e+03 | 1.33e+01 | 0.00e+00 | 2.24e+00 | 2.84e+04(+) | 1.33e+01(+) |
| randn-500-1000 | 2.1e+03 | 100 | 3.63e+04 | 2.90e+03 | 1.12e+00 | 0.00e+00 | 1.09e+00 | 2.05e+04(+) | 2.67e+02(+) |
| $\lambda = 300.00$, time limit=20 | | | | | | | | | |
| w1a-2477-300 | 1.3e+03 | 20 | 2.35e+03 | 3.90e+03 | 0.00e+00 | 0.00e+00 | 0.00e+00 | 0.00e+00(+) | 0.00e+00(+) |
| TDT2-500-1000 | 6.0e+03 | 20 | 1.00e+00 | 6.82e-01 | 1.03e-02 | 0.00e+00 | 0.00e+00 | 2.19e-02(+) | 2.19e-02(+) |
| 20News-8000-1000 | 6.0e+03 | 20 | 3.71e-01 | 3.41e-01 | 7.62e-05 | 0.00e+00 | 0.00e+00 | 4.76e-03(+) | 4.76e-03(+) |
| sector-6412-1000 | 6.0e+03 | 20 | 6.20e+02 | 3.06e+02 | 1.08e+00 | 0.00e+00 | 0.00e+00 | 3.01e+02(+) | 3.01e+02 |
| E2006-2000-1000 | 6.0e+03 | 20 | 1.20e+03 | 1.45e-01 | 0.00e+00 | 1.61e-04 | 3.36e-04 | 6.00e+02(+) | 1.73e-03(+) |
| MNIST-60000-784 | -1.8e+05 | 20 | 4.54e+04 | 1.87e+05 | 0.00e+00 | 1.28e+04 | 4.01e+04 | 2.09e+04(+) | 5.36e+03(+) |
| Gisette-3000-1000 | -7.7e+05 | 20 | 2.75e+04 | 4.32e+05 | 1.04e+05 | 0.00e+00 | 1.18e+04 | 1.50e+04(+) | 3.16e+05(+) |
| CnnCaltech-3000-1000 | 5.4e+03 | 20 | 2.59e+03 | 1.42e+04 | 3.00e+00 | 2.62e-01 | 0.00e+00 | 1.15e+03(+) | 9.73e+03(+) |
| Cifar-1000-1000 | 2.0e+03 | 20 | 4.15e+04 | 1.10e+03 | 9.33e-01 | 0.00e+00 | 9.09e-13 | 3.91e+04 | 1.81e+00(+) |
| randn-500-1000 | 1.4e+02 | 20 | 1.62e+04 | 1.51e+04 | 6.40e+00 | 0.00e+00 | 0.00e+00 | 5.09e+03(+) | 3.93e+03(+) |
| w1a-2477-300 | 6.8e+03 | 50 | 5.26e+03 | 4.14e+03 | 1.82e-12 | 1.82e-12 | 1.82e-12 | 0.00e+00(+) | 1.82e-12(+) |
| TDT2-500-1000 | 1.5e+04 | 50 | 9.01e+02 | 1.17e+00 | 0.00e+00 | 2.74e-03 | 4.80e-03 | 3.00e+02(+) | 1.65e-02(+) |
| 20News-8000-1000 | 1.5e+04 | 50 | 2.76e+04 | 3.86e-01 | 6.77e-04 | 3.98e-04 | 0.00e+00 | 1.11e+04(+) | 8.13e-04(+) |
| sector-6412-1000 | 1.5e+04 | 50 | 2.12e+03 | 1.51e+03 | 6.12e-02 | 1.66e-01 | 0.00e+00 | 2.10e+03 | 1.20e+03(+) |
| E2006-2000-1000 | 1.5e+04 | 50 | 2.40e+03 | 1.42e-01 | 1.61e-04 | 0.00e+00 | 6.30e-05 | 2.10e+03(+) | 1.95e-04(+) |
| MNIST-60000-784 | -2.2e+05 | 50 | 6.61e+04 | 1.12e+05 | 0.00e+00 | 8.39e+03 | 4.46e+04 | 1.61e+04(+) | 2.12e+04(+) |
| Gisette-3000-1000 | -7.9e+05 | 50 | 2.09e+05 | 7.34e+05 | 9.61e+04 | 0.00e+00 | 8.37e+03 | 1.80e+05(+) | 1.07e+05(+) |
| CnnCaltech-3000-1000 | 1.4e+04 | 50 | 1.14e+03 | 2.47e+03 | 3.68e-01 | 0.00e+00 | 3.68e-01 | 6.01e+02(+) | 2.09e+03(+) |
| Cifar-1000-1000 | 5.1e+03 | 50 | 3.63e+04 | 2.38e+03 | 3.78e+00 | 1.82e-12 | 0.00e+00 | 3.34e+04 | 9.66e+00(+) |
| randn-500-1000 | 7.6e+02 | 50 | 2.93e+04 | 2.02e+03 | 2.34e+00 | 0.00e+00 | 1.31e+00 | 1.28e+04(+) | 5.59e+02(+) |
| w1a-2477-300 | 1.8e+04 | 100 | 5.87e+03 | 6.01e+03 | 3.48e-05 | 3.48e-05 | 3.48e-05 | 3.48e-05(+) | 0.00e+00(+) |
| TDT2-500-1000 | 3.0e+04 | 100 | 9.90e+03 | 1.20e+03 | 0.00e+00 | 6.86e-04 | 2.06e-03 | 8.70e+03(+) | 6.00e+02(+) |
| 20News-8000-1000 | 3.0e+04 | 100 | 1.20e+04 | 9.01e+02 | 5.17e-04 | 0.00e+00 | 9.74e-04 | 7.20e+03(+) | 9.00e+02 |
| sector-6412-1000 | 3.0e+04 | 100 | 1.23e+04 | 6.02e+03 | 2.12e-02 | 0.00e+00 | 3.71e-01 | 1.05e+04(+) | 3.90e+03(+) |
| E2006-2000-1000 | 3.0e+04 | 100 | 1.17e+04 | 1.31e-01 | 6.55e-05 | 0.00e+00 | 2.58e-05 | 9.90e+03(+) | 2.39e-04(+) |
| MNIST-60000-784 | -2.6e+05 | 100 | 1.08e+05 | 2.33e+05 | 0.00e+00 | 1.92e+04 | 4.84e+04 | 2.68e+04(+) | 3.88e+04(+) |
| Gisette-3000-1000 | -8.4e+05 | 100 | 7.16e+04 | 7.20e+05 | 9.50e+04 | 0.00e+00 | 9.73e+03 | 9.05e+04(+) | 1.05e+05(+) |
| CnnCaltech-3000-1000 | 2.8e+04 | 100 | 1.54e+03 | 4.88e+03 | 3.33e-01 | 2.94e-01 | 0.00e+00 | 6.01e+02(+) | 3.52e+03(+) |
| Cifar-1000-1000 | 1.1e+04 | 100 | 3.43e+04 | 3.82e+03 | 4.20e+00 | 0.00e+00 | 4.58e+00 | 2.85e+04(+) | 1.77e+01(+) |
| randn-500-1000 | 2.1e+03 | 100 | 3.63e+04 | 2.91e+03 | 9.40e-01 | 0.00e+00 | 2.20e+00 | 2.05e+04(+) | 2.72e+02(+) |
| $\lambda = 300.00$, time limit=30 | | | | | | | | | |
| w1a-2477-300 | 1.3e+03 | 20 | 2.35e+03 | 3.90e+03 | 0.00e+00 | 0.00e+00 | 0.00e+00 | 0.00e+00(+) | 0.00e+00(+) |
| TDT2-500-1000 | 6.0e+03 | 20 | 1.00e+00 | 6.82e-01 | 6.86e-03 | 0.00e+00 | 3.43e-03 | 2.19e-02(+) | 2.19e-02(+) |
| 20News-8000-1000 | 6.0e+03 | 20 | 3.71e-01 | 3.41e-01 | 0.00e+00 | 0.00e+00 | 0.00e+00 | 4.76e-03(+) | 4.76e-03(+) |
| sector-6412-1000 | 6.0e+03 | 20 | 6.20e+02 | 3.06e+02 | 0.00e+00 | 0.00e+00 | 0.00e+00 | 3.01e+02(+) | 3.01e+02 |
| E2006-2000-1000 | 6.0e+03 | 20 | 1.20e+03 | 1.45e-01 | 0.00e+00 | 3.36e-04 | 0.00e+00 | 6.00e+02(+) | 1.73e-03(+) |
| MNIST-60000-784 | -1.8e+05 | 20 | 4.69e+04 | 1.88e+05 | 0.00e+00 | 2.80e+04 | 4.05e+04 | 2.14e+04(+) | 6.85e+03(+) |
| Gisette-3000-1000 | -7.7e+05 | 20 | 2.71e+04 | 4.32e+05 | 1.01e+05 | 0.00e+00 | 9.38e+03 | 1.53e+04(+) | 3.14e+05(+) |
| CnnCaltech-3000-1000 | 5.4e+03 | 20 | 2.59e+03 | 1.42e+04 | 1.91e+00 | 0.00e+00 | 2.62e-01 | 1.15e+03(+) | 9.73e+03(+) |
| Cifar-1000-1000 | 2.0e+03 | 20 | 4.15e+04 | 1.10e+03 | 1.38e-01 | 0.00e+00 | 0.00e+00 | 3.91e+04 | 1.81e+00(+) |
| randn-500-1000 | 1.4e+02 | 20 | 1.62e+04 | 1.51e+04 | 9.61e-01 | 0.00e+00 | 1.82e-12 | 5.09e+03(+) | 3.93e+03(+) |
| w1a-2477-300 | 6.8e+03 | 50 | 5.26e+03 | 4.14e+03 | 1.82e-12 | 1.82e-12 | 1.82e-12 | 0.00e+00(+) | 1.82e-12(+) |
| TDT2-500-1000 | 1.5e+04 | 50 | 9.01e+02 | 1.17e+00 | 0.00e+00 | 0.00e+00 | 6.86e-03 | 3.00e+02(+) | 1.71e-02(+) |
| 20News-8000-1000 | 1.5e+04 | 50 | 2.76e+04 | 3.86e-01 | 4.66e-04 | 0.00e+00 | 0.00e+00 | 1.11e+04(+) | 8.13e-04(+) |
| sector-6412-1000 | 1.5e+04 | 50 | 2.12e+03 | 1.51e+03 | 2.60e-02 | 0.00e+00 | 1.41e-01 | 2.10e+03 | 1.20e+03(+) |
| E2006-2000-1000 | 1.5e+04 | 50 | 2.40e+03 | 1.42e-01 | 6.30e-05 | 0.00e+00 | 6.30e-05 | 2.10e+03(+) | 1.63e-04(+) |
| MNIST-60000-784 | -2.2e+05 | 50 | 7.01e+04 | 1.16e+05 | 0.00e+00 | 2.20e+04 | 4.82e+04 | 2.15e+04(+) | 2.62e+04(+) |
| Gisette-3000-1000 | -8.0e+05 | 50 | 2.17e+05 | 7.41e+05 | 9.95e+04 | 0.00e+00 | 1.21e+04 | 1.89e+05(+) | 1.13e+05(+) |
| CnnCaltech-3000-1000 | 1.4e+04 | 50 | 1.14e+03 | 2.47e+03 | 3.54e-01 | 0.00e+00 | 0.00e+00 | 6.01e+02(+) | 2.09e+03(+) |
| Cifar-1000-1000 | 5.1e+03 | 50 | 3.63e+04 | 2.38e+03 | 3.65e+00 | 3.64e-12 | 0.00e+00 | 3.34e+04 | 9.66e+00(+) |
| randn-500-1000 | 7.6e+02 | 50 | 2.93e+04 | 2.02e+03 | 7.15e-01 | 9.00e-01 | 0.00e+00 | 1.28e+04(+) | 5.59e+02(+) |
| w1a-2477-300 | 1.8e+04 | 100 | 5.87e+03 | 6.01e+03 | 4.46e-06 | 4.46e-06 | 4.46e-06 | 4.46e-06(+) | 0.00e+00(+) |
| TDT2-500-1000 | 3.0e+04 | 100 | 9.90e+03 | 1.20e+03 | 0.00e+00 | 2.06e-03 | 2.74e-03 | 8.70e+03(+) | 6.00e+02(+) |
| 20News-8000-1000 | 3.0e+04 | 100 | 1.20e+04 | 9.01e+02 | 2.62e-04 | 0.00e+00 | 9.48e-04 | 7.20e+03(+) | 9.00e+02 |
| sector-6412-1000 | 3.0e+04 | 100 | 1.23e+04 | 6.02e+03 | 0.00e+00 | 6.26e-02 | 1.73e-01 | 1.05e+04(+) | 4.20e+03(+) |
| E2006-2000-1000 | 3.0e+04 | 100 | 1.17e+04 | 1.31e-01 | 0.00e+00 | 1.10e-04 | 2.39e-04 | 9.90e+03(+) | 2.42e-04(+) |
| MNIST-60000-784 | -2.6e+05 | 100 | 1.09e+05 | 2.34e+05 | 0.00e+00 | 2.08e+04 | 4.91e+04 | 2.94e+04(+) | 4.34e+04(+) |
| Gisette-3000-1000 | -8.4e+05 | 100 | 7.56e+04 | 7.24e+05 | 9.50e+04 | 0.00e+00 | 1.18e+04 | 1.52e+04(+) | 1.12e+05(+) |
| CnnCaltech-3000-1000 | 2.8e+04 | 100 | 1.54e+03 | 4.88e+03 | 2.36e-01 | 1.54e-02 | 0.00e+00 | 6.02e+02(+) | 3.52e+03(+) |
| Cifar-1000-1000 | 1.1e+04 | 100 | 3.43e+04 | 3.82e+03 | 3.34e+00 | 0.00e+00 | 6.39e-01 | 2.88e+04(+) | 1.97e+01(+) |
| randn-500-1000 | 2.1e+03 | 100 | 3.63e+04 | 2.91e+03 | 8.94e-01 | 0.00e+00 | 3.11e+00 | 2.17e+04(+) | 2.80e+02(+) |

Table 3: Comparisons of objective values $(F(\mathbf{X}) - F_{\min})$ of $L_0$ norm-based SPCA for all the compared methods with $\lambda = 300$. The $1^{st}$, $2^{nd}$, and $3^{rd}$ best results are colored with red, green and blue, respectively. If the objective values of 'LADMM+OBCD-R' (or 'SPM+OBCD-R') are smaller than those of 'LADMM' (or 'SPM') by a margin of $0.1 \times a$, where $a$ represents the objective values of 'LADMM' (or 'ManPG'), they will be marked with $(+)$.

| data-m-n | $F_{\min}$ | r | LADMM | SPM | OBCD-R | OBCD-SV | OBCD-OR | LADMM + OBCD-R | SPM +OBCD-R |
|---|---|---|---|---|---|---|---|---|---|
| $\lambda = 1000.00$, time limit=10 | | | | | | | | | |
| w1a-2477-300 | 1.5e+04 | 20 | 2.64e+03 | 3.90e+03 | 0.00e+00 | 0.00e+00 | 0.00e+00 | 0.00e+00(+) | 0.00e+00(+) |
| TDT2-500-1000 | 2.0e+04 | 20 | 1.00e+03 | 6.82e-01 | 1.85e-02 | 0.00e+00 | 1.71e-02 | 1.00e+03 | 2.95e-02(+) |
| 20News-8000-1000 | 2.0e+04 | 20 | 3.00e+03 | 3.00e+03 | 2.19e-02 | 0.00e+00 | 0.00e+00 | 3.00e+03 | 1.00e+03(+) |
| sector-6412-1000 | 2.0e+04 | 20 | 3.01e+03 | 3.00e+03 | 1.65e+00 | 0.00e+00 | 2.60e-01 | 3.00e+03 | 3.00e+03 |
| E2006-2000-1000 | 2.0e+04 | 20 | 1.00e+03 | 1.15e-01 | 9.69e-04 | 0.00e+00 | 2.75e-04 | 1.00e+03 | 9.69e-04(+) |
| MNIST-60000-784 | -6.3e+04 | 20 | 9.86e+04 | 8.25e+04 | 0.00e+00 | 7.62e+03 | 1.73e+04 | 3.81e+04(+) | 2.88e+02(+) |
| Gisette-3000-1000 | -2.2e+05 | 20 | 6.15e+05 | 2.20e+05 | 1.19e+03 | 0.00e+00 | 1.50e+03 | 5.26e+05(+) | 1.10e+04(+) |
| CnnCaltech-3000-1000 | 1.9e+04 | 20 | 1.41e+04 | 3.09e+04 | 4.63e+00 | 0.00e+00 | 0.00e+00 | 7.94e+03(+) | 1.99e+04(+) |
| Cifar-1000-1000 | 1.6e+04 | 20 | 1.81e+04 | 1.10e+03 | 3.03e+00 | 0.00e+00 | 0.00e+00 | 1.40e+04(+) | 3.03e+00(+) |
| randn-500-1000 | 1.4e+04 | 20 | 2.34e+04 | 5.81e+04 | 1.57e+01 | 0.00e+00 | 1.24e+00 | 1.27e+04(+) | 1.86e+04(+) |
| w1a-2477-300 | 4.2e+04 | 50 | 5.09e+03 | 4.79e+03 | 0.00e+00 | 0.00e+00 | 5.00e-01 | 0.00e+00(+) | 0.00e+00(+) |
| TDT2-500-1000 | 5.0e+04 | 50 | 9.00e+03 | 4.00e+03 | 4.12e-03 | 1.51e-02 | 0.00e+00 | 7.00e+03(+) | 4.00e+03 |
| 20News-8000-1000 | 5.0e+04 | 50 | 2.60e+03 | 2.00e+03 | 2.16e-03 | 3.05e-04 | 0.00e+00 | 8.00e+03(+) | 2.00e+03 |
| sector-6412-1000 | 5.0e+04 | 50 | 7.02e+03 | 3.00e+03 | 0.00e+00 | 7.31e-02 | 7.31e-02 | 5.00e+03(+) | 3.00e+03 |
| E2006-2000-1000 | 5.0e+04 | 50 | 8.00e+03 | 1.00e-01 | 2.92e-04 | 2.11e-04 | 0.00e+00 | 8.00e+03 | 2.79e-04(+) |
| MNIST-60000-784 | -8.6e+04 | 50 | 1.01e+05 | 1.92e+05 | 0.00e+00 | 8.98e+03 | 2.03e+04 | 1.81e+02(+) | 6.62e+04(+) |
| Gisette-3000-1000 | -2.3e+05 | 50 | 7.24e+05 | 2.08e+05 | 1.16e+04 | 0.00e+00 | 2.81e+03 | 8.56e+05 | 1.01e+04(+) |
| CnnCaltech-3000-1000 | 4.9e+04 | 50 | 6.46e+03 | 3.23e+04 | 4.09e-01 | 0.00e+00 | 0.00e+00 | 3.97e+03(+) | 2.39e+04(+) |
| Cifar-1000-1000 | 4.0e+04 | 50 | 2.64e+04 | 2.38e+03 | 8.20e+00 | 0.00e+00 | 0.00e+00 | 2.40e+04 | 8.20e+00(+) |
| randn-500-1000 | 3.6e+04 | 50 | 4.59e+04 | 9.36e+03 | 5.69e+00 | 2.43e+00 | 0.00e+00 | 3.45e+04(+) | 5.89e+03(+) |
| w1a-2477-300 | 8.8e+04 | 100 | 5.95e+03 | 9.30e+03 | 0.00e+00 | 0.00e+00 | 5.00e-01 | 0.00e+00(+) | 2.50e+00(+) |
| TDT2-500-1000 | 1.0e+05 | 100 | 5.00e+03 | 4.00e+03 | 0.00e+00 | 1.37e-02 | 1.92e-02 | 5.00e+03 | 4.00e+03 |
| 20News-8000-1000 | 1.0e+05 | 100 | 1.80e+03 | 1.00e+03 | 0.00e+00 | 1.24e-03 | 2.54e-03 | 1.70e+04 | 1.19e-04(+) |
| sector-6412-1000 | 1.0e+05 | 100 | 2.60e+04 | 2.40e+04 | 0.00e+00 | 6.34e-01 | 4.86e-01 | 1.90e+04(+) | 2.40e+04 |
| E2006-2000-1000 | 1.0e+05 | 100 | 5.20e+04 | 1.07e-01 | 2.16e-04 | 0.00e+00 | 4.25e-04 | 3.80e+04(+) | 1.46e-04(+) |
| MNIST-60000-784 | -1.2e+05 | 100 | 1.65e+05 | 2.02e+05 | 0.00e+00 | 2.03e+04 | 3.03e+04 | 1.14e+04(+) | 7.02e+04(+) |
| Gisette-3000-1000 | -2.4e+05 | 100 | 2.96e+04 | 2.01e+05 | 1.25e+04 | 0.00e+00 | 6.70e+02 | 5.66e+03(+) | 1.38e+04(+) |
| CnnCaltech-3000-1000 | 9.8e+04 | 100 | 6.84e+03 | 1.67e+04 | 0.00e+00 | 1.40e+00 | 1.87e+00 | 5.99e+03(+) | 1.59e+04 |
| Cifar-1000-1000 | 8.1e+04 | 100 | 7.16e+04 | 3.82e+03 | 1.51e+01 | 0.00e+00 | 5.56e+00 | 3.39e+04(+) | 1.51e+01(+) |
| randn-500-1000 | 7.2e+04 | 100 | 1.05e+05 | 8.62e+03 | 3.29e+00 | 0.00e+00 | 4.35e+00 | 9.80e+04 | 3.91e+03(+) |
| $\lambda = 1000.00$, time limit=20 | | | | | | | | | |
| w1a-2477-300 | 1.5e+04 | 20 | 2.64e+03 | 3.90e+03 | 0.00e+00 | 0.00e+00 | 0.00e+00 | 0.00e+00(+) | 0.00e+00(+) |
| TDT2-500-1000 | 2.0e+04 | 20 | 1.00e+03 | 6.82e-01 | 1.03e-02 | 1.51e-02 | 0.00e+00 | 1.00e+03 | 2.19e-02(+) |
| 20News-8000-1000 | 2.0e+04 | 20 | 3.00e+03 | 3.00e+03 | 7.62e-05 | 0.00e+00 | 0.00e+00 | 3.00e+03 | 1.00e+03(+) |
| sector-6412-1000 | 2.0e+04 | 20 | 3.01e+03 | 3.00e+03 | 1.08e+00 | 0.00e+00 | 0.00e+00 | 3.00e+03 | 3.00e+03 |
| E2006-2000-1000 | 2.0e+04 | 20 | 1.00e+03 | 1.16e-01 | 0.00e+00 | 6.66e-04 | 4.34e-04 | 1.00e+03 | 1.73e-03(+) |
| MNIST-60000-784 | -6.6e+04 | 20 | 1.02e+05 | 8.60e+04 | 0.00e+00 | 6.08e+03 | 1.80e+04 | 4.05e+04(+) | 2.39e+03(+) |
| Gisette-3000-1000 | -2.2e+05 | 20 | 6.09e+05 | 2.21e+05 | 0.00e+00 | 2.07e+03 | 1.52e+03 | 5.20e+05(+) | 1.13e+04(+) |
| CnnCaltech-3000-1000 | 1.9e+04 | 20 | 1.41e+04 | 3.09e+04 | 3.00e+00 | 0.00e+00 | 0.00e+00 | 7.94e+03(+) | 1.89e+04(+) |
| Cifar-1000-1000 | 1.6e+04 | 20 | 1.81e+04 | 1.10e+03 | 9.33e-01 | 0.00e+00 | 0.00e+00 | 1.00e+04(+) | 1.81e+00(+) |
| randn-500-1000 | 1.4e+04 | 20 | 2.34e+04 | 5.81e+04 | 6.92e+00 | 0.00e+00 | 1.82e-12 | 1.27e+04(+) | 1.47e+04(+) |
| w1a-2477-300 | 4.2e+04 | 50 | 5.09e+03 | 4.79e+03 | 0.00e+00 | 0.00e+00 | 0.00e+00 | 0.00e+00(+) | 0.00e+00(+) |
| TDT2-500-1000 | 5.0e+04 | 50 | 9.00e+03 | 4.00e+03 | 0.00e+00 | 0.00e+00 | 4.80e-03 | 7.00e+03(+) | 4.00e+03 |
| 20News-8000-1000 | 5.0e+04 | 50 | 2.60e+04 | 2.00e+03 | 5.76e-04 | 3.73e-04 | 0.00e+00 | 8.00e+03(+) | 2.00e+03 |
| sector-6412-1000 | 5.0e+04 | 50 | 7.02e+03 | 3.00e+03 | 0.00e+00 | 8.58e-02 | 2.87e-02 | 5.00e+03(+) | 3.00e+03 |
| E2006-2000-1000 | 5.0e+04 | 50 | 8.00e+03 | 1.00e-01 | 1.25e-04 | 0.00e+00 | 2.81e-04 | 8.00e+03 | 3.17e-04(+) |
| MNIST-60000-784 | -9.2e+04 | 50 | 1.06e+05 | 1.17e+05 | 0.00e+00 | 1.03e+04 | 1.68e+04 | 6.41e+03(+) | 1.87e+04(+) |
| Gisette-3000-1000 | -2.3e+05 | 50 | 5.82e+05 | 2.09e+05 | 9.98e+03 | 0.00e+00 | 2.49e+03 | 4.98e+05(+) | 1.22e+04(+) |
| CnnCaltech-3000-1000 | 4.9e+04 | 50 | 6.46e+03 | 1.93e+04 | 3.68e-01 | 0.00e+00 | 0.00e+00 | 3.97e+03(+) | 1.19e+04(+) |
| Cifar-1000-1000 | 4.0e+04 | 50 | 2.64e+04 | 2.38e+03 | 3.78e+00 | 0.00e+00 | 7.28e-12 | 2.40e+04 | 9.66e+00(+) |
| randn-500-1000 | 3.6e+04 | 50 | 4.59e+04 | 9.36e+03 | 2.49e+00 | 5.36e-01 | 0.00e+00 | 3.45e+04(+) | 5.89e+03(+) |
| w1a-2477-300 | 8.8e+04 | 100 | 5.95e+03 | 9.30e+03 | 0.00e+00 | 0.00e+00 | 0.00e+00 | 0.00e+00(+) | 2.50e+00(+) |
| TDT2-500-1000 | 1.0e+05 | 100 | 5.00e+03 | 4.00e+03 | 6.86e-04 | 2.06e-03 | 0.00e+00 | 5.00e+03 | 4.00e+03 |
| 20News-8000-1000 | 1.0e+05 | 100 | 1.80e+03 | 1.00e+03 | 0.00e+00 | 8.04e-04 | 6.10e-04 | 1.70e+04 | 9.48e-04(+) |
| sector-6412-1000 | 1.0e+05 | 100 | 2.60e+04 | 2.40e+04 | 0.00e+00 | 3.08e-01 | 4.34e-01 | 1.90e+04(+) | 2.40e+04 |
| E2006-2000-1000 | 1.0e+05 | 100 | 5.20e+04 | 1.07e-01 | 0.00e+00 | 9.01e-05 | 4.69e-05 | 3.80e+04(+) | 2.54e-04(+) |
| MNIST-60000-784 | -1.2e+05 | 100 | 1.69e+05 | 2.05e+05 | 0.00e+00 | 2.34e+04 | 2.16e+04 | 1.53e+04(+) | 1.09e+04(+) |
| Gisette-3000-1000 | -2.5e+05 | 100 | 3.28e+04 | 2.04e+05 | 1.26e+04 | 0.00e+00 | 1.35e+03 | 8.83e+03(+) | 1.66e+04(+) |
| CnnCaltech-3000-1000 | 9.8e+04 | 100 | 6.84e+03 | 1.67e+04 | 0.00e+00 | 7.36e-02 | 1.20e-01 | 5.99e+03(+) | 1.59e+04 |
| Cifar-1000-1000 | 8.1e+04 | 100 | 7.16e+04 | 3.82e+03 | 4.20e+00 | 0.00e+00 | 4.40e+00 | 3.29e+04(+) | 1.77e+01(+) |
| randn-500-1000 | 7.2e+04 | 100 | 1.05e+05 | 8.63e+03 | 0.00e+00 | 8.01e-01 | 1.94e+00 | 9.80e+04 | 3.91e+03(+) |
| $\lambda = 1000.00$, time limit=30 | | | | | | | | | |
| w1a-2477-300 | 1.5e+04 | 20 | 2.64e+03 | 3.90e+03 | 0.00e+00 | 0.00e+00 | 0.00e+00 | 0.00e+00(+) | 0.00e+00(+) |
| TDT2-500-1000 | 2.0e+04 | 20 | 1.00e+03 | 6.82e-01 | 1.03e-02 | 0.00e+00 | 0.00e+00 | 1.00e+03 | 2.95e-02(+) |
| 20News-8000-1000 | 2.0e+04 | 20 | 3.00e+03 | 3.00e+03 | 7.62e-05 | 0.00e+00 | 0.00e+00 | 3.00e+03 | 1.00e+03(+) |
| sector-6412-1000 | 2.0e+04 | 20 | 3.01e+03 | 3.00e+03 | 8.73e-01 | 0.00e+00 | 0.00e+00 | 3.00e+03 | 3.00e+03 |
| E2006-2000-1000 | 2.0e+04 | 20 | 1.00e+03 | 1.16e-01 | 9.75e-05 | 4.33e-04 | 0.00e+00 | 1.00e+03 | 1.83e-03(+) |
| MNIST-60000-784 | -6.7e+04 | 20 | 1.03e+05 | 8.70e+04 | 0.00e+00 | 1.17e+04 | 2.34e+04 | 4.14e+04(+) | 4.76e+03(+) |
| Gisette-3000-1000 | -2.2e+05 | 20 | 6.09e+05 | 2.22e+05 | 3.02e+02 | 0.00e+00 | 1.81e+03 | 5.21e+05(+) | 1.25e+04(+) |
| CnnCaltech-3000-1000 | 1.9e+04 | 20 | 1.41e+04 | 3.09e+04 | 2.22e+00 | 0.00e+00 | 0.00e+00 | 7.94e+03(+) | 1.99e+04(+) |
| Cifar-1000-1000 | 1.6e+04 | 20 | 1.81e+04 | 1.10e+03 | 1.38e-01 | 1.82e-12 | 0.00e+00 | 1.40e+04(+) | 3.03e+00(+) |
| randn-500-1000 | 1.4e+04 | 20 | 2.34e+04 | 5.81e+04 | 2.17e+00 | 0.00e+00 | 1.82e-12 | 1.37e+04(+) | 1.86e+04(+) |
| w1a-2477-300 | 4.2e+04 | 50 | 5.09e+03 | 4.79e+03 | 0.00e+00 | 0.00e+00 | 0.00e+00 | 0.00e+00(+) | 0.00e+00(+) |
| TDT2-500-1000 | 5.0e+04 | 50 | 9.00e+03 | 4.00e+03 | 0.00e+00 | 7.54e-03 | 6.86e-04 | 7.00e+03(+) | 4.00e+03 |
| 20News-8000-1000 | 5.0e+04 | 50 | 2.60e+04 | 2.00e+03 | 5.33e-04 | 6.27e-04 | 0.00e+00 | 9.00e+03(+) | 2.00e+03 |
| sector-6412-1000 | 5.0e+04 | 50 | 7.02e+03 | 3.00e+03 | 1.83e-02 | 1.00e-01 | 0.00e+00 | 5.00e+03(+) | 3.00e+03 |
| E2006-2000-1000 | 5.0e+04 | 50 | 8.00e+03 | 1.00e-01 | 9.50e-05 | 2.09e-05 | 0.00e+00 | 8.00e+03 | 3.53e-04(+) |
| MNIST-60000-784 | -9.2e+04 | 50 | 1.07e+05 | 1.17e+05 | 0.00e+00 | 7.67e+03 | 1.77e+04 | 7.37e+03(+) | 1.98e+04(+) |
| Gisette-3000-1000 | -2.3e+05 | 50 | 6.32e+05 | 2.10e+05 | 1.02e+04 | 0.00e+00 | 2.81e+03 | 4.04e+05(+) | 1.27e+04(+) |
| CnnCaltech-3000-1000 | 4.9e+04 | 50 | 6.46e+03 | 1.93e+04 | 3.68e-01 | 0.00e+00 | 6.14e-01 | 3.97e+03(+) | 1.39e+04(+) |
| Cifar-1000-1000 | 4.0e+04 | 50 | 2.64e+04 | 2.38e+03 | 3.65e+00 | 0.00e+00 | 7.28e-12 | 2.40e+04 | 1.50e+01(+) |
| randn-500-1000 | 3.6e+04 | 50 | 4.59e+04 | 9.36e+03 | 8.90e-01 | 0.00e+00 | 7.13e-01 | 3.55e+04(+) | 7.86e+03(+) |
| w1a-2477-300 | 8.8e+04 | 100 | 5.95e+03 | 9.30e+03 | 0.00e+00 | 0.00e+00 | 0.00e+00 | 0.00e+00(+) | 9.73e+02(+) |
| TDT2-500-1000 | 1.0e+05 | 100 | 5.00e+03 | 4.00e+03 | 0.00e+00 | 1.37e-03 | 6.86e-04 | 5.00e+03 | 4.00e+03 |
| 20News-8000-1000 | 1.0e+05 | 100 | 1.80e+03 | 1.00e+03 | 2.46e-02 | 0.00e+00 | 2.54e-05 | 1.70e+04 | 1.00e+03 |
| sector-6412-1000 | 1.0e+05 | 100 | 2.60e+04 | 2.40e+04 | 1.89e-02 | 0.00e+00 | 2.12e-01 | 2.00e+04(+) | 2.40e+04 |
| E2006-2000-1000 | 1.0e+05 | 100 | 5.20e+04 | 1.07e-01 | 2.37e-05 | 0.00e+00 | 1.58e-04 | 3.90e+04(+) | 3.69e-04(+) |
| MNIST-60000-784 | -1.2e+05 | 100 | 1.69e+05 | 2.06e+05 | 0.00e+00 | 1.51e+04 | 2.11e+04 | 1.73e+04(+) | 1.39e+04(+) |
| Gisette-3000-1000 | -2.5e+05 | 100 | 3.35e+04 | 2.05e+05 | 1.30e+04 | 0.00e+00 | 2.21e+03 | 9.74e+03(+) | 1.89e+04(+) |
| CnnCaltech-3000-1000 | 9.8e+04 | 100 | 6.84e+03 | 1.67e+04 | 1.49e-01 | 0.00e+00 | 1.14e+00 | 5.99e+03(+) | 1.59e+04 |
| Cifar-1000-1000 | 8.1e+04 | 100 | 7.16e+04 | 3.82e+03 | 3.34e+00 | 0.00e+00 | 0.00e+00 | 3.49e+04(+) | 1.97e+01(+) |
| randn-500-1000 | 7.2e+04 | 100 | 1.05e+05 | 8.63e+03 | 1.45e+00 | 0.00e+00 | 4.08e+00 | 1.00e+05 | 3.92e+03(+) |

Table 4: Comparisons of objective values ($F(\mathbf{X}) - F_{\min}$) of $L_0$ norm-based SPCA for all the compared methods with $\lambda = 1000$. The $1^{st}$, $2^{nd}$, and $3^{rd}$ best results are colored with red, green and blue, respectively. If the objective values of 'LADMM+OBCD-R' (or 'SPM+OBCD-R') are smaller than those of 'LADMM' (or 'SPM') by a margin of $0.1 \times a$, where $a$ represents the objective values of 'LADMM' (or 'ManPG'), they will be marked with $(+)$.

| data-m-n | $F_{\min}$ | r | LADMM | SPM | OBCD-R | OBCD-SV | OBCD-OR | LADMM + OBCD-R | SPM +OBCD-R |
|---|---|---|---|---|---|---|---|---|---|
| | | | | | time limit=10 | | | | |
| w1a-2477-300 | -5.2e+03 | 10 | 5.14e+02, 2e-13 | 1.29e+03, 1e-05 | 2.77e+02, 6e-14 | 5.00e+02, 2e-14 | 1.77e+02, 2e-14 | 0.00e+00, 2e-13(+) | 1.93e+02, 1e-05(+) |
| TDT2-500-1000 | -3.5e+00 | 10 | 9.22e-01, 1e-14 | 9.39e-01, 5e-05 | 1.59e+00, 1e-14 | 0.00e+00, 3e-14 | 2.67e-02, 3e-14 | 6.50e-01, 4e-14(+) | 6.56e-01, 5e-05(+) |
| 20News-8000-1000 | -1.5e+00 | 10 | 5.03e-01, 4e-14 | 5.16e-01, 1e-04 | 8.90e-01, 9e-14 | 0.00e+00, 5e-14 | 7.71e-03, 4e-14 | 2.87e-01, 1e-13(+) | 3.46e-01, 1e-04(+) |
| sector-6412-1000 | -3.4e+01 | 10 | 2.03e+00, 2e-08 | 2.28e+00, 5e-04 | 1.52e+01, 7e-14 | 0.00e+00, 5e-14 | 6.37e-02, 5e-14 | 1.03e+00, 2e-08(+) | 1.13e+00, 3e-04(+) |
| E2006-2000-1000 | -6.2e-01 | 10 | 1.18e-01, 4e-15 | 1.20e-01, 6e-05 | 4.19e-01, 2e-14 | 0.00e+00, 1e-13 | 2.68e-05, 6e-14 | 8.37e-02, 4e-14(+) | 1.01e-01, 6e-05(+) |
| MNIST-60000-784 | -2.5e+05 | 10 | 4.23e+04, 9e-10 | 6.51e+04, 5e-04 | 5.98e+02, 7e-15 | 4.60e+03, 2e-15 | 3.83e+03, 1e-15 | 0.00e+00, 9e-10(+) | 2.14e+04, 5e-04(+) |
| Gisette-3000-1000 | -1.0e+06 | 10 | 4.55e+04, 8e-10 | 3.89e+04, 7e-04 | 0.00e+00, 6e-15 | 3.18e+03, 2e-15 | 3.64e+03, 2e-15 | 7.47e+03, 8e-10(+) | 4.05e+03, 7e-04(+) |
| CnnCaltech-3000-1000 | -3.4e+03 | 10 | 9.82e+02, 1e-14 | 7.56e+02, 5e-04 | 0.00e+00, 6e-15 | 8.11e+01, 2e-15 | 1.09e+02, 2e-15 | 4.91e+02, 4e-14(+) | 1.89e+02, 5e-04(+) |
| Cifar-1000-1000 | -1.4e+05 | 10 | 1.63e+04, 2e-09 | 2.27e+04, 6e-04 | 0.00e+00, 5e-15 | 4.17e+02, 2e-15 | 3.73e+02, 2e-15 | 9.84e+03, 2e-09(+) | 4.20e+03, 6e-04(+) |
| randn-500-1000 | -6.8e+03 | 10 | 6.38e+02, 2e-11 | 2.80e+02, 5e-04 | 3.05e+02, 6e-15 | 5.60e+02, 2e-15 | 5.50e+02, 4e-15 | 2.68e+02, 2e-11(+) | 0.00e+00, 5e-04(+) |
| w1a-2477-300 | -6.6e+03 | 20 | 1.94e+03, 1e-12 | 2.62e+03, 9e-05 | 3.13e+02, 5e-14 | 1.85e+02, 2e-14 | 2.96e+02, 7e-15 | 0.00e+00, 1e-12(+) | 4.21e+02, 4e-05(+) |
| TDT2-500-1000 | -3.9e+00 | 20 | 1.04e+00, 2e-08 | 1.05e+00, 3e-04 | 1.65e+00, 8e-15 | 0.00e+00, 5e-14 | 5.94e-02, 1e-14 | 6.78e-01, 2e-08(+) | 7.34e-01, 3e-04(+) |
| 20News-8000-1000 | -1.7e+00 | 20 | 6.52e-01, 3e-08 | 6.68e-01, 7e-05 | 7.88e-01, 5e-14 | 0.00e+00, 5e-14 | 5.28e-02, 3e-14 | 4.56e-01, 3e-08(+) | 5.13e-01, 7e-05(+) |
| sector-6412-1000 | -4.6e+01 | 20 | 4.84e+00, 1e-07 | 4.86e+00, 7e-04 | 2.03e+01, 5e-14 | 0.00e+00, 5e-14 | 3.92e-01, 3e-14 | 2.11e+00, 1e-07(+) | 2.43e+00, 7e-04(+) |
| E2006-2000-1000 | -6.5e-01 | 20 | 1.42e-01, 7e-15 | 1.45e-01, 9e-05 | 3.17e-01, 1e-14 | 0.00e+00, 5e-14 | 5.53e-04, 2e-14 | 1.00e-01, 2e-14(+) | 1.22e-01, 9e-05(+) |
| MNIST-60000-784 | -2.7e+05 | 20 | 6.64e+04, 7e-12 | 2.49e+05, 6e-04 | 0.00e+00, 5e-15 | 2.51e+03, 1e-15 | 3.92e+03, 1e-15 | 4.75e+04, 7e-12(+) | 1.13e+04, 6e-04(+) |
| Gisette-3000-1000 | -1.0e+06 | 20 | 2.22e+05, 1e-08 | 3.51e+04, 1e-03 | 0.00e+00, 3e-15 | 8.37e+03, 1e-15 | 1.09e+04, 7e-16 | 5.96e+04, 1e-08(+) | 8.33e+03, 1e-03(+) |
| CnnCaltech-3000-1000 | -3.6e+03 | 20 | 1.43e+03, 4e-12 | 1.30e+03, 1e-03 | 0.00e+00, 3e-15 | 1.28e+02, 1e-15 | 1.40e+02, 1e-15 | 4.13e+02, 4e-12(+) | 3.26e+02, 1e-03(+) |
| Cifar-1000-1000 | -1.4e+05 | 20 | 2.74e+04, 3e-09 | 5.00e+04, 2e-03 | 0.00e+00, 3e-15 | 6.34e+02, 2e-16 | 5.78e+02, 8e-16 | 1.62e+04, 3e-09(+) | 1.41e+04, 2e-03(+) |
| randn-500-1000 | -1.1e+04 | 20 | 6.96e+02, 2e-10 | 5.08e+02, 1e-03 | 1.92e+02, 4e-15 | 9.19e+02, 9e-16 | 9.03e+02, 1e-15 | 3.62e+02, 2e-10(+) | 0.00e+00, 1e-03(+) |
| w1a-2477-300 | -1.2e+04 | 100 | 6.69e+03, 1e-07 | 8.92e+03, 9e-04 | 7.44e+01, 3e-14 | 1.18e+02, 2e-14 | 0.00e+00, 7e-15 | 1.12e+01, 1e-07(+) | 3.96e+01, 9e-04(+) |
| TDT2-500-1000 | -4.5e+00 | 100 | 1.25e+00, 3e-05 | 1.29e+00, 2e-04 | 1.96e+00, 1e-15 | 0.00e+00, 5e-14 | 2.09e-01, 4e-15 | 1.01e+00, 3e-05(+) | 1.06e+00, 2e-04(+) |
| 20News-8000-1000 | -1.9e+00 | 100 | 5.92e-01, 2e-05 | 6.01e-01, 2e-04 | 9.31e-01, 3e-15 | 0.00e+00, 4e-15 | 1.29e-02, 8e-15 | 4.90e-01, 2e-05(+) | 5.11e-01, 2e-04(+) |
| sector-6412-1000 | -8.5e+01 | 100 | 1.41e+01, 1e-04 | 2.59e+01, 5e-03 | 4.80e+01, 7e-15 | 0.00e+00, 6e-14 | 2.31e+00, 1e-14 | 9.98e+00, 1e-04(+) | 1.03e+01, 5e-03(+) |
| E2006-2000-1000 | -6.9e-01 | 100 | 1.63e-01, 3e-07 | 1.66e-01, 4e-06 | 3.64e-01, 2e-15 | 0.00e+00, 1e-14 | 2.41e-03, 1e-14 | 1.42e-01, 3e-07(+) | 1.42e-01, 4e-06(+) |
| MNIST-60000-784 | -3.3e+05 | 100 | 1.45e+05, 3e-06 | 3.20e+05, 2e-03 | 1.72e+04, 2e-15 | 0.00e+00, 2e-15 | 2.59e+03, 2e-15 | 3.88e+04, 3e-06(+) | 1.74e+03, 2e-03(+) |
| Gisette-3000-1000 | -1.1e+06 | 100 | 8.19e+05, 5e-06 | 5.37e+05, 5e-02 | 0.00e+00, 2e-15 | 2.62e+04, 1e-15 | 2.74e+04, 8e-16 | 5.68e+05, 5e-06(+) | 3.18e+05, 3e-02(+) |
| CnnCaltech-3000-1000 | -4.5e+03 | 100 | 3.28e+03, 8e-05 | 2.02e+03, 2e-02 | 0.00e+00, 2e-15 | 1.33e+02, 1e-15 | 1.74e+02, 1e-15 | 1.91e+03, 8e-05(+) | 5.63e+02, 2e-02(+) |
| Cifar-1000-1000 | -1.4e+05 | 100 | 1.18e+05, 3e-05 | 4.92e+04, 3e-02 | 0.00e+00, 2e-15 | 2.78e+02, 9e-16 | 1.26e+03, 1e-15 | 5.10e+04, 3e-05(+) | 3.77e+04, 3e-02(+) |
| randn-500-1000 | -3.4e+04 | 100 | 4.75e+03, 5e-05 | 1.63e+03, 2e-02 | 2.83e+02, 2e-15 | 6.24e+02, 1e-15 | 0.00e+00, 1e-15 | 2.82e+03, 5e-05(+) | 8.40e+02, 2e-02(+) |
| | | | | | time limit=30 | | | | |
| w1a-2477-300 | -5.2e+03 | 10 | 5.14e+02, 2e-13 | 1.28e+03, 1e-05 | 2.77e+02, 1e-13 | 4.96e+02, 4e-14 | 1.61e+02, 4e-14 | 0.00e+00, 2e-13(+) | 1.45e+02, 1e-05(+) |
| TDT2-500-1000 | -3.5e+00 | 10 | 9.22e-01, 1e-14 | 9.39e-01, 3e-05 | 6.60e-01, 6e-14 | 0.00e+00, 4e-14 | 2.65e-02, 3e-14 | 6.70e-01, 4e-14(+) | 6.98e-01, 3e-05(+) |
| 20News-8000-1000 | -1.6e+00 | 10 | 5.09e-01, 2e-14 | 5.22e-01, 1e-05 | 3.08e-01, 1e-13 | 6.47e-03, 8e-14 | 0.00e+00, 6e-14 | 2.84e-01, 1e-13(+) | 3.25e-01, 1e-05(+) |
| sector-6412-1000 | -3.4e+01 | 10 | 2.04e+00, 1e-11 | 2.29e+00, 1e-04 | 4.46e+00, 1e-13 | 0.00e+00, 9e-14 | 6.42e-02, 6e-14 | 1.09e+00, 1e-11(+) | 1.04e+00, 1e-04(+) |
| E2006-2000-1000 | -6.2e-01 | 10 | 1.18e-01, 3e-15 | 1.20e-01, 2e-05 | 1.15e-01, 6e-14 | 0.00e+00, 2e-13 | 2.68e-05, 6e-14 | 8.32e-02, 3e-14(+) | 9.95e-02, 2e-05(+) |
| MNIST-60000-784 | -2.5e+05 | 10 | 4.47e+04, 6e-10 | 6.46e+04, 3e-04 | 0.00e+00, 3e-14 | 5.73e+02, 1e-14 | 5.11e+02, 4e-15 | 3.82e+03, 6e-10(+) | 2.49e+04, 3e-04(+) |
| Gisette-3000-1000 | -1.0e+06 | 10 | 4.49e+04, 9e-10 | 3.32e+04, 5e-04 | 0.00e+00, 2e-14 | 1.04e+03, 4e-15 | 1.09e+03, 4e-15 | 6.41e+03, 9e-10(+) | 4.03e+03, 5e-04(+) |
| CnnCaltech-3000-1000 | -3.4e+03 | 10 | 1.01e+03, 1e-14 | 7.85e+02, 1e-04 | 0.00e+00, 2e-14 | 1.75e+01, 5e-15 | 6.63e+01, 4e-15 | 4.70e+02, 3e-14(+) | 2.05e+02, 1e-04(+) |
| Cifar-1000-1000 | -1.4e+05 | 10 | 1.61e+04, 2e-09 | 2.09e+04, 3e-04 | 0.00e+00, 2e-14 | 1.48e+02, 5e-15 | 6.74e+01, 4e-15 | 9.37e+03, 2e-09(+) | 3.27e+03, 3e-04(+) |
| randn-500-1000 | -6.8e+03 | 10 | 6.34e+02, 2e-11 | 2.82e+02, 1e-04 | 6.97e+01, 2e-14 | 2.66e+02, 6e-15 | 2.74e+02, 7e-15 | 2.23e+02, 2e-11(+) | 0.00e+00, 1e-04(+) |
| w1a-2477-300 | -6.6e+03 | 20 | 1.95e+03, 1e-12 | 2.63e+03, 2e-05 | 3.10e+02, 1e-13 | 1.65e+02, 3e-14 | 2.73e+02, 2e-14 | 0.00e+00, 1e-12(+) | 3.42e+02, 2e-05(+) |
| TDT2-500-1000 | -3.9e+00 | 20 | 1.04e+00, 8e-14 | 1.06e+00, 6e-05 | 7.34e-01, 4e-14 | 0.00e+00, 3e-14 | 9.44e-03, 3e-14 | 6.71e-01, 1e-13(+) | 7.16e-01, 6e-05(+) |
| 20News-8000-1000 | -1.7e+00 | 20 | 6.52e-01, 1e-13 | 6.69e-01, 1e-05 | 3.75e-01, 1e-13 | 0.00e+00, 5e-14 | 5.20e-02, 5e-14 | 4.58e-01, 2e-13(+) | 5.01e-01, 1e-05(+) |
| sector-6412-1000 | -4.6e+01 | 20 | 4.86e+00, 6e-09 | 4.89e+00, 2e-04 | 7.49e+00, 1e-13 | 0.00e+00, 6e-14 | 4.06e-01, 4e-14 | 2.13e+00, 6e-09(+) | 2.59e+00, 2e-04(+) |
| E2006-2000-1000 | -6.5e-01 | 20 | 1.42e-01, 8e-15 | 1.45e-01, 4e-05 | 2.48e-02, 4e-14 | 0.00e+00, 6e-14 | 5.54e-04, 3e-14 | 1.04e-01, 1e-14(+) | 1.18e-01, 4e-05(+) |
| MNIST-60000-784 | -2.7e+05 | 20 | 7.33e+04, 7e-12 | 2.27e+05, 4e-04 | 0.00e+00, 2e-14 | 2.19e+03, 3e-15 | 3.53e+03, 3e-15 | 5.49e+04, 7e-12(+) | 1.73e+04, 4e-04(+) |
| Gisette-3000-1000 | -1.0e+06 | 20 | 2.11e+05, 8e-09 | 3.36e+04, 8e-04 | 0.00e+00, 9e-15 | 3.21e+03, 2e-15 | 4.36e+03, 2e-15 | 6.19e+04, 8e-09(+) | 1.06e+04, 8e-04(+) |
| CnnCaltech-3000-1000 | -3.7e+03 | 20 | 1.50e+03, 4e-12 | 1.38e+03, 3e-04 | 0.00e+00, 1e-14 | 4.37e+01, 2e-15 | 7.19e+01, 2e-15 | 4.98e+02, 4e-12(+) | 3.24e+02, 3e-04(+) |
| Cifar-1000-1000 | -1.4e+05 | 20 | 2.74e+04, 3e-09 | 5.00e+04, 5e-04 | 0.00e+00, 1e-14 | 3.07e+02, 2e-15 | 3.97e+02, 2e-15 | 1.65e+04, 3e-09(+) | 9.86e+03, 5e-04(+) |
| randn-500-1000 | -1.1e+04 | 20 | 8.52e+02, 2e-11 | 6.83e+02, 3e-04 | 0.00e+00, 1e-14 | 5.34e+02, 3e-15 | 4.87e+02, 2e-15 | 5.21e+02, 2e-11(+) | 1.80e+02, 3e-04(+) |
| w1a-2477-300 | -1.2e+04 | 100 | 6.75e+03, 4e-09 | 8.98e+03, 2e-04 | 1.40e+01, 3e-14 | 1.47e+02, 2e-14 | 0.00e+00, 9e-15 | 4.21e-02, 4e-09(+) | 8.87e+01, 2e-04(+) |
| TDT2-500-1000 | -4.8e+00 | 100 | 1.50e+00, 3e-14 | 1.53e+00, 7e-05 | 1.62e+00, 2e-15 | 0.00e+00, 9e-15 | 1.45e-02, 8e-15 | 1.23e+00, 4e-14(+) | 1.28e+00, 7e-05(+) |
| 20News-8000-1000 | -2.0e+00 | 100 | 7.14e-01, 2e-07 | 7.23e-01, 7e-05 | 3.83e-01, 1e-14 | 0.00e+00, 6e-14 | 6.30e-03, 2e-14 | 5.85e-01, 2e-07(+) | 6.18e-01, 7e-05(+) |
| sector-6412-1000 | -8.5e+01 | 100 | 1.49e+01, 6e-08 | 2.67e+01, 2e-03 | 3.17e+01, 3e-14 | 0.00e+00, 4e-14 | 6.21e-02, 2e-14 | 9.81e+00, 6e-08(+) | 1.10e+01, 2e-03(+) |
| E2006-2000-1000 | -6.9e-01 | 100 | 1.64e-01, 5e-14 | 1.67e-01, 2e-06 | 1.19e-01, 3e-15 | 0.00e+00, 4e-14 | 3.06e-04, 4e-14 | 1.42e-01, 5e-14(+) | 1.45e-01, 2e-06(+) |
| MNIST-60000-784 | -3.5e+05 | 100 | 1.60e+05, 3e-08 | 3.39e+05, 6e-04 | 7.01e+03, 3e-15 | 0.00e+00, 2e-15 | 6.68e+03, 2e-15 | 5.71e+04, 3e-08(+) | 1.92e+04, 6e-04(+) |
| Gisette-3000-1000 | -1.1e+06 | 100 | 7.91e+05, 6e-07 | 5.95e+05, 5e-03 | 0.00e+00, 3e-15 | 2.16e+04, 1e-15 | 2.12e+04, 1e-15 | 5.06e+05, 6e-07(+) | 3.53e+05, 5e-03(+) |
| CnnCaltech-3000-1000 | -4.9e+03 | 100 | 3.62e+03, 7e-07 | 2.44e+03, 3e-03 | 0.00e+00, 3e-15 | 1.78e+02, 1e-15 | 1.61e+02, 2e-15 | 2.09e+03, 7e-07(+) | 8.27e+02, 3e-03(+) |
| Cifar-1000-1000 | -1.5e+05 | 100 | 1.20e+05, 2e-07 | 5.97e+04, 3e-03 | 0.00e+00, 3e-15 | 5.10e+02, 1e-15 | 7.80e+02, 1e-15 | 2.98e+04, 2e-07(+) | 4.22e+04, 3e-03(+) |
| randn-500-1000 | -3.6e+04 | 100 | 6.37e+03, 1e-08 | 3.53e+03, 2e-03 | 0.00e+00, 4e-15 | 1.10e+03, 2e-15 | 3.46e+02, 1e-15 | 4.32e+03, 1e-08(+) | 2.50e+03, 2e-03(+) |
| | | | | | time limit=60 | | | | |
| w1a-2477-300 | -5.2e+03 | 10 | 5.15e+02, 2e-13 | 1.27e+03, 1e-05 | 2.78e+02, 2e-13 | 4.98e+02, 9e-14 | 1.56e+02, 1e-13 | 0.00e+00, 2e-13(+) | 1.46e+02, 1e-05(+) |
| TDT2-500-1000 | -3.5e+00 | 10 | 9.22e-01, 1e-14 | 9.39e-01, 1e-05 | 5.16e-02, 1e-13 | 0.00e+00, 6e-14 | 2.66e-02, 5e-14 | 6.43e-01, 5e-14(+) | 6.49e-01, 1e-05(+) |
| 20News-8000-1000 | -1.6e+00 | 10 | 5.09e-01, 1e-14 | 5.22e-01, 9e-07 | 1.83e-02, 2e-13 | 6.49e-03, 9e-14 | 0.00e+00, 7e-14 | 2.71e-01, 9e-14(+) | 3.09e-01, 9e-07(+) |
| sector-6412-1000 | -3.4e+01 | 10 | 2.04e+00, 1e-11 | 2.29e+00, 5e-05 | 1.46e+00, 2e-13 | 0.00e+00, 2e-13 | 5.85e-02, 1e-13 | 9.61e-01, 1e-11(+) | 1.00e+00, 5e-05(+) |
| E2006-2000-1000 | -6.2e-01 | 10 | 1.18e-01, 3e-15 | 1.20e-01, 8e-06 | 1.67e-03, 7e-14 | 0.00e+00, 2e-13 | 2.67e-05, 6e-14 | 7.55e-02, 4e-14(+) | 9.28e-02, 8e-06(+) |
| MNIST-60000-784 | -2.5e+05 | 10 | 4.52e+04, 3e-11 | 6.19e+04, 2e-04 | 3.11e+02, 9e-14 | 0.00e+00, 2e-14 | 1.25e+02, 1e-14 | 1.41e+03, 3e-11(+) | 2.76e+04, 2e-04(+) |
| Gisette-3000-1000 | -1.0e+06 | 10 | 4.23e+04, 9e-10 | 2.62e+04, 4e-04 | 0.00e+00, 7e-14 | 6.86e+02, 1e-14 | 6.03e+02, 9e-15 | 7.00e+03, 9e-10(+) | 2.26e+03, 4e-04(+) |
| CnnCaltech-3000-1000 | -3.4e+03 | 10 | 1.03e+03, 1e-14 | 8.05e+02, 4e-05 | 0.00e+00, 5e-14 | 1.58e+01, 1e-14 | 5.04e+01, 9e-15 | 4.62e+02, 4e-14(+) | 2.08e+02, 4e-05(+) |
| Cifar-1000-1000 | -1.4e+05 | 10 | 1.56e+04, 2e-09 | 1.79e+04, 2e-04 | 5.10e+01, 3e-14 | 8.57e+01, 1e-14 | 0.00e+00, 1e-14 | 9.32e+03, 2e-09(+) | 2.44e+03, 2e-04(+) |
| randn-500-1000 | -6.8e+03 | 10 | 6.74e+02, 2e-11 | 3.23e+02, 3e-05 | 0.00e+00, 5e-14 | 1.95e+02, 2e-14 | 2.04e+02, 2e-14 | 2.36e+02, 2e-11(+) | 2.77e+01, 3e-05(+) |
| w1a-2477-300 | -6.6e+03 | 20 | 1.94e+03, 1e-12 | 2.62e+03, 1e-05 | 3.08e+02, 2e-13 | 1.60e+02, 7e-14 | 2.67e+02, 4e-14 | 0.00e+00, 1e-12(+) | 3.08e+02, 1e-05(+) |
| TDT2-500-1000 | -3.9e+00 | 20 | 1.05e+00, 1e-14 | 1.06e+00, 1e-05 | 8.31e-02, 7e-14 | 3.93e-03, 4e-14 | 0.00e+00, 4e-14 | 6.53e-01, 4e-14(+) | 7.16e-01, 1e-05(+) |
| 20News-8000-1000 | -1.7e+00 | 20 | 6.53e-01, 2e-14 | 6.69e-01, 6e-06 | 1.13e-01, 1e-13 | 0.00e+00, 7e-14 | 3.17e-02, 6e-14 | 4.35e-01, 8e-14(+) | 4.82e-01, 6e-06(+) |
| sector-6412-1000 | -4.6e+01 | 20 | 4.87e+00, 2e-13 | 4.90e+00, 1e-04 | 2.10e+00, 2e-13 | 0.00e+00, 1e-13 | 4.08e-01, 5e-14 | 2.13e+00, 3e-13(+) | 2.38e+00, 1e-04(+) |
| E2006-2000-1000 | -6.5e-01 | 20 | 1.42e-01, 5e-15 | 1.45e-01, 5e-06 | 2.18e-03, 8e-14 | 0.00e+00, 6e-14 | 5.42e-04, 3e-14 | 1.02e-01, 2e-14(+) | 1.13e-01, 5e-06(+) |
| MNIST-60000-784 | -2.8e+05 | 20 | 7.55e+04, 7e-12 | 1.08e+05, 3e-04 | 8.77e+02, 6e-14 | 0.00e+00, 2e-14 | 1.04e+03, 6e-15 | 5.67e+04, 7e-12(+) | 5.17e+04, 3e-04(+) |
| Gisette-3000-1000 | -1.1e+06 | 20 | 1.90e+05, 8e-09 | 3.23e+04, 7e-04 | 0.00e+00, 3e-15 | 2.06e+03, 5e-15 | 1.59e+03, 4e-15 | 5.05e+04, 8e-09(+) | 8.92e+03, 7e-04(+) |
| CnnCaltech-3000-1000 | -3.7e+03 | 20 | 1.53e+03, 4e-12 | 1.41e+03, 3e-04 | 6.78e+00, 3e-14 | 0.00e+00, 5e-15 | 4.81e+01, 4e-15 | 5.00e+02, 4e-12(+) | 3.26e+02, 2e-04(+) |
| Cifar-1000-1000 | -1.4e+05 | 20 | 2.70e+04, 2e-09 | 4.88e+04, 4e-04 | 0.00e+00, 3e-14 | 1.76e+02, 5e-15 | 2.54e+02, 5e-15 | 1.65e+04, 2e-09(+) | 8.28e+03, 4e-04(+) |
| randn-500-1000 | -1.1e+04 | 20 | 1.07e+03, 2e-11 | 9.02e+02, 9e-05 | 0.00e+00, 3e-14 | 4.35e+02, 6e-15 | 4.49e+02, 5e-15 | 6.91e+02, 2e-11(+) | 3.55e+02, 9e-05(+) |
| w1a-2477-300 | -1.2e+04 | 100 | 6.75e+03, 1e-09 | 8.98e+03, 6e-05 | 1.83e+01, 5e-14 | 1.45e+02, 2e-14 | 0.00e+00, 1e-14 | 5.73e+01, 1e-09(+) | 8.76e+01, 6e-05(+) |
| TDT2-500-1000 | -4.8e+00 | 100 | 1.52e+00, 3e-14 | 1.55e+00, 5e-05 | 4.46e-01, 5e-15 | 0.00e+00, 1e-14 | 1.41e-02, 9e-15 | 1.22e+00, 4e-14(+) | 1.27e+00, 5e-05(+) |
| 20News-8000-1000 | -2.0e+00 | 100 | 7.18e-01, 1e-08 | 7.26e-01, 3e-05 | 1.28e-01, 5e-14 | 0.00e+00, 4e-14 | 1.65e-03, 3e-14 | 5.91e-01, 1e-08(+) | 6.26e-01, 3e-05(+) |
| sector-6412-1000 | -8.6e+01 | 100 | 1.52e+01, 3e-09 | 2.69e+01, 8e-04 | 1.09e+01, 8e-14 | 8.85e-02, 5e-14 | 0.00e+00, 4e-14 | 9.94e+00, 3e-09(+) | 1.13e+01, 8e-04(+) |
| E2006-2000-1000 | -6.9e-01 | 100 | 1.64e-01, 5e-14 | 1.67e-01, 5e-07 | 2.06e-02, 8e-15 | 0.00e+00, 4e-14 | 6.25e-05, 4e-14 | 1.34e-01, 5e-14(+) | 1.43e-01, 5e-07(+) |
| MNIST-60000-784 | -3.6e+05 | 100 | 1.63e+05, 1e-08 | 3.45e+05, 4e-04 | 0.00e+00, 5e-15 | 1.94e+03, 2e-15 | 8.98e+03, 2e-14 | 6.47e+04, 1e-08(+) | 2.92e+04, 4e-04(+) |
| Gisette-3000-1000 | -1.1e+06 | 100 | 6.45e+05, 4e-07 | 5.89e+05, 5e-03 | 0.00e+00, 5e-15 | 1.49e+04, 2e-15 | 1.60e+04, 1e-15 | 3.93e+05, 4e-07(+) | 3.26e+05, 3e-03(+) |
| CnnCaltech-3000-1000 | -5.0e+03 | 100 | 3.74e+03, 4e-08 | 2.57e+03, 8e-04 | 0.00e+00, 5e-15 | 1.85e+02, 2e-15 | 1.77e+02, 2e-15 | 2.19e+03, 4e-08(+) | 9.76e+02, 8e-04(+) |
| Cifar-1000-1000 | -1.5e+05 | 100 | 1.21e+05, 5e-08 | 6.10e+04, 1e-03 | 0.00e+00, 5e-15 | 4.59e+02, 2e-15 | 7.68e+02, 2e-15 | 5.51e+04, 5e-08(+) | 4.07e+04, 1e-03(+) |
| randn-500-1000 | -3.7e+04 | 100 | 7.17e+03, 1e-10 | 4.36e+03, 9e-04 | 0.00e+00, 6e-15 | 1.26e+03, 2e-15 | 4.94e+02, 2e-15 | 5.02e+03, 1e-10(+) | 3.32e+03, 9e-04(+) |

Table 5: Comparisons of objective values and the violation of the constraints $(F(\mathbf{X}) - F_{\min}, \|\min(\mathbf{0},\mathbf{X})\|_{\mathsf{F}} + \|\mathbf{X}^{\mathsf{T}}\mathbf{X} - \mathbf{I}_r\|_{\mathsf{F}})$ for nonnegative PCA for all the compared methods. The $1^{st}$, $2^{nd}$, and $3^{rd}$ best results are colored with red, green and blue, respectively. If the objective values of 'LADMM+OBCD-R' (or 'SPM+OBCD-R') are smaller than those of 'LADMM' (or 'SPM') by a margin of $0.1 \times a$, where $a$ represents the objective values of 'LADMM' (or 'ManPG'), they will be marked with $(+)$.

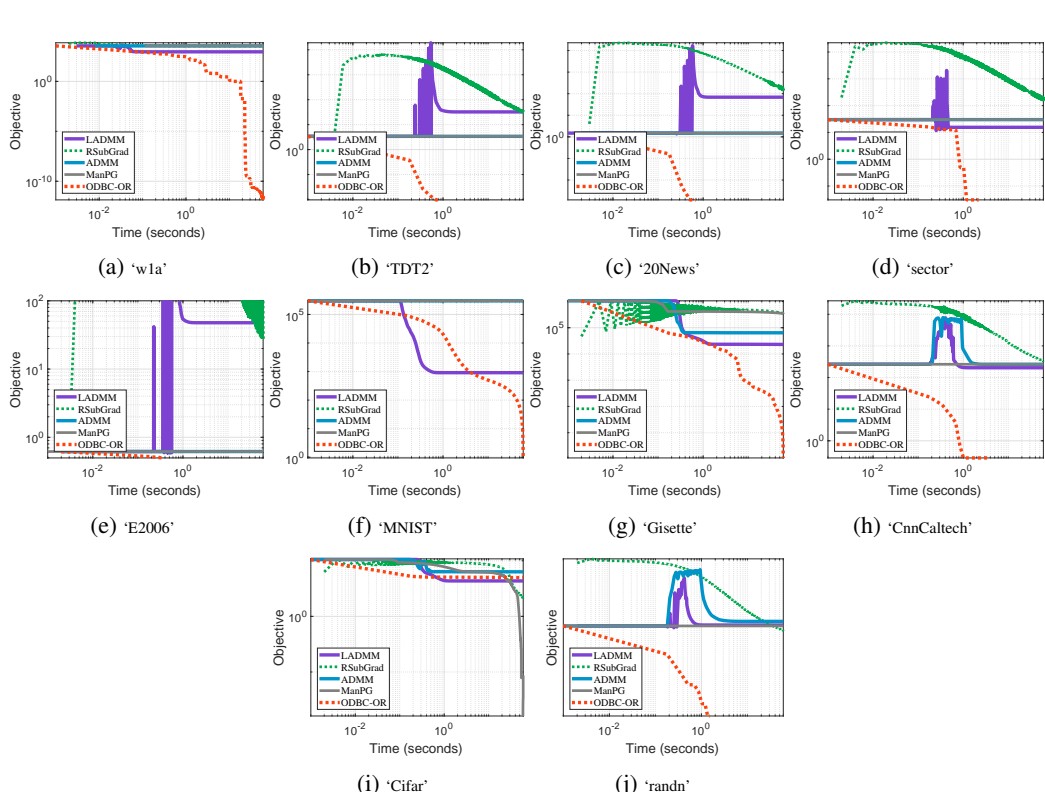

Figure 4: The convergence curve of the compared methods for solving $L_1$ norm-based SPCA with $\lambda = 100$.

| data-m-n | $F_{\min}$ | r | LADMM | RSubGrad | ADMM | ManPG | OBCD-OR | LADMM + OBCD-R | ManPG +OBCD-R |
|---|---|---|---|---|---|---|---|---|---|
| $\lambda = 1.00$, time limit=10 | | | | | | | | | |
| w1a-2477-300 | -5.7e+03 | 10 | 2.52e+01 | 4.71e+03 | 4.85e+03 | 4.71e+03 | 0.00e+00 | 1.99e+01(+) | 4.45e+01(+) |
| TDT2-500-1000 | 6.6e+00 | 10 | 3.40e+00 | 3.40e+00 | 1.51e+00 | 3.40e+00 | 0.00e+00 | 3.09e-02(+) | 9.81e-02(+) |
| 20News-8000-1000 | 8.5e+00 | 10 | 1.49e+00 | 1.62e+00 | 1.30e+00 | 1.49e+00 | 0.00e+00 | 3.07e-02(+) | 3.07e-02(+) |
| sector-6412-1000 | -2.2e+01 | 10 | 3.00e+01 | 3.00e+01 | 1.62e+01 | 3.00e+01 | 0.00e+00 | 2.73e-01(+) | 2.73e-01(+) |
| E2006-2000-1000 | 9.4e+00 | 10 | 6.17e-01 | 6.37e-01 | 1.60e-01 | 6.17e-01 | 2.32e-04 | 0.00e+00(+) | 0.00e+00(+) |
| MNIST-60000-784 | -3.1e+05 | 10 | 1.91e+02 | 3.13e+05 | 3.13e+05 | 3.13e+05 | 0.00e+00 | 1.60e+02(+) | 1.23e+04(+) |
| Gisette-3000-1000 | -1.1e+06 | 10 | 2.24e+04 | 4.78e+05 | 7.31e+04 | 3.43e+05 | 0.00e+00 | 7.94e+03(+) | 1.39e+04(+) |
| CnnCaltech-3000-1000 | -4.2e+03 | 10 | 2.87e+01 | 1.50e+00 | 9.33e+02 | 8.62e-03 | 2.18e+01 | 2.47e+01(+) | 0.00e+00(+) |
| Cifar-1000-1000 | -1.4e+05 | 10 | 2.75e+03 | 4.28e+04 | 1.22e+04 | 1.26e+00 | 2.54e+03 | 1.53e+03(+) | 0.00e+00(+) |
| randn-500-1000 | -1.3e+04 | 10 | 2.78e+01 | 1.42e+00 | 2.51e+02 | 1.10e-04 | 3.62e+01 | 2.63e+01(+) | 0.00e+00(+) |
| w1a-2477-300 | -7.1e+03 | 20 | 8.04e+01 | 5.95e+03 | 6.05e+03 | 5.94e+03 | 0.00e+00 | 4.91e+01(+) | 1.42e+02(+) |
| TDT2-500-1000 | 1.6e+01 | 20 | 3.75e+00 | 3.91e+00 | 1.89e+00 | 3.75e+00 | 0.00e+00 | 7.54e-03(+) | 7.54e-03(+) |
| 20News-8000-1000 | 1.8e+01 | 20 | 1.66e+00 | 1.94e+00 | 5.55e+00 | 1.66e+00 | 0.00e+00 | 1.08e-02(+) | 1.08e-02(+) |
| sector-6412-1000 | -2.5e+01 | 20 | 4.25e+01 | 2.97e+01 | 3.04e+01 | 4.25e+01 | 0.00e+00 | 9.39e-01(+) | 9.39e-01(+) |
| E2006-2000-1000 | 1.9e+01 | 20 | 6.37e-01 | 8.10e-01 | 7.62e-01 | 6.37e-01 | 0.00e+00 | 1.06e-03(+) | 1.06e-03(+) |
| MNIST-60000-784 | -3.5e+05 | 20 | 1.26e+02 | 3.54e+05 | 3.54e+05 | 3.54e+05 | 1.45e+03 | 0.00e+00(+) | 4.33e+04(+) |
| Gisette-3000-1000 | -1.1e+06 | 20 | 1.31e+04 | 5.01e+05 | 7.75e+04 | 3.57e+05 | 0.00e+00 | 1.90e+03(+) | 2.23e+04(+) |
| CnnCaltech-3000-1000 | -5.1e+03 | 20 | 6.73e+01 | 5.75e+00 | 1.46e+03 | 7.21e-03 | 3.08e+02 | 6.04e+01(+) | 0.00e+00(+) |
| Cifar-1000-1000 | -1.5e+05 | 20 | 3.31e+03 | 4.95e+04 | 1.37e+04 | 1.56e+02 | 6.46e+03 | 2.66e+03(+) | 0.00e+00(+) |
| randn-500-1000 | -2.6e+04 | 20 | 2.56e+01 | 5.82e+00 | 4.11e+02 | 1.51e-03 | 2.61e+02 | 2.44e+01(+) | 0.00e+00(+) |
| w1a-2477-300 | -1.2e+04 | 100 | 1.01e+03 | 5.73e+02 | 4.31e+03 | 3.97e+03 | 0.00e+00 | 8.59e+02(+) | 8.63e+02(+) |
| TDT2-500-1000 | 9.5e+01 | 100 | 4.44e+00 | 9.22e+00 | 3.90e+00 | 4.44e+00 | 0.00e+00 | 0.00e+00(+) | 0.00e+00(+) |
| 20News-8000-1000 | 9.8e+01 | 100 | 9.25e-01 | 4.15e+00 | 7.41e+01 | 9.25e-01 | 2.50e-03 | 0.00e+00(+) | 0.00e+00(+) |
| sector-6412-1000 | 1.6e+01 | 100 | 7.42e+01 | 6.64e+01 | 6.09e+01 | 7.42e+01 | 1.04e+00 | 2.84e-14(+) | 0.00e+00(+) |
| E2006-2000-1000 | 9.9e+01 | 100 | 6.71e-01 | 4.47e+00 | 3.73e+02 | 6.71e-01 | 5.06e-05 | 0.00e+00(+) | 0.00e+00(+) |
| MNIST-60000-784 | -3.9e+05 | 100 | 2.89e+03 | 1.09e+05 | 1.93e+05 | 2.24e+05 | 1.60e+04 | 0.00e+00(+) | 7.58e+04(+) |
| Gisette-3000-1000 | -1.2e+06 | 100 | 3.60e+04 | 8.49e+05 | 1.13e+05 | 3.11e+05 | 2.43e+04 | 0.00e+00(+) | 5.83e+04(+) |
| CnnCaltech-3000-1000 | -6.7e+03 | 100 | 5.35e+02 | 0.00e+00 | 4.43e+03 | 1.40e+03 | 2.24e+03 | 5.33e+02(+) | 1.36e+03(+) |
| Cifar-1000-1000 | -1.5e+05 | 100 | 1.08e+02 | 7.04e+04 | 1.00e+04 | 2.00e+04 | 7.70e+03 | 0.00e+00(+) | 8.15e+03(+) |
| randn-500-1000 | -1.0e+05 | 100 | 1.14e+02 | 0.00e+00 | 1.50e+03 | 6.03e+01 | 4.76e+04 | 1.14e+02(+) | 6.03e+01 |
| $\lambda = 1.00$, time limit=30 | | | | | | | | | |
| w1a-2477-300 | -5.7e+03 | 10 | 2.60e+01 | 4.71e+03 | 4.85e+03 | 4.71e+03 | 0.00e+00 | 2.07e+01(+) | 4.53e+01(+) |
| TDT2-500-1000 | 6.6e+00 | 10 | 3.40e+00 | 3.40e+00 | 1.51e+00 | 3.40e+00 | 0.00e+00 | 3.09e-02(+) | 3.09e-02(+) |
| 20News-8000-1000 | 8.5e+00 | 10 | 1.49e+00 | 1.53e+00 | 1.30e+00 | 1.49e+00 | 0.00e+00 | 3.07e-02(+) | 3.07e-02(+) |
| sector-6412-1000 | -2.2e+01 | 10 | 3.00e+01 | 3.00e+01 | 1.62e+01 | 3.00e+01 | 0.00e+00 | 2.73e-01(+) | 2.73e-01(+) |
| E2006-2000-1000 | 9.4e+00 | 10 | 6.17e-01 | 6.23e-01 | 1.60e-01 | 6.17e-01 | 6.54e-04 | 0.00e+00(+) | 0.00e+00(+) |
| MNIST-60000-784 | -3.1e+05 | 10 | 1.97e+02 | 3.13e+05 | 3.13e+05 | 3.13e+05 | 0.00e+00 | 1.65e+02(+) | 1.19e+04(+) |
| Gisette-3000-1000 | -1.1e+06 | 10 | 2.53e+04 | 4.35e+05 | 7.61e+04 | 3.24e+05 | 0.00e+00 | 1.06e+04(+) | 1.59e+04(+) |
| CnnCaltech-3000-1000 | -4.2e+03 | 10 | 2.97e+01 | 1.96e+00 | 9.34e+02 | 7.40e-07 | 1.63e+01 | 2.56e+01(+) | 0.00e+00 |
| Cifar-1000-1000 | -1.4e+05 | 10 | 2.79e+03 | 0.00e+00 | 1.22e+04 | 1.20e+01 | 1.96e+02 | 1.57e+03(+) | 1.18e+01(+) |
| randn-500-1000 | -1.3e+04 | 10 | 2.87e+01 | 1.88e+00 | 2.52e+02 | 6.80e-10 | 9.32e+00 | 2.72e+01(+) | 0.00e+00 |
| w1a-2477-300 | -7.1e+03 | 20 | 8.24e+01 | 5.95e+03 | 6.05e+03 | 5.95e+03 | 0.00e+00 | 5.07e+01(+) | 1.43e+02(+) |
| TDT2-500-1000 | 1.6e+01 | 20 | 3.75e+00 | 3.81e+00 | 1.89e+00 | 3.75e+00 | 0.00e+00 | 8.92e-03(+) | 8.92e-03(+) |
| 20News-8000-1000 | 1.8e+01 | 20 | 1.66e+00 | 1.76e+00 | 5.55e+00 | 1.66e+00 | 0.00e+00 | 1.08e-02(+) | 1.08e-02(+) |
| sector-6412-1000 | -2.5e+01 | 20 | 4.25e+01 | 2.92e+01 | 3.04e+01 | 4.25e+01 | 0.00e+00 | 9.61e-01(+) | 9.61e-01(+) |
| E2006-2000-1000 | 1.9e+01 | 20 | 6.38e-01 | 6.45e-01 | 7.61e-01 | 6.38e-01 | 0.00e+00 | 1.30e-03(+) | 1.30e-03(+) |
| MNIST-60000-784 | -3.5e+05 | 20 | 1.25e+02 | 3.54e+05 | 3.54e+05 | 3.54e+05 | 2.02e+02 | 0.00e+00(+) | 4.33e+04(+) |
| Gisette-3000-1000 | -1.1e+06 | 20 | 2.50e+04 | 4.55e+05 | 8.95e+04 | 3.55e+05 | 0.00e+00 | 1.38e+04(+) | 3.72e+04(+) |
| CnnCaltech-3000-1000 | -5.1e+03 | 20 | 7.47e+01 | 1.17e+01 | 1.47e+03 | 1.07e-05 | 4.52e+01 | 6.77e+01(+) | 0.00e+00(+) |
| Cifar-1000-1000 | -1.5e+05 | 20 | 4.40e+03 | 1.02e+04 | 1.48e+04 | 3.16e+01 | 2.75e+03 | 3.73e+03(+) | 0.00e+00(+) |
| randn-500-1000 | -2.6e+04 | 20 | 2.78e+01 | 6.97e+00 | 4.13e+02 | 5.23e-04 | 7.67e+01 | 2.66e+01(+) | 0.00e+00(+) |
| w1a-2477-300 | -1.2e+04 | 100 | 1.11e+03 | 6.63e+02 | 4.41e+03 | 2.20e+03 | 0.00e+00 | 9.60e+02(+) | 1.14e+03(+) |
| TDT2-500-1000 | 9.5e+01 | 100 | 4.45e+00 | 5.91e+00 | 3.90e+00 | 4.45e+00 | 0.00e+00 | 3.43e-03(+) | 3.43e-03(+) |
| 20News-8000-1000 | 9.8e+01 | 100 | 9.26e-01 | 4.77e+00 | 7.06e+01 | 9.26e-01 | 0.00e+00 | 9.23e-04(+) | 9.23e-04(+) |
| sector-6412-1000 | 1.6e+01 | 100 | 7.45e+01 | 6.46e+01 | 6.12e+01 | 7.45e+01 | 0.00e+00 | 2.65e-01(+) | 2.65e-01(+) |
| E2006-2000-1000 | 9.9e+01 | 100 | 6.72e-01 | 1.79e+00 | 3.67e+02 | 6.72e-01 | 0.00e+00 | 3.03e-04(+) | 3.03e-04(+) |
| MNIST-60000-784 | -4.0e+05 | 100 | 1.38e+04 | 7.60e+04 | 2.04e+05 | 2.09e+05 | 0.00e+00 | 1.10e+04(+) | 9.39e+04(+) |
| Gisette-3000-1000 | -1.2e+06 | 100 | 3.95e+04 | 2.65e+05 | 1.16e+05 | 3.13e+05 | 0.00e+00 | 3.52e+03(+) | 6.23e+04(+) |
| CnnCaltech-3000-1000 | -6.8e+03 | 100 | 5.46e+02 | 0.00e+00 | 4.43e+03 | 4.48e+02 | 1.55e+03 | 5.44e+02(+) | 4.47e+02(+) |
| Cifar-1000-1000 | -1.5e+05 | 100 | 1.08e+02 | 4.95e+04 | 9.99e+03 | 1.13e+04 | 4.78e+03 | 0.00e+00(+) | 9.25e+03(+) |
| randn-500-1000 | -1.0e+05 | 100 | 1.31e+02 | 0.00e+00 | 1.50e+03 | 1.08e+00 | 2.23e+04 | 1.31e+02(+) | 1.08e+00 |
| $\lambda = 1.00$, time limit=60 | | | | | | | | | |
| w1a-2477-300 | -5.7e+03 | 10 | 2.54e+01 | 4.71e+03 | 4.85e+03 | 4.71e+03 | 0.00e+00 | 2.01e+01(+) | 4.35e+01(+) |
| TDT2-500-1000 | 6.6e+00 | 10 | 3.40e+00 | 3.40e+00 | 1.51e+00 | 3.40e+00 | 0.00e+00 | 3.09e-02(+) | 3.09e-02(+) |
| 20News-8000-1000 | 8.5e+00 | 10 | 1.49e+00 | 1.50e+00 | 1.30e+00 | 1.49e+00 | 0.00e+00 | 3.07e-02(+) | 3.07e-02(+) |
| sector-6412-1000 | -2.2e+01 | 10 | 3.00e+01 | 3.01e+01 | 1.63e+01 | 3.00e+01 | 0.00e+00 | 3.54e-01(+) | 3.54e-01(+) |
| E2006-2000-1000 | 9.4e+00 | 10 | 6.18e-01 | 6.21e-01 | 1.61e-01 | 6.18e-01 | 0.00e+00 | 1.59e-03(+) | 1.59e-03(+) |
| MNIST-60000-784 | -3.1e+05 | 10 | 1.98e+02 | 3.13e+05 | 3.13e+05 | 3.13e+05 | 0.00e+00 | 1.66e+02(+) | 1.15e+04(+) |
| Gisette-3000-1000 | -1.1e+06 | 10 | 2.51e+04 | 3.21e+05 | 7.62e+04 | 4.52e+05 | 0.00e+00 | 1.06e+04(+) | 1.92e+04(+) |
| CnnCaltech-3000-1000 | -4.2e+03 | 10 | 2.97e+01 | 1.77e+00 | 9.34e+02 | 3.44e-08 | 1.19e+01 | 2.56e+01(+) | 0.00e+00 |
| Cifar-1000-1000 | -1.4e+05 | 10 | 2.79e+03 | 0.00e+00 | 1.22e+04 | 4.22e+00 | 6.36e+01 | 1.60e+03(+) | 4.19e+00(+) |
| randn-500-1000 | -1.3e+04 | 10 | 2.87e+01 | 1.76e+00 | 2.52e+02 | 8.33e-09 | 6.36e+00 | 2.72e+01(+) | 0.00e+00 |
| w1a-2477-300 | -7.1e+03 | 20 | 9.14e+01 | 5.96e+03 | 6.06e+03 | 5.95e+03 | 0.00e+00 | 5.89e+01(+) | 1.49e+02(+) |
| TDT2-500-1000 | 1.6e+01 | 20 | 3.75e+00 | 3.76e+00 | 1.89e+00 | 3.75e+00 | 0.00e+00 | 1.10e-02(+) | 1.10e-02(+) |
| 20News-8000-1000 | 1.8e+01 | 20 | 1.66e+00 | 1.70e+00 | 5.55e+00 | 1.66e+00 | 0.00e+00 | 1.08e-02(+) | 1.08e-02(+) |
| sector-6412-1000 | -2.6e+01 | 20 | 4.27e+01 | 2.93e+01 | 3.07e+01 | 4.27e+01 | 0.00e+00 | 1.22e+00(+) | 1.22e+00(+) |
| E2006-2000-1000 | 1.9e+01 | 20 | 6.38e-01 | 6.41e-01 | 7.62e-01 | 6.38e-01 | 0.00e+00 | 1.70e-03(+) | 1.70e-03(+) |
| MNIST-60000-784 | -3.5e+05 | 20 | 3.33e+02 | 3.55e+05 | 3.55e+05 | 3.55e+05 | 0.00e+00 | 2.09e+02(+) | 4.30e+04(+) |
| Gisette-3000-1000 | -1.1e+06 | 20 | 3.01e+04 | 3.92e+05 | 9.47e+04 | 5.14e+05 | 0.00e+00 | 1.86e+04(+) | 3.99e+04(+) |
| CnnCaltech-3000-1000 | -5.1e+03 | 20 | 7.53e+01 | 1.16e+00 | 1.47e+03 | 1.57e-05 | 3.28e+01 | 6.83e+01(+) | 0.00e+00(+) |
| Cifar-1000-1000 | -1.5e+05 | 20 | 4.44e+03 | 0.00e+00 | 1.48e+04 | 2.72e+01 | 1.53e+03 | 3.80e+03(+) | 2.71e+01(+) |
| randn-500-1000 | -2.6e+04 | 20 | 2.85e+01 | 7.30e+00 | 4.14e+02 | 1.32e-07 | 2.91e+01 | 2.74e+01(+) | 0.00e+00 |
| w1a-2477-300 | -1.2e+04 | 100 | 1.16e+03 | 7.00e+02 | 4.46e+03 | 1.55e+03 | 0.00e+00 | 1.01e+03(+) | 9.42e+02(+) |
| TDT2-500-1000 | 9.5e+01 | 100 | 4.45e+00 | 5.14e+00 | 3.90e+00 | 4.45e+00 | 0.00e+00 | 4.80e-03(+) | 4.80e-03(+) |
| 20News-8000-1000 | 9.8e+01 | 100 | 9.26e-01 | 2.71e+00 | 7.06e+01 | 9.26e-01 | 0.00e+00 | 1.20e-03(+) | 1.20e-03(+) |
| sector-6412-1000 | 1.6e+01 | 100 | 7.45e+01 | 6.25e+01 | 6.13e+01 | 7.45e+01 | 0.00e+00 | 2.94e-01(+) | 2.94e-01(+) |
| E2006-2000-1000 | 9.9e+01 | 100 | 6.72e-01 | 1.24e+00 | 3.67e+02 | 6.72e-01 | 0.00e+00 | 4.65e-04(+) | 4.65e-04(+) |
| MNIST-60000-784 | -4.1e+05 | 100 | 2.29e+04 | 6.95e+04 | 2.13e+05 | 2.14e+05 | 0.00e+00 | 2.00e+04(+) | 9.80e+04(+) |
| Gisette-3000-1000 | -1.2e+06 | 100 | 5.38e+04 | 4.49e+05 | 1.30e+05 | 3.25e+05 | 0.00e+00 | 1.79e+04(+) | 7.67e+04(+) |
| CnnCaltech-3000-1000 | -6.8e+03 | 100 | 5.51e+02 | 0.00e+00 | 4.43e+03 | 2.67e+02 | 1.13e+03 | 5.49e+02(+) | 2.67e+02 |
| Cifar-1000-1000 | -1.5e+05 | 100 | 1.08e+02 | 4.01e+04 | 9.99e+03 | 1.99e+04 | 2.42e+03 | 0.00e+00(+) | 1.71e+03(+) |
| randn-500-1000 | -1.0e+05 | 100 | 1.53e+02 | 1.56e+01 | 1.53e+03 | 3.45e-03 | 1.04e+04 | 1.53e+02(+) | 0.00e+00(+) |

Table 6: Comparisons of objective values $(F(\mathbf{X}) - F_{\min})$ of $L_1$ norm-based SPCA for all the compared methods with $\lambda = 1$. The $1^{st}$, $2^{nd}$, and $3^{rd}$ best results are colored with red, green and blue, respectively. If the objective values of 'LADMM+OBCD-R' (or 'ManPG+OBCD-R') are smaller than those of 'LADMM' (or 'ManPG') by a margin of $0.1 \times a$, where $a$ represents the objective values of 'LADMM' (or 'ManPG'), they will be marked with $(+)$.

| data-m-n | $F_{\min}$ | r | LADMM | RSubGrad | ADMM | ManPG | OBCD-OR | LADMM + OBCD-R | ManPG +OBCD-R |
|---|---|---|---|---|---|---|---|---|---|
| | | | | $\lambda = 100.00$, time limit=10 | | | | | |
| w1a-2477-300 | -3.0e+03 | 10 | 9.05e+02 | 3.48e+03 | 3.57e+03 | 3.57e+03 | 0.00e+00 | 3.74e+01(+) | 1.36e+01(+) |
| TDT2-500-1000 | 1.0e+03 | 10 | 3.24e+01 | 1.64e+02 | 3.40e+00 | 3.40e+00 | 0.00e+00 | 2.90e+01(+) | 9.81e-02(+) |
| 20News-8000-1000 | 1.0e+03 | 10 | 6.88e+01 | 1.11e+02 | 1.49e+00 | 1.49e+00 | 0.00e+00 | 3.76e+01(+) | 3.07e-02(+) |
| sector-6412-1000 | 9.7e+02 | 10 | 1.55e+01 | 8.08e+02 | 3.00e+01 | 3.00e+01 | 0.00e+00 | 3.21e-01(+) | 3.68e-01(+) |
| E2006-2000-1000 | 1.0e+03 | 10 | 4.77e+01 | 4.69e+02 | 6.18e-01 | 6.18e-01 | 0.00e+00 | 1.01e+01(+) | 1.59e-03(+) |
| MNIST-60000-784 | -3.0e+05 | 10 | 7.53e+02 | 3.00e+05 | 3.00e+05 | 3.00e+05 | 0.00e+00 | 6.65e+02(+) | 7.27e+03(+) |
| Gisette-3000-1000 | -1.1e+06 | 10 | 2.35e+04 | 4.70e+05 | 6.42e+04 | 3.99e+05 | 0.00e+00 | 1.27e+04(+) | 1.43e+04(+) |
| CnnCaltech-3000-1000 | 5.9e+02 | 10 | 2.32e+02 | 1.06e+03 | 2.90e+02 | 2.90e+02 | 0.00e+00 | 7.90e+01(+) | 3.80e+00(+) |
| Cifar-1000-1000 | -1.3e+05 | 10 | 2.93e+02 | 3.78e+04 | 8.94e+03 | 7.45e+03 | 1.90e+03 | 0.00e+00(+) | 5.76e+03(+) |
| randn-500-1000 | -2.0e+03 | 10 | 5.40e+02 | 1.17e+03 | 6.68e+02 | 5.12e+02 | 0.00e+00 | 1.57e+02(+) | 5.72e+00(+) |
| w1a-2477-300 | -3.3e+03 | 20 | 1.72e+03 | 4.45e+03 | 4.45e+03 | 4.45e+03 | 5.59e+01 | 1.07e+02(+) | 0.00e+00(+) |
| TDT2-500-1000 | 2.0e+03 | 20 | 2.71e+02 | 8.34e+02 | 3.75e+00 | 3.75e+00 | 0.00e+00 | 2.09e+02(+) | 1.10e-02(+) |
| 20News-8000-1000 | 2.0e+03 | 20 | 1.59e+02 | 2.90e+03 | 1.66e+00 | 1.66e+00 | 0.00e+00 | 7.94e+01(+) | 1.30e-02(+) |
| sector-6412-1000 | 2.0e+03 | 20 | 9.94e+01 | 2.45e+03 | 4.27e+01 | 4.27e+01 | 0.00e+00 | 4.21e+01(+) | 1.19e+00(+) |
| E2006-2000-1000 | 2.0e+03 | 20 | 1.21e+02 | 1.11e+03 | 6.38e-01 | 6.38e-01 | 0.00e+00 | 8.05e+01(+) | 1.83e-03(+) |
| MNIST-60000-784 | -3.3e+05 | 20 | 3.28e+03 | 3.31e+05 | 3.31e+05 | 3.31e+05 | 0.00e+00 | 2.11e+03(+) | 3.82e+04(+) |
| Gisette-3000-1000 | -1.1e+06 | 20 | 2.30e+04 | 4.72e+05 | 7.61e+04 | 3.72e+05 | 0.00e+00 | 1.27e+04(+) | 3.33e+04(+) |
| CnnCaltech-3000-1000 | 1.4e+03 | 20 | 2.46e+02 | 3.72e+02 | 2.66e+03 | 3.90e+02 | 0.00e+00 | 3.32e+01(+) | 4.63e+00(+) |
| Cifar-1000-1000 | -1.3e+05 | 20 | 3.42e+02 | 5.36e+04 | 1.35e+04 | 1.30e+04 | 1.96e+03 | 0.00e+00(+) | 1.06e+04(+) |
| randn-500-1000 | -3.9e+03 | 20 | 1.25e+03 | 3.69e+03 | 8.33e+02 | 8.19e+02 | 0.00e+00 | 5.13e+02(+) | 1.47e+01(+) |
| w1a-2477-300 | -1.7e+03 | 100 | 5.04e+03 | 4.60e+03 | 7.36e+03 | 5.29e+03 | 1.07e+02 | 5.22e+01(+) | 0.00e+00(+) |
| TDT2-500-1000 | 1.0e+04 | 100 | 1.09e+03 | 2.73e+04 | 4.45e+00 | 4.45e+00 | 1.37e-03 | 8.51e+02(+) | 0.00e+00(+) |
| 20News-8000-1000 | 1.0e+04 | 100 | 8.15e+02 | 5.60e+04 | 9.25e-01 | 9.25e-01 | 3.90e-04 | 6.68e+02(+) | 0.00e+00(+) |
| sector-6412-1000 | 9.9e+03 | 100 | 6.26e+01 | 5.12e+04 | 7.43e+01 | 7.43e+01 | 0.00e+00 | 8.32e+00(+) | 3.75e-02(+) |
| E2006-2000-1000 | 1.0e+04 | 100 | 1.09e+03 | 3.39e+04 | 6.71e-01 | 6.71e-01 | 7.81e-04 | 9.16e+02(+) | 1.09e+00(+) |
| MNIST-60000-784 | -3.4e+05 | 100 | 3.32e+04 | 1.24e+05 | 1.57e+05 | 1.75e+05 | 0.00e+00 | 2.70e+04(+) | 6.24e+04(+) |
| Gisette-3000-1000 | -1.1e+06 | 100 | 3.90e+04 | 5.50e+05 | 1.05e+05 | 3.07e+05 | 9.59e+02 | 0.00e+00(+) | 6.87e+04(+) |
| CnnCaltech-3000-1000 | 7.8e+03 | 100 | 9.37e+02 | 3.93e+04 | 1.81e+03 | 9.92e+02 | 1.88e+00 | 6.63e+01(+) | 0.00e+00(+) |
| Cifar-1000-1000 | -1.2e+05 | 100 | 2.86e+02 | 1.32e+05 | 3.72e+04 | 5.68e+04 | 2.65e+03 | 0.00e+00(+) | 2.62e+04(+) |
| randn-500-1000 | -1.8e+04 | 100 | 3.78e+03 | 7.11e+04 | 2.74e+03 | 2.74e+03 | 2.73e+00 | 1.67e+03(+) | 0.00e+00(+) |
| | | | | $\lambda = 100.00$, time limit=30 | | | | | |
| w1a-2477-300 | -3.0e+03 | 10 | 9.10e+02 | 3.49e+03 | 3.58e+03 | 3.58e+03 | 0.00e+00 | 4.31e+01(+) | 1.73e+01(+) |
| TDT2-500-1000 | 1.0e+03 | 10 | 3.24e+01 | 7.93e+01 | 3.40e+00 | 3.40e+00 | 0.00e+00 | 2.90e+01(+) | 3.09e-02(+) |
| 20News-8000-1000 | 1.0e+03 | 10 | 6.88e+01 | 3.86e+02 | 1.49e+00 | 1.49e+00 | 0.00e+00 | 3.76e+01(+) | 3.07e-02(+) |
| sector-6412-1000 | 9.7e+02 | 10 | 1.55e+01 | 3.52e+02 | 3.00e+01 | 3.00e+01 | 0.00e+00 | 3.21e-01(+) | 3.68e-01(+) |
| E2006-2000-1000 | 1.0e+03 | 10 | 4.77e+01 | 2.10e+02 | 6.18e-01 | 6.18e-01 | 0.00e+00 | 1.01e+01(+) | 1.59e-03(+) |
| MNIST-60000-784 | -3.0e+05 | 10 | 1.33e+03 | 3.01e+05 | 3.01e+05 | 3.01e+05 | 0.00e+00 | 1.25e+03(+) | 7.85e+03(+) |
| Gisette-3000-1000 | -1.1e+06 | 10 | 2.27e+04 | 4.22e+05 | 6.36e+04 | 2.31e+05 | 0.00e+00 | 1.21e+04(+) | 1.58e+04(+) |
| CnnCaltech-3000-1000 | 5.9e+02 | 10 | 2.32e+02 | 4.55e+02 | 2.90e+02 | 2.90e+02 | 0.00e+00 | 7.90e+01(+) | 3.80e+00(+) |
| Cifar-1000-1000 | -1.3e+05 | 10 | 1.37e+03 | 2.98e+01 | 1.00e+04 | 6.75e+01 | 2.99e+03 | 1.07e+03(+) | 0.00e+00(+) |
| randn-500-1000 | -2.0e+03 | 10 | 5.41e+02 | 4.89e+02 | 6.69e+02 | 5.13e+02 | 0.00e+00 | 1.57e+02(+) | 7.08e+00(+) |
| w1a-2477-300 | -3.3e+03 | 20 | 1.72e+03 | 4.44e+03 | 4.45e+03 | 4.45e+03 | 3.07e+01 | 1.07e+02(+) | 0.00e+00(+) |
| TDT2-500-1000 | 2.0e+03 | 20 | 2.71e+02 | 3.75e+02 | 3.75e+00 | 3.75e+00 | 0.00e+00 | 2.09e+02(+) | 1.10e-02(+) |
| 20News-8000-1000 | 2.0e+03 | 20 | 1.59e+02 | 1.00e+03 | 1.66e+00 | 1.66e+00 | 0.00e+00 | 7.93e+01(+) | 1.08e-02(+) |
| sector-6412-1000 | 2.0e+03 | 20 | 9.94e+01 | 7.47e+02 | 4.27e+01 | 4.27e+01 | 0.00e+00 | 4.21e+01(+) | 1.19e+00(+) |
| E2006-2000-1000 | 2.0e+03 | 20 | 1.21e+02 | 5.19e+02 | 6.38e-01 | 6.38e-01 | 0.00e+00 | 8.05e+01(+) | 1.83e-03(+) |
| MNIST-60000-784 | -3.3e+05 | 20 | 4.26e+03 | 3.32e+05 | 3.32e+05 | 3.32e+05 | 0.00e+00 | 3.09e+03(+) | 3.80e+04(+) |
| Gisette-3000-1000 | -1.1e+06 | 20 | 2.66e+04 | 4.51e+05 | 7.96e+04 | 3.60e+05 | 0.00e+00 | 1.60e+04(+) | 3.52e+04(+) |
| CnnCaltech-3000-1000 | 1.4e+03 | 20 | 2.46e+02 | 1.16e+03 | 2.65e+03 | 3.90e+02 | 0.00e+00 | 3.29e+01(+) | 4.37e+00(+) |
| Cifar-1000-1000 | -1.3e+05 | 20 | 3.42e+02 | 1.24e+04 | 1.24e+04 | 1.35e+04 | 7.90e+03 | 1.94e+03 | 0.00e+00(+) | 7.12e+03(+) |
| randn-500-1000 | -3.9e+03 | 20 | 1.26e+03 | 1.31e+03 | 8.40e+02 | 8.26e+02 | 0.00e+00 | 4.98e+02(+) | 2.17e+01(+) |
| w1a-2477-300 | -1.6e+03 | 100 | 4.99e+03 | 2.58e+03 | 7.31e+03 | 5.24e+03 | 3.09e+01 | 0.00e+00(+) | 3.58e+01(+) |
| TDT2-500-1000 | 1.0e+04 | 100 | 1.08e+03 | 1.13e+04 | 4.45e+00 | 4.45e+00 | 0.00e+00 | 9.69e+02(+) | 4.80e-03(+) |
| 20News-8000-1000 | 1.0e+04 | 100 | 8.15e+02 | 2.28e+04 | 9.26e-01 | 9.26e-01 | 0.00e+00 | 6.68e+02(+) | 1.41e-03(+) |
| sector-6412-1000 | 9.9e+03 | 100 | 6.25e+01 | 2.17e+04 | 7.45e+01 | 7.45e+01 | 0.00e+00 | 8.27e+00(+) | 2.97e-01(+) |
| E2006-2000-1000 | 1.0e+04 | 100 | 1.08e+03 | 1.15e+04 | 6.72e-01 | 6.72e-01 | 0.00e+00 | 9.06e+02(+) | 4.94e-04(+) |
| MNIST-60000-784 | -3.5e+05 | 100 | 4.43e+04 | 8.22e+04 | 1.68e+05 | 1.69e+05 | 0.00e+00 | 3.81e+04(+) | 7.46e+04(+) |
| Gisette-3000-1000 | -1.1e+06 | 100 | 5.71e+04 | 4.81e+05 | 1.23e+05 | 3.23e+05 | 0.00e+00 | 1.81e+04(+) | 8.55e+04(+) |
| CnnCaltech-3000-1000 | 7.8e+03 | 100 | 9.39e+02 | 2.52e+04 | 1.76e+03 | 9.93e+02 | 0.00e+00 | 6.75e+01(+) | 1.29e+00(+) |
| Cifar-1000-1000 | -1.2e+05 | 100 | 2.97e+02 | 8.07e+04 | 3.72e+04 | 5.66e+04 | 2.28e+03 | 0.00e+00(+) | 2.58e+04(+) |
| randn-500-1000 | -1.8e+04 | 100 | 3.80e+03 | 2.57e+04 | 2.75e+03 | 2.75e+03 | 0.00e+00 | 1.64e+03(+) | 1.20e+01(+) |
| | | | | $\lambda = 100.00$, time limit=60 | | | | | |
| w1a-2477-300 | -3.0e+03 | 10 | 9.08e+02 | 3.49e+03 | 3.58e+03 | 3.58e+03 | 0.00e+00 | 4.10e+01(+) | 1.54e+01(+) |
| TDT2-500-1000 | 1.0e+03 | 10 | 3.23e+01 | 2.86e+01 | 3.40e+00 | 3.40e+00 | 0.00e+00 | 2.90e+01(+) | 3.09e-02(+) |
| 20News-8000-1000 | 1.0e+03 | 10 | 6.88e+01 | 1.03e+02 | 1.49e+00 | 1.49e+00 | 0.00e+00 | 3.76e+01(+) | 3.07e-02(+) |
| sector-6412-1000 | 9.7e+02 | 10 | 1.55e+01 | 1.53e+02 | 3.00e+01 | 3.00e+01 | 0.00e+00 | 3.21e-01(+) | 3.68e-01(+) |
| E2006-2000-1000 | 1.0e+03 | 10 | 4.77e+01 | 8.87e+01 | 6.18e-01 | 6.18e-01 | 0.00e+00 | 1.01e+01(+) | 1.59e-03(+) |
| MNIST-60000-784 | -3.0e+05 | 10 | 1.18e+03 | 3.01e+05 | 3.01e+05 | 3.01e+05 | 0.00e+00 | 1.09e+03(+) | 7.17e+03(+) |
| Gisette-3000-1000 | -1.1e+06 | 10 | 1.28e+04 | 3.01e+05 | 5.40e+04 | 3.00e+05 | 0.00e+00 | 2.18e+03(+) | 6.93e+03(+) |
| CnnCaltech-3000-1000 | 6.0e+02 | 10 | 2.28e+02 | 3.15e+02 | 2.86e+02 | 2.86e+02 | 1.57e+01 | 6.79e+01(+) | 0.00e+00(+) |
| Cifar-1000-1000 | -1.3e+05 | 10 | 1.57e+03 | 4.72e+01 | 1.02e+04 | 3.31e+01 | 3.11e+03 | 1.28e+03(+) | 0.00e+00(+) |
| randn-500-1000 | -2.0e+03 | 10 | 5.41e+02 | 3.71e+02 | 6.69e+02 | 5.13e+02 | 0.00e+00 | 1.58e+02(+) | 7.08e+00(+) |
| w1a-2477-300 | -3.3e+03 | 20 | 1.72e+03 | 4.44e+03 | 4.45e+03 | 4.45e+03 | 3.85e+01 | 1.07e+02(+) | 0.00e+00(+) |
| TDT2-500-1000 | 2.0e+03 | 20 | 2.71e+02 | 7.21e+01 | 3.75e+00 | 3.75e+00 | 0.00e+00 | 2.09e+02(+) | 1.10e-02(+) |
| 20News-8000-1000 | 2.0e+03 | 20 | 1.59e+02 | 4.86e+02 | 1.66e+00 | 1.66e+00 | 0.00e+00 | 7.93e+01(+) | 1.08e-02(+) |
| sector-6412-1000 | 2.0e+03 | 20 | 9.93e+01 | 4.84e+02 | 4.27e+01 | 4.27e+01 | 0.00e+00 | 4.20e+01(+) | 1.19e+00(+) |
| E2006-2000-1000 | 2.0e+03 | 20 | 1.20e+02 | 2.25e+02 | 6.38e-01 | 6.38e-01 | 0.00e+00 | 8.05e+01(+) | 1.70e-03(+) |
| MNIST-60000-784 | -3.3e+05 | 20 | 4.84e+03 | 3.32e+05 | 3.32e+05 | 3.32e+05 | 0.00e+00 | 3.65e+03(+) | 3.83e+04(+) |
| Gisette-3000-1000 | -1.1e+06 | 20 | 2.49e+04 | 3.82e+05 | 7.81e+04 | 3.36e+05 | 0.00e+00 | 1.44e+04(+) | 3.52e+04(+) |
| CnnCaltech-3000-1000 | 1.4e+03 | 20 | 2.46e+02 | 7.54e+02 | 2.65e+03 | 3.90e+02 | 0.00e+00 | 3.32e+01(+) | 4.63e+00(+) |
| Cifar-1000-1000 | -1.3e+05 | 20 | 1.18e+02 | 0.00e+00 | 1.43e+04 | 5.38e+02 | 2.66e+03 | 8.43e+02(+) | 3.75e+02(+) |
| randn-500-1000 | -3.9e+03 | 20 | 1.26e+03 | 9.29e+02 | 8.40e+02 | 8.26e+02 | 0.00e+00 | 5.05e+02(+) | 2.17e+01(+) |
| w1a-2477-300 | -1.6e+03 | 100 | 4.99e+03 | 2.04e+03 | 7.31e+03 | 5.24e+03 | 1.13e+01 | 0.00e+00(+) | 2.47e+01(+) |
| TDT2-500-1000 | 1.0e+04 | 100 | 1.08e+03 | 3.86e+03 | 4.45e+00 | 4.45e+00 | 0.00e+00 | 9.69e+02(+) | 4.80e-03(+) |
| 20News-8000-1000 | 1.0e+04 | 100 | 8.15e+02 | 1.19e+04 | 9.26e-01 | 9.26e-01 | 0.00e+00 | 6.68e+02(+) | 1.43e-03(+) |
| sector-6412-1000 | 9.9e+03 | 100 | 6.25e+01 | 1.09e+04 | 7.45e+01 | 7.45e+01 | 0.00e+00 | 8.27e+00(+) | 2.97e-01(+) |
| E2006-2000-1000 | 1.0e+04 | 100 | 1.08e+03 | 5.83e+03 | 6.72e-01 | 6.72e-01 | 0.00e+00 | 9.06e+02(+) | 4.94e-04(+) |
| MNIST-60000-784 | -3.6e+05 | 100 | 5.10e+04 | 6.43e+04 | 1.75e+05 | 1.62e+05 | 0.00e+00 | 4.48e+04(+) | 7.91e+04(+) |
| Gisette-3000-1000 | -1.1e+06 | 100 | 6.56e+04 | 4.70e+05 | 1.32e+05 | 3.29e+05 | 0.00e+00 | 2.66e+04(+) | 9.34e+04(+) |
| CnnCaltech-3000-1000 | 7.8e+03 | 100 | 9.39e+02 | 1.39e+04 | 1.76e+03 | 9.93e+02 | 0.00e+00 | 6.77e+01(+) | 1.42e+00(+) |
| Cifar-1000-1000 | -1.2e+05 | 100 | 2.86e+02 | 6.01e+04 | 3.72e+04 | 5.64e+04 | 2.23e+03 | 0.00e+00(+) | 2.61e+04(+) |
| randn-500-1000 | -1.8e+04 | 100 | 3.78e+03 | 1.49e+04 | 2.74e+03 | 2.74e+03 | 7.43e+00 | 1.62e+03(+) | 0.00e+00(+) |

Table 7: Comparisons of objective values $(F(\mathbf{X}) - F_{\min})$ of $L_1$ norm-based SPCA for all the compared methods with $\lambda = 100$. The $1^{st}$, $2^{nd}$, and $3^{rd}$ best results are colored with red, green and blue, respectively. If the objective values of 'LADMM+OBCD-R' (or 'ManPG+OBCD-R') are smaller than those of 'LADMM' (or 'ManPG') by a margin of $0.1 \times a$, where $a$ represents the objective values of 'LADMM' (or 'ManPG'), they will be marked with (+).

| data-m-n | $F_{\min}$ | r | LADMM | RSubGrad | ADMM | ManPG | OBCD-OR | LADMM + OBCD-R | ManPG +OBCD-R |
|---|---|---|---|---|---|---|---|---|---|
| $\lambda = 1000.00$, time limit=10 | | | | | | | | | |
| w1a-2477-300 | 7.0e+03 | 10 | 3.03e+02 | 2.81e+03 | 2.59e+03 | 2.59e+03 | 0.00e+00 | 0.00e+00(+) | 0.00e+00(+) |
| TDT2-500-1000 | 1.0e+04 | 10 | 1.24e+03 | 1.39e+04 | 3.40e+00 | 3.40e+00 | 0.00e+00 | 5.21e+02(+) | 3.09e-02(+) |
| 20News-8000-1000 | 1.0e+04 | 10 | 8.97e+02 | 7.55e+04 | 1.49e+00 | 1.49e+00 | 0.00e+00 | 3.20e+02(+) | 3.07e-02(+) |
| sector-6412-1000 | 1.0e+04 | 10 | 2.88e+01 | 7.59e+04 | 3.00e+01 | 3.00e+01 | 0.00e+00 | 1.29e-01(+) | 3.68e-01(+) |
| E2006-2000-1000 | 1.0e+04 | 10 | 3.88e+02 | 3.77e+04 | 6.18e-01 | 6.18e-01 | 0.00e+00 | 3.87e+02(+) | 1.59e-03(+) |
| MNIST-60000-784 | -2.2e+05 | 10 | 1.69e+02 | 2.27e+05 | 2.27e+05 | 2.27e+05 | 1.71e+04 | 0.00e+00(+) | 9.07e+03(+) |
| Gisette-3000-1000 | -1.0e+06 | 10 | 2.26e+04 | 5.38e+05 | 6.58e+04 | 4.70e+05 | 0.00e+00 | 1.61e+04(+) | 4.39e+04(+) |
| CnnCaltech-3000-1000 | 9.6e+03 | 10 | 1.06e+02 | 7.01e+04 | 1.20e+03 | 2.90e+02 | 0.00e+00 | 0.00e+00(+) | 3.80e+00(+) |
| Cifar-1000-1000 | -9.2e+04 | 10 | 1.89e+03 | 1.14e+05 | 0.00e+00 | 1.00e+05 | 9.99e+04 | 1.80e+03(+) | 9.99e+04(+) |
| randn-500-1000 | 7.0e+03 | 10 | 3.66e+02 | 8.30e+04 | 3.14e+04 | 5.11e+02 | 0.00e+00 | 8.04e+00(+) | 4.69e+00(+) |
| w1a-2477-300 | 1.5e+04 | 20 | 5.30e+02 | 5.72e+03 | 3.90e+03 | 3.90e+03 | 0.00e+00 | 0.00e+00(+) | 0.00e+00(+) |
| TDT2-500-1000 | 2.0e+04 | 20 | 1.40e+03 | 7.01e+04 | 3.75e+00 | 3.75e+00 | 0.00e+00 | 9.85e+02(+) | 1.10e-02(+) |
| 20News-8000-1000 | 2.0e+04 | 20 | 1.45e+03 | 2.56e+05 | 1.66e+00 | 1.66e+00 | 0.00e+00 | 1.45e+03(+) | 1.30e-02(+) |
| sector-6412-1000 | 2.0e+04 | 20 | 6.06e+02 | 2.06e+05 | 4.27e+01 | 4.27e+01 | 0.00e+00 | 1.37e+00(+) | 1.19e+00(+) |
| E2006-2000-1000 | 2.0e+04 | 20 | 1.65e+03 | 8.70e+04 | 6.38e-01 | 6.38e-01 | 0.00e+00 | 7.18e+02(+) | 1.83e-03(+) |
| MNIST-60000-784 | -2.2e+05 | 20 | 3.85e+03 | 2.41e+05 | 2.41e+05 | 2.41e+05 | 2.71e+04 | 0.00e+00(+) | 3.00e+04(+) |
| Gisette-3000-1000 | -9.9e+05 | 20 | 1.41e+04 | 6.98e+05 | 1.15e+05 | 3.45e+05 | 0.00e+00 | 2.26e+03(+) | 9.40e+04(+) |
| CnnCaltech-3000-1000 | 1.9e+04 | 20 | 2.01e+02 | 2.05e+05 | 8.77e+02 | 3.90e+02 | 0.00e+00 | 8.32e-01(+) | 4.37e+00(+) |
| Cifar-1000-1000 | -8.1e+04 | 20 | 1.52e+02 | 2.22e+05 | 1.04e+05 | 9.83e+04 | 9.72e+04 | 0.00e+00(+) | 9.72e+04(+) |
| randn-500-1000 | 1.4e+04 | 20 | 4.49e+02 | 1.98e+05 | 3.61e+04 | 8.21e+02 | 0.00e+00 | 1.10e+01(+) | 1.69e+01(+) |
| w1a-2477-300 | 8.8e+04 | 100 | 3.43e+03 | 2.56e+05 | 8.92e+03 | 6.89e+03 | 3.00e+00 | 0.00e+00(+) | 0.00e+00(+) |
| TDT2-500-1000 | 1.0e+05 | 100 | 6.91e+03 | 1.10e+06 | 4.45e+00 | 4.45e+00 | 4.80e-03 | 5.95e+03(+) | 0.00e+00(+) |
| 20News-8000-1000 | 1.0e+05 | 100 | 3.61e+03 | 1.88e+06 | 9.25e-01 | 9.25e-01 | 1.01e-03 | 3.53e+03(+) | 0.00e+00(+) |
| sector-6412-1000 | 1.0e+05 | 100 | 4.16e+03 | 1.79e+06 | 7.42e+01 | 7.42e+01 | 8.04e-01 | 3.46e+03(+) | 0.00e+00(+) |
| E2006-2000-1000 | 1.0e+05 | 100 | 4.18e+03 | 1.19e+06 | 6.71e-01 | 6.71e-01 | 1.50e-03 | 4.18e+03(+) | 0.00e+00(+) |
| MNIST-60000-784 | -1.7e+05 | 100 | 5.23e+04 | 1.05e+06 | 1.02e+05 | 2.72e+05 | 1.05e+04 | 0.00e+00(+) | 5.25e+03(+) |
| Gisette-3000-1000 | -9.9e+05 | 100 | 5.95e+04 | 2.41e+06 | 4.30e+05 | 5.84e+05 | 3.49e+04 | 0.00e+00(+) | 3.06e+05(+) |
| CnnCaltech-3000-1000 | 9.8e+04 | 100 | 7.61e+02 | 1.90e+06 | 1.40e+03 | 9.92e+02 | 3.59e+00 | 0.00e+00(+) | 4.00e-02(+) |
| Cifar-1000-1000 | -2.5e+03 | 100 | 1.28e+03 | 1.94e+06 | 8.69e+04 | 8.69e+04 | 8.31e+04 | 0.00e+00(+) | 8.31e+04(+) |
| randn-500-1000 | 7.2e+04 | 100 | 2.91e+03 | 1.90e+06 | 2.76e+06 | 2.74e+03 | 7.14e+00 | 6.87e+02(+) | 0.00e+00(+) |
| $\lambda = 1000.00$, time limit=30 | | | | | | | | | |
| w1a-2477-300 | 7.0e+03 | 10 | 3.03e+02 | 2.64e+03 | 2.59e+03 | 2.59e+03 | 0.00e+00 | 0.00e+00(+) | 0.00e+00(+) |
| TDT2-500-1000 | 1.0e+04 | 10 | 1.24e+03 | 5.07e+03 | 3.40e+00 | 3.40e+00 | 0.00e+00 | 5.20e+02(+) | 3.09e-02(+) |
| 20News-8000-1000 | 1.0e+04 | 10 | 8.96e+02 | 3.10e+04 | 1.49e+00 | 1.49e+00 | 0.00e+00 | 3.20e+02(+) | 3.07e-02(+) |
| sector-6412-1000 | 1.0e+04 | 10 | 2.88e+01 | 2.42e+04 | 3.00e+01 | 3.00e+01 | 0.00e+00 | 1.29e-01(+) | 3.68e-01(+) |
| E2006-2000-1000 | 1.0e+04 | 10 | 3.87e+02 | 1.26e+04 | 6.18e-01 | 6.18e-01 | 0.00e+00 | 3.87e+02(+) | 1.59e-03(+) |
| MNIST-60000-784 | -2.2e+05 | 10 | 1.72e+02 | 2.27e+05 | 2.27e+05 | 2.27e+05 | 1.68e+04 | 0.00e+00(+) | 8.98e+03(+) |
| Gisette-3000-1000 | -1.0e+06 | 10 | 2.41e+04 | 4.51e+05 | 6.72e+04 | 2.94e+05 | 0.00e+00 | 1.76e+04(+) | 3.24e+04(+) |
| CnnCaltech-3000-1000 | 9.6e+03 | 10 | 1.06e+02 | 2.33e+04 | 1.19e+03 | 2.90e+02 | 0.00e+00 | 0.00e+00(+) | 3.80e+00(+) |
| Cifar-1000-1000 | -9.2e+04 | 10 | 1.89e+03 | 2.35e+04 | 0.00e+00 | 1.00e+05 | 9.99e+04 | 1.80e+03(+) | 9.99e+04(+) |
| randn-500-1000 | 7.0e+03 | 10 | 3.66e+02 | 2.66e+04 | 3.10e+04 | 5.11e+02 | 0.00e+00 | 5.13e+00(+) | 4.69e+00(+) |
| w1a-2477-300 | 1.5e+04 | 20 | 5.30e+02 | 4.27e+03 | 3.90e+03 | 3.90e+03 | 0.00e+00 | 0.00e+00(+) | 0.00e+00(+) |
| TDT2-500-1000 | 2.0e+04 | 20 | 1.40e+03 | 2.96e+04 | 3.75e+00 | 3.75e+00 | 0.00e+00 | 9.84e+02(+) | 1.10e-02(+) |
| 20News-8000-1000 | 2.0e+04 | 20 | 1.45e+03 | 9.38e+04 | 1.66e+00 | 1.66e+00 | 0.00e+00 | 1.45e+03(+) | 1.30e-02(+) |
| sector-6412-1000 | 2.0e+04 | 20 | 6.05e+02 | 6.35e+04 | 4.27e+01 | 4.27e+01 | 0.00e+00 | 1.37e+00(+) | 1.19e+00(+) |
| E2006-2000-1000 | 2.0e+04 | 20 | 1.65e+03 | 3.21e+04 | 6.38e-01 | 6.38e-01 | 0.00e+00 | 7.18e+02(+) | 1.83e-03(+) |
| MNIST-60000-784 | -2.2e+05 | 20 | 3.90e+03 | 2.41e+05 | 2.41e+05 | 2.41e+05 | 2.42e+04 | 0.00e+00(+) | 2.78e+04(+) |
| Gisette-3000-1000 | -1.0e+06 | 20 | 1.91e+04 | 5.27e+05 | 1.20e+05 | 3.38e+05 | 0.00e+00 | 7.17e+03(+) | 1.00e+05(+) |
| CnnCaltech-3000-1000 | 1.9e+04 | 20 | 2.02e+02 | 6.88e+04 | 8.74e+02 | 3.90e+02 | 0.00e+00 | 1.09e+00(+) | 4.63e+00(+) |
| Cifar-1000-1000 | -8.1e+04 | 20 | 1.56e+02 | 9.10e+04 | 1.04e+05 | 9.83e+04 | 9.72e+04 | 0.00e+00(+) | 9.72e+04(+) |
| randn-500-1000 | 1.4e+04 | 20 | 4.49e+02 | 8.47e+04 | 3.54e+04 | 8.21e+02 | 0.00e+00 | 1.10e+01(+) | 1.69e+01(+) |
| w1a-2477-300 | 8.8e+04 | 100 | 3.43e+03 | 1.06e+05 | 8.92e+03 | 6.89e+03 | 0.00e+00 | 0.00e+00(+) | 0.00e+00(+) |
| TDT2-500-1000 | 1.0e+05 | 100 | 6.78e+03 | 7.04e+05 | 4.45e+00 | 4.45e+00 | 0.00e+00 | 5.55e+03(+) | 2.74e-03(+) |
| 20News-8000-1000 | 1.0e+05 | 100 | 3.61e+03 | 1.38e+06 | 9.26e-01 | 9.26e-01 | 0.00e+00 | 3.53e+03(+) | 1.41e-03(+) |
| sector-6412-1000 | 1.0e+05 | 100 | 4.10e+03 | 1.28e+06 | 7.45e+01 | 7.45e+01 | 0.00e+00 | 3.40e+03(+) | 3.12e-01(+) |
| E2006-2000-1000 | 1.0e+05 | 100 | 4.13e+03 | 7.75e+05 | 6.72e-01 | 6.72e-01 | 0.00e+00 | 4.13e+03(+) | 4.60e-04(+) |
| MNIST-60000-784 | -1.7e+05 | 100 | 5.23e+04 | 6.81e+05 | 1.02e+05 | 2.72e+05 | 3.46e+04 | 0.00e+00(+) | 5.25e+03(+) |
| Gisette-3000-1000 | -9.9e+05 | 100 | 5.98e+04 | 1.91e+06 | 4.30e+05 | 5.83e+05 | 2.23e+04 | 0.00e+00(+) | 3.06e+05(+) |
| CnnCaltech-3000-1000 | 9.8e+04 | 100 | 7.62e+02 | 1.31e+06 | 1.39e+03 | 9.93e+02 | 0.00e+00 | 1.41e+00(+) | 1.23e+00(+) |
| Cifar-1000-1000 | -1.9e+03 | 100 | 6.77e+02 | 1.48e+06 | 8.63e+04 | 8.63e+04 | 8.25e+04 | 0.00e+00(+) | 8.25e+04(+) |
| randn-500-1000 | 7.2e+04 | 100 | 2.91e+03 | 1.35e+06 | 2.22e+05 | 2.74e+03 | 0.00e+00 | 6.90e+02(+) | 3.33e+00(+) |
| $\lambda = 1000.00$, time limit=60 | | | | | | | | | |
| w1a-2477-300 | 7.0e+03 | 10 | 3.03e+02 | 2.58e+03 | 2.59e+03 | 2.59e+03 | 0.00e+00 | 0.00e+00(+) | 0.00e+00(+) |
| TDT2-500-1000 | 1.0e+04 | 10 | 1.24e+03 | 1.98e+03 | 3.40e+00 | 3.40e+00 | 0.00e+00 | 5.19e+02(+) | 3.09e-02(+) |
| 20News-8000-1000 | 1.0e+04 | 10 | 8.94e+02 | 1.32e+04 | 1.49e+00 | 1.49e+00 | 0.00e+00 | 3.19e+02(+) | 3.07e-02(+) |
| sector-6412-1000 | 1.0e+04 | 10 | 2.88e+01 | 1.18e+04 | 3.00e+01 | 3.00e+01 | 0.00e+00 | 1.29e-01(+) | 3.68e-01(+) |
| E2006-2000-1000 | 1.0e+04 | 10 | 3.87e+02 | 5.62e+03 | 6.18e-01 | 6.18e-01 | 0.00e+00 | 3.87e+02(+) | 1.59e-03(+) |
| MNIST-60000-784 | -2.2e+05 | 10 | 1.71e+02 | 2.27e+05 | 2.27e+05 | 2.27e+05 | 1.16e+04 | 0.00e+00(+) | 8.33e+03(+) |
| Gisette-3000-1000 | -1.0e+06 | 10 | 2.62e+04 | 3.30e+05 | 6.93e+04 | 2.65e+05 | 0.00e+00 | 1.97e+04(+) | 4.01e+04(+) |
| CnnCaltech-3000-1000 | 9.6e+03 | 10 | 1.06e+02 | 1.29e+04 | 1.19e+03 | 2.90e+02 | 0.00e+00 | 0.00e+00(+) | 3.80e+00(+) |
| Cifar-1000-1000 | -9.2e+04 | 10 | 1.89e+03 | 1.30e+04 | 0.00e+00 | 1.00e+05 | 9.99e+04 | 1.80e+03(+) | 9.99e+04(+) |
| randn-500-1000 | 7.0e+03 | 10 | 3.66e+02 | 1.44e+04 | 3.09e+04 | 5.11e+02 | 0.00e+00 | 8.04e+00(+) | 4.69e+00(+) |
| w1a-2477-300 | 1.5e+04 | 20 | 5.30e+02 | 4.03e+03 | 3.90e+03 | 3.90e+03 | 0.00e+00 | 0.00e+00(+) | 0.00e+00(+) |
| TDT2-500-1000 | 2.0e+04 | 20 | 1.40e+03 | 1.19e+04 | 3.75e+00 | 3.75e+00 | 0.00e+00 | 9.82e+02(+) | 1.10e-02(+) |
| 20News-8000-1000 | 2.0e+04 | 20 | 1.45e+03 | 5.00e+04 | 1.66e+00 | 1.66e+00 | 0.00e+00 | 1.45e+03(+) | 1.08e-02(+) |
| sector-6412-1000 | 2.0e+04 | 20 | 6.05e+02 | 3.08e+04 | 4.27e+01 | 4.27e+01 | 0.00e+00 | 1.37e+00(+) | 1.19e+00(+) |
| E2006-2000-1000 | 2.0e+04 | 20 | 1.65e+03 | 1.64e+04 | 6.38e-01 | 6.38e-01 | 0.00e+00 | 7.18e+02(+) | 1.70e-03(+) |
| MNIST-60000-784 | -2.2e+05 | 20 | 3.92e+03 | 2.41e+05 | 2.41e+05 | 2.41e+05 | 1.17e+04 | 0.00e+00(+) | 2.85e+04(+) |
| Gisette-3000-1000 | -1.0e+06 | 20 | 2.03e+04 | 4.31e+05 | 1.22e+05 | 4.61e+05 | 0.00e+00 | 7.84e+03(+) | 1.07e+05(+) |
| CnnCaltech-3000-1000 | 1.9e+04 | 20 | 2.02e+02 | 3.98e+04 | 8.74e+02 | 3.90e+02 | 0.00e+00 | 1.09e+00(+) | 4.63e+00(+) |
| Cifar-1000-1000 | -8.1e+04 | 20 | 1.55e+02 | 4.42e+04 | 1.04e+05 | 9.83e+04 | 9.72e+04 | 0.00e+00(+) | 9.72e+04(+) |
| randn-500-1000 | 1.4e+04 | 20 | 4.49e+02 | 4.65e+04 | 3.54e+04 | 8.21e+02 | 0.00e+00 | 1.10e+01(+) | 1.69e+01(+) |
| w1a-2477-300 | 8.8e+04 | 100 | 3.43e+03 | 4.16e+04 | 8.92e+03 | 6.89e+03 | 0.00e+00 | 0.00e+00(+) | 0.00e+00(+) |
| TDT2-500-1000 | 1.0e+05 | 100 | 6.78e+03 | 4.62e+05 | 4.45e+00 | 4.45e+00 | 0.00e+00 | 5.20e+03(+) | 4.80e-03(+) |
| 20News-8000-1000 | 1.0e+05 | 100 | 3.61e+03 | 9.50e+05 | 9.26e-01 | 9.26e-01 | 0.00e+00 | 3.53e+03(+) | 1.43e-03(+) |
| sector-6412-1000 | 1.0e+05 | 100 | 4.10e+03 | 8.69e+05 | 7.45e+01 | 7.45e+01 | 0.00e+00 | 3.40e+03(+) | 2.97e-01(+) |
| E2006-2000-1000 | 1.0e+05 | 100 | 4.13e+03 | 4.91e+05 | 6.72e-01 | 6.72e-01 | 0.00e+00 | 4.13e+03(+) | 4.94e-04(+) |
| MNIST-60000-784 | -1.8e+05 | 100 | 5.53e+04 | 4.36e+05 | 1.06e+05 | 2.75e+05 | 0.00e+00 | 2.94e+03(+) | 8.25e+03(+) |
| Gisette-3000-1000 | -9.9e+05 | 100 | 5.98e+04 | 1.45e+06 | 4.30e+05 | 6.84e+05 | 1.43e+04 | 0.00e+00(+) | 3.07e+05(+) |
| CnnCaltech-3000-1000 | 9.8e+04 | 100 | 7.63e+02 | 9.73e+05 | 1.39e+03 | 9.93e+02 | 0.00e+00 | 1.58e+00(+) | 1.62e+00(+) |
| Cifar-1000-1000 | -1.9e+03 | 100 | 6.75e+02 | 1.16e+06 | 8.63e+04 | 8.63e+04 | 8.25e+04 | 0.00e+00(+) | 8.25e+04(+) |
| randn-500-1000 | 7.2e+04 | 100 | 2.92e+03 | 9.34e+05 | 2.22e+05 | 2.75e+03 | 0.00e+00 | 6.92e+02(+) | 5.80e+00(+) |

Table 8: Comparisons of objective values $(F(\mathbf{X}) - F_{\min})$ of $L_1$ norm-based SPCA for all the compared methods with $\lambda = 1000$. The $1^{st}$, $2^{nd}$, and $3^{rd}$ best results are colored with red, green and blue, respectively. If the objective values of 'LADMM+OBCD-R' (or 'ManPG+OBCD-R') are smaller than those of 'LADMM' (or 'ManPG') by a margin of $0.1 \times a$, where $a$ represents the objective values of 'LADMM' (or 'ManPG'), they will be marked with $(+)$.

