# OpenReview forum: "A Block Coordinate Descent Method for Nonsmooth Composite Optimization under Orthogonality Constraints"
_ICLR.cc/2025/Conference — Submitted to ICLR 2025_

### Official Review · Reviewer_eCrN · 2024-10-31

**Soundness:** 3
**Presentation:** 2
**Contribution:** 3
**Rating:** 6
**Confidence:** 4

**Summary:**

The paper presents OBC to tackle nonsmooth composite optimization with orthogonality constraints. OBCD updates multiple rows of the matrix variable while globally solving a small nonsmooth optimization problem. The authors demonstrate that OBCD converges to block-k stationary points, providing stronger optimality guarantees than traditional critical points, and is the first greedy descent method in this domain to exhibit monotonicity. They establish strong limit-point convergence under the Kurdyka-Lojasiewicz inequality, solve subproblems with breakpoint searching techniques, and provide greedy strategies to improve computational efficiency. Extensive experiments validate OBCD's superior performance in both accuracy and efficiency compared to existing methods.

**Strengths:**

1. This paper proposes a very interesting row-wise BCD method for composite optimization over orthogonality constraints, which enables low-cost update.
2. The authors show that their algorithm converges to a BSk-point, which is sharper than usual critical points. This is a great point.
3. The proposed algorithm outperforms other existing methods in various problems.

**Weaknesses:**

1. The paper is not well-written. The organization requires a thorough revising to make this paper more readable.
2. The Asm-iii in Line 49 is restrictive. As the problem can be hard to solve. This can hinder the practical meaning of the theoretical guarantee. The authors propose to use the majorization surrogate and minimize the majorization surrogate only in k=2. Did the authors considered solving subproblems inexactly?

**Questions:**

1. In Figure 1, the objective function values of other methods seem unchanged. Could the authors explain the reason and elaborate more about the performance of other methods? Besides, sharper notions of stationarity do not necessarily lead to lower objective function values. It can be better to present some average results based on multiple tests.
2. Could the authors add some discussion on the possible parallel implementation of OBCD? Or explain possible difficulties of parallel implementation. Because this aspect would greatly strengthen the practical popularity of OBCD.
3. From Proposition 4.6, the KL holds locally. To apply the KL property, it is required that the iterates stay in the local region. However, the randomness in the algorithm can make the iterates fail to stay in the local region. Could the authors explain how to overcome this difficulty?
4. Could the authors revise the presentation and flow of this paper to make it more readable?
5. Could the authors add some discussion on how this BCD framework can be generalized to other manifold?
7. There are also other papers that tackle nonsmooth manifold opt using BCD (e.g., https://arxiv.org/pdf/2409.01770). Could the authors add some related references?

---

> ### Author Response · Authors · 2024-11-22
>
> Thank you for your efforts in evaluating our manuscript and providing encouraging feedback.
>
> **Question 1: In Figure 1, the objective function values of other methods seem unchanged. Could the authors explain the reason and elaborate more about the performance of other methods?**
>
> **Response.** This is because, while other methods often become trapped in poor local minima, OBCD successfully escapes these minima and generally achieves lower objective values. This observation aligns with our theory, which suggests that our methods converge to stronger stationary points.
>
> **Question 2: Besides, sharper notions of stationarity do not necessarily lead to lower objective function values. It can be better to present some average results based on multiple tests.**
>
> **Response.** Since OBCD is the first greedy descent method developed for this class of composite problems, we have demonstrated the clear advantages of the proposed approach. For simplicity, we have reported results from a single run, as we thought that multiple-run comparisons are not always necessary.
>
> However, upon request, we will include average results based on multiple tests in the revised manuscript. Thank you for your thoughtful suggestions.
>
> **Question 3: Could the authors add some discussion on the possible parallel implementation of OBCD? Or explain possible difficulties of parallel implementation. Because this aspect would greatly strengthen the practical popularity of OBCD.**
>
> **Response.**
>
> 1. We have discussed the potential parallel implementation of OBCD, such as the Jacobi update strategy, as mentioned in Lines 108–109 of the paper.
>
> 2. Another possible avenue for parallel acceleration of the proposed method is the incorporation of variance reduction techniques to further decrease its oracle complexity.
>
>
>
> **Question 4: From Proposition 4.6, the KL holds locally. To apply the KL property, it is required that the iterates stay in the local region. However, the randomness in the algorithm can make the iterates fail to stay in the local region. Could the authors explain how to overcome this difficulty?**
>
> **Response.**
>
> 1. OBCD is a greedy descent algorithm. Specifically, when $X^t$ is sufficiently close to $\ddot{X}$, all subsequent solutions remain sufficiently close to $\ddot{X}$ as well.
>
> 2. We have added a strongly convex term $0.5 \alpha ||V-I||_F^2$ to the majorization function, as shown in Equation (9), to ensure that the iterates do not increase excessively.
>
>
>
> **Question 5: Could the authors revise the presentation and flow of this paper to make it more readable?**
>
> **Response.** We will make every effort to improve the presentation and logical flow of this paper in future revisions.
>
>
> **Question 6: Could the authors add some discussion on how this BCD framework can be generalized to other manifold?**
>
> **Response.**
>
> 1. The proposed method can be extended to the generalized Stiefel manifold (https://arxiv.org/abs/2405.01702), defined as $\Omega =\{X | X^TBX = I\}$. Assume that $X \in \Omega$. We denote $C = B^{1/2}$ and $D = B^{-1/2}$. It is straightforward to verify that the following update rule:
>
> $X^+=X + D U_B (V-I) U_B^T CX$
>
>
> ensures that the next iterate remains on the generalized Stiefel manifold, i.e., $X^+\in \Omega$.
>
> 2. The proposed method is also adaptable to other manifolds, such as the J-orthogonal manifold, the symplectic Stiefel manifold (https://arxiv.org/abs/2006.15226), and others, demonstrating its flexibility in addressing a broader class of constrained optimization problems.
>
>
> **Question 7: There are also other papers that tackle nonsmooth manifold opt using BCD (e.g., https://arxiv.org/pdf/2409.01770). Could the authors add some related references?**
>
> **Response.** As this paper is highly relevant to our work, we will include this reference in the updated version of our paper.

---

> > ### Comment · Reviewer_eCrN · 2024-11-23
> > **Reply**
> >
> > I appreciate the algorithm idea. However, I think the whole organization of this paper need to be extensively revised, which can better attract audiences.

---

> > > ### Author Response · Authors · 2024-11-28
> > >
> > > We have extensively revised the manuscript in accordance with the reviewers' suggestions, including improvements to the writing and some technical results.
> > >
> > > We hope the reviewers will take the time to review the updated version.

---

### Official Review · Reviewer_FMaS · 2024-11-03

**Soundness:** 2
**Presentation:** 2
**Contribution:** 2
**Rating:** 3
**Confidence:** 4

**Summary:**

The paper is concerned with solving a composite optimization problem, whose objective is the sum of a smooth and non-smooth function, over the set of $n\times r$ orthogonal matrices, called the Stiefel manifold. The main innovation of the proposed work is to consider a form of **block-coordinate update**, in which only some, $k\leq n$, rows are updated.

Having a method that updates only $k$-rows would have great computationally practical advanteges: 1) the gradient of the function does not have to be evaluated on all of the $nr$ entries, 2) only $kr$ entries have to be updated, 3) data can be distributed based on the updated blocks. Moreover, if the update consists of only orthogonal rotation by a $k\times k$ on the selected columns, the update automatically belongs to the feasible set.

The main contributions of the paper are:
* the concept of $k$-row update,
* strong limit point convergence of the method under KL-inequality
* greedy strategies for selecting the $k$ indices which are to be updated

**Strengths:**

I find the concept of updating only $k$-selected rows in the composite optimization over the Stiefel manifold to be an important and useful problem. Having good algorithms in this setting allows for all kind of additional applications, including distributional and privacy aware settings. The idea of the algorithm is straightforward to understand. The authors also focus on $k=2$ case when the iteration subproblems have closed form solution and connect these with classical notions of Givens rotations and Jacobi reflections.

**Weaknesses:**

While I enjoyed reading the paper, there are several points that are concerning to me.

**Convoluted presentation of some results**
At some points in the paper, there are basic linear algebra concepts presented in an unusual convoluted way. For example the start of Section 2, lines 136 to 186 basically say, that if we have an update $X^+ = V_B X$, where the matrix $V_B$ is an identity apart from a diagonal block indexed by $B$ where it has an orthogonal $k\times k$ matrix, then the next iterate will satisfy $(X^+)^\top X^+ = X^\top V_B^\top V_B X = X^\top X$ and that the distance will depend on the magnitude of the rotation $V_B$. However, since the authors write the update in an unusual form in eq. (4), this fact is, in my opinion, less apparent, than by using orthogonality of $V_B$.

**Inaccuracies**
Similarly, there is a discussion of using not only Givens rotations, stating the advantage of using Jacobi reflections since it will generate whole Stiefel manifold. This is also not entirely accurate I think. Givens rotations generate the whole SO(n) which is sufficient in solving PCA since there is an inherent ambiguity in the solution up to $\pm1$ sign anyway.

**Other minor typos/writing fixes**
* Another difficulty I had was understanding the notation, for example: $[\cdot]_BB$ appearing for the first time in line 199, becomes slightly clear only later in line 224. Overall, I would also suggest to bring more focus on the computational advantages of the formulation.
* The notation of $\mathcal{M}$ in line 187 should be Stiefel manifold I think.

**Questions:**

* What is the impact of different strategies for choosing $B_k$?
* How is the problem in (10) solved when $Q$ is not diagonal?
* In the proof of the convergence Theorem 4.2, in line 1495, how come you can upper bound the inequality with the stationary point of the algorithm?

---

> ### Author Response · Authors · 2024-11-22
>
> Thank you for your efforts in evaluating our manuscript.
>
> **Weakness 1: The authors write the update in an unusual form in eq. (4), this fact is, in my opinion, less apparent, than by using orthogonality of $V_B$.**
>
> **Response:** We present equivalent expressions using matrix multiplication and matrix addition. These two forms of expressing the update rule play a crucial role in our analysis, particularly the matrix addition rule in (4). This specific rule is indispensable for theoretical analysis and for performing the majorization-minimization procedure.
>
> **Weakness 2: Inaccuracies: Similarly, there is a discussion of using not only Givens rotations, stating the advantage of using Jacobi reflections since it will generate whole Stiefel manifold. This is also not entirely accurate I think. Givens rotations generate the whole SO(n) which is sufficient in solving PCA since there is an inherent ambiguity in the solution up to $\pm 1$ sign anyway.**
>
> **Response:**
>
> 1. For the PCA problem, there is no additional benefit since the objective function is symmetric (without a linear term).
>
> 2. However, for certain quadratic functions that include a linear term, consider the example in Line 1008: $\min_V ||V-A||_F^2, s.t. V\in St(2,2)$, where $A=[1, 0;-1, -1]$ is a reflection matrix. In this simple example, the reflection matrix plays a crucial role and leads to improved optimality, achieving a **strictly** lower objective value compared to using only a rotation matrix.
>
> **Other minor 1: Other minor typos/writing fixes Another difficulty I had was understanding the notation, for example:  $[\cdot]_{BB}$ appearing for the first time in line 199, becomes slightly clear only later in line 224.**
>
> **Response:** We adopt the Matlab colon notation to denote indices that describe submatrices. See Line 726-727 in the paper. In our revision, we will explicitly mention this notation for clarity.
>
> **Other minor 2: Overall, I would also suggest to bring more focus on the computational advantages of the formulation.**
>
> **Response:**
>
> 1. OBCD operates on k rows of the solution matrix, offering lower computational complexity per iteration for $k\geq 2$.
>
> 2. OBCD is a greedy descent method with monotonicity for this problem class, offering provable convergence guarantees.
>
>
>
> **Other minor 3: The notation of M in line 187 should be Stiefel manifold I think.**
>
>
> **Response:** $M = St(n, r)$ denotes the Stiefel manifold. See Line37.
>
>
> **Questions 1: What is the impact of different strategies for choosing $B_k$?**
>
> **Response:**
>
> 1. When the subgradient provides a good estimate, the greedy strategy often results in faster convergence. Otherwise, random strategies are recommended.
>
> 2. Random strategies frequently achieve faster convergence compared to cyclic strategies.
>
> 3. The cyclic strategy has the advantage of being a deterministic approach.
>
>
> **Questions 2: How is the problem in (10) solved when Q is not diagonal?**
>
> 1. When $h(\cdot)=0$ and $Q$ is not diagonal, Problem (10) can be solved to local optimality using gradient projection methods.
>
> 2. When $h(\cdot)\neq 0$ and $k=2$, Problem (10) can be addressed using one-dimensional search strategies, such as the break-point search method.
>
>
> **Questions 3: In the proof of the convergence Theorem 4.2, in line 1495, how come you can upper bound the inequality with the stationary point of the algorithm?**
>
> Note that $\ddot{X}$ is a limit point of the sequence $(X^0,X^1,X^2,X^3,\ldots,X^{\infty})$. The proposed algorithm is a descent method, and it is evident that $F(\ddot{X}) =F(\ddot{X}^{\infty}) \leq F(\ddot{X}^k)$ for all $k$.

---

> > ### Comment · Reviewer_FMaS · 2024-11-25
> >
> > I thank the authors for their reply to my review. While I like the algorithm's idea, I share my concerns with the other reviewers regarding the novelty and writing of the paper, and my score remains unchanged.

---

> ### Author Response · Authors · 2024-11-28
>
> We have extensively revised the manuscript based on the reviewers' suggestions, including improvements to both the writing and the technical content. We have made every effort to enhance its quality and sincerely hope the reviewers will take the time to review the updated version.

---

> > ### Author Response · Authors · 2024-12-02
> >
> > Dear Reviewer FMaS,
> >
> > As the discussion concludes, we hope our response has addressed your concerns.
> >
> > If so, we kindly ask for your reconsideration of the score.
> >
> > If any issues remain, please let us know, and we will work to improve our submission.

---

### Official Review · Reviewer_DbfY · 2024-11-04

**Soundness:** 3
**Presentation:** 1
**Contribution:** 2
**Rating:** 3
**Confidence:** 3

**Summary:**

The authors study a the problem of minimizing a particular class nonconvex nonsmooth function $f$ over the space of n x r orthogonal matrices (a Stiefel manifold $St(n,r)$ ). Three standard approaches for such problems are as follows: (1) Use gradient descent-type updates in the ambient Euclidean space but use projection onto $St(n,r)$;  (2) Make gradient descent update within the tangent space and retract onto the manifold, see [2]; and (3) Iteratively construct a majorizing surrogate $\hat{f}$ of the objective $f$ and minimize it within the manifold, see [1,3]. The last approach has been particularly successful for optimization problems on Stiefel manifolds, which is the topic of the current paper, see [1]. Due to the non-convex nature of both the objective and the constraint set, researchers typically aim at obtaining first-order convergence guarantees (asymptotically or non-asymptotically with sub-linear rate), suitably defined. Convergence rate can be improved by imposing additional assumptions such as the KL property on the objective.

The authors use an approach under the umbrella of block majorization-minimization, which progressively minimizes a majorizing surrogate of the block-restricted objective. There proposed method essentially reduces working with high-dimensional (n) orthonormal vectors to only k-dimensional (k fixed, but k=2 seems the only effective case) orthonormal columns, by subsampling k rows at random or according to a pre-determined schedule. The authors derive non-asymptotic iteration complexity to obtain approximate first-order optimal points and obtain improved convergence rate under the additional KL condition.

There are many existing works on block majorization-minimization in both the Euclidean and the Riemannian settings for nonconvex constrained problems, so the idea itself does not seem very novel. However, there are some interesting ideas and observations that the authors make in this work specifically for Stiefel manifold problems (I will say more on them in Strengths) that can be quite valuable for high-dimensional problems on Stiefel manifolds. But, unfortunately, the paper's presentation is very poor and it cannot be considered for a publication in top ML conferences such as ICLR. The authors fail to give a cohesive narrative of their work, arguments and evidences are fragmented across the main text and the appendix, numerous typos and undefined quantities, and so on (I will say more in the Weaknesses). I think the paper has a valuable contribution, but it needs a significant rewriting from ground-up. I will need to see the whole revised paper so that these issues are successfully addressed, which definitely goes beyond that is expected to be done during the rebuttal. Therefore, overall I recommend a rejection of the paper.


[1] Breloy, Kumar, Sun, and Palomar, "Majorization-Minimization on the Stiefel Manifold With Application to Robust Sparse PCA", IEEE TRANSACTIONS ON SIGNAL PROCESSING, VOL. 69, 2021

[2] Yuchen Li, Laura Balzano, Deanna Needell, Hanbaek Lyu, “Convergence and Complexity Guarantee for Inexact First-order Riemannian Optimization Algorithms.” ICML 2024

[3] Yuchen Li, Laura Balzano, Deanna Needell, Hanbaek Lyu, “Convergence and complexity of block majorization-minimization for constrained block-Riemannian optimization” arXiv:2312.10330 (2023)

**Strengths:**

1. Algorithm

The most novel part of their algorithm is that, instead of updating the entire orthogonal matrix $X\in \mathbb{R}^{n\times r}$ (here $n\gg 1$), they update only two chosen rows of $X$ by left-multiplying a suitable $2 x 2$ orthonormal matrix $V\in St(2,2)$ to those rows. The action of a large orthogonal matrix restricted on two rows does not need to be orthonormal. However, the authors make this simplifying assumption and shows that this does not loose any generality as far as first-order convergence is concerned. I think this is a nice observation. Practically, this brings down the per-iteration computational complexity to $O(n)$, which could be much less than the cost of computing the Riemannian gradient of some loss functions. The authors give an extensive details on how to solve the sub-problem involving $St(2,2)$ to make this approach executable. This algorithm could be quite practical for large-scale problems on Stiefel manifolds.

2. Optimality measure

The authors introduce a new measure of first-order optimality that is in general stronger than the usual (Riemannian) stationary points (block-k stationary points). Roughly speaking, the sub-problem should not be improved from the current parameter by an arbitrary adjustment using the 2 x 2 frame. This measure comes directly from the sub-problem of minimizing a majorizing surrogate over $St(2,2)$, so it depends on the particular surrogate $\mathcal{K}$ chosen. This is a somewhat unpleansent feature of their proposed optimality measure, which should in principle depend only on the information about the objective function at the given parameter (e.g., Riemannian subgradient norm of the objective). Nonetheless, the authors provide some examples (in Appendix C.1) that the number of critical points can be strictly larger than the number of their block-k stationary points.

3. Iteration complexity

The authors show that an epsilon-relaxation of their block-k stationarity decays is obtained after $O(\eps^{-1})$ iterations, matching  standard iteration complexity results in nonconvex optimization literature.

4. Improved iteration complexity under KL condition:

The authors show finite-length property (in Thm. 4.8)  under additional assumption.

5. Extensive computation and examples for the problems on Stiefel manifold.

The authors provide nice lemmas that are based on extensive computations using properties of Stiefel manifold along with numerical examples.

**Weaknesses:**

1. Presentation:

The paper does not read very well. This is partly because of the style of the paper writing, which puts most of texts within proposition, lemma, remarks and there are not much narratives that connects them. There are numerous places that quantities are not properly defined. Important equations are written in line within text, which makes it extremely hard to read. Space constraints granted, there are still well-written, easy-to-read theoretical papers in top ML conferences. The current manuscript looks like it is a math paper crammed into 10 pages without much effort to make it readable.

(1) In the very first paragraph, the authors state assumptions in text. This is extremely hard to read and almost impossible to refer back when reading further into the paper.

(2)

2. Assumptions:

In (2), the authors assume that the objective $f$ admits a quadratic surrogate of very specific form. This looks like some kind of restricted smoothness property but with the inner product depending on an additional kernel $$H$$. The authors should discuss if this is a reasonable assumption (e.g., if $f$ is $L$-smooth in the Euclidean or Riemannian sense, is it satisfied?) Otherwise, it is unclear if the paper's framework applies beyond a handful of objective functions (essentially $|| X - B||_{F}^{2}$ plus some nonsmooth regularization terms) and this makes the paper's scope very narrow.

3. Optimality measure:

While the optimality measure based on the authors' notion of block-k stationary points is interesting, it is hard to assess how it compares with the usual optimality measure based on (sub)gradient norms or the ones that directly relaxes the variational inequality defining stationary points. Also, another complication is that this measure involves the chosen surrogate as part of the definition, so it is not clear that if one has small BS_k measure then one has small subgradient norm or other standard stationarity measure.

Furthermore, the authors does not show asymptotically this proposed measure goes to zero. If this result is given, then using their hierarchy between optimality measures, one can deduce that any limit point of the algorithm is a BS_k-point, which is stationary point with additional potentially desirable properties.

4. Analysis assumes exact solver for the sub-problems:

While in Remark 2.5 the authors claim that "For general k and h(·), the subproblem may not be solved globally, but a critical point can still be reached. Although strong optimality may be lost, a critical point (discussed later) for the final configuration X∞ remains achievable." This claim is never justified in the paper. First, even with exact computation of sub-problems, they do not show that the iterates asymptotically converge to the stationary points. Second, to support their claim, the authors extend their analysis for Thm. 4.2 with the first point here under the assumption that the sub-problems are inexactly solved. Justifying this is in fact is a non-trivial problem. See [1] for such a result for block inexact Riemannian coordinate descent.

[1] Yuchen Li, Laura Balzano, Deanna Needell, Hanbaek Lyu, “Convergence and Complexity Guarantee for Inexact First-order Riemannian Optimization Algorithms.” ICML 2024

5. Analysis under KL:

Thm. 4.8 states that essentially the iterates converge after a finite number of iterations under the additional KL assumption. The authors do not specify the the exponent $\sigma\in [0,1)$ in the desingularization function $\varphi$. This cannot be true for all range of $\sigma$ and it should only hold when $\sigma=0$. See Thm. 2.9 in [1].

[1] Xu, Yangyang, and Wotao Yin. "A block coordinate descent method for regularized multiconvex optimization with applications to nonnegative tensor factorization and completion." SIAM Journal on imaging sciences 6.3 (2013): 1758-1789.

5. Minor comments:

(1) Below (2), the norm on $H$: is it the spectral norm?
(2) L47: $\lVert X \rVert_{p}$ is not coordinate-wise separable.
(3) L88: The authors imply that subgradient descent with dimihsing stepwise is a limitation. If the stepsizes diminish at a rate not too fast (e.g., $O(n^{-1/2})$), it still give an optimal iteration complexity. Please specify.
(4) The authors are recommended to cite the following references and discuss their relevance/differences.

[1] Breloy, Kumar, Sun, and Palomar, "Majorization-Minimization on the Stiefel Manifold With Application to Robust Sparse PCA", IEEE TRANSACTIONS ON SIGNAL PROCESSING, VOL. 69, 2021
[2] Yuchen Li, Laura Balzano, Deanna Needell, Hanbaek Lyu, “Convergence and Complexity Guarantee for Inexact First-order Riemannian Optimization Algorithms.” ICML 2024
[3] Yuchen Li, Laura Balzano, Deanna Needell, Hanbaek Lyu, “Convergence and complexity of block majorization-minimization for constrained block-Riemannian optimization” arXiv:2312.10330 (2023)

(5) Summary: The authors state that there are not much known results about block marjoziation-minimization or similar type of algorithms for nonconvex nonsmooth problems. This is simply not true. Please revise this assessment.
(6) In Lemma 2.1, the authors define $X^{+}$ and then specify $X$. They should first say "For any $X$, define $X^{+}=$.."
(7) In Algorithm 1, $\mathcal{K}$ is not defined nor referred.
(8) In eq. (6), remove comma.
(9) The construction of quadratic surrogate in (9) resembles very much the discussion in Breloy et a. [1]. The authors should discuss the relation.
(10) L215: The authors claim that the so-constructed quadratic surrogate can be exactly minimized over $St(k,k)$. But this is the case only if the nonsmooth part is zero (by SVD) or if $k=2$. This should be clearly stated.
(11) Table 1: The numbers are too small to read. There is no point of adding this kind of table since it is unreadable anyways.
(12) Appendix C.3: The authors compared computational complexity of computing the Riemannian gradient directly v.s. incrementally updating the gradient using their k-row approach.

**Questions:**

N/A

---

> ### Author Response · Authors · 2024-11-22
>
> Thank you for your careful reading and the effort you put into evaluating our manuscript.
>
> **Weakness 1: The paper does not read very well. This is partly because of the style of the paper writing, which puts most of texts within proposition, lemma, remarks and there are not much narratives that connects them. There are numerous places that quantities are not properly defined. Important equations are written in line within text, which makes it extremely hard to read. Space constraints granted, there are still well-written, easy-to-read theoretical papers in top ML conferences. The current manuscript looks like it is a math paper crammed into 10 pages without much effort to make it readable.(1) In the very first paragraph, the authors state assumptions in text. This is extremely hard to read and almost impossible to refer back when reading further into the paper.**
>
> **Response.** We will present the small-sized subproblem in a formal mathematical formulation.
>
>
> **Weakness 2: Assumptions: In (2), the authors assume that the objective $f$ admits a quadratic surrogate of very specific form. This looks like some kind of restricted smoothness property but with the inner product depending on an additional kernel $H$. The authors should discuss if this is a reasonable assumption (e.g., if $f$ is $L$-smooth in the Euclidean or Riemannian sense, is it satisfied?) Otherwise, it is unclear if the paper's framework applies beyond a handful of objective functions (essentially $||X-B||_F^2$ plus some nonsmooth regularization terms) and this makes the paper's scope very narrow.**
>
> **Response.**
>
> 1. Inequality (2) can be interpreted as a generalized smoothness condition. By setting $H = L \cdot I$, it reduces to the classical $L$-smoothness condition.
>
> 2. To achieve greater generality, we adopt the notion of $H$-smoothness, which provides a tighter upper bound and can potentially lead to improved optimality. (Note that for a quadratic function, inequality (2) holds with equality.)
>
>
> **Weakness 3: Optimality measure: While the optimality measure based on the authors' notion of block-k stationary points is interesting, it is hard to assess how it compares with the usual optimality measure based on (sub)gradient norms or the ones that directly relaxes the variational inequality defining stationary points. Also, another complication is that this measure involves the chosen surrogate as part of the definition, so it is not clear that if one has small BS_k measure then one has small subgradient norm or other standard stationarity measure.**
>
> **Response.** We have demonstrated in Lines 315–316 that any block-2 stationary point must be a critical point, where its (sub)gradient norm is zero.

---

> > ### Comment · Reviewer_DbfY · 2024-11-25
> >
> > >Weakness 2 on $\mathbf{H}$-smoothness:
> >
> > Thanks for the clarification. I suggest to define the notion of `$\mathbf{H}$-smoothness' of the objective $f$ as in (2) and remark that when $\mathbf{H}=L\mathbf{I}$, it reduces to the standard $L$-smoothness.
> >
> > >Weakness 3 on optimality measure:
> >
> > No, the authors did not get my point. The authors did show that their $BK_{k}=0$ implies stationarity. But they did not show that $BK_{k}\le \epsilon$ implies more standard stationary measure (e.g., gradient norm of the objective) is also small (e.g., $O(\epsilon)$). This discussion is important since Thm. 4.2 states that the proposed algorithm finds a point where $BK_{k}\le \epsilon$ within certain number of iterations, and it is unclear what would be the implication of this result in terms of the standard stationary measure. For instance, can the authors show $\textup{gradient norm}\le BK_{k}$? This is unclear since the LHS is a generic measure of stationarity but the RHS depends on the choice of surrogate.

---

> ### Author Response · Authors · 2024-11-22
>
> **Weakness 4: Furthermore, the authors does not show asymptotically this proposed measure goes to zero. If this result is given, then using their hierarchy between optimality measures, one can deduce that any limit point of the algorithm is a BS_k-point, which is stationary point with additional potentially desirable properties.**
>
> **Response.** We have shown in Theorem 4.2 that the proposed measure converges to zero. Specifically, after $T$ iterations, OBCD identifies an $\epsilon$-$BS_k$-point (see Lines 1507–1509 and 1535–1536).
>
> **Weakness 5: Analysis assumes exact solver for the sub-problems: While in Remark 2.5 the authors claim that "For general k and h(·), the subproblem may not be solved globally, but a critical point can still be reached. Although strong optimality may be lost, a critical point (discussed later) for the final configuration X∞ remains achievable." This claim is never justified in the paper.**
>
> **Response.** Assume that $\bar{V}^t$ is a local critical stationary point of the subproblem $\min_{V\in St(k,k)} K(V;X^t,B)$ and that the solution $\bar{V}^t$ satisfies $K(\bar{V}^t;X^t,B)\leq K(I_k;X^t,B)$, where $I_k$ is an identity matrix. By employing similar strategies as those outlined in Theorem 4.2, we can readily establish the sufficient condition $F(X^{t+1})-F(X^t) \leq - 0.5 \alpha ||V^t - I_k||_F^2$ and confirm the validity of this conclusion.
>
> **Weakness 6: First, even with exact computation of sub-problems, they do not show that the iterates asymptotically converge to the stationary points.**
>
> **Response.** In Theorem 4.2, we have demonstrated that OBCD asymptotically converges to the Block-k stationary points, regardless of whether the coordinates are selected randomly or cyclically.
>
> **Weakness 7: Second, to support their claim, the authors extend their analysis for Thm. 4.2 with the first point here under the assumption that the sub-problems are inexactly solved. Justifying this is in fact is a non-trivial problem. See [1] for such a result for block inexact Riemannian coordinate descent.
> [1] Yuchen Li, Laura Balzano, Deanna Needell, Hanbaek Lyu, “Convergence and Complexity Guarantee for Inexact First-order Riemannian Optimization Algorithms.” ICML 2024**
>
> **Response.** We assume that the objective function of the small-sized subproblem satisfies the descent property, expressed as $K(\bar{V}^t;X^t,B) \leq K(I_k;X^t,B)$. The suggested references will be cited in the updated manuscript.
>
>
> **Weakness 8: Analysis under KL: Thm. 4.8 states that essentially the iterates converge after a finite number of iterations under the additional KL assumption. The authors do not specify the the exponent $\sigma\in[0,1)$ in the desingularization function $\phi$. This cannot be true for all range of and it should only hold when $\sigma=0$. See Thm. 2.9 in [1]. [1] Xu, Yangyang, and Wotao Yin. "A block coordinate descent method for regularized multiconvex optimization with applications to nonnegative tensor factorization and completion." SIAM Journal on imaging sciences 6.3 (2013): 1758-1789.**
>
> **Response.** We note that our analysis establishes only the finite-length property of OBCD, which is independent of the KL exponent $\sigma$. In the future, we aim to extend this work by incorporating the convergence rate of OBCD through a deeper exploration of the KL exponent. Nonetheless, the finite-length property itself is a significant and state-of-the-art theoretical result.

---

> > ### Comment · Reviewer_DbfY · 2024-11-25
> >
> > >Weakness 4: Furthermore, the authors does not show asymptotically this proposed measure goes to zero. If this result is given, then using their hierarchy between optimality measures, one can deduce that any limit point of the algorithm is a BS_k-point, which is stationary point with additional potentially desirable properties.
> >
> > >Response. We have shown in Theorem 4.2 that the proposed measure converges to zero. Specifically, after  iterations, OBCD identifies an --point (see Lines 1507–1509 and 1535–1536).
> >
> > No, this is incorrect. What the authors have shown is an iteration complexity statement, that an approximate stationary point is achieved by the algorithm within a certain number of steps. This is essentially showing that there exists a subsequence of iterates that have diminishing stationary measure. What the reviewer asked is about asymptotic convergence to stationary points along the whole iterate sequence. For a simple counterexample, consider the sequence 1, 1, 1/2, 1, 1/3, 1, 1/4, 1, .... This odd subsequence $1/n$ converges to zero at a sublinear rate, but the even subsequence does not converge to zero.
> >
> >
> > >Response. Assume that $\bar{V}^{t}$ is a local critical stationary point ..
> >
> > This argument is unclear to me. It seem that the authors are trying to argue that if the approximate solution $V^{t}$ to the subproblem is close to the actual solution $\bar{V}^{t}$, then one still satisfies a descent lemma. This type of statement cannot be true unless one specifies the magnitude of approximation error. In fact, a main contribution in the following reference [1] is to show that for Riemannian block majorizaiton-minimizaiton, one does not need to solve the sub-problems exactly and still obtain the same guarantees for asymptotic convergence and iteration complexity, as long as the *sub-optimality gaps are summable*. Otherwise, the accumulated errors diverge.
> >
> > [1] Yuchen Li, Laura Balzano, Deanna Needell, Hanbaek Lyu, “Convergence and Complexity Guarantee for Inexact First-order Riemannian Optimization Algorithms.” ICML 2024
> >
> > Please provide more details on the sketch. If it can be justified, the authors should include this discussion in the appendix to support their claims. Otherwise, the claim should be omitted.
> >
> >
> > >Weakness 6: First, even with exact computation of sub-problems, they do not show that the iterates asymptotically converge to the stationary points.
> >
> > >Response. In Theorem 4.2, we have demonstrated that OBCD asymptotically converges to the Block-k stationary points, regardless of whether the coordinates are selected randomly or cyclically.
> >
> > Again, the authors should distinguish iteration complexity and asymptotic convergence.
> >
> >
> > > Response. We note that our analysis establishes only the finite-length property of OBCD, which is independent of the KL exponent . In the future, we aim to extend this work by incorporating the convergence rate of OBCD through a deeper exploration of the KL exponent. *Nonetheless, the finite-length property itself is a significant and state-of-the-art theoretical result.*
> >
> > The authors should clearly state in the statement of Theorem 4.8 that the KL exponent $\sigma$ is $0$. It is not surprising to achieve stronger statement under stronger assumption. Without justifying that the assumption being made (here KL with $\sigma=0$)  is not that restrictive and some important class of problems satisfies it, the significance of this result is not convincing.

---

> ### Author Response · Authors · 2024-11-22
>
> **Minor Comments 1: (1) Below (2), the norm on $H$: is it the spectral norm?**
>
> **Response.** The H-norm is not an operator norm; it is essentially a vector norm obtained by stacking the matrix into a vector.
>
> **Minor Comments 2: (2) L47: $||X||_p$ is not coordinate-wise separable.**
>
> **Response.** $||X\||_p$ is defined as $||X||_p= \sum_{i} \sum_j |X_{i,j}|_p$, where $p=0$ or $p=1$. Clearly, it can be decomposed into $n\times r$ components with $X\in R^{n\times r}$. Therefore, it is coordinate-wise separable.
>
> **Minor Comments 3: (3) L88: The authors imply that subgradient descent with dimihsing stepwise is a limitation. If the stepsizes diminish at a rate not too fast (e.g., $O^{-1/2}$), it still give an optimal iteration complexity. Please specify.**
>
> **Response.** Although it theoretically guarantees an optimal rate of $O^{-1/2}$, it may result in a conservative step size and slower convergence in practice.
>
> **Minor Comments 4: (4) The authors are recommended to cite the following references and discuss their relevance/differences. [1] Breloy, Kumar, Sun, and Palomar, "Majorization-Minimization on the Stiefel Manifold With Application to Robust Sparse PCA", IEEE TRANSACTIONS ON SIGNAL PROCESSING, VOL. 69, 2021 [2] Yuchen Li, Laura Balzano, Deanna Needell, Hanbaek Lyu, “Convergence and Complexity Guarantee for Inexact First-order Riemannian Optimization Algorithms.” ICML 2024 [3] Yuchen Li, Laura Balzano, Deanna Needell, Hanbaek Lyu, “Convergence and complexity of block majorization-minimization for constrained block-Riemannian optimization” arXiv:2312.10330 (2023)**
>
> **Response.**
>
> Generally speaking, none of the three references list above address Problem (1) when $h(X) \neq 0$. OBCD is the first greedy descent method with monotonicity for this class of nonsmooth composite problem.
>
>
> **Minor Comments 5: (5) Summary: The authors state that there are not much known results about block marjoziation-minimization or similar type of algorithms for nonconvex nonsmooth problems. This is simply not true. Please revise this assessment.**
>
> **Response.** We appreciate the reviewer for highlighting relevant references on block majorization-minimization algorithms. We will carefully review these works and include appropriate citations in the revised manuscript to provide a more comprehensive context.
>
>
> **Minor Comments 6: (6) In Lemma 2.1, the authors define $X^+$ and then specify $X$. They should first say "For any $X$, define $X^+=..$.."**
>
> **Response.** Thank you for your suggestion; we will make the necessary adjustments accordingly.
>
> **Minor Comments 7: (7) In Algorithm 1, $K$ is not defined nor referred.**
>
> **Response.** $K$ is defined in Equation (9) and referenced in the proof.
>
> **Minor Comments 8: (8) In eq. (6), remove comma.**
>
> **Response.** The comma in Equation (6) appears to be correctly placed.
>
> **Minor Comments 9: (9) The construction of quadratic surrogate in (9) resembles very much the discussion in Breloy et a. [1]. The authors should discuss the relation.**
>
> **Response.** We will include a citation to Breloy et al. [1] in the updated manuscript and discuss the relation in detail.
>
> **Minor Comments 10: (10) L215: The authors claim that the so-constructed quadratic surrogate can be exactly minimized over $St(k,k)$. But this is the case only if the nonsmooth part is zero (by SVD) or if $k=2$. This should be clearly stated.**
>
> **Response.** Please refer to the subsections "Solving the General OBCD Subproblems" and "Smallest Possible Subproblems When $k=2$" for a detailed explanation.
>
> **Minor Comments 11: (11) Table 1: The numbers are too small to read. There is no point of adding this kind of table since it is unreadable anyways.**
>
> **Response.** We will retain only some part of the table to ensure readability and clarity.
>
> **Minor Comments 12: (12) Appendix C.3: The authors compared computational complexity of computing the Riemannian gradient directly v.s. incrementally updating the gradient using their k-row approach.**
>
> **Response.** Yes, this comparison highlights the merits of coordinate descent, which is known for its reduced computational overhead and efficiency in handling large-scale problems.

---

> > ### Comment · Reviewer_DbfY · 2024-11-25
> >
> > >Generally speaking, none of the three references list above address Problem (1) when . OBCD is the first greedy descent method with monotonicity for this class of nonsmooth composite problem.
> >
> > This statement is incorrect. Reference [2] handles nonsmooth optimization over general Riemannian manifolds using block tangential majorization-minimization, which is a more general setting than the current paper considers. The authors makes the same incorrect claim in the paper (line 109) "no existing BCD methods can address Problem (1) when $h(X)\ne 0$", which should be fixed. Also, other references (especially [1]) is highly relevant (MM on Stiefel manifold). The paper does not give a through literature review, especially on BCD/BMM on Riemannian manifolds. This should help strengthen the positioning the work.
> >
> > >Response. $\mathcal{K}$ is defined in Equation (9) and referenced in the proof.
> >
> > I think this response very well illustrates the authors' style of writing that makes the paper hard to read. Simply imagine a reader first reading Algorithm 1 and tries to understand how it works. There is an undefined function $\mathcal{K}$ and may wonder where it is defined. The authors say it is ok since it is defined in eq. (9), but it appears almost after a full a page and especially within a proof. So a reader must have read the detailed computation first in order to parse the algorithm. It is ok that it is not fully defined within the algorithm, but there should at least be a pointer to a location where it is defined. The authors should try to improve the flow of the paper.
> >
> > >Response. Please refer to the subsections "Solving the General OBCD Subproblems" and "Smallest Possible Subproblems When "$k=2$" for a detailed explanation.
> >
> > Again, the authors' response is missing the point. First, the statement in line 215 that "..the surrogate minimization subproblem  can be  efficiently and exactly solved due to our assumption" is not entirely correct, since in Lemma 2.4, it is reformulated as a quadratic minimization over $St(k,k)$, which is exactly solvable only when $h=0$ (by SVD) or $k=2$ by the proposed method (the authors state this in Remark 2.5). I suggest revising it to something like "..the surrogate minimization subproblem  can be  efficiently and exactly solved when $h=0$ or $k=2$ due to our assumption, see Remark 2.5." Second, again, the way that the authors write is not reader-friendly. How can a reader know what forthcoming subsection to look at when there is a simple claim without any reference? I think the paper's technical contribution is nice but the presentation hinders it very much.

---

> ### Author Response · Authors · 2024-11-28
>
> The reviewers have provided a series of constructive and valuable comments that have greatly helped improve the quality of our paper, for which we are deeply grateful.
>
> Below, we address the key points raised by the reviewers.
>
> **K1. Complexity of OBCD using the optimality measure of Riemannian subgradient.**
>
> **Response**. In the updated manuscript (see attached), we have established the ergodic convergence rates of OBCD using the optimality measure based on the Riemannian subgradient. See Theorem 4.6 in L368-370.
>
>
> **K2. The paper does not give a through literature review, especially on BCD/BMM on Riemannian manifolds**
>
> **Response**.  In the updated manuscript (see attached), we have expanded the literature review to include discussions on BCD/BMM methods on Riemannian manifolds. See lines 92-98.
>
> **K4. the authors does not show asymptotically this proposed measure goes to zero**
>
> **Response**.  We have established the non-ergodic/last-iterate convergence rate for OBCD in Theorem 4.11.
>
> **K5. The subproblem is solved approximately.**
>
> **Response**.  We show that, as long as the proposed algorithm exhibits some descent property that $\mathcal{K}(\bar{V}^t;X^t,B) \leq \mathcal{K}(I_k;X^t,B)$, sufficient descent condition is still guaranteed (See Theorem 4(a)). Thus, convergence to a weaker optimality condition for the ﬁnal solution $X^{\infty}$ is achievable. This condition can be achieved with reasonable initialization, and differs slightly from the condition of the approximate solution in Reference [1] mentioned by the reviewer.
>
> **K6. Other Important Writing Issues**
>
> 1. "OBCD is the first greedy descent method with monotonicity for this class of nonsmooth composite problem". We have removed this sentence (Note that our small-sized subproblem can be solved exactly, while most existing methods can only solve it approximately).
>
> 2. "No existing BCD methods can address Problem (1) when $h(X)\neq 0$". We have removed this sentence.
>
> 3. We have added a pointer to the location where the notation $\mathcal{K}$ is defined.
>
> 4. We use $||X||sp$ to denote the spectral norm of the matrix $X$, and $||X||p=\sum_{i,j}|X_{i,j}|_p$ with $p=1$ or $p=0$.

---

> > ### Author Response · Authors · 2024-12-02
> >
> > Dear Reviewer DbfY,
> >
> > As the discussion phase comes to a close, we sincerely hope that our additional response and the updated manuscript have addressed your concerns. If so, we would greatly appreciate your consideration in raising the score.
> >
> > If there are any remaining concerns, we would be grateful if you could let us know, and we will make every effort to further refine and enhance our work.
> >
> > Best regards,

---

### Meta-Review · Area_Chair_tCER · 2024-12-14

**Metareview:**

This paper studies block coordinate descent method (BCD) for nonsmooth composite optimization with orthogonality constraints. The authors proposed a new method that minimizes a majorizing surrogate of the block restricted objective, which is obtained by sampling k rows and formulating the subproblem accordingly. Though there are some interesting ideas, the paper is not well written, making it hard to read and difficult to verify the results and claims.

**Additional Comments On Reviewer Discussion:**

Discussed the novelty. The reviewers were not convinced.

---

### Decision · Program_Chairs · 2025-01-22

Reject